# How catchment characteristics influence hydrological pathways and travel times in a boreal landscape

Elin Jutebring Sterte[1,2], Fredrik Lidman[1], Emma Lindborg[2], Ylva Sjöberg[3], Hjalmar Laudon[1]

[1]Department of Forest Ecology and Management, Swedish University of Agricultural Sciences, SE-901 83 Umeå, SWEDEN

[2]DHI Sweden AB, Skeppsbron 28, SE- 111 30 Stockholm, SWEDEN

[3]Center for Permafrost (CENPERM), Department of Geosciences and Natural Resource Management, University of Copenhagen, Øster Voldgade 10, 1350 Copenhagen, Denmark

*Corresponding author*: Elin Jutebring Sterte (eljs@dhigroup.com)

**Key Points:**

- A numerical model was used to estimate annual and seasonal mean travel times across 14 long-term monitored nested catchments in the boreal region of northern Sweden.
- The estimated travel times and young water fractions were consistent with observed variations of base cation concentration and stable water isotopes, $\delta^{18}O$.
- Soil type was the most important factor regulating the variation in mean travel times between different sub-catchments
- The areal coverage of mires increased the young water fraction, especially during the spring.
- The greatest seasonality in mean travel times was found in catchments dominated by silty soils because of the long travel times in winter and relatively short travel times in spring.

**Abstract**

Understanding travel times and hydrological pathways of rain and snowmelt water transported through the landscape to recipient surface waters is critical in many hydrological and biogeochemical investigations. In this study, a particle tracking model approach in Mike SHE was used to investigate the pathway and its associated travel time of water in 14 partly nested, long-term monitored boreal sub-catchments of the Krycklan catchment (0.12-68 km$^2$). This region is characterised by long and snow-rich winters with little groundwater recharge and highly dynamic runoff during spring snowmelt. The geometric mean of the annual travel time distribution (MTT$_{geo}$) for the studied sub-catchments varied from 0.8-2.7 years. The variations were related to the different landscape types and their varying hydrological response during different seasons. Winter MTT$_{geo}$ ranged from 1.2-7.7 years, while spring MTT$_{geo}$ varied from 0.5-1.9 years. The modelled variation in annual and seasonal MTT$_{geo}$ and the fraction of young water (<3 months) was supported by extensive observations of both $\delta^{18}$O and base cation concentrations in the different streams. The travel time of water to streams was positively correlated to the area coverage of low conductive silty sediments (r=0.90, P<0.0001). Catchments with mixed soil-landscape settings typically displayed larger variability in seasonal MTT$_{geo}$, as contrasting hydrological responses between different soil types (e.g., peat in mires, till and silty sediments) are integrated. The areal coverage of mires was especially important for the young water contribution in spring (r=0.96, P<0.0001). The main factor for this was attributed to extensive soil frost in mires, causing considerable overland flow during the snowmelt period. However, this lower groundwater recharge during snowmelt caused mire-dominated catchments to have longer stream runoff MTT$_{geo}$ than comparable forest catchments in winter. Boreal landscapes are sensitive to climate change, and our results suggest that changes in seasonality are likely to cause contrasting responses in different catchments depending on the dominating landscape type.

# 1 Introduction

The pathways and associated travel times of water through the terrestrial landscape to stream networks is a widely discussed topic in contemporary hydrology. This interest has emerged because of the significant role travel time and routing of water through various subsurface environments have on hydrological and biogeochemical processes (McDonnell et al., 2010; Sprenger et al., 2018). This includes fundamental implications for weathering rates (Burns et al., 2003), transport and dispersal of contaminants (Bosson et al., 2013; Kralik, 2015), and accumulation and mobilisation of organic carbon and associated solutes (Tiwari et al., 2017). The travel time, from precipitation input to the outflow into streams, provides valuable information about catchment sensitivity to changes in land use and climate and the fate of long-range transport of contaminants and nutrients deposited with precipitation (van der Velde et al., 2012). The travel time distribution can vary substantially in time and space, depending on catchment characteristics and hydrological conditions, including, for example, slope, catchment size, soil heterogeneity, and seasonality (Botter et al., 2010; Lin, 2010; Heidbüchel et al., 2012; Hrachowitz et al., 2013). Therefore, estimating travel times for contrasting landscape elements is challenging, but when successful, it will enhance our ability to understand and predict catchment functioning more adequately.

Stream water consists of a blend of overland flow and groundwater of different ages. The mean travel time (MTT) to streams is calculated as the average age of this mix (McGuire et al., 2006). The baseflow is the part of stream groundwater contribution that generally has travelled the furthest and is the oldest (Klaus et al., 2013; Hrachowitz et al., 2016). In contrast, young stream water is typically connected to overland flow or shallow subsurface pathways, which mainly can be seen at times with large rain or snowmelt inputs (Peters et al., 2014; Hrachowitz et al., 2016). The variability of water sources makes the travel time distribution difficult to quantify, especially on intra-annual time scales, as they vary in time and space depending on numerous scale-dependent and scale-independent processes (Botter et al., 2010). A better understanding of the seasonal variability in the fraction of young and old waters can help provide insights into the fundamental role catchment characteristics play in regulating the hydrology and biogeochemistry of streams and rivers.

Stable water isotopes and biogeochemical tracers are common tools applied in field investigations to locate water sources and follow their pathways through the landscape (Maulé and Stein, 1990; Rodhe et al., 1996; Goller et al., 2005; Tetzlaff and Soulsby, 2008). Isotopic tracer signal dampening can provide an estimate of MTT (Uhlenbrook et al., 2002; McGuire et al., 2005; Peralta-Tapia et al., 2016), and more elaborate time-series analysis can offer quantitative assessments of travel times (Harman, 2015; Danesh-Yazdi et al., 2016). However, the isotope amplitude signal used to estimate MTT in many transfer functions is lost after approximately four to five years because of effective mixing (Kirchner, 2016), limiting the use of isotopes in catchments with long travel times. The young water fraction, often defined as water younger than two to three months, can, however, still be quantifiable in such catchments (von Freyberg et al., 2018; Lutz et al., 2018; Stockinger et al., 2019). The main advantage of water isotopes is that they are relatively conservative and fractionate primarily because of evaporation. Hence, once in the subsurface environment, the signal is only affected by mixing different water sources. In contrast, many biogeochemical tracers react and transform on their route to streams (Lidman et al., 2017; Ledesma et al., 2018). Such transformation and reactions depend on the specific solute and soil environment that water encounters and, therefore, give qualitative information about groundwater flow pathways (Wolock et al., 1997; Frisbee et al., 2011; Zimmer et al., 2012). Combined information from conservative and reactive tracers can hence provide an enhanced understanding of hydrological processes as their concentrations and dynamics can tell complementary stories about the specific pathways water take from the source to the recipient stream (Laudon et al., 2011).

A complementary approach to field experiments is numerical modelling, which can help achieve a more complete system understanding. Lumped hydrological models often describe catchments as single integrated entities. In contrast, distributed numerical models can include spatial heterogeneity in input parameters and therefore have the potential to represent catchment processes more mechanistically. In turn, this can lead to a more process-based understanding of hydrology and biogeochemistry at the catchment-scale (Brirhet and Benaabidate, 2016; Soltani, 2017). Common methods to calculate travel times using numerical methods include models using solute transport routines and particle tracking (Hrachowitz et al., 2013; Ameli et al., 2016; Kaandorp et al., 2018; Remondi et al., 2018; Yang et al., 2018; Heidbüchel et al. 2020). Models, however, need – as far as possible – proper tests against empirical observations to build confidence in their output. Stream discharge, groundwater levels, and tracer data are examples of such validation data that can provide vital information (McGuire et al., 2007; Hrachowitz et al., 2015; Wang et al., 2017). The collection of such field data is, however, costly and time-consuming. Therefore, data for calibration and validation is often limited, and the minimum length and types in data-sparse catchments is currently a topic of increasing interest (Bjerklie et al., 2003; Jian et al., 2017; Li et al., 2018).

Snow-dominated landscapes have received increasing attention in the last decades due to their importance as water resources (Barnett et al., 2005) and their vulnerability to climate change (Tremblay et al., 2011; Aubin et al., 2018). Landscapes with long-lasting snow cover that often melts rapidly in the spring create both opportunities and challenges for determining the pathways and travel time of water discharging to streams. The long, snow-rich winters do not only cause protracted periods of winter baseflow with little or no recharge (Spence et al., 2011; Spence and Phillips, 2015; Lyon et al., 2018), they also cause considerable amounts of water during the often short and intensive snowmelt in the spring. Although attempts to assess travel times generally have provided useful results using, for example, models to reconstruct isotope signal dampening in snow-dominated catchments, the winter season has proven to be especially challenging, suggesting that other methods to assess travel times may be required (Heidbüchel et al., 2012; Peralta-Tapia et al., 2016). The boreal region also consists of numerous patches of lakes and mires, interspersed in a landscape dominated by coniferous forests on different soil types, which makes this task even more challenging. Hence, accounting for the unique circumstances of both baseflow with long travel times and those of the intensive spring snowmelt with potential large overland flow components in heterogeneous landscapes requires models that can handle the complexity and separation of various flow components across scales, soil types, and landscape patches.

To overcome previous limitations, this study used particle tracking in the physically based distributed numerical model, Mike SHE (Graham and Butts, 2005), to enhance our understanding of stream water contribution in boreal landscapes across seasons and landscape configurations. The water movement model in Mike SHE calculates saturated (3D) groundwater flow and unsaturated (1D) flow and is fully integrated with the surface water and evapotranspiration. The water flow model setup and results previously presented by Jutebring Sterte et al. (2018) were used as the study platform for this work. The model has been calibrated and validated to 14 sub-catchments using daily stream-discharge observations and periodical measured groundwater levels in 15 wells throughout the Krycklan catchment in the boreal region of northern Sweden (Laudon et al., 2013; Jutebring Sterte et al., 2018). The model complexity allows for an in-depth investigation of advective travel times by non-reactive particle tracking simulations in a transient flow field.

The main objective of this study was to quantify annual and seasonal (winter, spring, and summer) travel time distributions and calculate MTT of water runoff to streams of the Krycklan sub-catchments to disentangle how these are related to physical landscape

characteristics and variation in groundwater recharge. Firstly, the credibility of the model results was tested by comparing calculated travel times for the 14 sub-catchments to ten-year observational records from Krycklan, including average seasonal changes of stream isotope signatures and base cation concentrations. The usefulness of stream isotopic composition and chemistry record has previously been demonstrated for understanding the connection of hydrological flow pathways and travel times for this site (Laudon et al., 2007; Peralta-Tapia et al., 2015), but with the limitation of studies on only short periods or single catchments. Secondly, the purpose was to go beyond what was previously done by identifying the connection between travel times and different catchment characteristics and test how this varies depending on the hydrological conditions. This was accomplished by capturing contrasting seasons such as the low flow conditions in winter with limited input of new precipitation, high flow in spring when the system is still partly frozen, and summer when evapotranspiration (ET) becomes a significant process. We focused especially on the catchment characteristics that have been suggested to be important factors for regulating stream chemistry of the Krycklan sub-catchments, including the areal coverage of mires, catchment size, soil properties, and seasonal changes in groundwater recharge (Karlsen et al., 2016; Klaminder et al., 2011; Laudon et al., 2007; Peralta-Tapia et al., 2015; Tiwari et al., 2017).

## 2 Method

### 2.1 Site Description

The Krycklan study catchment, located in the boreal region at the transition of the temperate/subarctic climate zone of northern Sweden, spans elevations from 114 to 405 m.a.s.l (Fig. 1, Table 1). The characteristic vegetation of this boreal landscape is the dominance of Scots pine (*Pinus sylvestris*) and Norway spruce (*Picea abies*), covering most of the catchment (Laudon et al., 2013). In this study, we refer to soil as all unconsolidated material above the bedrock.

Krycklan has a landscape distinctively formed by the last ice age (Ivarsson and Johnsson, 1988; Lidman et al., 2016). At the higher elevations to the northwest, located above the highest postglacial coastline, the soils can reach up to 15-20 m in thickness. Here, the soil primarily consists of glacial till, and the landscape is intertwined with lakes and peatlands. The deeper soils consist of basal till which was deposited and compacted under the moving ice. In contrast, the shallower till layers consists primarily of ablation till, which is less compact since it mainly has been compacted by its own weight (Goldthwait, 1971). This causes a decreasing hydraulic conductivity with depth, which is characteristic for glacial till in northern Sweden (Bishop et al., 2011; Nyberg, 1995; Seibert et al., 2009). At lower elevations, the soils consist of fluvial and glaciofluvial deposits of primarily sandy and silty sediments. Compared to the soil at higher elevations in the catchment, these deposits can reach thicknesses up to approximately 40 to 50 m and have a hydrological conductivity that is more constant with depth because these soil types have mainly been compacted only by their own weight.

For more than 30 years, multi-disciplinary biogeochemical and hydrological studies have been conducted in Krycklan (e.g., Laudon and Sponseller, 2018). Streamflow is monitored in 14 nested sub-catchments, called C1 to C20, with the longest continuously monitored time-series starting at the beginning of the 1980s. Connected by a network of streams, the different sub-catchments allow an evaluation of the effects of catchment characteristics on hydrologic transport, including soil type, vegetation, and differences in topography (Table 1).

**Table 1: Sub-catchment characteristics.** The list includes all 14 monitored sub-catchments in Krycklan, called C1 to C20, including the entire Krycklan catchment, C16. Different branches of the stream network are gathered in the table and illustrated in distinct colours in Fig. 1. The table includes the sub-catchment area, average elevation, and average slope. Further descriptions of these characteristics can be found in Karlsen et al. (2016). The table also includes soil proportion based on the soil map (1:100,000) from the Swedish Geological Survey (2014).

| | Catchment size (km²) | Average elevation (m.a.s.l.) | Slope (°) | Till (%) | Mire (%) | Sandy sediments (%) | Silty sediments (%) | Lake (%) |
|---|---|---|---|---|---|---|---|---|
| **C2** | 0.12 | 273 | 4.75 | 79 | 0 | 0 | 0 | 0.0 |
| **C4** | 0.18 | 287 | 4.24 | 29 | 42 | 0 | 0 | 0.0 |
| **C5** | 0.65 | 292 | 2.91 | 47 | 46 | 0 | 0 | 6.4 |
| **C6** | 1.10 | 283 | 4.53 | 51 | 29 | 0 | 0 | 3.8 |
| **C7** | 0.47 | 275 | 4.98 | 68 | 16 | 0 | 0 | 0.0 |
| **C9** | 2.88 | 251 | 4.25 | 64 | 14 | 7 | 4 | 1.5 |
| **C13** | 7.00 | 251 | 4.52 | 60 | 10 | 9 | 9 | 0.7 |
| **C1** | 0.48 | 279 | 4.87 | 91 | 0 | 0 | 0 | 0.0 |
| **C10** | 3.36 | 296 | 5.11 | 64 | 28 | 1 | 0 | 0.0 |
| **C12** | 5.44 | 277 | 4.90 | 70 | 18 | 6 | 0 | 0.0 |
| **C14** | 14.10 | 228 | 6.35 | 46 | 6 | 24 | 15 | 0.7 |

| | | | | | | | |
|---|---|---|---|---|---|---|---|
| **C20** | 1.45 | 214 | 5.96 | 55 | 9 | 0 | 28 | 0.0 |
| **C15** | 19.13 | 277 | 6.38 | 64 | 15 | 8 | 2 | 2.4 |
| **C16** | 67.90 | 239 | 6.35 | 51 | 9 | 21 | 10 | 1.0 |

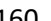

(a)

(b)

(c)

(d)

**Figure 1: The Krycklan catchment.** (a) Location of sub-catchment and their outlets. The areas are colour-coded based on their stream network connections, e.g., all sub-catchments of one colour connect before reaching the white area. For further details of the catchment characteristics, see Table 1. (b) The figure shows the soil map used in the Mike SHE flow model which is based on data from the Swedish

 Geological Survey soil map (1:100,000) and field investigations. (c) Soil depth to bedrock map taken from the Swedish Geological Survey, 2014, and is shown in metres below the ground surface (m.b.g.s.). (d) Catchment topography, shown as metres above sea level (m.a.s.l.).

## 2.2 Linking seasonal base cation concentration and isotopic signature to travel times to stream water

This study was focused on three seasons in Krycklan: winter, spring, and summer (Table 2, Table A1, Appendix). For evaluation of stream chemistry, we defined the winter as late early December to late February, from the time air temperatures were below zero °C, until the air temperature started to rise above freezing temperatures again, causing snowmelt. This season is characterised by an extensive and permanent snow cover with little or no groundwater recharge. We assumed that the winter stream composition reflects the chemistry of deeper groundwater (Fig. 2). Similarly, we defined spring as the hydrological period directly influenced by the snowmelt. The main part of the snowmelt and spring flood occurs in April-May. During snowmelt, ca 50% of the annual precipitation leaves the system in a short period of time, diluting baseflow with new input of water. Finally, we defined the summer season as the period between July and September when the hydrology is characterised by rain, high ET, and relatively little runoff. March, June, October, and November were excluded because, hydrologically, they are typically transition months between the three distinct seasons. This is because snowmelt influences runoff in March and June. October and November are transitional months between summer and winter conditions with irregularly occurring snowfall and soil frost events.

In this study, stable water isotopes ($\delta^{18}O$) were used to track pathways of precipitation inputs to stream networks (see Appendix for the $\delta^{18}O$ definition). Ten years of $\delta^{18}O$ measurements for 13 of the 14 sub-catchments were used. Isotopic fractionation caused by lake surface evaporation affects the isotopic signal of some of the sub-catchments (Leach and Laudon, 2019). This fractionation was corrected by accounting for the percentage of lakes in each sub-catchment (Table 2), using the same principle as Peralta-Tapia et al. (2015) but adjusted to newly acquired $\delta^{18}O$ observations (Eq. A7, Appendix).

The comparison of the modelling results to observations of $\delta^{18}O$ was based on a conceptual model of the seasonal variability and differences between precipitation and runoff (Fig. 2a). The precipitation signal varies on a seasonal basis, creating an amplitude difference (Fig 2b). This amplitude is reduced due to groundwater mixing until complete mixing is reached and the groundwater receives the same signal as the long-term precipitation average. There is no or little groundwater recharge during winter because almost all precipitation inputs arrive and accumulates as snow. Hence, we assume that the stream isotopic signature originates from groundwater only (Laudon et al., 2007; Peralta et al., 2015). Consequently, the closer the stream signature comes to the long-term precipitation average, the more the groundwater has been mixed. The groundwater isotopic signature, in turn, should be correlated to the travel time to stream until full mixing of the precipitation signal is reached. The closer the signature is to the long-term precipitation average (which is equal to the deep groundwater measurements in Krycklan (Laudon et al., 2007)), the more well-mixed and, consequently, the longer travel times will be found. We used the average annual winter signature for the evaluation. In spring, previous studies have shown that the young water fraction can be distinguished by comparing the change in the isotopic signature to the preceding winter because the snow is much lighter (depleted in $^{18}O$) (Laudon et al., 2007; Tetzlaff et al., 2015). We calculated the difference between the average winter and average spring signature for all years. The mean difference we hereafter refer to as the $\Delta\delta^{18}O_{spring}$, which we assumed to be negatively correlated to the young water fraction (Fig. 2a). Similarly, we refer to the mean difference between annual averages of winter and summer signature as $\Delta\delta^{18}O_{summer}$, which similarly should be related to the young water fraction during the summer. However, in summer, precipitation is heavier (more enriched in $^{18}O$) than in winter, which hence should give the young water a heavier signal. Therefore, we assumed a positive relationship between the young water fraction and the $\Delta\delta^{18}O_{summer}$ (Fig. 2a).

Another indicator of travel times to stream that we used was the sum of base cations (BC) concentration (Fig. 2b) (Abbott et al., 2016). Previous attempts to follow the chemical development of groundwater in the Krycklan catchment and other streams have shown that the BC concentration increases along the groundwater flow pathway (Klaminder et al., 2011). Therefore, a correlation between the stream concentration of BCs on the one hand and modelled soil contact time on the other were assumed in this study. The BCs are mainly derived from the weathering of local soils in the Krycklan catchment, with only a minor contribution from atmospheric deposition (Lidman et al., 2014). Our assumption is further based on modelling studies of weathering rates in a soil

transect in the Krycklan catchment, which indicates that there is a kinetic control of the release of BCs in the soils (Erlandsson et al., 2016). Since all BCs behave relatively conservatively in these environments (Ledesma et al., 2013; Lidman et al., 2014), we used their combined concentration as a proxy for soil contact time. However, the assumption is only valid when the water is in contact with mineral soils, not with peat in mires, which are abundant in some of the investigated sub-catchments. There are little minerals present in the peat and, therefore, the BC concentration cannot be expected to increase during the time the water spends

there. Therefore, the BC concentrations were adjusted for the influence of mire, using the sub-catchment mire proportion as a scaling factor to allow a fair comparison to water soil contact time (Lidman et al., 2014) (Table 2).

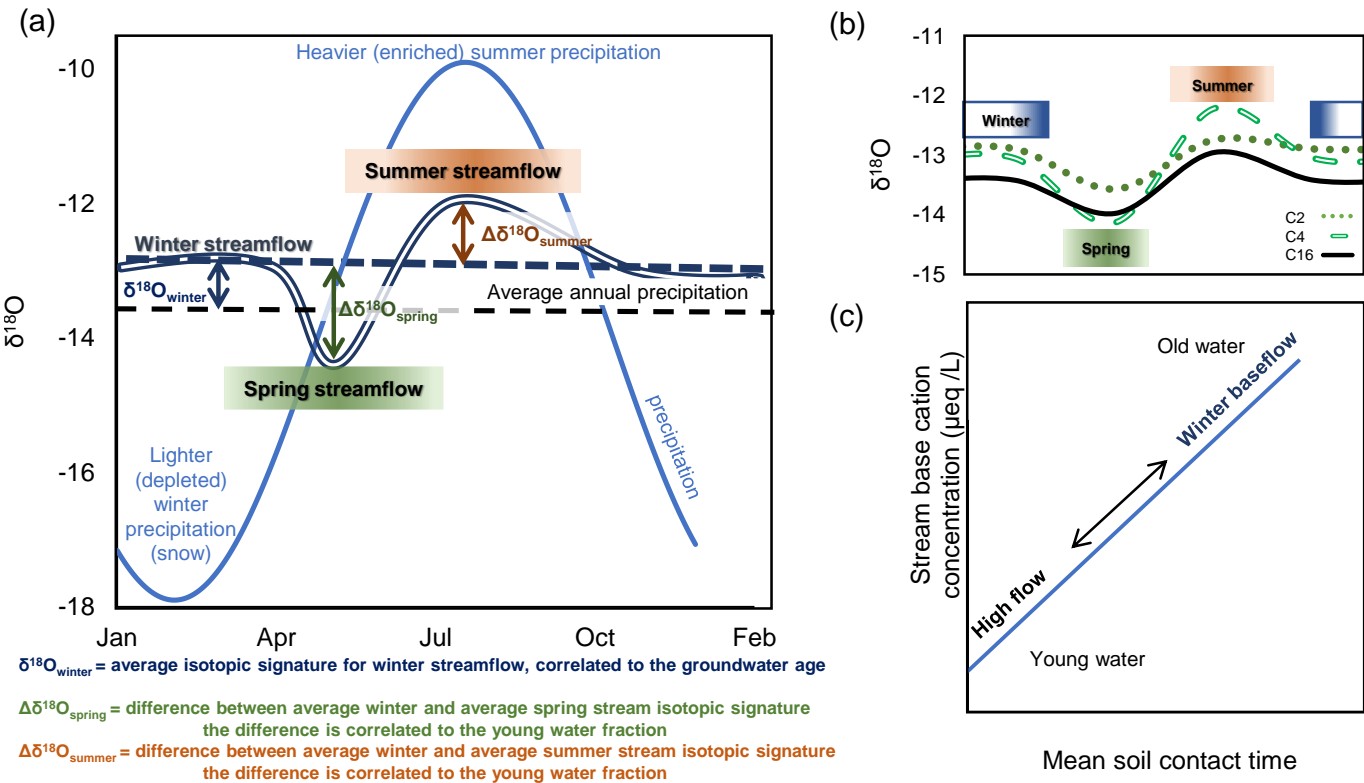

**Figure 2: Conceptual figure of travel time to stream vs stream isotopic signature (a and b), and stream base cation concentration (c).**
(a) The connection between $\delta^{18}O$ and travel time to stream, where the sine curve shows the annual variations of $\delta^{18}O$ in precipitation, and approximate seasonal winter, spring, and summer stream compositions are marked and exemplified by the annual changes of C4. In winter, the travel times are related to the average deviation in the isotopic signature between the winter baseflow and the long-term precipitation. In spring, the fraction of young water is correlated to the difference between the average spring stream signature and the average winter baseflow. In summer, the fraction of young water is correlated to the difference between the average summer stream signature and
average winter baseflow. (b) Seasonal $\delta^{18}O$ averages for three example streams: C2, C4 and C16. (c) The connection between base cation (BC) concentration and soil contact time. The longer time the water spent in the mineral soil, the higher the stream concentrations of BCs will be due to soil weathering.

All stream chemistry data comes from the online open Krycklan database (Table 2) (Krycklan Database, 2021). The isotopic
signatures contain approximately ten years of field observations (2008 to mid-2018), approximately 25 samples per year for each

site. Parts of the dataset have been published by Peralta-Tapia et al. (2016), where sampling and analyses are described in detail. It has since been expanded using the same methodology. We used the average winter isotope signatures from these years as a representation of baseflow. These averages were also compared to the volume-weighted average of the long-term precipitation, calculated using approximately 1000 precipitation measurements of $\delta^{18}O$ between 2007 and 2016. The precipitation was measured throughout the year, both as rain and as snow. The long-term precipitation average is -13.4 ‰, which is close to equal to observations of the isotopic signature at the deep groundwater wells of Krycklan (ca. 10 m depth). The BC data collection methodology is reported in Ledesma et al. (2013).

**Table 2: Seasonal stream chemistry.** The table includes average winter signatures (‰) and the average difference between annual winter-spring and winter - summer signatures ($\Delta\delta^{18}O$). The table also includes average winter, spring, and summer BC concentrations.

| | $\delta^{18}O$[a] | | | | | | Base cations (BC)[b] | | | | | |
| | Winter | | Spring | | Summer | | Winter concentration | | Spring concentration | | Summer concentration | |
| | ‰ | SD/SEM[c] | $\Delta\delta^{18}O$ | SD/SEM | $\Delta\delta^{18}O$ | SD/SEM | µeq/L | SD/SEM | µeq/L | SD/SEM | µeq/L | SD/SEM |
|---|---|---|---|---|---|---|---|---|---|---|---|---|
| **C2** | -12.9 | 0.46/0.07 | -0.68 | 0.52/0.16 | 0.15 | 0.45/0.16 | 264 | 99/20 | 189 | 43/6 | 267 | 58/9 |
| **C4** | -13.1 | 0.36/0.06 | -1.08 | 0.66/0.20 | 0.82 | 0.48/0.21 | 263 | 71/16 | 120 | 48/8 | 306 | 77/12 |
| **C5** | -13.0 | 0.47/0.08 | -1.80 | 0.66/0.20 | 0.72 | 0.65/0.21 | 267 | 67/16 | 202 | 59/8 | 231 | 34/5 |
| **C6** | -13.1 | 0.35/0.06 | -1.27 | 0.55/0.16 | 0.52 | 0.47/0.17 | 321 | 61/12 | 233 | 104/14 | 322 | 120/16 |
| **C7** | -13.0 | 0.22/0.04 | -0.73 | 0.56/0.17 | 0.42 | 0.37/0.18 | 271 | 40/8 | 191 | 57/8 | 270 | 38/5 |
| **C9** | -13.1 | 0.29/0.05 | -0.98 | 0.46/0.14 | 0.57 | 0.44/0.15 | 349 | 57/11 | 231 | 73/10 | 327 | 61/8 |
| **C13** | -13.1 | 0.26/0.05 | -0.83 | 0.55/0.16 | 0.60 | 0.48/0.17 | 338 | 57/11 | 223 | 50/6 | 309 | 43/6 |
| **C1** | -12.9 | 0.28/0.05 | -0.53 | 0.60/0.18 | 0.10 | 0.38/0.19 | 272 | 36/7 | 229 | 44/6 | 285 | 31/4 |
| **C10** | -13.3 | 0.28/0.05 | -0.80 | 0.61/0.18 | 0.53 | 0.39/0.19 | 314 | 50/10 | 209 | 82/11 | 332 | 72/10 |
| **C12** | -13.1 | 0.30/0.05 | -0.88 | 0.48/0.15 | 0.36 | 0.43/0.16 | 319 | 43/8 | 211 | 58/7 | 316 | 45/6 |
| **C14** | -13.4 | 0.23/0.04 | -0.70 | 0.55/0.17 | 0.48 | 0.45/0.18 | 358 | 34/7 | 272 | 69/9 | 376 | 74/10 |
| **C20** | - | - | - | - | - | - | 519 | 65/13 | 398 | 108/14 | 526 | 60/8 |
| **C15** | -13.4 | 0.40/0.07 | -0.73 | 0.69/0.21 | 0.63 | 0.44/0.22 | 347 | 41/8 | 258 | 60/8 | 349 | 45/6 |
| **C16** | -13.4 | 0.44/0.08 | -0.56 | 0.64/0.64 | 0.46 | 0.33/0.20 | 480 | 68/13 | 272 | 70/9 | 441 | 76/10 |
| **Long term precipitation average - isotopes** | | | | | | | | | | | | |
| | | | | -13.4 ‰ [d] | | | | | | | | |

[a] $\delta^{18}O$ signature (2008-2018), data have been adjusted according to the lake proportion according to Eq. A7, Appendix

[b] Base cation concentration (2008-2016), data has been adjusted according to the mire proportion

[c] SD = standard deviation, SEM = standard error of the mean

[d] Measured precipitation average for isotopes (2007-2016). The precipitation average is close to equal to isotope measurements of groundwater below ca. 10 m.

## 2.3 Water flow model setup

We applied the Mike SHE/Mike-11 hydrological modelling tools to quantify travel times in a pre-calculated 3D transient flow field. The simulated terrestrial hydrological system for the Krycklan catchment includes: the saturated and unsaturated flow, ET, snowmelt, overland flow, and streamflow processes. The fully distributed 3D modelling tool uses topography, soil properties, and time-varying climate inputs to calculate the water fluxes throughout a catchment (Rahim et al., 2012; Sishodia et al., 2017; Wang et al., 2012; Wijesekara et al., 2014). The ET processes include canopy interception, open surface evaporation, root uptake, sublimation, and soil evaporation from the unsaturated zone based on a methodology developed by Kristensen and Jensen (1975). Flow in the saturated zone (SZ) is calculated in 3D by the Darcy equation. The flow in the unsaturated zone (UZ) is calculated in vertical 1D using the Richards equation, and overland flow (OL) is calculated using a horizontal 2D diffusive wave approximation in the Saint-Venant equations (Fig. 3). Streams are modelled in 1D using a high-order dynamic wave formulation of the Saint-

Venant equations. The river model (Mike 11) is not restricted to the grid size of Mike SHE and allows for a more precise calculation of stream water levels and flow rates. The different model compartments OL, UZ, SZ, and rivers are fully integrated, and water fluxes between and within the compartments are calculated in each time step of the simulation. More in-depth documentation and manuals of Mike SHE and Mike 11 is provided by DHI (DHI, 2021).

For the Krycklan model, the horizontal grid was set to 50*50 m. Vertically, the model is divided into ten calculation layers (CL) and extends to a depth of 100 m below ground. The SZ-CLs vary with depth and are thinner closer to the soil surface; the first CLs extend to 2.5 m, 3 m, 4 m, and 5 m, respectively, below the ground surface, with the soil properties and depth extension following the stratigraphy (Table 3). The UZ and SZ interact throughout the soil. If the soil is unsaturated, the UZ discretisation and equations are used. The influence from ET and UZ processes on the SZ is only fully active to the depth of the uppermost SZ-CL. Here, the ET and UZ are calculated at a finer resolution, leading to a detailed calculation of the groundwater table level. The first SZ-CL depth was set to 2.5 m and was calibrated using the influence of the CL thickness on groundwater table level, UZ, and ET dynamics.

Following the thickness of the SZ-CL in the Krycklan model, all soils above 2.5 m depth are prescribed as one soil type, with hydraulic properties being an average of all the soil types throughout the vertical profile from the ground surface to 2.5 m depth. In Mike SHE, horizontal hydraulic conductivity (Kh) is averaged using the thickness of each soil layer. Vertical flows are more dependent on the lowest vertical hydraulic conductivity (Kv). Therefore, the harmonic weighted mean value is used to calculate the new Kv instead (Table 3). A drain function was used in this model and several previous studies (Bosson et al. 2012, 2013, Johansson et al. 2015 and Jutebring et al. 2018) to account for the higher hydraulic conductivity in the uppermost part of the first CL. In the Krycklan model, the function was activated whenever the groundwater reached 0.5 m below the ground surface, above which higher K-values have been observed (Table 3) (Bishop et al., 2011; Nyberg, 1995; Seibert et al., 2009). The model also accounted for soil freezing processes, which in Krycklan have been shown to have a strong influence on the water turnover in mires (Laudon et al., 2011). Based on a methodology presented in Johansson et al. (2015), soil freeze and thawing processes were described using time-varying K and infiltration capacity.

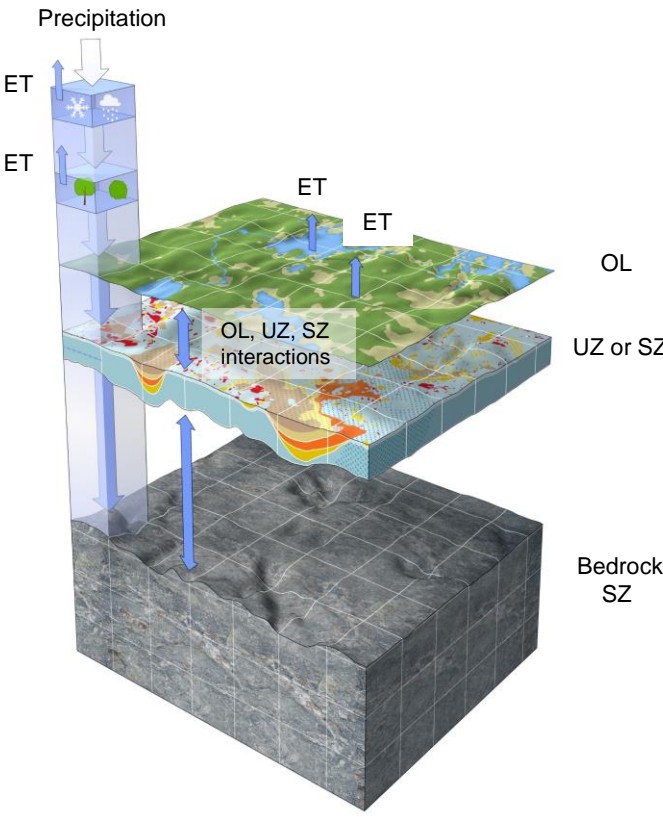

**Figure 3: Schematic of a general Mike SHE model set up.** Precipitation falls on the ground as rain or snow. Evapotranspiration (ET) processes include canopy interception, open surface evaporation, root uptake, and soil evaporation from the unsaturated zone (UZ). The overland flow (OL), saturated zone (SZ), and UZ interact depending on the saturation level. The SZ is divided into ten calculation layers (CL), while the UZ has a much finer description. Streamflow is modelled through Mike 11 and is not restricted to the Mike SHE resolution. The figure is used on the courtesy of SKB. Figure illustrator: LAJ.

The Krycklan flow model was able to reproduce daily accumulated stream discharge, groundwater levels, and timing of precipitation events (Jutebring Sterte et al., 2018). This includes daily discharge observations (14 streams) and weekly to monthly observed groundwater levels (15 wells) for 2009-2014. The accumulated error in stream discharge was on average 11%, and highest for sub-catchments with few observation points (<25%). For this study, a few changes were made to the Krycklan model (Jutebring Sterte et al., 2018). Most importantly, new field data from the Krycklan database gave a more precise location of the observation station at C5 (red circle in Fig. 1). The Kh of silt was also increased from $1*10^{-8}$ m/s to $1*10^{-7}$ m/s due to new soil property samples, which gave a slightly better flow representation of the sites affected silty sediments. However, the corrections and additions did not influence the model results in any substantial way. The improvements were small but were made to better represent the site with the hydrological flow model (Jutebring Sterte et al., 2021).

**Table 3: Flow model setup.** Flow model setup from the calibrated and validated Mike SHE model presented in Jutebring Sterte et al. (2018). The "soil type surface" corresponds to the soil type shown in Fig. 1b. A drain constant was used to account for coarser material of the upper half metre of the soil.

| Soil type surface | Depth below ground (m)[a] | Soil type | Horizontal hydraulic conductivity (m/s) | Vertical hydraulic conductivity (m/s) |
|---|---|---|---|---|
| Till | 2.5 | Till | $2*10^{-5}$ | $2*10^{-6}$ |
| | To bedrock | Fine till | $1*10^{-6}$ | $1*10^{-7}$ |
| | Bedrock | | $1*10^{-9}$ | $1*10^{-9}$ |
| Peat | 5 | Peat | $1*10^{-5}$ | $5*10^{-5}$ |
| | 7 | Clay | $1*10^{-9}$ | $1*10^{-9}$ |
| | To bedrock | Fine till | $1*10^{-6}$ | $1*10^{-7}$ |
| | Bedrock | | $1*10^{-9}$ | $1*10^{-9}$ |
| Silty sediments | 3 | Silt/clay | $1*10^{-7}$ | $1*10^{-7}$ |
| | To bedrock | Fine till | $1*10^{-6}$ | $1*10^{-7}$ |
| | Bedrock | | $1*10^{-9}$ | $1*10^{-9}$ |
| Sandy Sediments | 4 | Silt/Sand | $2*10^{-5}$ | $2*10^{-5}$ |
| | 0.9*max depth | Sand | $3*10^{-4}$ | $3*10^{-5}$ |
| | To bedrock | Gravel | $1*10^{-4}$ | $1*10^{-4}$ |
| | Bedrock | | $1*10^{-9}$ | $1*10^{-9}$ |
| **Drain constant** | | | | |
| Peat | $1*10^{-6}$ | | | |
| Till | $4*10^{-7}$ | | | |
| Silty sediments | $1*10^{-7}$ | | | |

[a] The table shows the depth down to which the same description extends. For example, the first description of peat extends down to 5 m, while the first calculation layer is 2.5 m.

**2.4 Establishing travel times - particle tracking**

Particle tracking in Mike SHE enables investigations of groundwater travel time from the recharge to the SZ until the discharge into the streams, as described in detail in Bosson et al. (2010, 2013). The model calculates the location and age of separate particles added with infiltrating water along their flow lines. The particles move by advection governed by the pre-calculated groundwater flow field from the Mike SHE model (Jutebring et al., 2018, 2021). This method allows for long-term transport calculations where particle tracking can be run for several annual cycles based on the same transient or steady-state flow field. The advection-dispersion equation governs the transportation of particles for a porous medium. The Darcy velocity is divided by the porosity to calculate the groundwater velocity. Therefore, the only complementary input data needed to run the particle was porosity values (Table 4).

Particle tracking was used to assess groundwater travel times from groundwater recharge to stream runoff for each sub-catchment. The model was run for 1000 years to capture the travel times of all discharging groundwater for each sub-catchment. One year of simulated flow results was cycled 1000 times to extend the particle tracking simulation. The year 2010 was selected, as the water balance was close to the long-term annual averages observed for the Krycklan catchment. All particles were released at the top of the transient groundwater table the first year. Numerical constraints restricted the number of particles released to 0.5 particles/10 mm modelled groundwater recharge per grid cell, which corresponds to a total of approximately 0.6 million particles for the entire modelled area in the first year. This number of particles was assumed to be enough to capture the timing of recharge patterns (Fig. 4).

**Table 4: Porosity values for different soil types used in the Mike SHE model.**

| Soil type | Porosity (-) |
|---|---|
| Gravel [a] | 0.32 |
| Sand [b] | 0.35 |
| Silt [c] | 0.45 |
| Clay [b] | 0.55 |
| Silt-clay [d] | 0.50 |
| Till [b] | 0.30 |
| Peat [b] | 0.50 |
| Bedrock [b] | 0.0001 |
| Bedrock fractures/deformation zones [b] | 0.001 |

[a] Average of Morris and Johnson. (1967). [b]Joyce et al. (2010). [c] Average value between sand and clay. [d] Average value between silt and clay

## 2.5 Analysis of modelled travel times, relationship to stream chemistry and landscape characteristics

The time it took for particles to reach a stream or lake via groundwater (hereafter called 'travel time') was calculated for each sub-catchment. The calculated travel time distributions were based on all particles arriving in a stream within a certain period of time, either annually or for a specific season, for the entire modelling period. The distributions were analysed using four statistical measurement tools, the arithmetic mean, the geometric mean, the median, and the standard deviation (SD). The arithmetic mean, the geometric mean, and the median are common choices to describe the central tendency of a distribution (Destouni et al., 2001; Kaandorp et al., 2018; Massoudieh et al., 2012, 2017; Unlu et al., 2004), which all have their strengths and weaknesses. If the distribution is not significantly skewed, the SD is smaller than half of the average (Taagepera, 2008). In the case of the observed $\delta^{18}O$ and BC concentrations (Table 2), the SD is much smaller than half of the average. Therefore, the arithmetic mean was used to describe the central tendency of the data set. However, if the travel time distribution becomes skewed, the arithmetic mean becomes highly sensitive to the tail of the distribution and produces considerable uncertainty. In these cases, the median and the geometric mean are often better as a measure of the central tendency of mean travel time (MTT) than the average. However, to compare the MTT of discharged water of different streams, we still wanted the metric to account for the length of the tail. Therefore, we used the geometric mean because the median only states the middle value of a distribution regardless of the tail length (Taagepera, 2008; Unlu et al., 2004; Zhang et al., 1996). However, we provide all metrics, including the arithmetic mean, the geometric mean, median, and SD, in the Appendix, Table A2.

The MTT was compared to stream chemistry, which is a mix of both groundwater and surface water. In winter, all streamflow contributions originate from groundwater. Here the results from the particle tracking reflect the actual travel time to the streams. However, in summer and especially in spring, some water will reach the streams via overland flow (OL), which has not spent any time in the ground. Since the particle tracking does not take surface flow into account, OL was accounted for by reducing the MTT by using the OL fraction as a scaling factor (Appendix, Table A2). The young water, young water fraction was also used as an evaluation criterion. Like previous studies (Kirchner., 2016; von Freyberg et al., 2018; Lutz et al., 2018; Stockinger et al., 2019), we assumed young water fraction to be the sum of all water less than three month old. In our case, this includes all water reaching streams as overland flow and as young groundwater (<three months). The modelled MTT and young water fraction were also used to identify the main factors determining the travel times to stream. The catchment characteristics tested included important terrain factors such as catchment size, slope, and main soil types (Table 1).

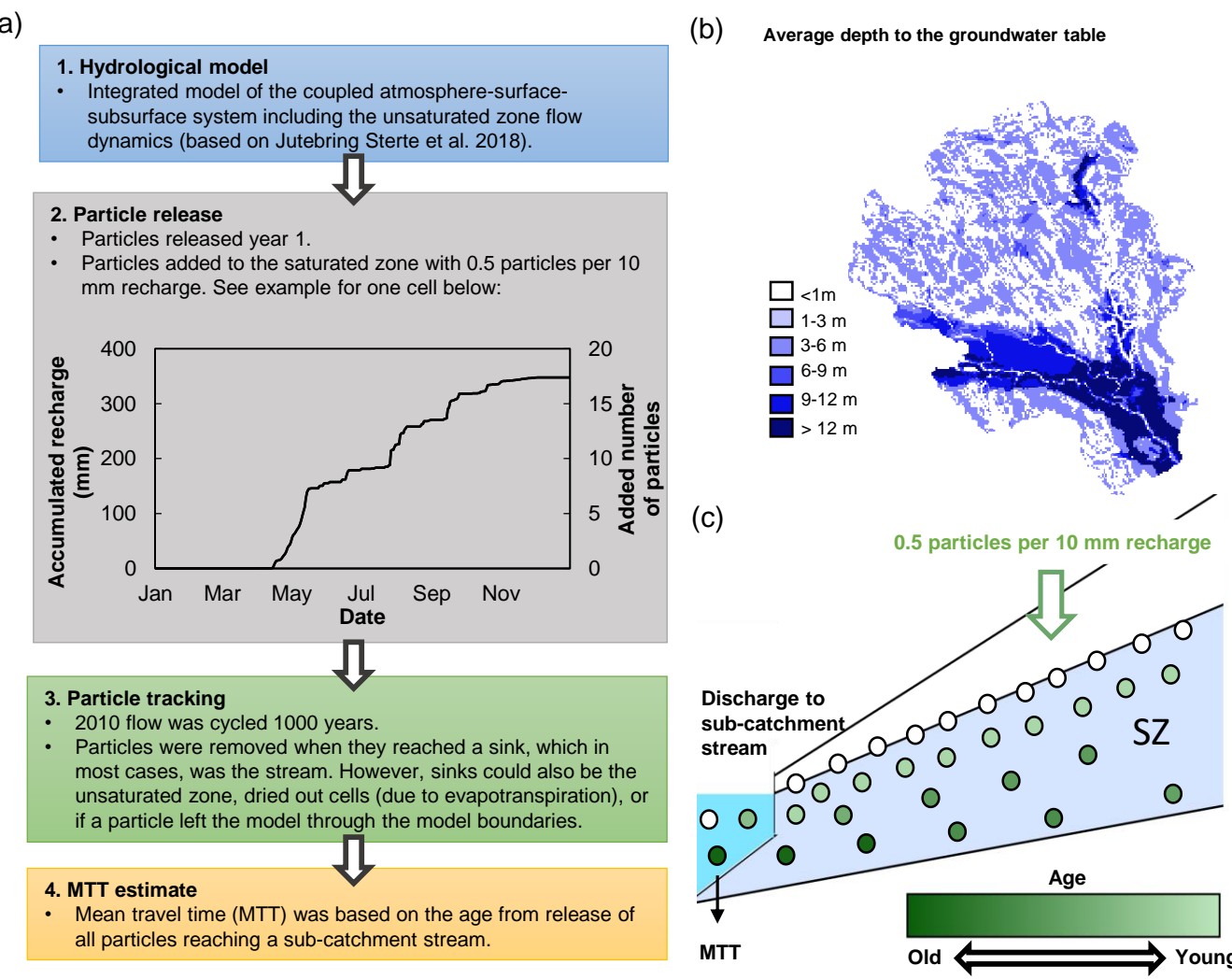

**Figure 4: Particle model setup.** (a) Steps of particle tracking (b) Average depth to the groundwater table. The main part of the model area has a calculated depth to the groundwater table between 0-3 m and varied daily. (c) Schematic illustration of particle tracking set up. Particles were added to groundwater recharge at the transient groundwater table. The age of these particles was zero at the time of recharge. Thereafter, they followed the groundwater flow, increasing in age until reaching a stream or lake.

## 3. Results

### 3.1 Travel time results

The particle tracking results were used to establish travel time distributions and MTT of water to the streams of the 14 sub-catchments in Krycklan. Since the travel time distributions were significantly skewed, we assumed that the geometric mean of the travel time distributions provided the best representation of MTT (Table 5, Fig. 5). However, all metrics are stated in the Appendix, Table A2. The annual $MTT_{geo}$ for all sub-catchments ranged from 0.8 to 3.1 years (Table 5). Most groundwater discharging to a stream had a travel time of less than one year in all sub-catchments (34% to 54%). The longest stream MTTs were connected to

the larger catchments, such as C16, and the silt dominated catchments such as C20. We used some sub-catchments for result representation, but all results are provided in Table 5 and Appendix A1. The displayed sub-catchments were: C2 (small till and forest dominated catchment), C4 (small mire dominated catchment), C20 (small silt dominated catchment), and C16 (the full-scale Krycklan catchment).

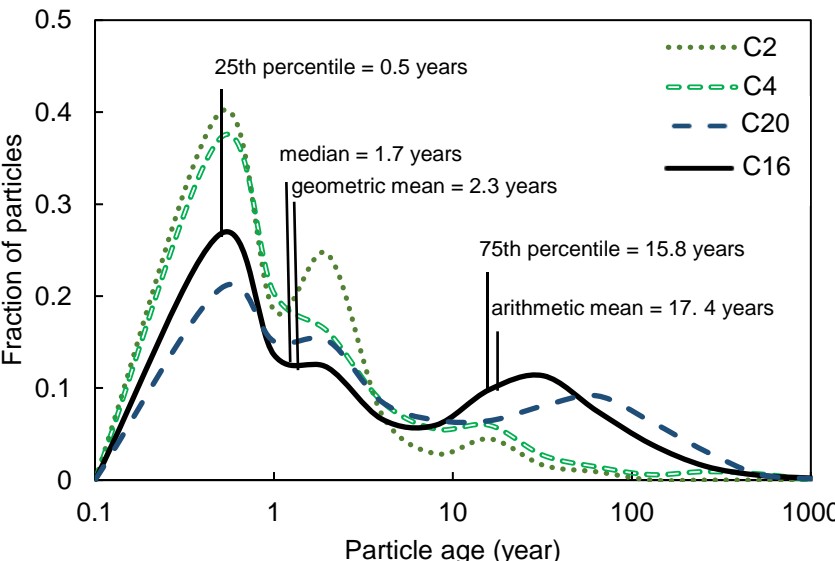

**Figure 5: Examples of particle tracking results.** The figure shows the distribution of all particles reaching the different streams for the entire modelling period. The solid line shows the statistics for C16, including the 25th percentile, the median, the geometric mean, the arithmetic mean, and the 75th percentile (Appendix, A1). Moreover, the figure shows three other example distributions, including C2 (small forest and till dominated catchment), C4 (small mire dominated catchment), C20 (small silt dominated catchment).

On an annual basis, a fraction of water reached the streams as overland flow. A major part of the overland flow occurred during the snowmelt in spring, especially in sub-catchments with mires such as C4 (Fig. 6). Both the fraction of young water reaching the streams and the $MTT_{geo}$ displayed strong seasonal trends. The longest seasonal $MTT_{geo}$, 1.2-7.7 years, and the smallest young water fraction were found during the winter season. In winter, the fraction of older water successively increased until the spring snowmelt began in early April. Conversely, the smallest fraction of old discharging water and short $MTT_{geo}$, 0.5-1.9 years, were connected

to events of larger groundwater recharge, such as the spring snowmelt and heavy summer rains.

In spring, mire sub-catchments had the shortest $MTT_{geo}$. However, as exemplified by the similar-sized C2 and C4 sub-catchments, groundwater was not renewed to the same extent in mire dominated systems due to a larger fraction of surface runoff (Fig. 6). Mire dominated sub-catchments (like C4) displayed stronger seasonal variations in $MTT_{geo}$, with shorter $MTT_{geo}$ than till dominated

sub-catchments (like C2) in spring and longer $MTT_{geo}$ than C2 in winter (Table 5). In C4, the $MTT_{geo}$ decreased from 1.5 years to 0.7 years from winter to spring, while the corresponding change in C2 was 1.2 years to 0.7 years. The seasonality of $MTT_{geo}$ was even more pronounced for catchments with a larger areal coverage of mires combined with a larger areal coverage of silt. For example, C20 had an $MTT_{geo}$ that decreased from 7.7 years to 1.9 years from winter to spring (Table 5).

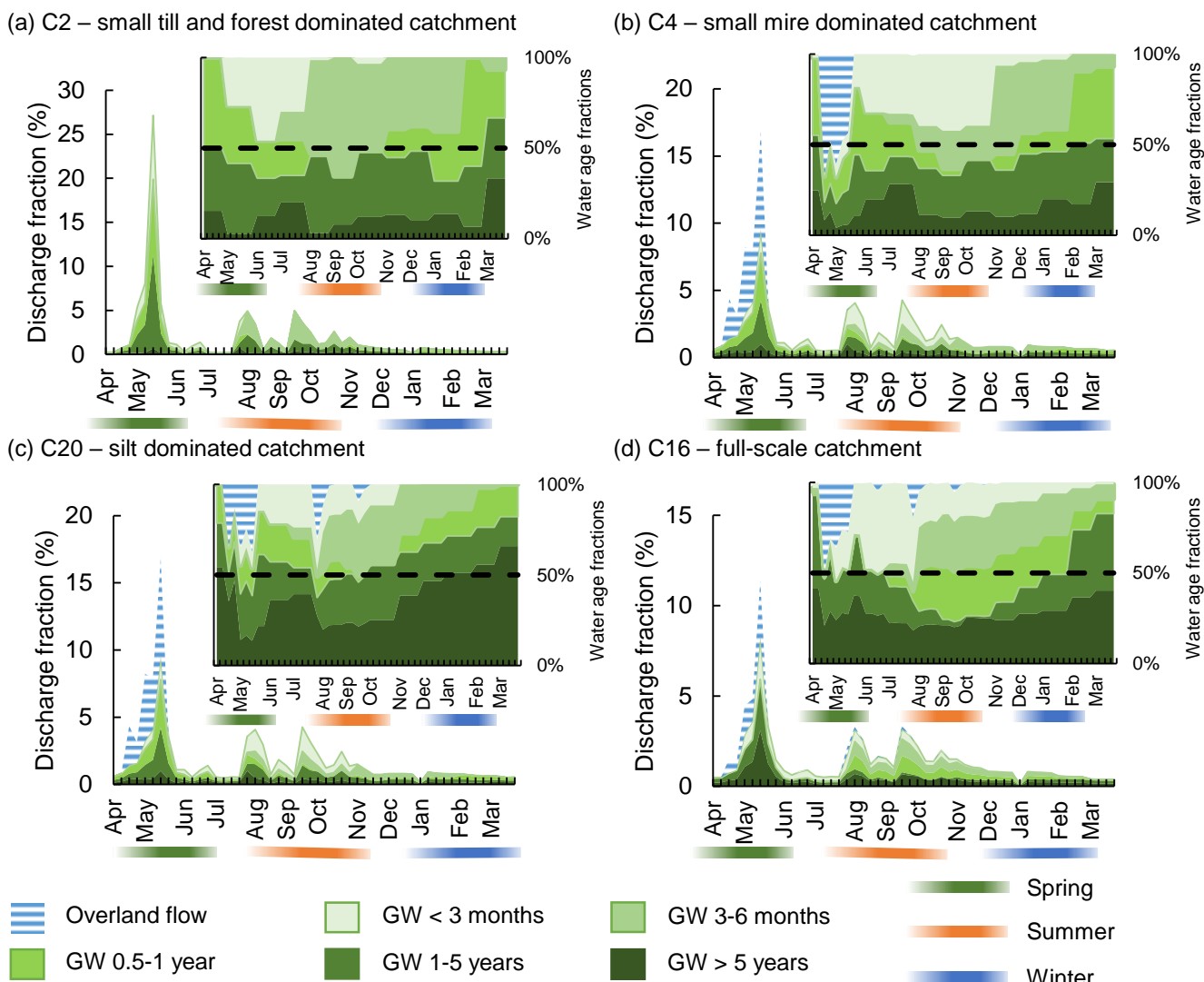

**Figure 6: Seasonal fraction of discharge to streams.** The figure shows the proportion of annual stream discharge arriving as groundwater and overland flow. Four sub-catchments are exemplified, including (a) the small till and forest dominated C2, (b) the small mire dominated C4, (c) the silt dominated C20, and (d) the full-scale Krycklan catchment C16 with mixed mires and forests (extended version in Appendix Fig. A1). The figure showcases the travel time fraction of water discharging to the streams. The fractions are both shown as part of the total annual discharge as well as the water composition. The bands below the months highlight the three investigated seasons, spring, summer, and winter.

**Table 5: Annual and seasonal (winter, spring, and summer) travel times**
The geometric mean of the travel time distribution ($MTT_{geo}$) is adjusted for the overland flow. The young water fraction (YWF) includes overland flow and groundwater younger than three months (%). An extended version of the results, including arithmetic mean, median, and SD, is included in the Appendix (Table A2).

| | Annual | | Season - Winter | | Season - Spring | | Season - Summer | |
|---|---|---|---|---|---|---|---|---|
| | $MTT_{geo}$ | YWF | $MTT_{geo}$ | YWF | $MTT_{geo}$ | YWF | $MTT_{geo}$ | YWF |
| unit | year | % | year | % | year | % | year | % |
| C2 | 0.8 | 16 | 1.2 | 0 | 0.7 | 26 | 0.7 | 6 |
| C4 | 0.8 | 40 | 1.5 | 2 | 0.7 | 53 | 0.7 | 39 |
| C5 | 0.8 | 49 | 2.9 | 1 | 0.5 | 66 | 0.8 | 38 |
| C6 | 0.9 | 42 | 2.8 | 2 | 0.6 | 58 | 0.8 | 34 |
| C7 | 1.1 | 28 | 2.2 | 4 | 0.9 | 37 | 0.9 | 27 |
| C9 | 1.4 | 28 | 3.4 | 3 | 1.0 | 41 | 1.1 | 24 |
| C13 | 1.4 | 26 | 3.3 | 3 | 1.0 | 37 | 1.2 | 23 |
| C1 | 1.3 | 20 | 3.0 | 6 | 1.0 | 25 | 0.9 | 19 |
| C10 | 1.1 | 33 | 2.5 | 3 | 0.8 | 47 | 0.9 | 31 |
| C12 | 1.3 | 28 | 2.8 | 5 | 0.9 | 39 | 1.1 | 26 |
| C14 | 2.4 | 20 | 5.6 | 2 | 1.6 | 32 | 1.6 | 21 |
| C20 | 2.7 | 23 | 7.7 | 0 | 1.9 | 36 | 1.5 | 24 |
| C15 | 1.5 | 28 | 3.8 | 4 | 0.9 | 41 | 1.1 | 27 |
| C16 | 2.3 | 23 | 5.3 | 4 | 1.4 | 35 | 1.6 | 23 |

### 3.2 Testing model results to stream isotopic composition and chemistry

In addition to investigating the annual $MTT_{geo}$, three distinct seasons were evaluated regarding the stream chemistry: winter, spring, and summer. The isotopic composition was available for 13 out of 14 sub-catchments (C20 excluded because of short time-series), while the base cation (BC) data was available for all sites. In winter, the modelled $MTT_{geo}$ was correlated to the isotopic composition (r=-0.80, P<0.01), with older stream water being closer to the long-term precipitation average (Fig. 7a). Some of the sub-catchments had an isotopic signature close to the precipitation average, suggesting almost complete mixing (e.g., C16). The negative correlation between the $\Delta\delta^{18}O_{spring}$ and the young water fraction was also significant (r=-0.90, P<0.0001, Fig. 7c), following the conceptual model (Fig. 2a). The same was also true for the summer season, but with a weaker positive correlation compared to the spring (r=0.80, P<0.001, Fig. 7e), again agreeing with the conceptual model (Fig. 2b). The opposite sign of the slope was due to the heavier summer precipitation compared to the winter baseflow. The correlation between the BC concentration and $MTT_{geo}$ was strongest in winter (r=0.90 P<0.0001) and weakest in summer (r=0.79, P<0.001). The sub-catchments with the longest travel times to streams and highest BC concentration included the sub-catchments with larger areal coverage of silt, for example, C16 and C20. The shortest travel times and lowest BC concentrations were connected to smaller sub-catchments in till dominated areas, such as C2 and C4.

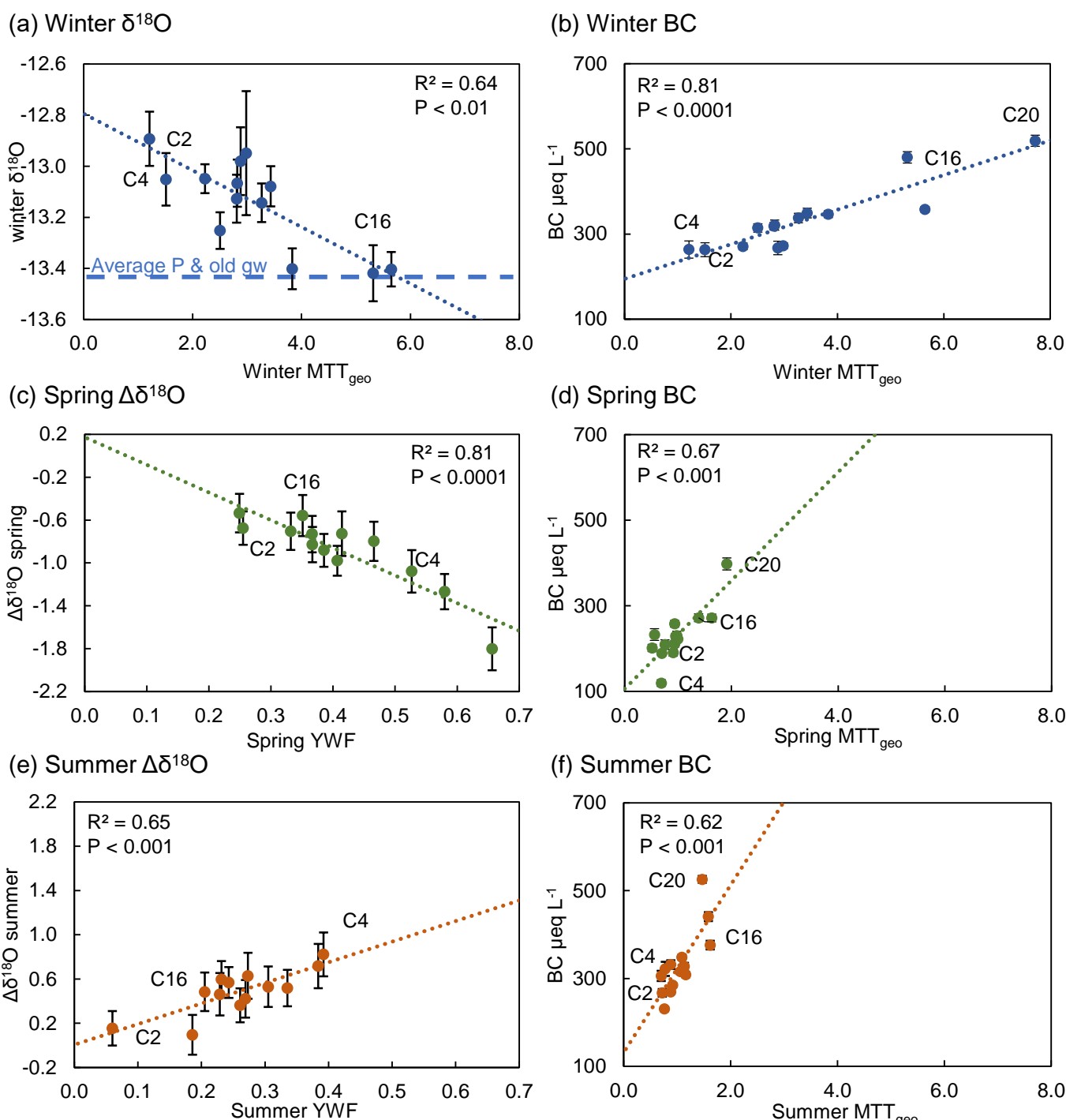

**Figure 7: Relationships of seasonal MTT$_{geo}$ and young water fractions (YWF) with seasonal stream isotopic composition and base cation concentration.** Note that δ$^{18}$O results are for 13 sites, while the BC record comprises all 14. The sub-plots (a) to (f) show the δ$^{18}$O (winter) or Δδ$^{18}$O$_{spring/summer}$ and BC concentrations as a function of the MTT$_{geo}$ in winter, spring, and summer, respectively. The standard error of the mean (SEM) shown as whiskers denotes variations in field observations.

### 3.3 Model results compared to catchment characteristics

The main catchment characteristics correlated to MTT$_{geo}$ and young water fraction were catchment size, the areal coverage of low conductive silty sediments, and the areal coverage of mires. The strongest positive correlation was found between the young water fraction and the areal coverage of mires (r=0.96, P<0.0001). There was also a strong positive correlation between MTT$_{geo}$ and the

areal coverage of silt (r=0.90, P<0.0001) (Fig. 8). A positive correlation between catchment size and $MTT_{geo}$ was also found, albeit weak due to one catchment, C20, yet significant (r=0.63, P<0.05) (Fig. 8). However, the catchment size was also correlated to the areal coverage of silt, which may be the underlying reason for this correlation (Table 6) as C20 is the only relatively small monitored sub-catchment located in the area with sorted sediments. The annual and seasonal patterns were similar (Table 6). However, the positive correlation between mires and the young water fraction was lost in winter due to a lack of new precipitation input into the system. A weak negative correlation between $MTT_{geo}$ and the young water fraction was found for the annual and spring seasonal results but were lost for the summer and winter.

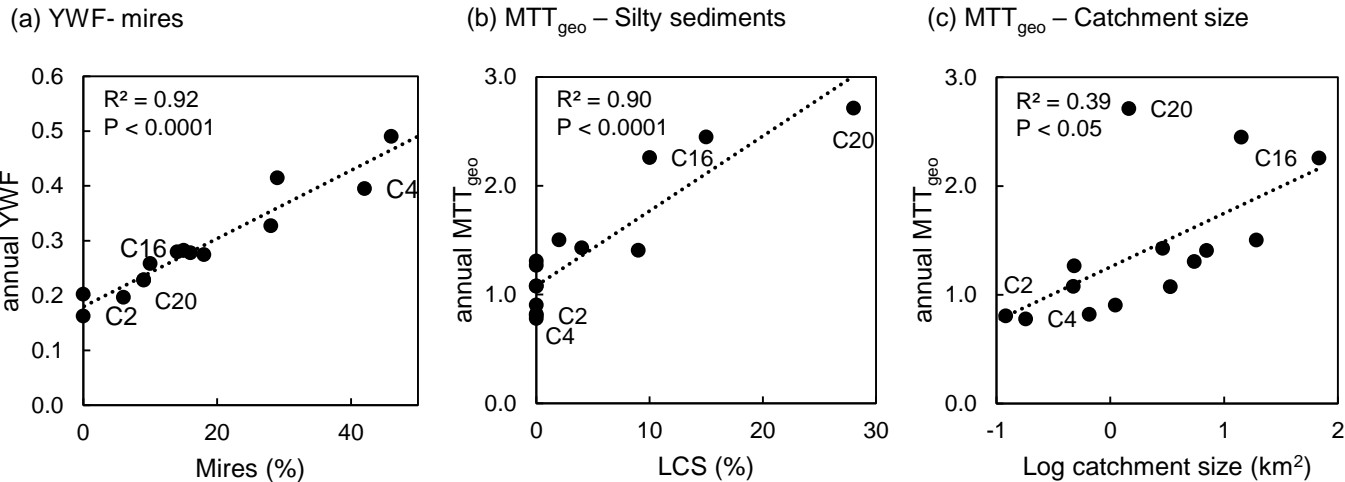

**Figure 8: Catchment characteristics are important for travel times.** The figure shows the annual averages: (a) the areal coverage of mires and the young water fraction (YWF), (b) areal coverage of silt and $MTT_{geo}$, and (c) catchment size and $MTT_{geo}$.

**Table 6: Correlation matrix – young water fraction (YWF), geometric mean travel time (MTT$_{geo}$), and catchment characteristics.** The catchment characteristics include the log catchment size (log A), the areal coverage of mires (Mire), and the areal coverage of silt (Silt). The table includes annual, winter, spring, and summer results. For |r|> 0.5, the p-value is shown according to a p<0.05 and b p>0.05.

| | Winter season | | | | | Summer season | | | | |
|---|---|---|---|---|---|---|---|---|---|---|
| | Log A (km$^2$) | Mire (%) | Silt (%) | MTT$_{geo}$ (year) | YWF (%) | Log A (km$^2$) | Mire (%) | Silt (%) | MTT$_{geo}$ (year) | YWF (%) |
| **Log A (km$^2$)** | 1 | 0.02 | 0.58 [a] | 0.64 [a] | -0.08 | 1 | 0.02 | 0.58 [a] | 0.68 [a] | 0.20 |
| **Mire (%)** | 0.02 | 1 | -0.37 | -0.34 | -0.14 | 0.02 | 1 | .0.37 | -0.50 [b] | 0.91 [a] |
| **Silt (%)** | 0.58 [a] | -0.37 | 1 | 0.92 [a] | -0.43 | 0.58 [a] | -0.37 | 1 | 0.80 [a] | -0.20 |
| **MTT$_{geo}$ (year)** | 0.63 [b] | -0.51 [b] | 0.90 [a] | 1 | -0.21 | 0.55 [a] | -0.55 [a] | 0.92 [a] | 1 | -0.28 |
| **YWF (%)** | -0.02 | 0.96 [a] | -0.39 | -0.53 [b] | 1 | 0.11 | 0.95 [a] | -0.29 | -0.52 [b] | 1 |
| | **Annual** | | | | | **Spring season** | | | | |

**4 Discussion**

Particle tracking in the Mike SHE model provided valuable insights into the annual and seasonal mean travel times (MTT$_{geo}$) across the 14 Krycklan sub-catchments. The modelled MTT$_{geo}$ and the young water fractions were strongly correlated to observed stream $\delta^{18}O_{winter}$ signatures, seasonal variation in $\delta^{18}O$, and base cation (BC) concentrations. This model validation suggests that particle tracking is a useful complementary tool to tracer-based travel time studies, at least in snow-dominated catchments, areas with pronounced seasonality, and streams dominated by older groundwater (> four years). Overall, we found that soil type was the most important variable explaining MTT$_{geo}$ and that mires are an important landscape feature regulating the young water fraction in spring (Fig.8).

**4.1 Model assumptions and limitations of estimated travel times**

Comparing the results from this modelling study to previous Krycklan investigations of MTT conducted in the C7 sub-catchment demonstrates that different model approaches have provided similar results. While our study suggested a MTT$_{geo}$ of 1.1 years and a median of 0.8 years (Appendix, Table A2), Peralta-Tapia et al. (2016) calculated a MTT of 1.8 (minimum 0.8 and maximum 3.3) years by applying a mathematical method for isotopic dampening to fit a model to the observed stream isotopic response. In another recent study using the Spatially distributed Tracer-Aided Rainfall-Runoff (STARR) model for the same stream, the median age was estimated to 0.9 years (Ala-aho et al., 2017). The close agreement between the different studies strengthens the overall reliability of the results. However, like all modelling techniques, particle tracking in Mike SHE is associated with some uncertainties and limitations.

In contrast to the Mike SHE flow model, which estimates groundwater and overland flow pathways, the particle tracking model is restricted to the subsurface hydrological component. This is a limitation in the modelling approach as water reaches the streams as a mix of groundwater and overland flow. Therefore, to allow for actual MTT$_{geo}$ estimates, we corrected the results by reducing the estimated MTT$_{geo}$ using the overland flow from the flow model as a scaling factor (Appendix, Table A2). This uncertainty primarily affects the mire dominated sub-catchments that have a large fraction of overland flow, especially during the spring.

Another uncertainty related to the particle tracking model in Mike SHE is related to the travel time from the point of infiltration through the unsaturated soil horizons to the saturated groundwater. Due to technical limitations, this travel time cannot be accounted for in the particle tracking calculations. Particles are placed at the groundwater table proportionally to the groundwater recharge (Fig. 4). Therefore, the main fraction of particles introduced to the model occurs at high infiltration rates when the groundwater level is close to the soil surface. Under these conditions, the water has, in most cases, spent a relatively short time in the unsaturated zone. However, some particles are also introduced when the groundwater level is lower, such as early snowmelt or following extended dry periods. Under such conditions, the model uncertainty increases. In this context, the smallest potential uncertainty occurs in mires, where the groundwater table always is close to the ground surface. The uncertainty becomes somewhat larger in the till areas where the unsaturated zone on average is above 1 m but can extend down to 3 m below the ground during low flow. C14 and the lower part of C16 are exceptions to these relatively shallow saturated conditions as a deep esker traverses the sub-catchments resulting in a groundwater level up to 10 m below the soil surface (Fig. 1). Accounting for the travel time from infiltration to recharge could impact the results and provide, especially for C14 and C16, longer MTT than if the groundwater level were at the same level throughout the whole catchment. This limitation primarily affects catchments with the longest MTTs and, therefore, does not seriously question the general pattern observed. The distance

from the ground surface to the groundwater table is, for most model cells, much shorter than the distance to the nearest stream, so most of the transit time should be related to the groundwater flow rather than to percolation. Although water, especially during dry conditions, no doubt can spend considerable time in the unsaturated zone, it must also be acknowledged that this water volume is small compared to the groundwater inventory in the saturated zone. Therefore, its impact on the average MTTs should be relatively small.

### 4.2 Seasonality of isotopic composition

Following the conceptual model (Fig. 2), patterns in stream isotopic signatures can be explained by seasonal changes in travel times. The modelling results show that all sub-catchments discharged water with the longest travel times in winter, somewhat shorter travel times in summer, and water with the shortest travel times in spring. When winter arrived, the main precipitation was snow, resulting in a cessation of the groundwater recharge. This caused an increasing proportion of old groundwater discharging into the streams (Fig. 6). In agreement with our conceptual model (Fig. 2), a strong negative correlation between winter $MTT_{geo}$ and the isotopic stream signatures during winter baseflow was observed (Fig. 7a). At an average travel time older than four years, it can be expected that the groundwater has reached full mixing. Hence, older water can no longer be accurately quantified using amplitude dampening of the water isotope signal (Kirchner., 2016). These theoretical considerations strengthen the results of a winter $MTT_{geo}$ older than four years for some sub-catchments since their stream isotopic signatures were close to the long-term precipitation average and, therefore, should have reached complete mixing.

When snowmelt began in late April or early May, the MTTs consistently decreased in all sub-catchments. The fraction of young groundwater in different sub-catchments was well reflected in the change of the isotope signal (Fig. 7). For snowmelt in spring, the calculated young water fraction was used to evaluate the proportion of water reaching the stream through rapid pathways, including overland flow. It is well established that the difference in stream isotopic signature between winter baseflow and spring peak flow at snowmelt ($\Delta\delta^{18}O_{spring}$) is mechanistically linked to the amount of new water reaching the stream (Tetzlaff et al., 2009). In agreement with this, we found a strong statistical relationship between $\Delta\delta^{18}O_{spring}$ and the calculated young water fraction (Fig. 7c). These results are well in line with previous work in Krycklan using end-member mixing of new and old water in the same streams (Laudon et al., 2004, 2007, 2011).

Similar to the conditions in spring, the conceptual model predicted that the difference in stream isotopic signature between winter baseflow and summer flow, $\Delta\delta^{18}O_{summer}$, should be correlated to the young water fraction in summer, but with the opposite sign, due to isotopically heavier summer rains (Fig. 2). A larger inter-annual variation in precipitation and high ET likely caused the relationship to be less evident compared to the spring results as the snowmelt conditions are more consistent from year to year. The groundwater signal reaching the streams during the summer season may also be affected by a lingering signal from the snowmelt. However, although less evident than compared to the $\Delta\delta^{18}O_{spring}$, there was still a significant correlation between the average $\Delta\delta^{18}O_{summer}$ and the modelled young water fraction (Fig. 7e).

### 4.3 Controls of travel times on base cation concentrations

The annual and seasonal average BC concentrations were positively correlated to the $MTT_{geo}$ (Fig. 7b, 7d, and 7f). Since the weathering rates were assumed to be kinetically controlled and hence related to the exposure time of water to minerals, spatial and temporal variability in BCs can be used as a relative indicator for transit time (Erlandsson Lampa et al., 2020). However, reducing

weathering to travel times may be an oversimplification as the rate is also affected by differences in mineralogy, particle size distributions and the chemical conditions in the groundwater. However, previous research in the Krycklan catchment has suggested that the chemical composition of the local mineral soils is surprisingly homogeneous, even when comparing till and sorted
sediments (Klaminder et al., 2011; Peralta-Tapia et al., 2015; Erlandsson et al., 2016; Lidman et al., 2016). Therefore, we did not expect mineralogical differences between soil types to impact the release of cations significantly. However, one exception is peat deposits, which strongly affect the cation concentrations on the landscape scale. The effect of the peat was accounted for by adjusting the concentrations following Lidman et al. (2014). Differences in particle size distribution may be important because coarser soils will have less surface area per volume unit, allowing for less weathering. However, such soils can also be expected
to have higher hydraulic conductivities, leading to higher flow velocities and, consequently, less time available for weathering. Therefore, differences in area-volume ratios between different soil types would not counteract the effect of travel times on the weathering, rather enhance it.

Despite arguments that can be made against the use of BCs as tracers, they still offer a complementary possibility to test the model
performance (Abbott et al., 2016). As McDonnell and Beven (2014) emphasised, the inclusion of tracers in hydrological models are necessary to ensure that a model reproduces the speed of flow, which is an important parameter when assessing travel time distributions. For catchment-scale models, this could be an isotopic tracer or a solute transported with the water (Hooper et al., 1988; Seibert et al., 2003; Fenicia et al., 2010; Hrachowitz et al., 2013). Although neither the travel time distribution nor the kinetics of weathering is fully understood, the strong agreement between the calculated travel times and the observed stream water
chemistry provides additional support that our modelling of these processes – and thus the entire system – was reasonable and consistent with the empirical data.

### 4.4 Mean travel times, young water fractions, and catchment characteristics

All sub-catchments showed similar seasonal patterns in $MTT_{geo}$ and young water fraction, manifested as water with shorter travel
550 times discharging in spring and water with longer travel times discharging in winter. Some of the catchment characteristics influenced the magnitude of these seasonal patterns across the landscape. On a landscape level, the main causal mechanism determining the annual $MTT_{geo}$ was the areal coverage of silty sediments (Table 1), which largely overshadowed the importance of other catchment characteristics (Fig. 8, Table 6). This finding partially stands in contrast to earlier studies in Krycklan by Peralta-Tapia et al. (2015) and Tiwari et al. (2017) that suggested that the groundwater travel times are nonlinearly linked to the catchment
size. However, a correlation of travel times to catchment size appears to be a spurious relationship since there is a correlation between the catchment size and the areal coverage of silty sediments (Table 6). The reason is that the silty sediments are located in the lower parts of the Krycklan catchment, which implies that all large catchments contain at least some proportion of silty sediments. The importance of the silt, rather than the catchment area, for the groundwater travel time is most clearly illustrated by the small silt dominated C20 catchment, which was a distinct outlier to such a scale-dependent pattern. This indicates that
catchment size may not be the primary factor determining the variability of travel times of different catchments (Fig. 6). Instead, the long travel times in C20 suggest that the groundwater flow velocity is slower in the silt areas than elsewhere in Krycklan, even though the average catchment slope is steeper than comparably sized sub-catchments in till areas (Table 1 and Fig. 1).

Similarly, silt may also explain the long travel times at C14 and C16. Although C14 is smaller than C15, which mostly lacks silt,
C14 still has a longer $MTT_{geo}$. In contrast, $MTT_{geo}$ in C15 is much closer to C12 and C13, even though the C15 catchment is twice

the size (Table 5). Hence, the results suggest that the critical difference between these sub-catchments and other sub-catchments is related to the soil hydraulic conductivity rather than the catchment size. The results further emphasise that one cannot assume that the travel time would increase with catchment size unless the distribution of different soils is comparable throughout the landscape. A larger catchment size does not necessarily imply that considerably more amounts of old groundwater is discharged directly to
570 their streams. Instead, it may just be mainly a confluence of stream water from smaller sub-catchments that determined variation, and in that case, no relationship between catchment area and groundwater travel times should be expected.

The effect of the areal coverage of silty sediments was especially prominent in winter when the range in $MTT_{geo}$ is between one and almost eight years. The change in seasonal $MTT_{geo}$ from winter to spring was also largest for the silt dominated catchments.
For example, there was a six-year difference for C20 compared to the two-year difference for the similar-sized till dominated sub-catchment C6. These intra-annual variations can also be linked to another landscape feature, namely the areal coverage of mires. Mires affected the young water fraction only when new precipitation or snowmelt input into the system occurred in spring and summer. The contrasting hydrological response of mire and silt areas, respectively, caused greater annual $MTT_{geo}$ variation for sub-catchments with both features. For example, the $MTT_{geo}$ for C4, dominated by mires, decreased from 1.5 years to 0.7 years
from winter to spring. In contrast, winter $MTT_{geo}$ for the C20 catchment dominated by silt was 7.7 years, which decreased to 1.5 years in spring. The results also show that groundwater recharge is affected by the soil frost in mires. For example, C4 showed more variations in its seasonal $MTT_{geo}$, although C2 (dominated by forest and till) and C4 (dominated by mires) had almost an equal annual $MTT_{geo}$ (Table 5). In spring, the $MTT_{geo}$ was shorter in C4 than in C2 due to the surface runoff from the frozen mire. Besides the slightly higher specific discharge (Karlsen et al., 2016), empirical studies suggest that the soil frost on mires causes a
large fraction of overland flow (Laudon et al. 2007; 2011). Looking only at the part of runoff originating from groundwater (Appendix, Table A2), the $MTT_{geo}$ for C4 decreases from 1.5 years to 1.2 years. However, C4 still showed more seasonal variation than C2, due to longer travel times in winter. The results suggest that the lack of recharge during spring effects the renewal of the groundwater in mire dominated catchments have long-term consequences on travel times.

Earlier studies have demonstrated that fluxes of old groundwater are more stable throughout the year than younger groundwater, which shows a more variable temporal pattern (Rinaldo et al., 2011; van der Velde et al., 2015; Kaandrop et al., 2018). In this system, such a pattern can mechanistically be linked to the till –soils that dominate most sub-catchments. The response of groundwater in till soils to precipitation events can be described by transmissivity feedback processes (Bishop, 1991), which are caused by the fact that the hydraulic conductivity increases exponentially towards the soil surface. When water infiltrates the
ground, the water table rises and activates more conductive soil layers, resulting in a rapid increase in the lateral flow. This implies that much of the water transport in till soils occurs relatively close to the surface. Simultaneously, the groundwater in deeper layers is more stagnant, further explaining the relatively short and consistent MTTs of till areas. Measurements of chlorofluorocarbons (CFCs) further support that deeper groundwater water transport in till soils in Krycklan is slow. Not far below the groundwater table, CFCs have indicated that the groundwater can be several decades old, suggesting that most groundwater transport occurs
close to the surface (Kolbe et al., 2020). Consistent with this explanation, silt dominated areas had much longer MTTs than comparatively sized sub-catchments underlain by till soils because till soils have a greater decline in hydrologic conductivity with soil depth (Fig 6, Fig. 8 and Appendix Fig. A1).

**5 Conclusions**

The combination of stable water isotopes, stream water chemistry, and particle tracking provided a consistent picture of the
hydrological functioning of a boreal catchment and what processes and factors are most important for regulating water pathways
and travel times. We identified specific landscape characteristics that impact the seasonal distribution of travel times by combining
a distributed hydrological model with empirical observations from 14 nested sub-catchments. In the wake of a changing climate
and intensified pressure from forestry and other types of land use, this study provides a useful baseline for assessing the intricate
connections and feedbacks between hydrological and biogeochemical processes throughout the boreal landscape. Our results
showed that travel times to stream could vary considerably on annual and seasonal scales between different types of catchments.
This was mainly related to soil properties, with low conductivity silty sediments leading to the longest travel times on annual and
seasonal timescales. In contrast, mires led to increased fractions of young water, and hence shorter travel times, but mainly in
spring when the soil was frozen. However, mire dominated catchments experience longer travel times than similar-sized forested
catchments in winter. Generally, for the boreal landscape, a warmer climate is predicted with reduced snow cover and snow
duration, accompanied by increases in the frequency of winter thawing episodes and reduction in soil frost (IPCC, 2014; Jungqvist
et al., 2014; Brown et al.,2017; Lyon et al., 2018). Our results suggest that these changes would reduce the intra-annual variations
of MTT created by the freezing of mires, while the impact on other parts of the landscape would remain relatively low.

**6 Data and code availability**

Data from the Hydrological Research at Krycklan Catchment Study is available in Svartbergets open database (Krycklan Database,
2021), which includes chemistry, environmental and GIS data.  The software and licence for Mike SHE and Mike 11 are available
online (DHI, 2021). Additionally, the Krycklan hydrological flow and particle tracking model setup and input files can be acquired
from the open database Safe Deposit, together with the main chemistry data used in this study (Jutebring Sterte et al., 2021).

**7 Author contribution**

Elin Jutebring Sterte and Hjalmar Laudon were responsible for designing, conceptualising, and evaluating results in collaboration
with the other co-authors. Elin Jutebring Sterte and Emma Lindborg were responsible for numerical modelling. Elin Jutebring
Sterte prepared the manuscript and figures. Elin Jutebring Sterte lead the writing of the paper with contributions from all co-
authors.

**8 Competing interests**

The authors declare that they have no conflict of interest.

**9 Acknowledgments**

We like to thank the anonymous reviewers and the editor, Conrad Jackisch, for their time and effort, improving the quality of our
manuscript. Their combined constructive criticism has helped to refine the text and figures of this study. The authors are also
thankful to the funding agency Svensk Kärnbränslehantering AB (SKB), the Danish Hydraulic Institute (DHI) for software access
and expert consultation, and the crew of the Krycklan Catchment Study (KCS) funded by SITES (VR) for advice and data
collection. KCS is funded by the Swedish University of Agricultural Sciences, Swedish Research Council (as part of the SITES

network and project funds), FORMAS, Knut and Alice Wallenberg Foundation through Branch-Point and Future Siliviculture, Kempe foundation and, SKB. Data is available from the open Krycklan database (Laudon et al., 2013). Several individuals have also helped with the creation of this work. Special acknowledgement goes to Patrik Vidstrand (SKB) for bedrock properties consultation, Jan-Olof Selroos (SKB) for constructive comments and criticism, Hanna Corell (DHI) for initial particle release consultation, and Anders Lindblom (SKB) with the design of Fig. 1.

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

## Appendix

The appendix includes the classification of the seasons for each year (Table A2). It also includes the statistical measurements for particle tracking and isotope data used in, e.g., Table 5 and Table A2. The statistics used in this study include arithmetic mean (Eq. A1), geometric mean (Eq. A2), standard deviation (Eq. A3), standard error of the mean (Eq. A4). The isotopic signature of
$\delta^{18}O$ has been calculated as Eq. A5. Eq. A6 shows the equation used for adjusting the MTT from the particle tracking (MTT$_{particle\ tracking}$) to the overland flow MTT shown in table 5. Eq. A7 shows the lake regression from Peralta-Tapia et al., 2015, adjusted according to new data.

$$\text{Arithmetic mean of the travel time distribution} = \left(\frac{1}{n}\right)\sum_{i=1}^{n} ai \qquad \text{(Eq. A1)}$$

$$\text{Geometric mean of the travel time distribution} = 10^{\left(\frac{1}{n}\right)\sum_{i=1}^{n}\log(ai)} \qquad \text{(Eq. A2)}$$

$$\text{SD} = \text{standard deviation} = \sqrt{\frac{\sum(ai - aMTT)^2}{N}} \qquad \text{(Eq. A3)}$$

$$\text{SEM} = \text{standard error of the mean} = \frac{SD}{\sqrt{n}} \qquad \text{(Eq. A4)}$$

$$\delta = \left(\frac{R_{sample}}{R_{standard}} - 1\right)\text{‰}, \ R = \frac{18O}{16O} \qquad \text{(Eq. A5)}$$

$$\text{MTT} = \text{MTT}_{Particle\ tracking}\ (1 - \text{fraction OL}) \qquad \text{(Eq. A6)}$$

$$\delta^{18}O = 0.2(\text{lake coverage, \%}) - 13.2 \qquad (R^2 = 0.9,\ p < 0.01) \qquad \text{(Eq. A7)}$$

ai= data set values, n=number of values, OL = overland flow

The start and end dates used in this study for the evaluation of isotopic signatures and BC are shown in Table A1 (Krycklan Database, 2021). The winter season generally occurs from early December to late February, from negative air temperatures until increasing temperatures cause small snowmelt events. Here, the isotopic signal is more sensitive to snowmelt than the BC. Spring is the period of the main part of the spring flood, mainly occurring in April to May. Moreover, the summer season
includes all observations occurring between July to September.

**Table A1: Dates used for chemistry investigation.** The dates are the start and end dates for observations within the seasons classified as winter, spring, and summer, respectively. Note that BC only includes dates from 2008-2016.

| Year | Season - Winter | | Season - Spring | | Season - summer | |
|---|---|---|---|---|---|---|
| | Start date (year-month-day) | End date (year-month-day) | Start date (month-day) | End date (month-day) | Start date (month-day) | End date (month-day) |
| **2008** | 2008-01-16 | 2008-02-13 | 04-18 | 05-12 | 07-01 | 09-28 |
| **2009** | 2008-12-09 | 2009-02-12 | 04-20 | 05-12 | 07-08 | 09-15 |
| **2010** | 2009-12-15 | 2010-02-10 | 04-15 | 05-14 | 07-06 | 09-28 |
| **2011** | 2010-12-16 | 2011-02-21 | 04-18 | 05-09 | 07-04 | 09-28 |
| **2012** | 2011-12-20 | 2012-02-14 | 04-17 | 05-14 | 07-03 | 09-25 |
| **2013** | 2012-12-18 | 2013-02-20 | 04-17 | 05-10 | 07-04 | 09-20 |
| **2014** | 2013-12-17 | 2014-02-25 | 04-22 | 05-13 | 07-08 | 09-29 |
| **2015** | 2014-12-16 | 2015-02-17 | 04-17 | 05-12 | 07-14 | 09-22 |
| **2016** | 2015-12-15 | 2016-02-15 | 04-18 | 05-12 | 07-12 | 09-20 |
| **2017** | 2016-12-06 | 2017-02-08 | 04-18 | 05-09 | 07-11 | 09-21 |
| **2018** | 2017-12-05 | 2018-02-13 | 04-17 | 05-07 | - | - |

The Appendix also includes an extended version of table 5 and Fig. 6, including all sub-catchments (Table A2 and Fig. A1). The figure shows the travel time fraction of water reaching the streams of the sub-catchment in Krycklan annually. The figure shows both the groundwater fraction (travel time fraction calculated using the particle tracking results) and the simulated direct runoff fraction. The table shows more statistical information regarding the travel time distribution, including the Skew, SD, and SEM, and the fraction of overland flow on an annual and seasonal timescale.

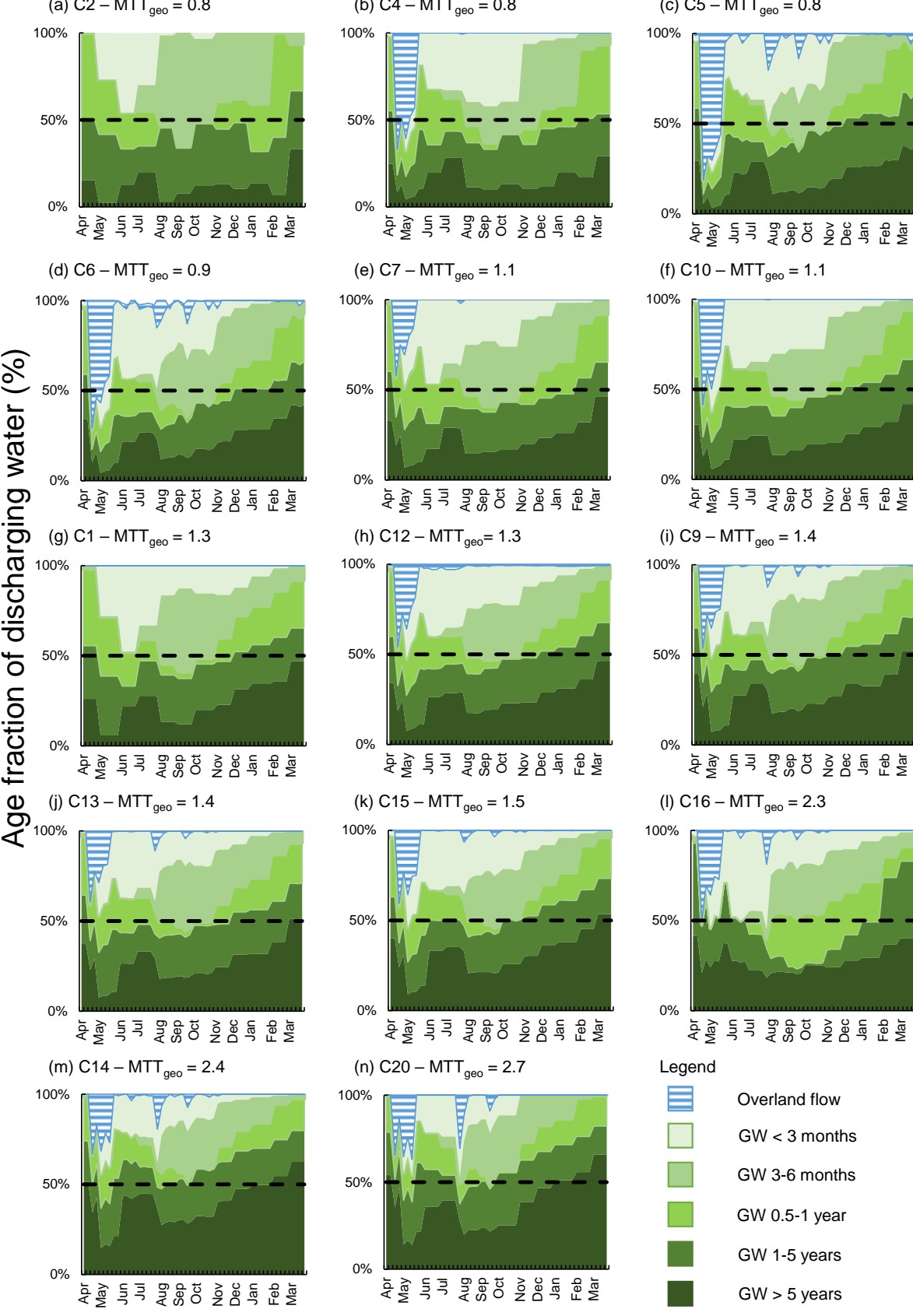

**Figure A1:** Travel time fraction of water discharging to the streams of Krycklan, in increasing annual geometric mean travel time, MTT$_{geo}$ (up left to down right). The black vertical line shows the 50 % mark for visual aid. All charts begin in spring (late March/early April) and end in winter (early March).

**Table A2: Extended version of Table 5 - Annual and seasonal (winter, spring, and summer) travel time results.**
The table includes the travel time results based on particle tracking ***before*** taking the overland flow into account. The results include arithmetic mean (A), median (M), geometric mean (Geo), skew, standard deviation (SD), and standard error of the mean (SEM). The table also shows the fraction of overland flow (OL) used as a scaling factor for the calculation of MTT.

| | Annual | | | | | | | Season - Winter | | | | | | |
|------|------|------|------|------|-----|-----|-----|------|------|------|------|-----|-----|-----|
| | A | M | Geo | Skew | SD | SEM | OL | A | M | Geo | Skew | SD | SEM | OL |
| unit | year | year | year | - | - | - | % | year | year | year | - | - | - | % |
| C2 | 2.2 | 0.9 | 0.8 | 5.0 | 5 | 0.2 | 0 | 2.7 | 0.7 | 1.2 | 2.4 | 4 | 0.5 | 0 |
| C4 | 7.7 | 0.8 | 1.0 | 7.5 | 34 | 1.0 | 21 | 10.5 | 1.1 | 1.5 | 6.6 | 42 | 2.7 | 0 |
| C5 | 15.2 | 1.0 | 1.3 | 6.4 | 61 | 1.0 | 35 | 30.4 | 1.4 | 2.9 | 4.1 | 84 | 3.2 | 0 |
| C6 | 13.7 | 0.8 | 1.2 | 6.8 | 51 | 0.6 | 23 | 25.9 | 1.4 | 2.8 | 4.6 | 69 | 1.8 | 0 |
| C7 | 8.0 | 0.8 | 1.2 | 7.4 | 25 | 0.4 | 8 | 13.2 | 1.3 | 2.2 | 5.6 | 32 | 1.1 | 0 |
| C9 | 13.2 | 1.0 | 1.6 | 6.7 | 38 | 0.3 | 12 | 21.6 | 1.6 | 3.4 | 5.0 | 47 | 0.7 | 0 |
| C13 | 13.3 | 1.0 | 1.5 | 8.0 | 43 | 0.2 | 9 | 21.6 | 1.6 | 3.3 | 6.4 | 53 | 0.5 | 0 |
| C1 | 10.1 | 1.0 | 1.3 | 4.1 | 27 | 0.4 | 0 | 18.8 | 1.4 | 3.0 | 2.7 | 36 | 1.2 | 0 |
| C10 | 10.9 | 0.9 | 1.2 | 6.4 | 35 | 0.2 | 14 | 16.5 | 1.4 | 2.5 | 4.5 | 40 | 0.6 | 0 |
| C12 | 11.9 | 1.0 | 1.4 | 5.5 | 33 | 0.2 | 9 | 17.6 | 1.0 | 2.8 | 4.0 | 37 | 0.4 | 0 |
| C14 | 18.3 | 1.9 | 2.7 | 7.8 | 54 | 0.2 | 9 | 26.4 | 6.6 | 5.6 | 6.8 | 60 | 0.4 | 0 |
| C20 | 21.9 | 1.8 | 3.1 | 6.0 | 52 | 0.6 | 13 | 32.9 | 9.7 | 7.7 | 5.8 | 55 | 1.1 | 0 |
| C15 | 14.3 | 1.0 | 1.7 | 8.7 | 43 | 0.1 | 10 | 21.9 | 2.4 | 3.8 | 6.7 | 49 | 0.3 | 0 |
| C16 | 17.4 | 1.7 | 2.5 | 8.5 | 50 | 0.2 | 8 | 25.3 | 6.7 | 5.3 | 7.3 | 57 | 0.2 | 0 |

| | Season - Spring | | | | | | | Season - Summer | | | | | | |
|------|------|------|------|------|-----|-----|-----|------|------|------|------|-----|-----|-----|
| | A | M | Geo | Skew | SD | SEM | OL | A | M | Geo | Skew | SD | SEM | OL |
| unit | year | year | year | - | - | - | % | year | year | year | - | - | - | % |
| C2 | 1.6 | 1.0 | 0.7 | 6.1 | 3 | 0.2 | 0 | 2.7 | 0.4 | 0.7 | 4.2 | 8 | 0.6 | 0 |
| C4 | 5.7 | 0.3 | 1.2 | 10.8 | 27 | 1.5 | 44 | 9.1 | 0.4 | 0.7 | 6.1 | 39 | 2.0 | 0 |
| C5 | 9.9 | 1.0 | 1.2 | 9.3 | 50 | 1.5 | 57 | 11.3 | 0.4 | 0.8 | 7.6 | 52 | 1.7 | 8 |
| C6 | 9.1 | 0.9 | 1.0 | 9.5 | 45 | 1.0 | 42 | 9.9 | 0.4 | 0.8 | 8.2 | 42 | 1.0 | 7 |
| C7 | 5.5 | 1.0 | 1.1 | 11.2 | 19 | 0.6 | 18 | 7.5 | 0.4 | 0.9 | 7.5 | 27 | 0.8 | 0 |
| C9 | 8.2 | 1.0 | 1.3 | 10.8 | 31 | 0.4 | 24 | 11.2 | 0.8 | 1.2 | 7.0 | 34 | 0.5 | 5 |
| C13 | 7.8 | 1.0 | 1.2 | 10.8 | 30 | 0.3 | 18 | 12.5 | 0.8 | 1.2 | 8.8 | 45 | 0.4 | 3 |
| C1 | 5.2 | 1.0 | 1.0 | 6.8 | 19 | 0.5 | 0 | 8.4 | 0.4 | 0.9 | 4.5 | 25 | 0.8 | 0 |
| C10 | 8.0 | 1.0 | 1.1 | 8.2 | 32 | 0.4 | 31 | 9.0 | 0.4 | 0.9 | 6.3 | 31 | 0.4 | 0 |
| C12 | 8.2 | 1.0 | 1.2 | 7.6 | 29 | 0.3 | 21 | 10.2 | 0.8 | 1.1 | 5.3 | 29 | 0.3 | 1 |
| C14 | 12.2 | 1.6 | 2.1 | 10.6 | 45 | 0.3 | 21 | 16.3 | 1.1 | 1.8 | 7.9 | 54 | 0.4 | 11 |
| C20 | 12.2 | 1.9 | 2.6 | 10.5 | 39 | 0.8 | 27 | 20.4 | 1.0 | 1.6 | 5.4 | 59 | 1.3 | 9 |
| C15 | 9.2 | 1.0 | 1.2 | 11.4 | 34 | 0.2 | 22 | 12.4 | 0.9 | 1.2 | 9.2 | 41 | 0.2 | 9 |
| C16 | 11.4 | 1.1 | 1.7 | 11.0 | 40 | 0.1 | 20 | 15.6 | 1.0 | 1.8 | 8.9 | 48 | 0.2 | 11 |