# Peer review of "Linking travel times and flow pathways to stream chemistry, isotopic composition, and catchment characteristics in a boreal landscape"

_Hydrology and Earth System Sciences, 2020_

## Referee Comment (RC1) · Anonymous Referee #1 · 6 May 2020

General comments:

In this study the authors used a physically-based distributed model in combination with a particle tracking approach to determine groundwater travel times in the well-studied Kryklan catchment in northern Sweden. They compared the modeled mean travel times (MTTs) with average winter values of stable isotopes, pH, base cations and found significant correlations for all of them. Furthermore they tried to relate MTTs to certain catchment characteristics. The only strong and significant correlation they found was between MTTs and the fraction of low conductive sediments.

The use of particle tracking approaches to determine travel time distributions with numerical models becomes more and more common in catchment hydrology. And although some of these approaches still suffer from certain simplifications (missing dispersion component, particles disappearing after temporary exfiltration, etc.) they can already shed light on general catchment dynamics. Having said that I am missing a more detailed description of both the model setup with Mike SHE and the particle tracking approach in particular (one or two additional figures would not hurt).

Language, style and structure throughout the manuscript are quite good and easy to follow.

In the introduction the authors present their hypotheses regarding the relationships between travel times and catchment characteristics. However, it seems that the review of relationships that have already been determined in former studies is a little short. I would like to point out that it is already quite well established that MTTs are time-variant and going along with this the strength of relationships between MTTs and certain physical catchment characteristics changes as well (this should be discussed at least in a little more depth than just mentioning it in line 49 once). Also research over at least the last 10 years has stressed the fact that it is in most cases a combination of multiple characteristics that control a catchment's MTTs. A comprehensive (short) summary of the state of art would be really helpful.

I am also missing a more detailed analysis of the modeled travel time distributions. I am especially intrigued by the very steep initial rise of the cumulative distributions. Given the fact that the TTDs are related to baseflow conditions, this seems particularly puzzling to me. Therefore I would like to see more explanation and discussion in this part. A figure of the modeled (not cumulative) TTDs would also be interesting/helpful.

The discussion and conclusion sections are straightforward and quite easy to comprehend. This may, unfortunately, be due to the fact that the results are not really new. Catchment size is not related to MTTs; hydraulic conductivity, slope and flow path length are related to MTTs. In the end I am left wondering whether there is enough

novelty (maybe relating pH and base cations to MTTs?). A more in depth discussion of the modeled TTDs could help or maybe additional analyses on the interdependencies of how different catchment characteristics combined control MTTs. Still, I believe that in the end the authors will manage to add more analysis and discussion to justify a publication in HESS.

Specific comments:

Line 23: '. . .to investigate. . .'

Line 23: Better write: '. . .the travel time of the input to 14 [. . .] sub-catchments via groundwater to the stream. . .'.

Line 30: Move this sentence up to Line 27 just after '. . .stream water.'.

Line 31: I would add whether these were positive or negative correlations.

Line 34: I would not call this a 'landscape characteristic'. Maybe better call it 'physical catchment characteristic'.

Line 35: '. . .to positively correlate with MTT.'

Line 50: These referenced papers deal mainly with mean travel times, not with travel time distributions. Appropriate references for variations in TTDs would be for example Botter et al. (2010), Heidbüchel et al. (2012), Hrachowitz et al., (2013).

Line 87: Again, I would not use 'landscape' factors since the term landscape refers to the visible landforms rather than to physical properties or subsurface features.

Introduction: This section lacks the mention/discussion of previous work that is very relevant to the study. Especially concerning research on the connection of travel time and catchment characteristics. Your hypothesis is that catchment size correlates with travel time, but what about any other catchment properties? So far travel times have been related to slope, flow path length, soil thickness, antecedent moisture content, hydraulic conductivity, just to name a few. I recommend expanding the short introduc-
tion a bit more to include a brief discussion of potential travel time controlling factors (Ameli et al., 2016; Birkel et al., 2016; Haitjema et al., 1995; Heidbüchel et al., 2013; Hrachowitz et al., 2010; McGuire et al., 2005; Yang et al., 2018).

Line 106: '...type OF regolith...'?

Line 123: I would like to see a figure of the stratigraphy and how it is displayed in the model.

Line 172: Are these winter GW travel times travel times of particles that entered the catchment during the winter or exited the catchment during the winter (are they from 'forward' or from 'backward' TTDs)? This is a very important and interesting question since it could be very different catchment characteristics that control forward or backward travel times.

Line 215: According to this equation a mire coverage of 100% would result in an infinitely large adjusted cation concentration which is quite unrealistic (this is equivalent to a scenario where mires do not contribute any cations whatsoever). How do you justify this relationship?

Line 220: What exactly do you mean by this ('...less impacted...', '...also considered.')?

Line 239: But the simulated specific baseflow is not included in Table 4.

Line 243: Again it would be important to know whether these are forward or backward times.

Line 243: I would somehow indicate that the means are geometric instead of arithmetic when you write MTT (maybe something like 'gMTT'?). Because one will automatically assume that MTT is the arithmetic mean of the travel times.

Line 245: This is not a complete sentence (verb missing).

Line 248: 'Over the course of a year...'; '...may enrich or dilute...'

Line 259: Winter baseflow or winter groundwater fraction?

Line 272: Instead of 'decreased' I would rather write '. . .became more negative/became more enriched in the lighter isotope. . .'

Line 277: '. . .averageS. . .'

Line 292-294: It would be good to add 'positive' or 'negative' before 'correlations', so you immediately know the direction of the relationship.

Line 297: How do these equations show that LCS is the most significant parameter?

Line 304: 'Percentage of low conductive sediments. . .'

Line 314: '. . .on the other. . .'?

Line 318: '. . .and a gamma distribution transfer function (convolution) method. . .'

Line 318: 'In a conceptual, non-distributed modeling study. . .'

Line 319: Some more details would be helpful ('another travel time distribution technique').

Line 361: This section is a bit unstructured. You start out by stating that slope and hydraulic conductivity are the main factors controlling MTTs, then you state that the fraction of LCS is the most important factor adding travel distance to the mix before arguing that the steeper small C20 behaves differently maybe also because of the fluvial deposits fraction. . . Since these are some of your main findings it would be good to clarify the section.

Also, what about the fact that you released the particles at the top of the transient groundwater table? That means that depending on the groundwater table a larger or smaller part of the regolith was not taken into account for the MTT calculations. If the water table was high a larger proportion of the regolith was considered, if it was low the particles started somewhere else in the profile. What are the implications of this?

Would that influence your results?

Line 380: It would be good to point out more the novel aspects of your work. What did you find out that was really new? The fact that Darcy's law works? Catchment size has been ruled out as a MTT control since quite a while now. Or is it mainly a study that confirms the applicability of the particle tracking of your model by comparing results to time series of pH, base cations and stable isotopes?

Figures:

Figure 1: Why do you call the sub-catchments subareas in this figure? Also, all sub-catchments (not only the green ones) connect before reaching the main outlet at C16.

Figure 2: I am confused by the y axis label. Why does it start with 100% and then decrease to 0%? Isn't this a cumulative TTD (that should start with 0%)? Also, you never mention how you construct this TTD. Do you record the time the particles that arrive in the stream in the winter needed to travel through the catchment (backward TTD) or do you record the time that particles that are released during winter need to travel through the catchment (forward TTD)? I am curious since the cumulative TTDs exhibit a shape I would not have expected since the initial rise is very steep although you use a logarithmic x axis. For a groundwater TTDs in particular, this extremely high initial values are quite unusual. How do you explain this?

Figure 3: So if you replace the geometric MTT with the commonly used arithmetic MTT, how does that change the correlations? In case they become weaker you could argue for the general use of the geometric MTT.

Supplements

S1: Check again for typos and misuse of words throughout the supplement (in particular in the table captions and the table foot notes).

S1, line 3: What does that mean that the table includes both non-transformed and log-normal transformed data? Which is which?

S1, line 4: What happened to the other 44% of particles?

S1, line 8: 'Artesian' mean? And why 'back-transformed'?

S1, line 19: '...as well as the precipitation (P)'

S2, line 3: What does 'The statistics are based on...' mean?

S2: The tables are not that helpful. Many of the abbreviations are not explained.

Literature

Ameli, A. A., Amvrosiadi, N., Grabs, T., Laudon, H., Creed, I. F., McDonnell, J. J., and Bishop, K.: Hillslope permeability architecture controls on subsurface transit time distribution and flow paths, J. Hydrol., 543, 17–30, https://doi.org/10.1016/j.jhydrol.2016.04.071, 2016.

Birkel, C., Geris, J., Molina, M. J., Mendez, C., Arce, R., Dick, J., et al.: Hydroclimatic controls on non-stationary stream water ages in humid tropical catchments, J. Hydrol., 542, 231–240, https://doi.org/10.1016/j.jhydrol.2016.09.006, 2016.

Botter, G., Bertuzzo, E., and Rinaldo, A.: Transport in the hydrologic response: Travel time distributions, soil moisture dynamics, and the old water paradox, Water Resour. Res., 46, https://doi.org/10.1029/2009WR008371, 2010.

Haitjema, H. M.: On the residence time distribution in idealized groundwatersheds, J. Hydrol., 172, 127–146, https://doi.org/10.1016/0022-1694(95)02732-5, 1995.

Heidbüchel, I., Troch, P. A., Lyon, S. W., and Weiler, M.: The master transit time distribution of variable flow systems, Water Resour. Res., 48, https://doi.org/10.1029/2011WR011293, 2012.

Heidbüchel, I., Troch, P. A., and Lyon, S. W.: Separating physical and meteorological controls of variable transit times in zero-order catchments, Water Resour. Res., 49, 7644–7657, https://doi.org/10.1002/2012WR013149, 2013.

[Figure]

Hrachowitz, M., Soulsby, C., Tetzlaff, D., Malcolm, I. A., and Schoups, G.: Gamma distribution models for transit time estimation in catchments: Physical interpretation of parameters and implications for time-variant transit time assessment, Water Resour. Res., https://doi.org/10.1029/2010WR009148, 2010.

Hrachowitz, M., Savenije, H., Bogaard, T. A., Tetzlaff, D., and Soulsby, C.: What can flux tracking teach us about water age distribution patterns and their temporal dynamics?, Hydrol. Earth Syst. Sci., 17, 533–564, https://doi.org/10.5194/hess-17-533-2013, 2013.

McGuire, K. J., McDonnell, J. J.,Weiler, M., Kendall, C., McGlynn, B. L., Welker, J. M., and Seibert, J.: The role of topography on catchment-scale water residence time, Water Resour. Res., 41, https://doi.org/10.1029/2004WR003657, 2005.

Yang, J., Heidbüchel, I., Musolff, A., Reinstorf, F., and Fleckenstein, J. H.: Exploring the dynamics of transit times and subsurface mixing in a small agricultural catchment, Water Resour. Res., https://doi.org/10.1002/2017WR021896, 2018.

---

## Referee Comment (RC2) · Anonymous Referee #2 · 28 May 2020

In the manuscript "Linking groundwater travel times to stream chemistry, isotopic composition and catchment characteristics," Sterte et al. analyze the drivers of catchment travel times across catchments in norther Sweden. They use a physical hydrology model combined with particle tracking to to generate transit time distributions and compare this to isotopic and stream chemistry observations. Overall, I think that the study is well done and the manuscript is well written and easy to follow. However, I do have some significant concerns about the manuscript in its present form.

1. My most serious concern is that it's not clear what the novelty of this study is that would warrant publication in HESS. All of the methods used here are well established

and the idea that catchment travel time distributions are driven by catchment characteristics is not new. The authors start from the hypothesis that catchment area is the main driver, however previous research has already indicated that many drivers will be important, so disproving this hypothesis does not seem to be the best angle to take here. I would suggest the authors consider what portions of their findings are the most novel addition to the body of literature in this area and organize the manuscript around this rather than the area hypothesis. Even if the area hypothesis is what guided the study in the first place, this does not need to be the storyline of the manuscript.

2. Along the same lines as my first comment I think the introduction could use significant revision. As it stands it is a very broad overview of the topic but I would like to see a more thorough review of previous finding directly relevant to this work that can clearly motivate the novelty of this study and the gap it is filling.

3. Similarly I think the discussion section would be more powerful if it provides a better evaluation of how and where results form this study add new information/disagree or provide additional corroboration to existing studies.

4. For the most part I think the paper is very clearly written, however the description of the modeling approach is a bit confusing and could use some more details. For example the term simulation is used to refer to both the hydrology model and the particle tracking portion which can be confusing. This section could be helped by a figure or a schematic to illustrate the approach I think.

Specific comments: 1. I think the catchment numbering could be done in a more intuitive way so its easier to separate unique outlets (i.e C12-15). I would suggest giving each of these outlets their own letter and then numbering points within them potentially by drainage area, that way it is easier to compare when points are within the same drainage or not.

2.Line 138-140: This is confusing are you trying to say that the hydrologic model is run first and then the particle tracking is applied to the outputs of that model?

[Figure]

3. Section 2.3 – this is really more a description of scenario design than numerical methods. I would consider renaming.

4. Line 148: 'several years' is very vague can you be more precise.

5. Line 154: at what time frequency were particles injected into the model? Just at the start of each year?

6. Line 159: I think the more standard reference for this would be heavy tailed rather than long tailed.

7. The term simulation is used to refer to both the hydrology model and the particle tracking model and this can make the methods confusing when you are talking about run times for example.

8. At the beginning of section 2.3 you say that you used several years of simulation but actually it looks like you use only one year of the hydrologic simulation but repeated it 1000 times (i.e. more than several). This description is confusing.

9. I think some of the tables could be converted to figures to better present the information. For example Table 4 could be presented as a series of maps.

---

## Author Comment (AC1) · 9 Jul 2020

Introduction comment from reviewer #1 In this study the authors used a physically-based distributed model in combination with a particle tracking approach to determine groundwater travel times in the well-studied Kryklan catchment in northern Sweden. They compared the modeled mean travel times (MTTs) with average winter values of stable isotopes, pH, base cations and found significant correlations for all of them. Furthermore they tried to relate MTTs to certain catchment characteristics. The only strong and significant correlation they found was between MTTs and the fraction of low conductive sediments.

[Figure]

- Authors response: On behalf of all authors, I, would like to thank reviewer #1 for his/her constructive criticism and suggestions which will improve the next version of the manuscript. The main concern raised was the need to improve the analysis and discussion part, which, we agree will progress the manuscript further. Our explanations and responses to all the reviewer's comments and questions are listed below. All the comments have been included unabridged in this document, together with our response following each comment.

The use of particle tracking approaches to determine travel time distributions with numerical models becomes more and more common in catchment hydrology. And although some of these approaches still suffer from certain simplifications (missing dispersion component, particles disappearing after temporary exfiltration, etc.) they can already shed light on general catchment dynamics. Having said that I am missing a more detailed description of both the model setup with Mike SHE and the particle tracking approach in particular (one or two additional figures would not hurt).

- Authors response: We understand the reviewers concerns regarding the method sections. A main reason for this is that the present model is based on the model that already has been thoroughly described in a previous paper by Jutebring Sterte et al. (2018), which is cited in the manuscript. However, we realize the need to provide the readers of this manuscript with a more proper background. We will incorporate a schematic figure in the method section as well as adjust the text to make the method description easier to follow and understand. Figure 2 is a preliminary example of an additional of the particle tracking procedure. We will also add a more descriptive text of the Mike She water flow model in the revised manuscript.

Language, style and structure throughout the manuscript are quite good and easy to follow.

- Authors response: We appreciate the specific language comments listed below which helps to increase the understandability and language quality.
In the introduction the authors present their hypotheses regarding the relationships between travel times and catchment characteristics. However, it seems that the review of relationships that have already been determined in former studies is a little short. I would like to point out that it is already quite well established that MTT saretime-variant and going along with this the strength of relationships between MTTs and certain physical catchment characteristics changes as well (this should be discussed at least in a little more depth than just mentioning it in line 49 once). Also research over at least the last 10 years has stressed the fact that it is in most cases a combination of multiple characteristics that control a catchment's MTTs. A comprehensive (short) summary of the state of art would be really helpful.

- Authors response: We appreciate the reviewer's comment and will add a section that better describes the state of art regarding this in the introduction of the revised manuscript. We have read the suggested literature and will use these to extend the introduction and discussion. Reviewer #2 also addressed this issue and we do agree that our hypothesis used in the original version of the manuscript was not the best way to present the results of our study. As responded to reviewer #2 comment, we decided to focus our study on the connection between baseflow travel times and stream chemistry (base cation and isotopes). We do believe we have a unique opportunity to distinguish the baseflow in the streams due to the prolonged winter, which gives little to no new input of water to the system during almost 6 months. The baseflow in turn, can be a very important signature for the amount and quality of groundwater.

I am also missing a more detailed analysis of the modeled travel time distributions. I am especially intrigued by the very steep initial rise of the cumulative distributions. Given the fact that the TTDs are related to baseflow conditions, this seems particularly puzzling to me. Therefore I would like to see more explanation and discussion in this part. A figure of the modeled (not cumulative) TTDs would also be interesting/helpful.

- Authors response: The groundwater in Krycklan is relatively shallow, especially around the mires. The till has a marked decreasing hydraulic conductivity with depth.
The main flow of water occur therefore in the shallow part of the till, resulting in fast groundwater flow especially during snow melt in the spring. TTD:s is the travel time of the groundwater for the whole simulated period, and thereby the steep initial rise of the TTD:s are mainly connected to the spring snowmelt. This emphasize the importance of conducting thorough investigations of TTDs and MTTs in different climate regions. We assume that the "true" baseflow can be represented by the groundwater reaching the streams during the winter period since there is no new input of precipitation to the soil during this time. This period also has an older age (gMTT). We will add a figure with monthly groundwater age distribution and the partition of groundwater and direct surface flow, and also change the TTD:s figure to show only the particles related to the baseflow (winter flow), see example figure 6.

The discussion and conclusion sections are straightforward and quite easy to comprehend. This may, unfortunately, be due to the fact that the results are not really new. Catchment size is not related to MTTs; hydraulic conductivity, slope and flow path length are related to MTTs. In the end I am left wondering whether there is enough novelty (maybe relating pH and base cations to MTTs?). A more in depth discussion of the modeled TTDs could help or maybe additional analyses on the interdependencies of how different catchment characteristics combined control MTTs. Still, I believe that in the end the authors will manage to add more analysis and discussion to justify a publication in HESS.

- Authors response: We will extend the evaluation regarding the TTDs and the chemistry data and give more focus to the unique opportunity to analysis the baseflow due to the prolonged winter conditions at the site. We will also add a figure that show the monthly groundwater age distribution to the streams and add a discussion towards the importance of baseflow for the different sub-catchment. Additionally, we will add a section in the discussion part that highlights the unique set of data and models available, which enables a coupled analysis of chemistry and hydrology in order to understand the complex integrated surface-subsurface water system of the Krycklan catchment. It

should be noted that in contrast to many previous studies this study was conducted in a snow-dominated catchment in the boreal forest region, which implies that one could expect differences in the hydrology and transit time distributions compared to more temperate regions where the majority of previous work has been conducted. We believe that the extended winter season conditions allow great opportunities to address the role of the base flow, its TTDs and relationship to water isotopes and water chemistry. Furthermore, we must also emphasize the comprehensive background data that was used for the setup and testing of the results, e.g. continuous discharge measurements and measurements of stable water isotopes and stream chemistry at 14 streams. Combined this should strengthen the reliability of the modelled TTDs.

Specific comments from reviewer #1

Authors disclosure: We agree with most specific suggestions and comments from reviewer #1. However, some sections of the introduction and discussion will be re-written with the help of this input and some sentences might end up being different in the revised manuscript.

Line 23: '...to investigate...'

- Authors response: We will change this according to the reviewer's suggestion.

Line 23: Better write: '...the travel time of the input to 14 [...] sub-catchments via groundwater to the stream...'.

- Authors response: We will change this according to the reviewer's suggestion.

Line 30: Move this sentence up to Line 27 just after '...stream water.'.

- Authors response: We agree to this change and we will change this according to the reviewer's suggestion.

Line 31: I would add whether these were positive or negative correlations.

- Authors response: We agree that this would make the manuscript easier to follow and

we will therefore change it accordingly.

Line 34: I would not call this a 'landscape characteristic'. Maybe better call it 'physical catchment characteristic'.

- Authors response: We will change it to "physical catchment characteristic" in the revised version of the manuscript to make the text clearer.

  Line 35: '...to positively correlate with MTT.' - Authors response: We agree to this change and we will change this according to the reviewer's suggestion.

Line 50: These referenced papers deal mainly with mean travel times, not with travel time distributions. Appropriate references for variations in TTDs would be for example Botter et al. (2010), Heidbüchel et al. (2012), Hrachowitz et al., (2013). Line 87: Again, I would not use 'landscape' factors since the term landscape refers to the visible landforms rather than to physical properties or subsurface features.

- Authors response: We like to thank the reviewer fort the suggested references. We have read and will add these to the introduction.

Introduction: This section lacks the mention/discussion of previous work that is very relevant to the study. Especially concerning research on the connection of travel time and catchment characteristics. Your hypothesis is that catchment size correlates with travel time, but what about any other catchment properties? So far travel times have been related to slope, flow path length, soil thickness, antecedent moisture content, hydraulic conductivity, just to name a few. I recommend expanding the short introduction a bit more to include a brief discussion of potential travel time controlling factors (Ameli et al., 2016; Birkel et al., 2016; Haitjema et al., 1995; Heidbüchel et al., 2013; Hrachowitz et al., 2010; McGuire et al., 2005; Yang et al., 2018).

- Authors response: We appreciate the reviewer's suggestion and will extend the introduction using the suggested material. We will also focus more on the connection between stream chemistry compared to model results. The long winter with no input

of precipitation to the groundwater makes it possible to distinguish the baseflow from the discharge affected by new input of water to the system. Krycklan has a significant amount of empirical data and long-term observations that provides a unique opportunity to analyze a northern snow-dominated catchment.

Line 106: '...type OF regolith...'?

- Authors response: We agree with this change and we will add "of" to this sentence.

Line 123: I would like to see a figure of the stratigraphy and how it is displayed in the model.

- Authors response: We will add a map over the typography and a soil map in the manuscript, as well as a more descriptive text about the Mike SHE flow model.

Line 172: Are these winter GW travel times travel times of particles that entered the catchment during the winter or exited the catchment during the winter (are they from 'forward' or from 'backward' TTDs)? This is a very important and interesting question since it could be very different catchment characteristics that control forward or backward travel times.

- Authors response: We agree that this could be better described to make it clearer. It is the age of particles exiting the catchment through stream discharge during winter regardless when they entered the groundwater as recharge, i.e. it is backwards TTDs. We will improve the clarity throughout the manuscript regarding weather it is forward or backward TTDs. Note that practically all precipitation falls as snow during the winter so typically there is little or no addition of water to the system during this period. Therefore, there is no addition of particles to the model during the winter months. Consequently, we doubt that it would be meaningful to trace the small amounts of water that might be added during the winters in these systems.

Line 215: According to this equation a mire coverage of 100% would result in an infinitely large adjusted cation concentration which is quite unrealistic (this is equivalent to a scenario where mires do not contribute any cations whatsoever). How do you justify this relationship?

- Authors response: The adjustment of the cation concentrations is based on observations of how mires affect the stream water concentrations of a wide range of elements. This has been described in detail by Lidman et al. (2014), which is cited in the manuscript. A general pattern was that the concentration of predominately weathering-derived elements such as the base cations decreased with the mire coverage. For the base cations, which do not sorb particularly strongly to peat, the decrease corresponded well with the decrease in mineral soils. This was therefore interpreted as dilution by mire water, since weathering does not occur in peat. For example, if 20% of the mineral soils in catchment was replaced by 20% peat, the weathering should be expected to decrease by approximately 20%, eventually causing approximately 20% lower concentrations in the runoff. (In practice, somewhat higher specific discharge from mires would increase the dilution effect, but then a slight addition of base cation with the precipitation would counteract this effect so the net effect is approximately 1% dilution by 1% wetland.) This is illustrated by Na in figure 3 with data from Lidman et al. (2014), where these matters are discussed in more detail for Na and other elements. We believe that this justifies the need to adjust the base cation concentrations for the effect of the mires in order to make a fair comparison to the modelled transit times, which include the mires. In principle, however, the Reviewer is correct in stating that this adjustment would be problematic in cases where the mire coverage approaches 100%. One reason, as the Reviewer points out, is that the adjustment would lead to unrealistically high concentrations if the mire coverage were close to 100%. This is, however, only a theoretical problem, since the mire coverage in the investigated sub-catchments never exceeds 44%. There is therefore no need to make such unrealistic and also unsupported adjustments. We find it hard even to envisage how a catchment with much higher mire coverage would be able to develop in this area, but since no such sub-catchments exist in the present study we believe that this discussion could be left for another context. Another reason why we would be reluctant to use this type

of adjustment for such extreme cases where the mire coverage was close to 100% (had it for some reason been necessary) is that it would require extrapolation far outside the observed range. In this case the relationship to the concentration is significant between in the range 0-44% mire coverage (as in figure 3). The observations were made in the same streams that were investigated in this study so there is clearly a justifiable relationship between base cation concentrations and mire coverage.

Line 220: What exactly do you mean by this ('...less impacted...', '...also considered.')?

- Authors response: What we meant to express was that there is a fundamental difference between the water isotopes, on the one hand, and the stream chemistry, on the other. The interpretation of the water isotopes is based on the characteristic variation in the input signal throughout the year. Consequently, the important parameter is the variation in the water isotope signature in the runoff, as given in the manuscript. The annual average, however, is not a meaningful parameter. In the case of pH and base cation concentrations, the major drivers are processes within the catchment, e.g. weathering rates and transit times, and it is therefore meaningful to look also at the annual averages in this case. Apparently, we will need to explain this clearer in a revised manuscript.

Line 239: But the simulated specific baseflow is not included in Table 4.

- Authors response: Information about the specific discharge was omitted from Table 4 at a late stage. It must however still be left in the text. This needs to be corrected.

Line 243: Again it would be important to know whether these are forward or backward times.

- Authors response: We agree with this comment and will check and adjust the entire text accordingly. To answer the question, it is backwards times, i.e. the age of a particle when it reaches a stream and exits the model, regardless of when they were first introduced to the groundwater. Since the system is frozen during the winters, there

is only negligible addition of water. Hence, it would not be very meaningful to study the forward times for the winter period.

Line243: I would some how indicate that the means are geometric instead of arithmetic when you write MTT (maybe something like 'gMTT'?). Because one will automatically assume that MTT is the arithmetic mean of the travel times.

- Authors response: Thank you for the comment, we will adjust this in the next version since it will improve the clarity of the text and use the suggested gMTT.

Line 245: This is not a complete sentence (verb missing).

- Authors response: Thank you, we will change this in the next version

Line 248: 'Over the course of a year...'; '...may enrich or dilute...'

- Authors response: We agree with this change and change this in the next version

Line 259: Winter baseflow or winter groundwater fraction?

- Authors response: Winter groundwater fraction is the winter discharge divided by total yearly discharge. We will change this text.

Line272: Instead of 'decreased' I would rather write'...became more negative/became more enriched in the lighter isotope...'

- Authors response: We agree with this and change this in the next version

Line 277: '... averages...'

- Authors response: Thank you, we will change this in the next version

Line 292-294: It would be good to add 'positive' or 'negative' before 'correlations', so you immediately know the direction of the relationship.

- Authors response: We agree that this would add to the manuscript and make it easier for the reader to directly understand the relationship without having to look at the

figures.

Line 297: How do these equations show that LCS is the most significant parameter?

- Authors response: The reviewer has a valid point here. It is hard to justify based only on the equations in questions that LCS is the single most important parameter for controlling the MTTs in the investigated catchment. This should clearly be rephrased in a more stringent way. The message we were trying to convey is that it seems hard to get around the impact of the LCS when looking at the MTTs throughout the landscape. As in this example, it is difficult to replace LCS by any other parameter and obtain equally good correlations, which indicates that it probably plays an important role for controlling MTTs. We will improve this explanation in the next manuscript version.

Line 304: 'Percentage of low conductive sediments...'

- Authors response: We agree with this change and will adjust it in the revised manuscript.

Line 314: '...on the other...'?

- Authors response: We checked this and "...on the other..." should be removed.

Line 318: '...and a gamma distribution transfer function (convolution) method...'

- Authors response: We agree with this edit and will change it in the revised manuscript.

Line 318: 'In a conceptual, non-distributed modeling study...'

- Authors response: We agree with this edit and will change it in the revised manuscript.

Line 319: Some more details would be helpful ('another travel time distribution technique').

- Authors response: We will add more details regarding this technique.

Line 361: This section is a bit unstructured. You start out by stating that slope and hydraulic conductivity are the main factors controlling MTTs, then you state that the

fraction of LCS is the most important factor adding travel distance to the mix before arguing that the steeper small C20 behaves differently maybe also because of the iňĆuvial deposits fraction... Since these are some of your main iňĄndings it would be good to clarify the section. Also, what about the fact that you released the particles at the top of the transient groundwater table? That means that depending on the groundwater table a larger or smaller part of the regolith was not taken into account for the MTT calculations. If the water table was high a larger proportion of the regolith was considered, if it was low the particles started somewhere else in the profile. What are the implications of this? Would that iniňĆuence your results?

- Authors response: Please check the new figure 2 and new figure 1. Most of the calculated groundwater table has a level between 0-3 m below the ground. The vertical discretization of the saturated zone follows the soil layers and are a few meters in depth close to the surface and increasing in thickness with depth. Note that the horizontal grid-size is 50*50 m and therefore the MTT will be bias to long travelling groundwater. Peatlands are generally the areas with an average groundwater table above 1 m, while the till areas have a depth between 2-3 m in average. However, at C14, the deep esker has a lower water table than on other locations in the catchments. The esker has the highest vertical hydraulic conductivity, and thereby the fastest route from infiltration to recharge. Although the horizontal distance is still larger than the distance to the groundwater table and the vertical hydraulic conductivity is large, one should note that this could impact the results and give C14 and C16 older MTT than if the phreatic surface was at the same level around the whole catchment. It can be noted that the areas were the MTTs mainly could be suppressed by this limitation in the modelling tools are areas with relatively long MTTs. This effect therefore somewhat counteracts the general patterns that were observed in this study. Hence, the observed patterns are not a result of the differences in groundwater depth. On the contrary, we would have expected patterns that are even more distinct if it had been possible to account for the time it takes for the water to percolate to the groundwater surface. We will add a discussion about this, since it is an interesting subject and relevant for the interpretation

of the results and try to make the section more structured.

Line 380: It would be good to point out more the novel aspects of your work. What did you find out that was really new? The fact that Darcy's law works? Catchment size has been ruled out as a MTT control since quite a while now. Or is it mainly a study that confirms the applicability of the particle tracking of your model by comparing results to time series of pH, base cations and stable isotopes?

- Authors response: Reviewer #2 also indicated that we need to highlight more clearly the novelty of this study so this is one of the major concerns we would have to address in a revised manuscript. The original manuscript was organized around the hypothesis that MTTs are controlled by catchment size, but based on the comments of both Reviewers and the available literature, we agree that the results of this study could be presented in a better way by omitting this hypothesis as a rationale for the study and instead focus on other more novel aspects of this work. This includes emphasizing the implication of the northern landscape setting where the hydrology processes are dominated by prolonged winter conditions, and a dominating spring flood. Krycklan is one of the most well-investigated and well-instrumented catchments in the boreal region with a large database of empirical observations that allow multiple ways to test the predictions of hydrological models. In our opinion, the discussion about MTTs, TTDs and their variation at different scales in the landscape is far from finished, and we believe that this study could contribute significantly to the understanding of the hydrology of the snow-dominated boreal forest landscape, the coupling between hydrology and water quality and, not least, the role of winter base flow in a changing climate. It is far from obvious that MTTs and TTDs would look the same and be controlled by the same factors in different types of landscapes and climates, so we believe that this type of studies from different areas of the world are necessary to improve our understanding of these matters. We further believe that should highlight the role of the base flow better in a revised version of the manuscript, since the long winters make these boreal systems particularly suitable for addressing their role. We should also try to emphasize

better the large amount of data that were used to set up and test the model, e.g. the continuous discharge measurements in the streams and the extensive monitoring of water isotopes and stream chemistry. Combined we believe that this background data significantly strengthens the credibility of the conclusions we present.

Figure specific comments: Figure 1: Why do you call the sub-catchments subareas in this ïñĄgure? Also, all subcatchments (not only the green ones) connect before reaching the main outlet at C16.

- Authors response: We will change sub-areas to sub-catchments in the figure. Yes, all streams connect before they reach C16. However, the color code was used to illustrate which areas connect before they reach the white area. We will try to clarify this in the figure text.

Figure 2: I am confused by the y axis label. Why does it start with 100% and then decrease to 0%? Isn't this a cumulative TTD (that should start with 0%)? Also, you never mention how you construct this TTD. Do you record the time the particles that arrive in the stream in the winter needed to travel through the catchment (backward TTD) or do you record the time that particles that are released during winter need to travel through the catchment (forward TTD)? I am curious since the cumulative TTDs exhibit a shape I would not have expected since the initial rise is very steep although you use a logarithmic x axis. For a groundwater TTDs in particular, this extremely high initial values are quite unusual. How do you explain this?

- Authors response: Thank you for pointing this out! There was an error in this figure, the number of particles should start at 0 % and increase to 100 %, so the y-axis should change, see figure 4. The TTD is the age of particles when they reached a stream for the whole simulated period, winter has not been selected in this figure. However, we will clarify the text, change the y-axis and pick out only the winter period. The reason for the initial rise is due to that it includes particles coming to the streams during the whole year (Jan-Dec). The snowmelt impacts the groundwater travel time age and therefore
there is a rapid start. Given that a substantial portion of the annual precipitation leaves the catchments during a few weeks of spring flood, we do not think that the initial steepness of the distribution is surprising. This is probably one of the characteristics of boreal systems compared to other types of systems and emphasizes the importance of conducting this type of studies in different climate regions.

Figure3: So if youre place the geometric MTT with thec ommonly used arithmetic MTT, how does that change the correlations? In case they become weaker you could argue for the general use of the geometric MTT.

- Authors response: The arithmetic MTT is often used when looking at the gamma convolution distributions using isotopic data. It has been argued that the MTT should not be used for MTT older than 5 years because the amplitude signals are lost (Kirchner, 2016). Therefore, the aMTT is less impacted by the "old tail". For particle tracking or modelling methods the median or the geometric MTT may describe the bulk of the particles better than aMTT, because one particle (even though there are hundreds of particles in total) with a very old age may cause significant differences in the MTT estimation. This is undesirable, since the age of these very old particles becomes increasingly uncertain and are very hard to validate. The gMTT is less dependent on a few old particles but does still take them into account. In our case, the aMTT correlates well with the gMTT (see figure 5) and aMTT is shown in the supporting information but we will move them to a table in the main manuscript. Kirchner, J. W.: Aggregation in environmental systems – Part 1: Seasonal tracer cycles quantify young water fractions, but not mean transit times, in spatially heterogeneous catchments, Hydrol. Earth Syst. Sci., 20, 279–297, https://doi.org/10.5194/hess-20-279-2016, 2016.

Supplements Supplements S1: Check again for typos and misuse of words throughout the supplement (in particular in the table captions and the table foot notes).

- Authors response: We will check this in the revised version

S1, line 3: What does that mean that the table includes both non-transformed and

log-normal transformed data? Which is which?

- Authors response: We will check this and make the table easier to understand.

S1, line 4: What happened to the other 44% of particles?

- Authors response: These particles reached a model sink, either the model boundary through groundwater outflow, or reached the unsaturated zone/surface water that is not connected to a stream or a lake.

S1, line 8: 'Artesian' mean? And why 'back-transformed'?

- Authors response: We will make this text and corresponding table easier to read. "Artesian" should be arithmetic

S1, line 19: '...as well as the precipitation (P)'

- Authors response: We will change this according to the suggestion

S2, line 3: What does 'The statistics are based on...' mean?

- Authors response: We will make this text and corresponding table easier to read and explain more in the text what the table show.

S2: The tables are not that helpful. Many of the abbreviations are not explained

- Authors response: We will make this text and corresponding table easier to read and explain more in the text what the table show as well as explain all abbreviations.

Add to literature: Literature Ameli, A. A., Amvrosiadi, N., Grabs, T., Laudon, H., Creed, I. F., McDonnell, J. J., and Bishop, K.: Hillslope permeability architecture controls on subsurface transit time distribution and flow paths, J. Hydrol., 543, 17–30, https://doi.org/10.1016/j.jhydrol.2016.04.071, 2016. Birkel, C., Geris, J., Molina, M. J., Mendez, C., Arce, R., Dick, J., et al.: Hydroclimatic controls on non-stationary stream water ages in humid tropical catchments, J. Hydrol., 542, 231–240, https://doi.org/10.1016/j.jhydrol.2016.09.006, 2016. Botter, G., Bertuzzo,

E., and Rinaldo, A.: Transport in the hydrologic response: Travel time distributions, soil moisture dynamics, and the old water paradox, Water Resour. Res., 46, https://doi.org/10.1029/2009WR008371, 2010. Haitjema, H. M.: On the residence time distribution in idealized groundwatersheds, J. Hydrol., 172, 127–146, https://doi.org/10.1016/0022-1694(95)02732-5, 1995. Heidbüchel, I., Troch, P. A., Lyon, S. W., and Weiler, M.: The master transit time distribution of variable flow systems, Water Resour. Res., 48, https://doi.org/10.1029/2011WR011293, 2012. Heidbüchel, I., Troch, P. A., and Lyon, S. W.: Separating physical and meteorological controls of variable transit times in zero-order catchments, Water Resour. Res., 49, 7644–7657, https://doi.org/10.1002/2012WR013149, 2013. Hrachowitz, M., Soulsby, C., Tetzlaff, D., Malcolm, I. A., and Schoups, G.: Gamma distribution models for transit time estimation in catchments: Physical interpretation of parameters and implications for time-variant transit time assessment, Water Resour. Res., https://doi.org/10.1029/2010WR009148, 2010. Hrachowitz, M., Savenije, H., Bogaard, T. A., Tetzlaff, D., and Soulsby, C.: What can flux tracking teach us about water age distribution patterns and their temporal dynamics?, Hydrol. Earth Syst. Sci., 17, 533–564, https://doi.org/10.5194/hess-17-533-2013, 2013. McGuire, K. J., McDonnell, J. J.,Weiler, M., Kendall, C., McGlynn, B. L., Welker, J. M., and Seibert, J.: The role of topography on catchment-scale water residence time, Water Resour. Res., 41, https://doi.org/10.1029/2004WR003657, 2005. Yang, J., Heidbüchel, I., Musolff, A., Reinstorf, F., and Fleckenstein, J. H.: Exploring the dynamics of transit times and subsurface mixing in a small agricultural catchment, Water Resour. Res., https://doi.org/10.1002/2017WR021896, 2018.

- Authors response: We thank the reviewer for these literature suggestions which can be used to especially improve the introduction section further. We will read these suggestions and incorporate them were they seem to be most fitting.

Please also note the supplement to this comment:
https://www.hydrol-earth-syst-sci-discuss.net/hess-2020-121/hess-2020-121-AC1-

supplement.pdf

[Figure]

[Figure]

**Fig. 1.** The Krycklan catchment

[Figure]

**Fig. 2.** Model setup

[Figure]

**Fig. 3.** Na concentrations

**Travel time distribution**

(a)

*All sub-catchments*

(b) C2, C4, C7, C5, C6, C9, C13, C16

(c) C1, C10, C12, C16

(d) C20, C14, C15, C16

Legend: C2, C4, C7, C1, C5, C6, C20, C9, C10, C12, C13, C14, C15, C16

**Fig. 4.** TTD

[Figure]

**Fig. 5.** aMTT and gMTT relationship

[Figure]

**Fig. 6.** Discharge distribution

Figure text

**Figure 1:** The Krycklan catchment

(a) Location of sub-catchment and their outlets (red circles). The areas are color-coded based on their stream network connections, e.g., all sub-catchments of one color connect before reaching the white area. For further details of the catchment characteristics, see Table 1.

(b) Soil map used in the hydrology model is based on the quaternary deposits map (1:100,000) and depth to bedrock map from the Swedish Geological Survey (2014), combined with field investigations.

(c) Catchment topography, shown as meter above sea level (m.a.s.l.).

**Figure 2:** Model setup

(a) Step by step of the particle tracking procedure.

(b) Average depth to groundwater table. The main part of the model area has a calculated depth to the groundwater table between 0-3 m. Note that the top vertical layering of the saturated zone is 2.5 m at the soil surface and the thickness thereafter follows the soil layers (thickness increasing with depth) while the horizontal grid-size is 50x50 m.

(c) Schematic illustration of particle tracking set up. Particles are added in each cell at the depth varying and transient groundwater table. The age of these particles is zero at the time of recharge. The particles then follow the groundwater flow while increasing in age. All particles that reaches a stream or lake receives an end age which is equal to time from recharge to discharge in the stream. MTT is calculated for each stream using these particles.

**Figure 3:** Average Na concentrations in the investigated streams as a function of the mire coverage. Data from Lidman et al. (2014).

**Figure 4:** Simulated travel time distribution of the groundwater in Krycklan. C16 is used as a visual reference in all panels. The figure includes all 14 investigated sub-catchments, color-coded as Fig. 1 and displayed in the legend in size order from left to right with C2 being the smallest and C16 the largest sub-catchment. The figure is divided into (a) all sub-catchments, (b) the sub-catchments of C13, (c) the sub-catchments of C12 and, (d) the sub-catchments of C14 and C15. The figure shows the full year distribution of particles.

**Figure 5**: aMTT and gMTT relationship

**Figure 6** – Yearly discharge distribution of C16 (full-catchment). The discharge is divided into different source fractions, ranging from surface runoff to groundwater flow. The groundwater has been further divided into age groups which been calculated through the particle distribution for each month. The groundwater categories shown are; less than 3 months, less than 1 year, less than 5 years and more than 5 years.

**Fig. 7.** Figure text

---

## Author Comment (AC2) · 9 Jul 2020

Introduction comment from reviewer #2 In the manuscript "Linking groundwater travel times to stream chemistry, isotopic composition and catchment characteristics," Sterte et al. analyze the drivers of catchment travel times across catchments in norther Sweden. They use a physical hydrology model combined with particle tracking to generate transit time distributions and compare this to isotopic and stream chemistry observations. Overall, I think that the study is well done and the manuscript is well written and easy to follow.

- Authors response: We would like to thank reviewer #2 for his/her constructive criticism

and suggestions, which we believe will help to improve the manuscript. The main concern seemed to be the introduction and hypothesis, which we, having read the reviewer's comments, agree should be re-worked to progress the manuscript further. Our explanations and responses to all the reviewer's comments and questions are listed below.

However, I do have some significant concerns about the manuscript in its present form. My most serious concern is that it's not clear what the novelty of this study is that would warrant publication in HESS. All of the methods used here are well established and the idea that catchment travel time distributions are driven by catchment characteristics is not new. The authors start from the hypothesis that catchment area is the main driver, however previous research has already indicated that many drivers will be important, so disproving this hypothesis does not seem to be the best angle to take here. I would suggest the authors consider what portions of their findings are the most novel addition to the body of literature in this area and organize the manuscript around this rather than the area hypothesis.

- Authors response: Thank you for this comment, it will help to make a better manuscript. We received a similar comment from reviewer #1. Based on these comments, we will re-focus the manuscript, and in the new version, better highlight the novel aspects of this study. This includes emphasizing the implication of processes in the northern landscape setting where the hydrology is dominated by prolonged winter conditions followed by snowmelt. We do that by studying one of the most well-investigated and well-instrumented catchment systems in the boreal region. Both reviewers have pointed out that the hypothesis that the catchment area is the main factor controlling the travel time distributions is obsolete and that the manuscript, therefore, should not be organized around this idea. Having considered the remarks of the reviewers and studied the literature thoroughly, we feel inclined to agree that this was a mistake and that the manuscript would benefit from a revision where this idea does not have such a central place. As Reviewer #2 remarks below, this should not be the

storyline of the manuscript, particularly as neither the modelling results nor the observations supported the area hypothesis. Therefore, we see that there are possibilities to improve the manuscript by emphasizing and developing other aspects of the results, for instance, the role of base flow in these boreal systems as this will make the novel contributions of this study clearer.

Even if the area hypothesis is what guided the study in the first place, this does not need to be the storyline of the manuscript. Along the same lines as my first comment I think the introduction could use significant revision. As it stands it is a very broad overview of the topic but I would like to see a more thorough review of previous finding directly relevant to this work that can clearly motivate the novelty of this study and the gap it is filling. Similarly I think the discussion section would be more powerful if it provides a better evaluation of how and where results form this study add new information/disagree or provide additional corroboration to existing studies.

- Authors response: Thank you for your comment. We understand that the introduction needs to be extended with at least a section regarding relevant works related to this study. We received some very helpful reference suggestions from reviewer #1, which we will incorporate to change and extend the introduction with regards to relevant research. We also agree that the catchment size hypothesis used in the original version of the manuscript is not the best way to present the results of this study. Both the introduction and the discussion would, therefore, have to be changed accordingly. We will focus the study on the connection between baseflow travel times and stream chemistry (base cations and isotopes). There is a unique opportunity to distinguish the baseflow in the streams due to the prolonged winter, which gives little to no new input of water to the system. The baseflow, in turn, can be a very important signature for the amount and quality of groundwater. The boreal systems are sensitive to climate change, and to have as much of a base understanding of these systems as possible is important to be able to evaluate changes in the future. Shorter snow seasons and changes in the amount of precipitation can change travel times in the future, which can

have an impact on weathering and biological processes. We have in this study show a strong relationship between isotope mixing and base cation concentrations, on the one hand, and groundwater travel times on the other. We have already seen changes in the climate in the last couple of years, making this a pressing issue.

For the most part I think the paper is very clearly written, however the description of the modeling approach is a bit confusing and could use some more details. For example the term simulation is used to refer to both the hydrology model and the particle tracking portion which can be confusing. This section could be helped by a ïñĄgure or a schematic to illustrate the approach I think.

- Authors response: Reviewer #1 also requested a schematic figure, and we agree with the reviewers and like the suggestion to better describe the modeling procedure, see a preliminary figure 2. We will also give more distinctive terms to the hydrology modelling part and the particle tracking part of the manuscript. One reason for the confusion is probably that much of the hydrological model was described in a previous paper. Although this paper was cited, we realize that we need to explain the model setup in more detail also in this manuscript so that potential readers can grasp the general idea of what has been done without reading the previous paper.

SpeciïñĄc comments from reviewer #2

Authors' disclosure: We agree with the most specific suggestions and comments from reviewer #2. However, some sections of the introduction and discussion will be re-written whit the help of this input, and some sentences might end up being different in the revised manuscript.

I think the catchment numbering could be done in a more intuitive way so its easier to separate unique outlets (i.e C12-15). I would suggest giving each of these outlets their own letter and then numbering points within them potentially by drainage area, that way itis easier to compare when points are within the same drainage or not.

- Authors response: We understand the concern regarding this comment. However, we hesitate to change the names of the sub-catchment since they all have been included in many previous peer-reviewed papers, which argues against introducing new names in this manuscript. It would make all comparisons with previous papers from the Krycklan catchment unnecessarily difficult and confusing. We hoped that the colors of figure 1 will help to distinguish which sub-catchments were connected. However, we will also add a color code in table 1, which will further connect the different sub-catchments in figure and text.

Line 138-140: This is confusing are you trying to say that the hydrologic model is run first and then the particle tracking is applied to the outputs of that model?

- Authors response: Yes, the model flow field is run first and, then one year of the flow field is cycled for 1000 years with particle tracking applied to the flow field results. We will add the schematic figure (see figure 2) to clarify these steps and be more direct that the hydrology model output used are the results from the model is used in Jutebring et al. 2018. We will also add a more in-depth description of the water movement model in the revised manuscript.

Section 2.3 – this is really more a description of scenario design than numerical methods. I would consider renaming.

- Authors response: We appreciate this suggestion from the reviewer and will incorporate the suggestion and change the name from numerical methods to scenario design.

Line 148: 'several years' is very vague can you be more precise.

- Authors response: We agree with the reviewer and will be more precise. It is one year of the Mike SHE flow results that has been cycled for 1000 years in order to allow for long-term particle tracking. We will clarify this throughout the manuscript.

Line 154: at what time frequency were particles injected into the model? Just at the start of each year?

- Authors response: We will clarify this section of the text. The particles are only introduced in the first year. The Mike SHE flow results of that year are then cycled 1000 times to let the particles travel for a longer period (1000 years).

Line 159: I think the more standard reference for this would be heavy tailed rather than long tailed.

- Authors response: A distribution with a long tail means that there is a large proportion of the distributions is far away from the main central tendency. A long-tailed distribution can also be heavy tailed, meaning that a distribution goes towards zero slower than an exponential distribution. In the section in question, we are talking about the use of the geometric mean instead of the arithmetic mean for long-tailed distributions because the geometric mean is less bias to the long tail and describes the central tendency more effectively.

The term simulation is used to refer to both the hydrology model and the particle tracking model and this can make the methods confusing when you are talking about run times for example.

- Authors response: We will look through the manuscript and try to be more distinct when using the term simulation so that it becomes clearer when we talk about the flow model or the particle tracking.

At the beginning of section 2.3 you say that you used several years of simulation but actually it looks like you use only one year of the hydrologic simulation but repeated it 1000 times (i.e. more than several). This description is confusing.

- Authors response: We have taken one year and repeated it 1000 times. We will be more distinct and not use the word several for better clarity in the revised manuscript.

I think some of the tables could be converted to figures to better present the information. For example Table 4 could be presented as a series of maps.

- Authors response: We do not understand exactly what the reviewer is asking for. Most

of the information in Table 4 are shown in the figure 2 in the manuscript. We attempted to make maps at the onset of this study, but the information was not clearly visible. However, we will add travel time distribution per month for each sub-catchment to visualize the impact of various fractions of travel times. See an example of C16, Figure 6.

Please also note the supplement to this comment:
https://www.hydrol-earth-syst-sci-discuss.net/hess-2020-121/hess-2020-121-AC2-supplement.pdf
* * *
(a)

(b)

Till
Peat
Water
Silty sediments
Sandy sediments
Shallow soils/bedrock outcrops

(c)

m.a.s.l.
> 300
< 100

1000 m

**Fig. 1.** The Krycklan catchment

[Figure]

**Fig. 2.** Model setup

[Figure]

Discharge proportion

■ > 5 year  ■ < 5 years  ■ < 1 year  □ < 3 month  ⊟ Surface runoff

**Fig. 3.** Discharge distribution

Figure text

**Figure 1:** The Krycklan catchment

(a) Location of sub-catchment and their outlets (red circles). The areas are color-coded based on their stream network connections, e.g., all sub-catchments of one color connect before reaching the white area. For further details of the catchment characteristics, see Table 1.
(b) Soil map used in the hydrology model is based on the quaternary deposits map (1:100,000) and depth to bedrock map from the Swedish Geological Survey (2014), combined with field investigations.
(c) Catchment topography, shown as meter above sea level (m.a.s.l.).

**Figure 2:** Model setup

(a) Step by step of the particle tracking procedure.
(b) Average depth to groundwater table. The main part of the model area has a calculated depth to the groundwater table between 0-3 m. Note that the top vertical layering of the saturated zone is 2.5 m at the soil surface and the thickness thereafter follows the soil layers (thickness increasing with depth) while the horizontal grid-size is 50x50 m.
(c) Schematic illustration of particle tracking set up. Particles are added in each cell at the depth varying and transient groundwater table. The age of these particles is zero at the time of recharge. The particles then follow the groundwater flow while increasing in age. All particles that reaches a stream or lake receives an end age which is equal to time from recharge to discharge in the stream. MTT is calculated for each stream using these particles.

**Figure 3** – Yearly discharge distribution of C16 (full-catchment). The discharge is divided into different source fractions, ranging from surface runoff to groundwater flow. The groundwater has been further divided into age groups which been calculated through the particle distribution for each month. The groundwater categories shown are; less than 3 months, less than 1 year, less than 5 years and more than 5 years.

**Fig. 4.** Figure Text

---

## Author Response (AR1)

**Authors' response**

Dear Editor and Reviewers,

We hereby enclose a new version of the manuscript after revising it using your valuable input and constructive criticism of the two Reviewers. The provided material starts with a letter to the Editor followed by a list with significant changes. Thereafter, there is a point-by-point response to each of the comments of the Reviewers on the previous version of this manuscript and a marked-up manuscript version of the old manuscript, where all revisions can be seen. For clarity all previous comments have been included unabridged in black text below, while our response is provided in red.

**Response to Editor Decision**

**Reconsider after major revisions (further review by editor and referees)** (09 Jul 2020) by Conrad Jackisch
Comments to the Author:
Dear Elin and co-workers,

Thank you again for submitting your manuscript to our special issue. Thank you also for your replies to the Reviewers' comments. You have drafted ways to take up their constructive criticism in the revisions.

I think it is also worthwhile to revisit our abstract of the special issue for that. We hope to find contributions which really aim at the intertwined relations of landscape properties and landscape functioning. As you have discussed in your replies, working out the novelties towards indicators of changes, methodological implications or restrictions etc. would be a way forward. So far, I feel reluctant to see a shifted focus on baseflow alone to be sufficient. But in combination with your suggestion to work out the role of snow coverage and the interrelation with the catchment's characteristics this appears more plausible to me.

You have stated that the Kryklan catchment is one of the best instrumented and understood boreal catchments. From the point of view of our special issue, this could be transformed into a large advantage to really track the value of this data to improve our understanding. Given the situation that MikeSHE and particle tracking is not the novelty and that it is neither the findings about MTTs/TTDs, maybe turning the argument towards what is the added information and where does it actually come from could be a way to also include a critical view regarding the state of our hypotheses and tools in hydrological sciences. During the discussions among the Guest Editors we found that much of our intentions with the special issue can be seen as closely related to the Critical Zone/Hydropedology concept of Henry Lin et al. but for the landscape/catchment in our case.

I hope you can align your revisions to these thoughts and would be pleased to receive your revised manuscript in due time.
All the best.
Conrad

**Reply:** We like to thank the Editor for these comments and suggestions. Reviewer #1 and #2 also indicated that we need to highlight more clearly the novelty of this study, so this is one of the major concerns we had to address in a revised manuscript. As stated in our initial reply to the Reviewers, the original manuscript was organized around the hypothesis that MTTs are controlled by catchment size. However, based on the comments of the Editor, both Reviewers and the available literature, we agree that the results of this study could be presented in a better way. We omitted the hypothesis as a rationale for the study and instead focused on other more novel aspects of this work.

In the new manuscript, we emphasized the implication of the northern landscape setting with prolonged dry winter conditions, followed by an intensive flow spring flood, dominating the hydrology. The seasonality of the catchment gives a unique opportunity to investigate seasonal changes in travel times and young water fractions and linking seasonal changes to catchment characteristics. We used isotope and base cation data to further test the credibility of the model results with respect to seasonal travel times and young water fractions. We linked the travel times to distinct catchment characteristics on a seasonal basis. We added the seasonality aspect in this version of the manuscript since we believe that this is a novel extension to the previous results. This extension will hopefully be of interest and give a new insight to intra-annual travel times for catchments with significant and distinct seasonality. Especially the winter and spring seasons have a distinct impact on stream chemistry in this northern setting and presumably the boreal landscape at large, and we believe that linking and explaining these effects and how they can be connected to catchment characteristics can give new insights to the functioning of boreal catchments.

To our knowledge, Krycklan is one of the most well-investigated and well-instrumented catchments in the boreal region with a large database of empirical observations that allow multiple ways to test the predictions of hydrological models. In our opinion, the discussion about MTTs, TTDs, and their variation at different scales in the landscape is far from finished. We believe that this study contributes significantly to the understanding of the hydrology of the snow-dominated boreal forest landscape, the coupling between hydrology and water quality, and, not least, the role of travel times in a changing climate. We argue that it is far from evident that MTTs and TTDs would look the same, and be controlled by the same factors, in different types of landscapes and seasons, so we believe that this type of studies from different areas of the world is necessary to improve our understanding of these matters.

We sincerely thank you and the Reviewers for your input and constructive criticism, which improved the new version of the manuscript. We hope that these changes will make you reconsider this manuscript for publication in Hydrology and Earth System Sciences (HESS).

**List of significant changes:**

- **The manuscript is no longer restricted to the winter baseflow but includes seasonal and annual travel times and young water fractions.**
  - The seasons included are three distinct seasons of the Krycklan catchment:
    - winter low flow - (no new recharge to the groundwater as the precipitation falls as snow and remains frozen)
    - spring high flow (snowmelt occurring late March to mid-May)
    - summer (season impacted by evapotranspiration, here defined as Jul to Sept, to reduce the impact of spring snowmelt (spring) and early snow (winter)).
  - New results for the 14 monitored sub-catchments:
    - Seasonal changes in travel times and young water fractions
    - The role of mires for young water and overland flow
    - Governing factors changes between seasons
  - Tables have been edited in the new manuscript:
    - Tables include seasonal observations and model results
    - Table 6 is new and includes seasonal correlations to catchment characteristics
  - Figures have been edited in the new manuscript to provide more in-depth analysis and presentation of the results:
    - Annual and seasonal results for catchment characteristics have been added
    - Figure 2 is new and is used to explain the theories used to connect seasonal stream chemistry to travel times and young water fractions.
    - Figure 5 is new, with an extended version in the Appendix. It shows the intra-annual variations of groundwater ages for the investigated sub-catchments. Compared to the annual mean times that were discussed in the previous version these graphs provide a more complete description of how the travel times vary both between different types of catchments and between different seasons

- **The introduction and discussion were changed.**
  - The introduction was extended using reference suggestions by Reviewer #1.
  - The introduction and discussion were updated to reflect the new scope of the manuscript and the new results that are presented. We hope that the extension of seasonality is a good step forward for a more interesting take on travel times. We believe that the snow-dominated Krycklan catchment with distinct seasonality gives an extra novelty aspect to these results.
  - The hypotheses regarding that catchment size is the mayor impacting characteristic has been omitted, and the manuscript is focused on explaining and understanding the seasonality of travel times and young water fractions in the snow-dominated boreal landscape.

- **Reviewer #1 and Reviewer #2 both requested an extension of the method section.** The method section in the new version of the manuscript was therefore extended to include a more in-depth description of the hydrological model and particle tracking. Figure 1 has been changed to include a soil map and a topography map. Figure 3 is new and is used to help explain the particle tracking method.

- **The abstract and title was changed accordingly to reflect the changes listed above**

**Response to Reviewer #1**

**Introduction comment from Reviewer #1**

- In this study the authors used a physically-based distributed model in combination with a particle tracking approach to determine groundwater travel times in the well-studied Kryklan catchment in northern Sweden. They compared the modeled mean travel times (MTTs) with average winter values of stable isotopes, pH, base cations and found significant correlations for all of them. Furthermore they tried to relate MTTs to certain catchment characteristics. The only strong and significant correlation they found was between MTTs and the fraction of low conductive sediments.

**- Reply:** We would like to thank Reviewer #1 for his/her constructive criticism and suggestions which helped improve the new version of the manuscript. The main concern raised was the need to improve the analysis and discussion part, which, we agree has improved the manuscript further. Our explanations and responses to all the Reviewer's comments and questions are listed below. All the comments have been included unabridged in this document, together with our response following each comment (red).

- The use of particle tracking approaches to determine travel time distributions with numerical models becomes more and more common in catchment hydrology. And although some of these approaches still suffer from certain simplifications (missing dispersion component, particles disappearing after temporary exfiltration, etc.) they can already shed light on general catchment dynamics. Having said that I am missing a more detailed description of both the model setup with Mike SHE and the particle tracking approach in particular (one or two additional figures would not hurt).

**- Reply:** We understand the Reviewers concerns regarding the method sections. A main reason for this is that the present model is based on the model that already has been thoroughly described in a previous paper by Jutebring Sterte et al. (2018), which is cited in the manuscript. However, we realize the need to provide the readers of this manuscript with a more proper background. We therefore incorporated a schematic figure in the method section as well as adjust the text to make the method description easier to follow and understand. Please see Figure 3 in the new manuscript which is an additional part of the particle tracking procedure. We have also added a more descriptive text of the Mike She water flow model in the revised manuscript and there is also an extended version of Figure 1.

- Language, style and structure throughout the manuscript are quite good and easy to follow.

**- Reply:** We appreciate the specific language comments listed below which helps to increase the understandability and language quality.

- In the introduction the authors present their hypotheses regarding the relationships between travel times and catchment characteristics. However, it seems that the review of relationships that have already been determined in former studies is a little short. I would like to point out that it is already quite well established that MTT saretime-variant and going along with this the strength of relationships between MTTs and certain physical catchment characteristics changes as well (this should be discussed at least in a little more depth than just mentioning it in line 49 once). Also research over at least the last 10 years has stressed the fact that it is in most cases a combination of multiple characteristics that control a catchment's MTTs. A comprehensive (short) summary of the state of art would be really helpful.

**- Reply:** We appreciate the Reviewer's comment and have added a section that better describes the state of art regarding this in the introduction of the revised manuscript. We have read the suggested literature and used these to extend the introduction and discussion. Reviewer #2 also addressed this issue and we do agree that our hypothesis used in the original version of the manuscript was not the best way to present the results of our study. We added the concept of seasonality in the new manuscript, i.e., we aimed to quantify the age of the water for three district seasons and try to find explanations to intra-annual changes in travel times for different sub-catchments.

- I am also missing a more detailed analysis of the modeled travel time distributions. I am especially intrigued by the very steep initial rise of the cumulative distributions. Given the fact that the TTDs are related to baseflow conditions, this seems particularly puzzling to me. Therefore I would like to see more explanation and discussion in this part. A figure of the modeled (not cumulative) TTDs would also be interesting/helpful.

**- Reply:** The groundwater in Krycklan is relatively shallow, especially around the mires. The till has a marked decreasing hydraulic conductivity with depth. The main flow of water occurs therefore in the shallow part of the till, resulting in fast groundwater flow especially during snow melt in the spring. TTD:s is the travel time of the groundwater for the whole simulated period, and thereby the steep initial rise of the TTD:s are mainly connected to the spring snowmelt. This emphasize the importance of conducting thorough investigations of TTDs and MTTs in different climate regions. We assume that the "true" baseflow can be represented by the groundwater reaching the streams during the winter period since there is no new input of precipitation to the soil during this time. This period also has an older age (gMTT). We added a figure with monthly groundwater age distribution and the partition of groundwater and overland flow (Figure 5) and changed the TTD figure (Figure 4) to show the particle age distribution for some streams.

- The discussion and conclusion sections are straightforward and quite easy to comprehend. This may, unfortunately, be due to the fact that the results are not really new. Catchment size is not related to MTTs; hydraulic conductivity, slope and flow path length are related to MTTs. In the end I am left wondering whether there is enough novelty (maybe relating pH and base cations to MTTs?). A more in depth discussion of the modeled TTDs could help or maybe additional analyses on the interdependencies of how different catchment characteristics combined control MTTs. Still, I believe that in the end the authors will manage to add more analysis and discussion to justify a publication in HESS.

**- Reply:** We extended the evaluation regarding the TTDs and the chemistry data and gave more focus to the seasonal change of mean travel time for different sub-catchments. We also added a figure that show the monthly groundwater age distribution to the streams Figure 4. Additionally, we added a section in the introduction part that highlights the unique set of data and models available, which enables a coupled analysis of chemistry and hydrology in order to understand the complex integrated surface-subsurface water system of the Krycklan catchment. It should be noted that in contrast to many previous studies this study was conducted in a snow-dominated catchment in the boreal forest region, which implies that one could expect differences in the hydrology and transit time distributions compared to more temperate regions where the majority of previous work has been conducted. We believe that the extended winter season conditions allow great opportunities to address the role of the base flow, its TTDs and relationship to water isotopes and water chemistry. Furthermore, we must also emphasize the comprehensive background data that was used for the setup and testing of the results, e.g. continuous discharge measurements and measurements of stable water isotopes and stream chemistry at 14 streams. Combined this should strengthen the reliability of the modelled TTDs.

**Specific comments from Reviewer #1**

**-Reply:** We agree with most specific suggestions and comments from Reviewer #1. However, some sections of the introduction and discussion were re-written with the help of this input and some sentences might end up being different in the revised manuscript.

Abstract:

- Line 23: '...to investigate...'
- Line 23: Better write: '...the travel time of the input to 14 [...] sub-catchments via groundwater to the stream...'.
- Line 30: Move this sentence up to Line 27 just after '...stream water.'.
- Line 31: I would add whether these were positive or negative correlations.
- Line 34: I would not call this a 'landscape characteristic'. Maybe better call it 'physical catchment characteristic'.
- Line 35: '...to positively correlate with MTT.'

**- Reply:** Major adjustment to the Abstract have been made to fit the new introduction, result and discussion section. Thereby, the sentences that the Reviewer have commented on has been changed. We have tried to account for the suggestions in the new text.

Introduction:

- Line 50: These referenced papers deal mainly with mean travel times, not with travel time distributions. Appropriate references for variations in TTDs would be for example Botter et al. (2010), Heidbüchel et al. (2012), Hrachowitz et al., (2013). Line 87: Again, I would not use 'landscape' factors since the term landscape refers to the visible landforms rather than to physical properties or subsurface features.

**- Reply:** We like to thank the Reviewer fort the suggested references. We have read them and used them to extend the introduction.

- Introduction: This section lacks the mention/discussion of previous work that is very relevant to the study. Especially concerning research on the connection of travel time and catchment characteristics. Your hypothesis is that catchment size correlates with travel time, but what about any other catchment properties? So far travel times have been related to slope, flow path length, soil thickness, antecedent moisture content, hydraulic conductivity, just to name a few. I recommend expanding the short introduction a bit more to include a brief discussion of potential travel time controlling factors (Ameli et al., 2016; Birkel et al., 2016; Haitjema et al., 1995; Heidbüchel et al., 2013; Hrachowitz et al., 2010; McGuire et al., 2005; Yang et al., 2018).

**- Reply:** We appreciate the Reviewer's suggestion and used them to extend the introduction using the suggested material. We also focused more on the connection between stream chemistry compared to model results. The long winter with no input of precipitation to the groundwater makes it possible to distinguish the baseflow from the discharge affected by new input of water to the system. Krycklan has a significant amount of empirical data and long-term observations that provides a unique opportunity to analyze a northern snow-dominated catchment.

Methods:

- Line 106: '...type OF regolith...'?

**Reply:** The sentence: "Connected by a network of streams, the different sub-catchments have distinct characteristics, which allow for an evaluation of the effects of catchment characteristics on hydrologic transport, **including type regolith**, vegetation, and differences in topography." was changed to: "Connected by a network of streams, the different sub-catchments have distinct characteristics, which allow for an evaluation of the effects of catchment characteristics on hydrologic transport, **including soil type**, vegetation, and differences in topography."

- Line 123: I would like to see a figure of the stratigraphy and how it is displayed in the model.

**- Reply:** We changed Figure 1 to include a map over the typography and a soil map. The manuscript also includes a more descriptive text about the Mike SHE model in the method section.

- Line 172: Are these winter GW travel times travel times of particles that entered the catchment during the winter or exited the catchment during the winter (are they from 'forward' or from 'backward' TTDs)? This is a very important and interesting question since it could be very different catchment characteristics that control forward or backward travel times.

**- Reply:** We agree that this could be better described to make it clearer. It is the age of particles exiting the catchment through stream discharge during winter regardless when they entered the groundwater as recharge, i.e. it is backwards TTDs. We hope that this was made clearer in the new version of the manuscript. Note that practically all precipitation falls as snow during the winter so typically there is little or no addition of water to the system during this period. Therefore, there is no addition of particles to the model during the winter months. Consequently, we doubt that it would be meaningful to trace the small amounts of water that might be added during the winters in these systems.  Figure 3 was added to make the explanation of the particle tracking easier.

- Line 215: According to this equation a mire coverage of 100% would result in an infinitely large adjusted cation concentration which is quite unrealistic (this is equivalent to a scenario where mires do not contribute any cations whatsoever). How do you justify this relationship?

**- Reply:** The adjustment of the cation concentrations is based on observations of how mires affect the stream water concentrations of a wide range of elements. This has been described in detail by Lidman et al. (2014), which is cited in the manuscript. A general pattern was that the concentration of predominately weathering-derived elements such as the base cations decreased with the mire coverage. For the base cations, which do not sorb particularly strongly to peat, the decrease corresponded well with the decrease in mineral soils. This was therefore interpreted as dilution by mire water, since weathering does not occur in peat. For example, if 20% of the mineral soils in catchment was replaced by 20% peat, the weathering should be expected to decrease by approximately 20%, eventually causing approximately 20% lower concentrations in the runoff. (In practice, somewhat higher specific discharge from mires would increase the dilution effect, but then a slight addition of base cation with the precipitation would counteract this effect so the net effect is approximately 1% dilution by 1% wetland.) This is illustrated by Na in Fig. X with data from Lidman et al. (2014), where these matters are discussed in more detail for Na and other elements. We believe that this justifies the need to adjust the base cation concentrations for the effect of the mires in order to make a fair comparison to the modelled transit times, which include the mires.

In principle, however, the Reviewer is correct in stating that this adjustment would be problematic in cases where the mire coverage approaches 100%. One reason, as the Reviewer points out, is that the adjustment would lead to unrealistically high concentrations if the mire coverage were close to 100%. This is, however, only a theoretical problem since the mire coverage in the investigated sub-catchments never exceeds 44%. There is therefore no need to make such unrealistic and also unsupported adjustments. We find it hard even to envisage how a catchment with much higher mire coverage would be able to develop in this area, but since no such sub-catchments exist in the present study we believe that this discussion could be left for another context. Another reason why we would be reluctant to use this type of adjustment for such extreme cases where the mire coverage was close to 100% (had it for some reason been necessary) is that it would require extrapolation far outside the observed range. In this case the relationship to the concentration is significant between in the range 0-44% mire coverage. The observations were made in the same streams that were investigated in this study so there is clearly a justifiable relationship between base cation concentrations and mire coverage.

[Figure]

Figure X - Average Na concentrations in the investigated streams as a function of the mire coverage. Data from Lidman et al. (2014).

- Line 220: What exactly do you mean by this ('...less impacted...', '...also considered.')?

**- Reply:** What we meant to express was that there is a fundamental difference between the water isotopes, on the one hand, and the stream chemistry, on the other. The interpretation of the water isotopes is based on the characteristic variation in the input signal throughout the year. Consequently, the important parameter is the variation in the water isotope signature in the runoff, as given in the manuscript. The annual average, however, is not a meaningful parameter. In the case of pH and base cation concentrations, the major drivers are processes within the catchment, e.g. weathering rates and transit times, and it is therefore meaningful to look also at the annual averages in this case. We have made major changes in method section 2.3 to describe this better.

- Line 239: But the simulated specific baseflow is not included in Table 4.

**- Reply:** Information about the specific discharge was omitted from Table 4 at a late stage. It must however still have been left in the text and this was corrected in the revised manuscript.

- Line 243: Again it would be important to know whether these are forward or backward times.

**- Reply:** We agree with this comment and checked and adjusted the entire text accordingly. To answer the question, it is backwards times, i.e. the age of a particle when it reaches a stream and exits the model, regardless of when they were first introduced to the groundwater. Since the system is frozen during the winters, there is only negligible addition of water. Hence, it would not be very meaningful to study the forward times for the winter period. The section regarding particle tracking (section 2.4) has been edited and the Figure 2 is new and is used to help to describe the particle tracking part of the study.

Results:

- Line243: I would some how indicate that the means are geometric instead of arithmetic when you write MTT (maybe something like 'gMTT'?). Because one will automatically assume that MTT is the arithmetic mean of the travel times.

**- Reply:** Thank you for the comment, we adjusted this in the new version since it will improve the clarity of the text and we decided to use the suggested gMTT.

- Line 245: This is not a complete sentence (verb missing).
- Line 248: 'Over the course of a year...'; '...may enrich or dilute...'
- Line 259: Winter baseflow or winter groundwater fraction?
- Line272: Instead of 'decreased' I would rather write'...became more negative/became more enriched in the lighter isotope...'
- Line 277: '... averages...'
- Line 304: 'Percentage of low conductive sediments...'

**- Reply:** The result section now includes seasonal travel time results. This change also changed most of the text in this section.

- Line 292-294: It would be good to add 'positive' or 'negative' before 'correlations', so you immediately know the direction of the relationship.

**- Reply:** We agree that this would add to the manuscript and make it easier for the reader to directly understand the relationship without having to look at the figures. This have been changed in the new version.

- Line 297: How do these equations show that LCS is the most significant parameter?

**- Reply:** The Reviewer has a valid point here. It is hard to justify based only on the equations in questions that LCS is the single most important parameter for controlling the MTTs in the investigated catchment.

This should clearly be rephrased in a more stringent way. The message we were trying to convey is that it seems hard to get around the impact of the LCS when looking at the MTTs throughout the landscape. As in this example, it is difficult to replace LCS by any other parameter and obtain equally good correlations, which indicates that it probably plays an important role for controlling MTTs. However, this section has been removed in the new manuscript to give space to seasonal travel time results and comparisons to catchment characteristics.

Discussion

- Line 314: '...on the other...'?
- Line 318: '...and a gamma distribution transfer function (convolution) method...'
- Line 318: 'In a conceptual, non-distributed modeling study...'
- Line 319: Some more details would be helpful ('another travel time distribution technique').

**- Reply:** The discussion section now includes an uncertainty/limitation part, a discussion towards seasonal travel time and chemistry results, as well as a discussion towards seasonal travel times and catchment characteristics. These changes required major changes of the text in the discussion section We agree with the comments from the Reviewer and had them in mind when writing the revised result section.

- Line 361: This section is a bit unstructured. You start out by stating that slope and hydraulic conductivity are the main factors controlling MTTs, then you state that the fraction of LCS is the most important factor adding travel distance to the mix before arguing that the steeper small C20 behaves differently maybe also because of the fluvial deposits fraction... Since these are some of your main findings it would be good to clarify the section. Also, what about the fact that you released the particles at the top of the transient groundwater table? That means that depending on the groundwater table a larger or smaller part of the regolith was not taken into account for the MTT calculations. If the water table was high a larger proportion of the regolith was considered, if it was low the particles started somewhere else in the profile. What are the implications of this? Would that influence your results?

**- Reply:** Please check the Figure 1, Figure 2, method section 2.1 and the new discussion 4.1 in the new manuscript. Most of the calculated groundwater table has a level between 0-3 m below the ground. The vertical discretization of the saturated zone follows the soil layers and are a few meters in depth close to the surface and increasing in thickness with depth. Note that the horizontal grid-size is 50*50 m and therefore the MTT will be bias to long travelling groundwater. Peatlands are generally the areas with an average groundwater table above 1 m, while the till areas have a depth between 2-3 m in average. However, at C14, the deep esker has a lower water table than on other locations in the catchments. The esker has the highest vertical hydraulic conductivity, and thereby the fastest route from infiltration to recharge. Although the horizontal distance is still larger than the distance to the groundwater table and the vertical hydraulic conductivity is large, one should note that this could impact the results and give C14 and C16 older MTT than if the phreatic surface was at the same level around the whole catchment. It can be noted that the areas were the MTTs mainly could be suppressed by this limitation in the modelling tools are areas with relatively long MTTs. This effect therefore somewhat counteracts the general patterns that were observed in this study. Hence, the observed patterns are not a result of the differences in groundwater depth. On the contrary, we would have expected patterns that are even more distinct if it had been possible to account for the time it takes for the water to percolate to the groundwater surface. We added a discussion about this (section 4.1), since it is an interesting subject and relevant for the interpretation of the results and try to make the section more structured.

- **Line 380:** It would be good to point out more the novel aspects of your work. What did you find out that was really new? The fact that Darcy's law works? Catchment size has been ruled out as a MTT control since quite a while now. Or is it mainly a study that confirms the applicability of the particle tracking of your model by comparing results to time series of pH, base cations and stable isotopes?

**- Reply:** Reviewer #2 also indicated that we need to highlight more clearly the novelty of this study so this is one of the major concerns we have tried to address in the revised manuscript. The original manuscript was organized around the hypothesis that MTTs are controlled by catchment size, but based on the comments of both Reviewers and the available literature, we agree that the results of this study could be presented in a better way by omitting this hypothesis as a rationale for the study and instead focus on other more novel aspects of this work. This includes emphasizing the implication of the northern landscape setting where the hydrology is dominated by prolonged winter conditions, and a dominating spring flood. Krycklan is one of the most well-investigated and well-instrumented catchments in the boreal region with a large database of empirical observations that allow multiple ways to test the predictions of hydrological models.

In our opinion, the discussion about MTTs, TTDs and their variation at different scales in the landscape is far from finished, and we believe that this study could contribute significantly to the understanding of the hydrology of the snow-dominated boreal forest landscape, the coupling between hydrology and water quality and, not least, the role of winter base flow in a changing climate. It is far from obvious that MTTs and TTDs would look the same and be controlled by the same factors in different types of landscapes and climates, so we believe that this type of studies from different areas of the world are necessary to improve our understanding of these matters. We further believe that the highlighting of the role of seasonal travel times in the revised version of the manuscript, is particularly suitable for the boreal landscape due to its pronounced seasonality. We also tried to emphasize better the large amount of data that we used to set up and test the model, e.g. the continuous discharge measurements in the streams and the extensive monitoring of water isotopes and stream chemistry. Combined we believe that this background data significantly strengthens the credibility of the conclusions we present.

Figure specific comments:
- Figure 1: Why do you call the sub-catchments subareas in this figure? Also, all subcatchments (not only the green ones) connect before reaching the main outlet at C16.

**- Reply:** We changed sub-areas to sub-catchments in Figure 1. Yes, all streams connect before they reach C16. However, the color code was used to illustrate which areas connect before they reach the white area. We tried to clarify this in the figure text.

- Figure 2: I am confused by the y axis label. Why does it start with 100% and then decrease to 0%? Isn't this a cumulative TTD (that should start with 0%)? Also, you never mention how you construct this TTD. Do you record the time the particles that arrive in the stream in the winter needed to travel through the catchment (backward TTD) or do you record the time that particles that are released during winter need to travel through the catchment (forward TTD)? I am curious since the cumulative TTDs exhibit a shape I would not have expected since the initial rise is very steep although you use a logarithmic x axis. For a groundwater TTDs in particular, this extremely high initial values are quite unusual. How do you explain this?

**- Reply:** Thank you for pointing this out! There was an error in this figure, the number of particles should start at 0 %, and increase to 100 %, so the y-axis should change, see Figure 4 in the new manuscript. The

figure now shows the annual age distribution of particles reaching the streams of four example sub-catchments. The snowmelt affects the groundwater travel time age and therefore there is a rapid start (Table 5). Given that a substantial portion of the annual precipitation leaves the catchments during a few weeks of spring flood, we do not think that the initial steepness of the distribution is surprising. This is probably one of the characteristics of boreal systems compared to other types of systems and emphasizes the importance of conducting this type of studies in different climate regions.

- Figure3: So if youre place the geometric MTT with thec ommonly used arithmetic MTT, how does that change the correlations? In case they become weaker you could argue for the general use of the geometric MTT.

**- Reply:** The arithmetic MTT is often used when looking at the gamma convolution distributions using isotopic data. It has been argued that the MTT should not be used for MTT older than 5 years because the amplitude signals are lost (Kirchner, 2016). Therefore, the aMTT is less impacted by the "old tail". For particle tracking or modelling methods the median or the geometric MTT may describe the bulk of the particles better than aMTT, because one particle (even though there are hundreds of particles in total) with a very old age may cause significant differences in the MTT estimation. This is undesirable, since the age of these very old particles becomes increasingly uncertain and are very hard to validate. The gMTT is less dependent on a few old particles but does still take them into account. The aMTT and gMTT are also well correlated and would probably give similar correlation results (Fig. Y).

[Figure]

**Figure Y** – Correlation between gMTT (geometric mean) and aMTT (arithmetic mean) of particle tracking results

Kirchner, J. W.: Aggregation in environmental systems – Part 1: Seasonal tracer cycles quantify young water fractions, but not mean transit times, in spatially heterogeneous catchments, Hydrol. Earth Syst. Sci., 20, 279–297, https://doi.org/10.5194/hess-20-279-2016, 2016.

Supplements:
- Supplements S1: Check again for typos and misuse of words throughout the supplement (in particular in the table captions and the table foot notes).

S1, line 3: What does that mean that the table includes both non-transformed and log-normal transformed data? Which is which?

- S1, line 4: What happened to the other 44% of particles?
- S1, line 8: 'Artesian' mean? And why 'back-transformed'?
- S1, line 19: '...as well as the precipitation (P)'
- S2, line 3: What does 'The statistics are based on...' mean?
- S2: The tables are not that helpful. Many of the abbreviations are not explained

**- Reply:** The appendix has been completely changed to fit with the new manuscript regarding seasonal changes. It now includes a description of the statistics used and an extended version of Figure 5.

**Add to literature:**

- Literature Ameli, A. A., Amvrosiadi, N., Grabs, T., Laudon, H., Creed, I. F., McDonnell, J. J., and Bishop, K.: Hillslope permeability architecture controls on subsurface transit time distribution and flow paths, J. Hydrol., 543, 17–30, https://doi.org/10.1016/j.jhydrol.2016.04.071, 2016.
- Birkel, C., Geris, J., Molina, M. J., Mendez, C., Arce, R., Dick, J., et al.: Hydroclimatic controls on non-stationary stream water ages in humid tropical catchments, J. Hydrol., 542, 231–240, https://doi.org/10.1016/j.jhydrol.2016.09.006, 2016.
- Botter, G., Bertuzzo, E., and Rinaldo, A.: Transport in the hydrologic response: Travel time distributions, soil moisture dynamics, and the old water paradox, Water Resour. Res., 46, https://doi.org/10.1029/2009WR008371, 2010.
- Heidbüchel, I., Troch, P. A., Lyon, S. W., and Weiler, M.: The master transit time distribution of variable flow systems, Water Resour. Res., 48, https://doi.org/10.1029/2011WR011293, 2012.
- Heidbüchel, I., Troch, P. A., and Lyon, S. W.: Separating physical and meteorological controls of variable transit times in zero-order catchments, Water Resour. Res., 49, 7644–7657, https://doi.org/10.1002/2012WR013149, 2013.
- Hrachowitz, M., Soulsby, C., Tetzlaff, D., Malcolm, I. A., and Schoups, G.: Gamma distribution models for transit time estimation in catchments: Physical interpretation of parameters and implications for time-variant transit time assessment, Water Resour. Res., https://doi.org/10.1029/2010WR009148, 2010.
- Hrachowitz, M., Savenije, H., Bogaard, T. A., Tetzlaff, D., and Soulsby, C.: What can flux tracking teach us about water age distribution patterns and their temporal dynamics?, Hydrol. Earth Syst. Sci., 17, 533–564, https://doi.org/10.5194/hess-17-533-2013, 2013.
- McGuire, K. J., McDonnell, J. J.,Weiler, M., Kendall, C., McGlynn, B. L., Welker, J. M., and Seibert, J.: The role of topography on catchment-scale water residence time, Water Resour. Res., 41, https://doi.org/10.1029/2004WR003657, 2005.
- Yang, J., Heidbüchel, I., Musolff, A., Reinstorf, F., and Fleckenstein, J. H.: Exploring the dynamics of transit times and subsurface mixing in a small agricultural catchment, Water Resour. Res., https://doi.org/10.1002/2017WR021896, 2018.

**- Reply:** We thank the Reviewer for these literature suggestions which was used to especially improve the introduction section. We read these suggestions and incorporated them were they seemed to be most fitting.

**Response to Reviewer #2**

**Introduction comment from Reviewer #2**

- In the manuscript "Linking groundwater travel times to stream chemistry, isotopic composition and catchment characteristics," Sterte et al. analyze the drivers of catchment travel times across catchments in norther Sweden. They use a physical hydrology model combined with particle tracking to generate transit time distributions and compare this to isotopic and stream chemistry observations. Overall, I think that the study is well done and the manuscript is well written and easy to follow.

**Reply:** We would like to thank Reviewer #2 for his/her constructive criticism and suggestions, which we helped to improve the manuscript. The main concern seemed to be the introduction and hypothesis, which we, having read the Reviewer's comments, agree benefited from being re-worked to progress the manuscript further. Our explanations and responses to all the Reviewer's comments and questions are listed below (red).

- However, I do have some significant concerns about the manuscript in its present form. My most serious concern is that it's not clear what the novelty of this study is that would warrant publication in HESS. All of the methods used here are well established and the idea that catchment travel time distributions are driven by catchment characteristics is not new. The authors start from the hypothesis that catchment area is the main driver, however previous research has already indicated that many drivers will be important, so disproving this hypothesis does not seem to be the best angle to take here. I would suggest the authors consider what portions of their findings are the most novel addition to the body of literature in this area and organize the manuscript around this rather than the area hypothesis.

**Reply:** Thank you for this comment, it helped to make a better manuscript. We received a similar comment from Reviewer #1. Based on these comments, we re-focused the manuscript, and in the new version, better highlight the novel aspects of this study. This includes emphasizing the implication of processes in the northern landscape setting where the hydrology is dominated by prolonged winter conditions followed by snowmelt. We do that by studying one of the most well-investigated and well-instrumented catchment systems in the boreal region. Both Reviewers have pointed out that the hypothesis that the catchment area is the main factor controlling the travel time distributions is obsolete and that the manuscript, therefore, should not be organized around this idea. Having considered the remarks of the Reviewers and studied the literature thoroughly, we feel inclined to agree that this was a mistake. We believe that the manuscript has benefited from the revision, where this idea does not have such a central place. As Reviewer #2 remarks below, this should not be the storyline of the manuscript, particularly as neither the modelling results nor the observations supported the area hypothesis.

- Even if the area hypothesis is what guided the study in the first place, this does not need to be the storyline of the manuscript. Along the same lines as my first comment I think the introduction could use significant revision. As it stands it is a very broad overview of the topic but I would like to see a more thorough review of previous finding directly relevant to this work that can clearly motivate the novelty of this study and the gap it is filling. Similarly I think the discussion section would be more powerful if it provides a better evaluation of how and where results form this study add new information/disagree or provide additional corroboration to existing studies.

**Reply:** Thank you for your comment. We have expanded the introduction to include a section regarding relevant works related to this study. We received some very helpful reference suggestions from Reviewer #1, which were incorporated in the introduction with regards to relevant research. We also agree that the

catchment size hypothesis used in the original version of the manuscript was not the best way to present the results of this study, and this hypothesis has therefore been omitted. Both the introduction and the discussion were changed accordingly. We focused the study on the connection between seasonal travel times and stream chemistry (base cations and isotopes). There is a unique opportunity to distinguish the baseflow in the streams due to the prolonged winter, which gives little to no new input of water to the system and then comparing it to spring (high flow) and summer. The boreal systems are sensitive to climate change, and to have as much of a base understanding of these systems as possible is important to be able to evaluate changes in the future. Shorter snow seasons and changes in the amount of precipitation can change travel times in the future, which can have an impact on weathering and biological processes. We have in this study showed a strong relationship between isotope mixing and base cation concentrations, on the one hand, and groundwater travel times on the other. We have already seen changes in the climate and in the hydrology in the last couple of years in Krycklan, making this a pressing issue.

- For the most part I think the paper is very clearly written, however the description of the modeling approach is a bit confusing and could use some more details. For example the term simulation is used to refer to both the hydrology model and the particle tracking portion which can be confusing. This section could be helped by a figure or a schematic to illustrate the approach I think.

**Reply:** Reviewer #1 also requested a schematic figure, and we agree with the Reviewers and like the suggestion to better describe the modeling procedure (Figure 3). We aimed to give more distinctive terms to the hydrology modelling part and the particle tracking part of the manuscript. One reason for the confusion is probably that much of the hydrological model was described in a previous paper. Although this paper was cited, we realized that we need to explain the model setup in more detail also in this manuscript so that potential readers can grasp the general idea of what has been done without reading the previous paper (see method section 2.2).

**Specific comments from Reviewer #2**
**Reply:** We agree with the most specific suggestions and comments from Reviewer #2. However, some sections of the introduction and discussion will be re-written whit the help of this input, and some sentences might end up being different in the revised manuscript.

- I think the catchment numbering could be done in a more intuitive way so its easier to separate unique outlets (i.e C12-15). I would suggest giving each of these outlets their own letter and then numbering points within them potentially by drainage area, that way itis easier to compare when points are within the same drainage or not.

**Reply:** We understand the concern regarding this comment. However, we hesitate to change the names of the sub-catchment since they all have been included in many previous peer-reviewed papers, which argues against introducing new names in this manuscript. It would make all comparisons with previous papers from the Krycklan catchment unnecessarily difficult and confusing. We hoped that the colors of figure 1 will help to distinguish which sub-catchments were connected. However, we also added a color code in table 1, which will further connect the different sub-catchments in figure and text.

- Line 138-140: This is confusing are you trying to say that the hydrologic model is run first and then the particle tracking is applied to the outputs of that model?

**Reply:** Yes, the model flow field is run first and, then one year of the flow field is cycled for 1000 years with particle tracking applied to the flow field results. We will add the schematic figure (see Figure 3) to

clarify these steps and be more direct that the hydrology model output used are the results from the model is used in Jutebring et al. (2018). We will also add a more in-depth description of the water movement model in the revised manuscript (method section 2.2.).

- Section 2.3 – this is really more a description of scenario design than numerical methods. I would consider renaming.

**Reply:** We appreciate this suggestion from the Reviewer and will incorporate the suggestion and change the name from "Numerical methods" to "Establishing travel times - Particle tracking"

- Line 148: 'several years' is very vague can you be more precise.

**Reply:** We agree with the Reviewer and have tried to be more precise in the revised manuscript. It is one year of the Mike SHE flow results that has been cycled for 1000 years in order to allow for long-term particle tracking. This has been clarified this throughout the manuscript.

- Line 154: at what time frequency were particles injected into the model? Just at the start of each year?

**Reply:** This has been clarified in section 2.4 and the new Figure 3. The particles are only introduced in the first year. The Mike SHE flow results of that year are then cycled 1000 times to let the particles travel for a longer period (1000 years).

- Line 159: I think the more standard reference for this would be heavy tailed rather than long tailed.

**Reply:** A distribution with a long tail means that there is a large proportion of the distributions is far away from the main central tendency. A long-tailed distribution can also be heavy tailed, meaning that a distribution goes towards zero slower than an exponential distribution. In the section in question, we are talking about the use of the geometric mean instead of the arithmetic mean for long-tailed distributions because the geometric mean is less bias to the long tail and describes the central tendency more effectively.

- The term simulation is used to refer to both the hydrology model and the particle tracking model and this can make the methods confusing when you are talking about run times for example.

**Reply:** We looked through the manuscript and try to be more distinct when using the term simulation so that it becomes clearer when we talk about the flow model or the particle tracking. We also made a distinction in the method section were we first talk about the site (section 2.1), then the hydrology model (section 2.2) and thereafter the particle tracking (section 2.4)

- At the beginning of section 2.3 you say that you used several years of simulation but actually it looks like you use only one year of the hydrologic simulation but repeated it 1000 times (i.e. more than several). This description is confusing.

**Reply:** We have taken one year and repeated it 1000 times. We tried to be more distinct and not use the word several for better clarity in the revised manuscript.

- I think some of the tables could be converted to figures to better present the information. For example Table 4 could be presented as a series of maps.

**Reply:** We do not understand exactly what the Reviewer is asking for. Most of the information in the old manuscript Table 4 was shown in Figure 2. We attempted to make maps at the onset of this study, but the information was not clearly visible. However, we added travel time distribution per month for each sub-catchment to visualize the impact of various fractions of travel times. See new figure 5 and Appendix.

**Linking  travel times and flow pathways to stream chemistry, isotopic composition, and catchment characteristics in a snow-dominated landscape**

**Elin Jutebring Sterte[1,2], Fredrik Lidman[1], Emma Lindborg[2], Ylva Sjöberg[3], Hjalmar Laudon[1]**

[1]Department of Forest Ecology and Management, Swedish University of Agricultural Sciences, SE-901 83 Umeå, SWEDEN

[2]DHI Sweden AB, Svartmangatan 18, SE- 111 29 Stockholm, SWEDEN

[3]Center for Permafrost (CENPERM), Department of Geosciences and Natural Resource Management, University of Copenhagen, Øster Voldgade 10, 1350 Copenhagen, Denmark

Corresponding author: Elin Jutebring Sterte (eljs@dhigroup.com)

**Key Points:**

- A numerical model was used to estimate annual and seasonal mean travel times and flow pathways across the Krycklan catchment in the boreal region of northern Sweden.
- The estimated annual mean travel times of  14 partly nested sub-catchments ranged between 0.8 and 2.7 years
- The estimated travel times and young water fractions were consistent with  observed stream chemistry (base cation concentration and  stable water isotopes  $\delta^{18}$O .
- Hydraulic conductivity of the soil was the most important factor regulating the variation in mean travel times , while mires mainly affected the youngest fraction of stream water.
- Although all sub-catchments showed seasonal changes in mean travel times and young water fractions, the greatest seasonality was found in sub-catchments with a substantial fraction of mires.

**Abstract**

Understanding travel times  of rain and snowmelt water transported through the  landscape to recipient surface waters is critical in many hydrological and biogeochemical investigations. In this study, a particle tracking model approach in Mike SHE was used to investigate the travel time and pathway of water in 14 partly nested, long-term monitored boreal sub-catchments characterized by long pH snow rich winters with little groundwater recharge and highly dynamic hydrology during the following snowmelt. The calculated annual mean travel times these sub-catchments varied from 0.8-2.7 years. The seasonality caused considerable variation in travel times between different seasons and landscape types, with winter meanranging from 1.2-7.7 years spring mean travel times ranging from 0.5-1.9 years. The modelled variation in annual and seasonal travel times and the fraction of young water (less than three months) was supported by extensive observations of both $\delta^{18}$O and base cation concentrations in the stream water.  The age of the groundwater was positively correlated to the abundance of low conductive soils (r=0.90, P<0.0001). As a result of lacking synchronicity and contrasting hydrological responses between different soil types (e.g., peat and low-conductive soils), mixed catchments typically displayed larger differences in travel times between winter baseflow and spring flood. Mires were found to affect

the young water fractions of the stream contribution (r=0.96, P<0.0001) by introducing larger differences in the mean travel times between the seasons compared to forest dominated sub-catchments. The main factor for this difference is likely related to the soil frost in mires, causing considerable overland flow in spring. The lower recharge during these periods caused mire-dominated catchments to have older stream water contribution than comparable forest catchments during other parts of the year. Boreal landscapes are sensitive to climate change, and our results show that changes in seasonality are likely to affect different landscape types in different ways.

**1 Introduction**

The age and pathways of  water through the terrestrial landscape to streams is a widely discussed topic in contemporary hydrology. This interest has emerged because of the important role residence time, and routing of water through various subsurface environments have on hydrological and biogeochemical processes (McDonnell et al., 2010; Sprenger et al., 2018). These include fundamental implications for weathering rates (Burns et al., 2003), transport and dispersal of contaminants (Bosson et al., 2013; Kralik, 2015), and accumulation and mobilization of organic carbon and associated solutes (Tiwari et al., 2017). The ~~et al., 2013). In the vast boreal region, the landscape often consists of heterogeneous patches of lakes, mires, and coniferous forests regulated by sometimes contrasting hydrologic mechanisms. This heterogeneity emphasizes the need for an enhanced understanding of hydrological and biogeochemical processes and their inter-linkage (Winnick et al., 2017; Waddington et al., 2015; Demers et al., 2010).to,forinput~~fate of long-range transport of contaminants and nutrients deposited with precipitation (van der Velde et al., 2012). The travel time distribution can vary substantially in time and space, depending on catchment characteristics and hydrological conditions, including, for example, slope, catchment size, soil heterogeneity, and seasonality (Botter et al., 2010; Lin., 2010; Heidbüchel et al., 2012; Hrachowitz et al., 2013). ~~numerous catchment characteristics and hydrological conditions (McGuire and McDonnell, 2006; Scanlon et al., 2001). Therefore, estimating travel times for various contrasting landscape elements may enhance our process understanding and ability to more correctly quantify the contribution of water and various solutes derived from catchments. Groundwater is an especially important component of the hydrology and a regulator of many biogeochemical processes. From a surface water perspective, groundwater is a source of recharge for streams, lakes and, wetlands. Groundwater is the part of stream water contribution which is not linked to a specific hydrological episode and can, therefore, be used as a reference point to study solute transport, water quality and other event-activated processes (Bergknut et al., 2010; Doyle et al., 2005; Olson & Hawkins, 2012; Pinder & Jones, 1969). Northern landscapes with long-lasting snow cover and prolonged frozen conditions without the input of new surface water can give a unique opportunity to investigate stream groundwater input conditions (Peralta-Tapia et al., 2015).~~Therefore, estimating travel times for contrasting landscape elements is a challenging task but may enhance our ability to understand catchment functioning more adequately.

Stream water consists of a blend of groundwater and overland flow of different ages. The mean travel time (MTT) to streams is calculated as the average age of this mix (McGuire et al., 2006). The baseflow is the part of stream groundwater contribution that is not linked to a specific hydrological episode and instead part of the runoff mix that generally has travelled the farthest and is the oldest (Klaus et al., 2013; Hrachowitz et al., 2016). In contrast, young waters are connected to overland flow or fast and shallow groundwater, which mainly can be seen at times when new precipitation input and/or snowmelt arrives (Peters et al., 2014; Hrachowitz et al., 2016). Travel times are complex to quantify, especially at intra-annual time scales, as they can vary in time and space (Botter et al., 2010). A better

understanding of the seasonal variability in the connection between young and old waters in various catchment systems can help provide insights into the fundamental role catchment characteristics play for the regulation of the hydrology and biogeochemistry of streams and rivers.

Stable water isotopes and biogeochemical tracers are some of the most common tools applied in field investigations to locate sources of water and follow its pathways through the landscape ( Maulé and Stein, 1990; Rodhe et al., 1996; Goller et al., 2005; Tetzlaff and Soulsby, 2008). Isotopic tracer dampening can provide an estimate of mean travel times (Uhlenbrook et al., 2002; McGuire et al., 2005; Peralta-Tapia et al., 2016), and more elaborate time-series analysis can provide quantitative assessments of water age (Harman, 2015; Danesh-Yazdi et al., 2016). Theoretical transfer functions, such as the gamma distribution model, can be related to input-output signals (for example, precipitation to discharge) of isotopes (McGuire et al., 2005; Hrachowitz et al., 2010; Heidbüchel et al., 2013; Birkel et al., 2016). The isotope amplitude signal used to estimate mean travel times in many transfer functions is, however, lost after approximately four to five years (Kirchner., 2016), which limits the use of isotopes for catchments with long travel times. The fraction of young water, often defined as water younger than two to three months, can, however, still be quantifiable in such catchments (von Freyberg et al., 2018; Lutz et al., 2018; Stockinger et al., 2019). Water isotopes mainly fractionate due to evaporation and are hence not affected by their subsurface pathways. In contrast, biogeochemical tracers may react and transform on their route to a stream ( Lidman et al., 2017; Ledesma et al., 2018). Such transformations and reactions depend on the specific solute and soil environment that water encounters and can hence give qualitative information about  groundwater flow pathways in mineral soils (Wolock et al., 1997; Frisbee et al., 2011;  Zimmer et al., 2012). Therefore, combined information from conservative and reactive tracers can provide an enhanced understanding of hydrological and biogeochemical processes  (Laudon et al. 2011).

A complementary approach to field experiments is numerical modelling, which can be useful for achieving a system understanding of catchment hydrology. Lumped hydrological models often describe catchments as single integrated entities. In contrast, distributed numerical models can include spatial heterogeneity in input parameters and therefore have the potential to represent catchment processes  more realistically. In turn, this can lead to a more process-based understanding of hydrology and biogeochemistry at the catchment-scale (Brirhet and Benaabidate, 2016; Soltani, 2017). A common method to calculate travel times using numerical methods includes isotope models and particle tracking (Hrachowitz et al., 2013; Ameli et al., 2016; Kaandorp et al., 2018, Yang et al., 2018). Models, however, need – as far as possible – proper tests against real observations to build confidence in their outputs. Stream discharge, groundwater, and tracer data are examples of such validation data that can provide important information to understand a catchment hydrological functioning (McGuire et al., 2007; Hrachowitz et al., 2015; Wang et al., 2017). Such empirical data are costly and time-consuming to collect. Therefore, data for calibration and

validation is often limited, and the minimum length of data sets and methods needed in data-sparse catchments is currently a topic for some debate (Bjerklie et al., 2003; Jian et al., 2017; Li et al., 2018).

In this study, advective travel times groundwater input

Snow-dominated landscapes have gained increasing attention due to streams were investigated in their importance as water resources (Barnett et al., 2005) and their vulnerability to climate change the well-studied Krycklan last decades (Tremblay et al., 2011; Aubin et al., 2018). One snow-dominated catchment with long continuous data sets for multiple monitoring stations in the boreal region catchment Krycklan in northern Sweden (Laudon et al.,., 2013). These data sets give a unique opportunity for investigation of the hydrological functioning of the heterogeneous boreal landscape. Boreal catchments with long-lasting snow cover that melts rapidly in the spring create both opportunities and challenges in the context of the determination of age and pathways of stream water. The boreal region consists of heterogeneous patches of lakes, mires, and mostly coniferous forests regulated by sometimes contrasting hydrologic mechanisms. This heterogeneity emphasizes the need for an enhanced understanding of hydrological and biogeochemical processes and their inter-linkage in these systems (Temnerud and Bishop, 2005). In high latitude snow-dominated catchments, little to no new input of water occurs to the soil during the several months of winter conditions, whereby the source of water to the streams is originating from baseflow (Peralta-Tapia et al. 2016).) using a These specific conditions provide unique opportunities to study the source of water that have spent the longest time in the sub-subsurface environment.

In contrast to the conditions of winter baseflow, significant amounts of water are added to the system during the often short and intensive spring snowmelt period (Spence et al., 2011; Spence and Phillips, 2015; Lyon et al., 2018). Although attempts to assess travel times generally have shown good fits for a gamma distribution function in snow-dominated catchments, particularly the winter season has proven to be challenging, which suggests that other methods to assess travel times may be required (Heidbüchel et al., 2012; Peralta-Tapia et al., 2016). To account for the unique circumstances of both baseflow with long transit times and those of the intensive spring snowmelt with potential large overland flow components requires the need of models that can handle the complexity and separation of various flow components.

In order to overcome previous model limitations, this study used particle tracking in the physically based distributed numerical model, Mike SHE (Graham and Butts, 2005). The model), with the purpose of enhancing our understanding of stream water contribution in boreal landscapes across seasons and landscape configurations. The water moment model in Mike SHE calculates saturated (3D) groundwater flow and unsaturated (1D) groundwater flow and is fully integrated with the surface water as well as evapotranspiration. The water movement model setup and results previously presented by Jutebring Sterte et al. (2018) was were used and as a platform in the present study. The model has been calibrated and validated to observed 14 sub-catchment of daily stream-discharge and observations and 15 groundwater wells of periodically measured groundwater levels throughout the Krycklan catchment. (Laudon et al. 2013; Jutebring Sterte et al., 2018). The complexity of the model allows for an in-depth investigation of advective travel times by non-reactive particle tracking simulations in a transient flow field.

Based on previous work in Krycklan (Peralta-Tapia et al., 2015), we hypothesize that

The main objective of this study was to quantify seasonal age distributions and calculate mean travel times of stream water contribution of the sub-catchment  in Krycklan in order to disentangle how these are related to physical landscape characteristics and seasonality. Firstly, the credibility of the model results was tested by comparing calculated travel times for the 14 long-term monitored sub-catchments to ten-year seasonal isotope signatures,  and base cation concentrations record from the Krycklan network. The usefulness of stream isotopic composition and chemistry has previously been demonstrated for understanding the connection of hydrological flow pathways and travel times for this site (Laudon et al., 2007; Peralta-Tapia et al., 2015), but with the limitation of studies on only short periods or single catchments. Secondly, the purpose was to go beyond what has previously been done by identifying the connection between travel times and different catchment characteristics and test how this varies between different hydrological conditions and seasons. This was accomplished by capturing contrasting seasons such as the low flow conditions in winter with limited input of new precipitation, high flow in spring when the system still is partly frozen, and summer when evaporation becomes a significant process.

**2 Method**

**2.1 Site description**

The Krycklan study catchment, located in the boreal region at the transition of the temperate/subarctic climate zone of northern Sweden, is spanning elevations from 114 to 405 m.a.s.l (Fig. 1, Table 1). The characteristic features of this boreal landscape are the dominance of Scots pine (*Pinus sylvestris*) and Norway spruce (*Picea abies*) covering most of the catchment (Laudon et al., 2013). Krycklan has a landscape distinctively formed by the last ice age (Ivarsson and Johnsson, 1988; Lidman et al., 2016). At the higher altitudes to the northwest, which are located above the highest postglacial coastline, the till soils can reach up to 15-20 m in thickness  Here, the soil primarily consists of sandy-silty till, and the landscape is intertwined with lakes and peatlands. In this study, we refer to soils as all unconsolidated material above the bedrock. The decreasing hydraulic conductivity with depth is characteristic for glacial tills in northern Sweden (Bishop et al., 2011; Seibert et al., 2009) with conductivities close to $5\cdot10^{-5}$ ms$^{-1}$ at the ground surface and exponentially decreasing downwards (Nyberg, 1995).  At lower altitudes, the soils mainly consist of fluvial deposits of silty clay and sand. Compared to the soils at higher altitudes, these deposits can reach thicknesses up to approximately 40-50 m  and are more homogeneous with depth.

The catchment is divided into 14 nested sub-catchments and has been included multi-disciplinary biogeochemical and hydrological research for more than 30 years (Laudon and Sponseller, 2018. The sub-catchments are called C1 to C20. The longest continuously monitored time-series  streamflow began at the beginning of the 1980s. Connected by a network of streams, the different sub-catchments have distinct characteristics, which allow for an evaluation of the effects of catchment characteristics on hydrologic transport, including soil type , vegetation, and differences in topography.

**Table 1: Sub-catchment characteristics.** The list includes all 14 monitored sub-catchments in Krycklan, called C1 to C20. Different branches of the stream network are illustrated in different colours in Fig. 1. The table includes sub-catchment area, average elevation, and average slope. Further description of these characteristics can be found in Karlsen et al. (2016). The table also includes soil proportion based on the soil.

|  | Catchment size (km²) | Average elevation (m.a.s.l.) | Slope (°) | Till (%) | Mire (%) | Sorted sediments (%) | Lakes (%) |
|---|---|---|---|---|---|---|---|
| C1 | 0.48 | 279 | 4.87 | 91 | 0 | 0 | 0.0 |
| C2 | 0.12 | 273 | 4.75 | 79 | 0 | 0 | 0.0 |
| C4 | 0.18 | 287 | 4.24 | 29 | 42 | 0 | 0.0 |
| C5 | 0.65 | 292 | 2.91 | 47 | 46 | 0 | 6.4 |
| C6 | 1.10 | 283 | 4.53 | 51 | 29 | 0 | 3.8 |

| | | | | | | | |
|---|---|---|---|---|---|---|---|
| **C7** | 0.47 | 275 | 4.98 | 68 | 16 | 0 | 0.0 |
| **C9** | 2.88 | 251 | 4.25 | 64 | 14 | 11 (4) | 1.5 |
| **C10** | 3.36 | 296 | 5.11 | 64 | 28 | 1 | 0.0 |
| **C12** | 5.44 | 277 | 4.90 | 70 | 18 | 6 | 0.0 |
| **C13** | 7.00 | 251 | 4.52 | 60 | 10 | 18 (9) | 0.7 |
| **C14** | 14.10 | 228 | 6.35 | 46 | 6 | 39 (15) | 0.7 |
| **C15** | 19.13 | 277 | 6.38 | 64 | 15 | 10 (2) | 2.4 |
| **C16** | 67.90 | 239 | 6.35 | 51 | 9 | 31 (10) | 1.0 |
| **C20** | 1.45 | 214 | 5.96 | 55 | 9 | 28 (28) | 0.0 |

Commented [EJS1]: Edited

[Figure]

**Commented [EJS2]:** Edited

**Figure 1:** **The Krycklan catchment.** (a) Location of sub-catchment and  their outlets . The areas are color-coded based on their stream network connections, e.g., all  sub-catchments of one colour connect before reaching the white area. For further details of the catchment characteristics, see Table 1.  (b) The soil map used in the hydrology model is based on the  soil map (1:100,000) from the Swedish Geological Survey (2014), combined with field investigations. (c) Depth to bedrock from the Swedish Geological Survey (2014) is shown in meter below the ground surface. (e) Catchment topography, shown as meter above sea level (m.a.s.l.).

**2.2 Water flow model setup**

**The – Mike SHE**

The Mike SHE model setup used as a platform in this study was a slightly updatedmodified version of the previously established and validated surface and groundwater model presented byin Jutebring Sterte et al. (2018). The model has a horizontal grid resolution of 50*50 m. Vertically the model is divided into ten calculation layers and reaches a depth of 100 m below ground. The calculation layers follow the regolith stratigraphy of the soil with one exception: the uppermost layer thickness was set to 2.5 m. (Fig. 1, Table 2). This exception was due to the numerical implementation of the unsaturated zone and the evapotranspiration processes in Mike SHE, which only are fully active in the uppermost calculation layer. Therefore, the uppermost layer has tomust be deep enough to cover the part of the regolithsoils influenced by evapotranspiration processes and capillary rise of groundwater. This depth averages several regolithsoils types into one calculation layer, which may underestimate the observed high horizontal hydrological conductivity in the shallowest parts of the till (Peralta-Tapia et al., 2015). Numerically this is accounted for by implementing a depth-dependent drainage function, which increases the flowgroundwater velocity in the shallowestuppermost part of the tillsoil (Bosson et al., 2008). For more information regarding the model-setup, see Jutebring Sterte et al. (2018). For this study, a few changes were made to the original Krycklan Mike SHE model. Most importantly, new field data from the Krycklan database gave a more precise location, and the threshold level of the lake outlet of C5 and the horizontal conductivity of the silty sand was increased from 1*10-8 m/s to 1*10-7 m/s. The corrections and additions did not influence the model results in any substantial way.

**Table 2: Flow model setup.** Flow model setup from the calibrated and validated Mike SHE model presented in Jutebring Sterte et al. (2018). The "soil type surface" corresponds to the soil type shown in Fig. 1b

| Soil type surface | Depth below ground (m) | Soil type | Horizontal hydraulic conductivity (m/s) | Vertical hydraulic conductivity (m/s) |
|---|---|---|---|---|
| Till | 2.5 | Till | $2*10^{-5}$ | $2*10^{-6}$ |
| | To bedrock | Fine till | $1*10^{-6}$ | $1*10^{-7}$ |
| | Bedrock | | $1*10^{-9}$ | $1*10^{-9}$ |
| Peat | 5 | Peat | $1*10^{-5}$ | $5*10^{-5}$ |
| | 7 | Clay | $1*10^{-9}$ | $1*10^{-9}$ |
| | To bedrock | Fine till | $1*10^{-6}$ | $1*10^{-7}$ |
| | Bedrock | | $1*10^{-9}$ | $1*10^{-9}$ |
| Silty sediments | 3 | Silt/clay | $1*10^{-7}$ | $1*10^{-7}$ |
| | To bedrock | Fine till | $1*10^{-6}$ | $1*10^{-7}$ |
| | Bedrock | | $1*10^{-9}$ | $1*10^{-9}$ |
| Sandy Sediments | 0.8 | Silt/Sand | $1*10^{-7}$ | $1*10^{-7}$ |
| | 2.8 | Silt/clay | $1*10^{-8}$ | $1*10^{-7}$ |
| | 0.9*max depth | Sand | $3*10^{-4}$ | $3*10^{-5}$ |
| | To bedrock | Gravel | $1*10^{-4}$ | $1*10^{-4}$ |
| | Bedrock | | $1*10^{-9}$ | $1*10^{-9}$ |
| **Soil frost** | | | | |
| Peat | Reduced vertical and horizontal flow during winter | | | |

Commented [EJS3]: New Table

[revised manuscript text omitted]

The underlying assumption in this approach is that the strong seasonal signature from precipitation will be reduced with travel time due to mixing in the soil. With infinitely long travel times, the stream water signature should equal the long term average of precipitation inputs, while short travel times should make the stream water signature reflect the input signal from the precipitation. There should, therefore, be a significant relationship between the simulated travel times and the observed winter isotopic stream signature provided that the model performs well. Some of the sub-catchments are, however, affected by evaporation from lake surfaces that result in isotopic fractionation (Leach and Laudon, 2019). This fractionationThese fractionations must be accounted for in order to use the signature as a representation of the groundwater. The isotopic composition was corrected with respect to the percentage of lakes in each sub-catchment, and a regression equation for each isotope$\delta^{18}$O was determined and applied to sub-catchments containing lakes. We used the same principle as in Peralta-Tapia et al. (2015) but adjusted it for newly acquired data with approximately 270 samples from each site (2008-2018).. The long-term regression equationequation for each isotope$\delta^{18}$O lake adjustment for sub-catchments are as follows:

$$18O = 0.18(lake\ coverage\ [\%]) - 13.20\ (r^2 = 0.87,\ p < 0.001) \qquad Eq.\ (1)$$

$$2H = 0.81(lake\ coverage\ [\%]) - 96.03\ (r^2 = 0.68,\ p < 0.001) \qquad Eq.\ (2)$$

2.3.2 Observations of stream chemistry

Comparisons were also made to the long-term annual and winter averages of stream chemistry.

$$\delta^{18}O = 0.18(lake\ coverage\ [\%]) - 13.20\ (p < 0.001, R^2 = 0.87) \qquad Eq.\ (1)$$

The comparison of the modelling results to observations of $\delta^{18}$O was based on a conceptual understanding of the seasonal variability of $\delta^{18}$O in precipitation and runoff (Fig. 2a). In spring, studies have shown that the young water fraction can be distinguished by comparing the dilution of the isotopic signature to the previous winter because the snow has a much more depleted signal (Laudon et al., 2007; Tetzlaff et al., 2009; Tetzlaff et al., 2015). The difference between winter and spring signature is referred to as the $\Delta\delta^{18}O_{spring}$ (Eq. (2)):

$$\Delta\delta O^{18}_{spring/summer} = average\ (Wn - Sn) \qquad Eq.\ (2)$$

Wn= Winter isotopic composition average of year n

Sn= Spring/summer isotopic composition average of year n

The more negative $\Delta\delta^{18}O_{spring}$ becomes, the larger the young water fraction is. Hypothetically, the same pattern should be distinguishable in summer but reversed. In summer, the precipitation is enriched compared to the winter signal, which in turn gives younger water an enriched isotopic signal. There should, therefore, be a positive relationship between the young water fraction and the $\Delta\delta^{18}O_{summer}$. In wintertime, there is no infiltration, whereby the isotopic signature can be directly related to the age of the groundwater. The closer the signature comes to the long-term precipitation average (which is equal to the measurements from the deep groundwater in Krycklan), the more well-mixed and, consequently, the older the water should be.

Other indicators of stream water age are base cation (BC) concentration (Fig. 2b). Previous attempts to follow the chemical development of groundwater in the Krycklan catchment have shown that  the BC concentration increases along the groundwater flow pathway because of weathering (Klaminder et al., 2011). Therefore, a general agreement between the concentration of BC on the one hand and modelled travel times should be possible to distinguish. The  BC is mainly derived from the weathering of local soils in the Krycklan catchment, with only a minor contribution from atmospheric deposition (Lidman et al., 2014).

Modelling of weathering rates in a soil transect in the Krycklan catchment has indicated that there is kinetic control of the release of BC in the soils (Erlandsson et al., 2016). The release of BC suggests that the longer the groundwater is in contact with the mineral soils, the higher BC concentrations can be expected, similarly to what was observed by Klaminder et al. (2011). Since BC are expected to behave relatively conservatively in these environments (Ledesma et al., 2013; Lidman et al., 2014), their combined concentration was used as a proxy for water age. Sub-catchments with longer travel times would  exhibit higher BC concentrations . It has been observed, however, that mires have a significant impact on the concentration of cations in the streams within the Krycklan catchment. The reason is that the peat does not contain any appreciable amounts of minerals, so groundwater passing through mires will not acquire cations to the same amounts as when it passes through mineral soils (Lidman et al., 2014). In practice, this will cause cations in specific subareas to be diluted by groundwater from the mires in a manner that is not related to the groundwater travel time. The cation concentrations were therefore adjusted for the influence of mires, according to Eq. (3):

Adjusted cation concentration = Observed cation concentration/ (1-fraction of mire coverage)        Eq. (3)

Commented [EJS4]: New Figure

[Figure]

**Figure 2: Conceptual figure of connection between water travel time and stream chemistry.** (a) The connection between $\delta^{18}O$ and stream water travel time. The sine curve shows the annual variations of $\delta^{18}O$ precipitation composition, and approximate seasonal winter, spring, and summer stream composition are marked. In winter, older travel times are proportional to winter baseflow isotopic signature closer to the long-term precipitation average. In spring, a greater fraction of young water is proportional to a greater difference between the spring (snowmelt) signature and the winter baseflow signature (negative sign). In summer, a greater fraction of young water is proportional to a larger difference between the summer signature and winter baseflow (positive sign). (b) The connection between base cations (BC) and travel time. The older the mean travel time, the higher concentrations of BC due to weathering.

All stream chemistry data comes from the online open Krycklan database at www.slu.se/Krycklan (Table 4). The isotopic signatures contain approximately ten years of field observations (2008 to mid-2018), approximately 25 samples per year for each site. A small part of the dataset has been published by Peralta-Tapia et al. (2016), where sampling and analyses are described in detail, and it has since been expanded using the same methodology. We used the average of the stable isotope signature from these years as a representation of baseflow.

$$Adjusted\ cation\ concentration = Observed\ cation\ concentration / (1 - mire\ coverage)$$
$$Eq.\ (3)$$

These averages were also compared to the volume-weighted average of the long-term precipitation, calculated using approximately 1160 precipitation measurements of $\delta^{18}O$ measured between 2007 and 2016. The long-term precipitation average is -13.5 ‰, which is equal to observations of the isotopic signature at the deep groundwater wells of Krycklan (10 m depth). The BC data collection methodology is reported in Ledesma et al. (2013).

**Table 4: Seasonal stream chemistry.**

 pH and base cation data were taken from the open Krycklan database, and the collection methodology and analysis are reported in Laudon et al., 2013. The base cation and pH data comprise approximately 25 (2008-2017) and 20 (2011-2018) observations for the winter period (December to February) for each stream, respectively (Table 3, an extended table can be found in Supporting information 1). Since pH and base cations are less impacted by precipitation, compared to water isotopes, the annual average was also considered. The annual base cation and pH data comprise approximately 235 (2008-2017) observations and 180 (2011-2018) observations, respectively.

**Table 3: Stream chemistry of Krycklan in order of catchment size**

| | δ¹⁸Oᵃ | | | | | | Base cations (BC)ᵇ | | | | | |
|---|---|---|---|---|---|---|---|---|---|---|---|---|
| | Winter | | Spring | | Summer | | Winter concentration | | Spring concentration | | Summer concentration | |
| | ‰ | SD/SEMᶜ | Δδ¹⁸O | SD/SEM | Δδ¹⁸O | SD/SEM | µeq/L | SD/SEM | µeq/L | SD/SEM | µeq/L | SD/SEM |
| C1 | -12.9 | 0.28/0.05 | -0.53 | 0.60/0.18 | 0.10 | 0.38/0.19 | 283 | 39/7 | 211 | 36/5 | 285 | 31/4 |
| C2 | -12.9 | 0.46/0.07 | -0.68 | 0.52/0.16 | 0.15 | 0.45/0.16 | 288 | 104/21 | 174 | 41/6 | 267 | 58/9 |
| C4 | -13.1 | 0.36/0.06 | -1.08 | 0.66/0.20 | 0.82 | 0.48/0.21 | 295 | 77/17 | 107 | 33/5 | 306 | 77/12 |
| C5 | -13.0 | 0.47/0.08 | -1.80 | 0.66/0.20 | 0.72 | 0.65/0.21 | 273 | 27/6 | 172 | 49/9 | 231 | 34/5 |
| C6 | -13.1 | 0.35/0.06 | -1.27 | 0.55/0.16 | 0.52 | 0.47/0.17 | 364 | 80/16 | 230 | 133/19 | 322 | 120/16 |
| C7 | -13.0 | 0.22/0.04 | -0.73 | 0.56/0.17 | 0.42 | 0.37/0.18 | 290 | 43/9 | 177 | 59/9 | 270 | 38/5 |
| C9 | -13.1 | 0.29/0.05 | -0.98 | 0.46/0.14 | 0.57 | 0.44/0.15 | 385 | 61/12 | 219 | 87/13 | 327 | 61/8 |
| C10 | -13.3 | 0.28/0.05 | -0.80 | 0.61/0.18 | 0.53 | 0.39/0.19 | 348 | 58/12 | 200 | 104/16 | 332 | 72/10 |
| C12 | -13.1 | 0.30/0.05 | -0.88 | 0.48/0.15 | 0.36 | 0.43/0.16 | 349 | 48/10 | 187 | 40/6 | 316 | 45/6 |
| C13 | -13.1 | 0.26/0.05 | -0.83 | 0.55/0.16 | 0.60 | 0.48/0.17 | 379 | 57/12 | 203 | 37/5 | 309 | 43/6 |
| C14 | -13.4 | 0.23/0.04 | -0.70 | 0.55/0.17 | 0.48 | 0.45/0.18 | 388 | 46/10 | 264 | 85/12 | 376 | 74/10 |
| C15 | -13.4 | 0.40/0.07 | -0.73 | 0.69/0.21 | 0.63 | 0.44/0.22 | 373 | 44/9 | 233 | 41/6 | 349 | 45/6 |
| C16 | -13.4 | 0.44/0.08 | -0.56 | 0.64/0.64 | 0.46 | 0.33/0.20 | 511 | 56/11 | 251 | 53/8 | 441 | 76/10 |
| C20 | | | | | | | 582 | 80/17 | 348 | 48/7 | 526 | 60/8 |
| Long term precipitation average | | | | | | | | | | | | |
| | -13.5 ‰ ᵈ | | | | | | 80 µeq L⁻¹ ᵈ | | | | | |

ᵃ δ¹⁸O Signature (2008-2018), data have been lake adjusted according to equation 2, and delta was calculated using Eq. (1).
ᵇ Base cation concentration (2008-2016), data have been mire adjusted according to  Eq. (3)

ᶜSD = standard deviation, SEM = standard error of the mean
ᵈ Measured precipitation average for isotopes (2007-2016) and measured BC concentration (the year 1997 to 2018)

Commented [EJS5]: Edited

**2.4 Establishing travel times - Particle tracking**

Particle tracking was used to assess travel times for each sub-catchment. The model was run 1000 years to capture the travel times from source to sink of most of the released particles in the area. One year of simulated flow results from Jutebring Sterte et al. (2018) was cycled multiple times to extend the particle tracking for 1000 years. The year 2010 was selected, as the water balance for this year was close to the long-term annual averages observed for the Krycklan catchment. The number of particles released had to be restricted due to numerical constraints, and particles were released at the top of the transient groundwater table during the first year. Approximately 0.5 particles/10 mm modelled groundwater recharge was released to capture the timing of recharge patterns (Fig. 3).

The time it took for particles to reach a stream or lake via groundwater (hereafter called 'travel time') was calculated for each sub-catchment. The calculated travel time distributions were analysed using five statistical measurement tools, the arithmetic mean, the geometric mean, the standard deviation, the standard error of the mean, and the skew

(Appendix). If the standard deviation is higher than half of the arithmetic mean, the geometric mean is a better measure of the central tendency of the data set (Taagepera, 2008). The geometric mean is defined as the back-transformed arithmetic mean of the log-transformed data. The standard deviation and skew were therefore used to evaluate which measure of central tendency was best for describing the simulated travel times. To identify the minimum particle tracking time needed for robust travel time estimates, we compared median travel times for varying lengths of particle tracking. We assumed that the calculation was run for enough time when the median of the travel time was stabilized for all sub-catchments. 2.3.3 The median stabilized after 500 years of simulation time, but in the end, we let the particle tracking run in total 1000 years to ensure that the results were stable for all parts of the catchments. Thereafter, we used all particles that reached a stream or lake to calculate mean travel times for each sub-catchment.

During winter, all simulated streamflow contribution comes from the groundwater. Here the particle tracking reflects the actual travel time to the streams. However, during summer and especially during spring, some water will reach the streams as overland flow, and therefore has spent zero days in the ground. Since the particle tracking does not take surface flow into account, two travel times were calculated for each site. The first is the groundwater age directly based on the particle tracking results (groundwater gMTT), and a second version where the surface flow component was assumed to have a very young age (zero days), which can be interpreted as the total time stream water contribution have spent in the ground (overall gMTT). To reduce the travel time according to calculate the overall travel time, we used Eq. (4).

$$\text{Overall gMTT} = \text{groundwater gMTT} * (1 - \text{fraction overland flow}) \qquad \text{Eq. (4)}$$

The young water fraction metric was also used as an evaluation criterion. Similar to previous studies (Kirchner., 2016; von Freyberg et al., 2018; Lutz et al., 2018; Stockinger et al., 2019), we assumed young water to be the sum of water reaching streams as overland flow and groundwater with age less than three months.

[Figure]

Commented [EJS6]: New Figure

**Figure 3: Particle model setup.** (a) Step by step of the particle tracking procedure. (b) Average depth to the groundwater table. The main part of the model area has a calculated depth to the groundwater table between 0-3 m and varied on a daily basis. Note that the top vertical layering of the saturated zone was set to 2.5 m below the ground surface, and the thickness thereafter follows the soil layers (thickness increasing with depth). The horizontal grid-size used was 50*50 m. (c) Schematic illustration of particle tracking set up. Particles were added to each cell at the transient groundwater table. The age of these particles was zero at the time of recharge. The particles then followed the groundwater flow while increasing in age. All particles that reach a stream or lake receives an end age, which is equal to the time from recharge to discharge in the stream. MTT is calculated for each stream using the age of all particles reaching it.

**2.5 Catchment characteristic investigation**

 Correlations between the calculated seasonal gMTT and different catchment characteristics were established to identify the main factors that affect the travel time of water to streams. The young water fraction was also tested against catchment characteristics. The characteristics tested included important terrain factors

such as size and slope as well as soil types. As many factors can affect the hydrology of a catchment,  the most important descriptive physical landscape characteristics are listed in Table 1  which together describe much of the landscape variability of Krycklan.

3 . Result
* * *
** 3.6 years. The geometric mean~~1 Travel time results**

The particle tracking model in Mike SHE was used to establish mean travel time in the 14 sub-catchments. The time from groundwater recharge until the groundwater reached a stream was used as an estimation of groundwater travel time. The geometric mean (gMTT) was used to describe the central tendency of travel times because of the skewed distribution (Table ** 5, Fig. 4). From the particle results, the calculated annual groundwater and overall gMTT for all sub-catchments ranged between 0.8 to 3.1 years and 0.8-2.7 years, respectively (Table 5). Most particles had a travel time of less than one year (34 to 54 %). The older groundwater was connected to the larger sub-catchments and sub-catchments with fluvial sediments of C13, C14, C15, C16, and C20. Particles with old ages were generally connected to deep and long flow pathways.

[Figure]

**Figure 4: Particle tracking results.** The figure shows the timing of particles reaching the sub-catchment outlet. The figure shows four examples, including C2 (forest dominated sub-catchment), C4 (mire dominated sub-catchment), C16 (Krycklan as a whole), and C20 (silt dominated sub-catchment). The same example

mean age and the largest proportion of young particles. In comparison, C20 had the oldest age and the largest proportion of old particles (Table 4, Fig. 2). sub-catchments are shown in Fig. 5.

Over On an annual basis, a year, a small fraction of the water reaches thereached a stream as surfaceoverland flow, which may enhance or dilute various stream solutes in different ways. Winter baseflow conditions may, therefore, be a better representationThe major part of the groundwater chemistry. From December to February, there was no input of precipitation due to freezing conditions, resulting in that the only input to the streams came from the groundwater. The mean simulated travel time of winter baseflow (MTTw) was older for all overland flow occurs during the snowmelt in April to May, especially in sub-catchments compared to MTTy. According to the simulation, winter baseflow (Dec-Feb) accounts for approximatelywith mires such as C4 (Fig. 5 15% of the total yearly stream contribution.

**Table 4: Particle tracking results for all sites in Krycklan.** Statistics of particle tracking results with a simulation time of 1000). Each site has the oldest age during the winter season (1.2-7.7 years. The table is ordered by increasing sub-catchment size. The statistics are calculated for each sub-catchment ) and includethe youngest age in spring and summer (0.5-1.9 years). The input of new water is also reflected in the seasonal groundwater gMTT. The groundwater is youngest in connection with the snowmelt during late spring, then increases during the MTT (MTTy is summer period with little groundwater recharge (Jun-Jul). The oldest groundwater travel times occur during the yearly mean travel time, and MTTw is the winter (Dec-Feb) mean travel time), skew, and standard deviation (SD). Further statistical information can be found in Supporting information 1. MTD is the mean travel distance of the particles. Winter baseflow is the fraction of the total annual runoff generated during Dec-Feb.winter, before the beginning of snowmelt in late March or early April.

Mire sub-catchments have the youngest mean travel time during spring snowmelt. However, as exemplified by the similar-sized sub-catchments of C2 (forest) and C4 (mire), groundwater is not renewed to the same extent in mires as in forested sub-catchment (Table 5). The groundwater gMTT of C2 was reduced from 1.2 years to 0.7 years from winter to spring. In C4, groundwater gMTT was reduced from 1.5 years to 1.2 years, despite a larger young water fraction. The overall gMTT of C4 decreased even more, from 1.5 years to 0.7 years. A more pronounced seasonality in mean travel times also occurs for catchments with a larger proportion of mires combined with low conductive soils (LCS). For example, C20 had an overall gMTT that reduced from 7.7 years to 1.9 years from winter to spring, while the overall gMTT of the similar-sized till sub-catchment C6 only changed from 2.8 to 0.6 years.

[Figure]

**Commented [EJS8]:** New Figure

**Figure 5: Seasonal fraction of runoff to streams.** The figure shows the proportion of stream water arriving as groundwater flow and as direct overland flow. Four sub-catchments are exemplified, including (a) the small forested C2, (b) the mire-dominated C4, (c) the entire Krycklan catchment C16, and (d) and the silt-rich C20 (extended version in Appendix Fig. A1).

**Table 5: Annual and seasonal (winter, spring, and summer) travel time results.**

| | Annual | | | | | | | | Season - Winter | | | | | | | |
|---|---|---|---|---|---|---|---|---|---|---|---|---|---|---|---|---|
| | aMTT | Skew | SD | SEM | gw gMTT | OL | gMTT | Yf | aMTT | Skew | SD | SEM | gw gMTT | OL | gMTT | Yf |
| unit | year | - | - | - | year | % | year | % | year | - | - | - | year | % | year | % |
| C1 | 10.1 | 4.1 | 27 | 0.4 | 1.3 | 0 | 1.3 | 20.2 | 18.8 | 2.7 | 36 | 1.2 | 3.0 | 0 | 3.0 | 5.6 |
| C2 | 2.2 | 5.0 | 5 | 0.2 | 0.8 | 0 | 0.8 | 16.3 | 2.7 | 2.4 | 4 | 0.5 | 1.2 | 0 | 1.2 | 0.0 |
| C4 | 7.7 | 7.5 | 34 | 1.0 | 1.0 | 21 | 0.8 | 39.5 | 10.5 | 6.6 | 42 | 2.7 | 1.5 | 0 | 1.5 | 2.1 |
| C5 | 15.2 | 6.4 | 61 | 1.0 | 1.3 | 35 | 0.8 | 49.1 | 30.4 | 4.1 | 84 | 3.2 | 2.9 | 0 | 2.9 | 1.1 |
| C6 | 13.7 | 6.8 | 51 | 0.6 | 1.2 | 23 | 0.9 | 41.5 | 25.9 | 4.6 | 69 | 1.8 | 2.8 | 0 | 2.8 | 1.9 |
| C7 | 8.0 | 7.4 | 25 | 0.4 | 1.2 | 8 | 1.1 | 27.8 | 13.2 | 5.6 | 32 | 1.1 | 2.2 | 0 | 2.2 | 3.5 |
| C9 | 13.2 | 6.7 | 38 | 0.3 | 1.6 | 12 | 1.4 | 28.0 | 21.6 | 5.0 | 47 | 0.7 | 3.4 | 0 | 3.4 | 2.9 |
| C10 | 10.9 | 6.4 | 35 | 0.2 | 1.2 | 14 | 1.1 | 32.8 | 16.5 | 4.5 | 40 | 0.6 | 2.5 | 0 | 2.5 | 3.2 |
| C12 | 11.9 | 5.5 | 33 | 0.2 | 1.4 | 9 | 1.3 | 27.5 | 17.6 | 4.0 | 37 | 0.4 | 2.8 | 0 | 2.8 | 5.2 |
| C13 | 13.3 | 8.0 | 43 | 0.2 | 1.5 | 9 | 1.4 | 25.8 | 21.6 | 6.4 | 53 | 0.5 | 3.3 | 0 | 3.3 | 3.1 |
| C14 | 18.3 | 7.8 | 54 | 0.2 | 2.7 | 9 | 2.4 | 19.7 | 26.4 | 6.8 | 60 | 0.4 | 5.6 | 0 | 5.6 | 1.5 |
| C15 | 14.3 | 8.7 | 43 | 0.1 | 1.7 | 10 | 1.5 | 28.2 | 21.9 | 6.7 | 49 | 0.3 | 3.8 | 0 | 3.8 | 4.0 |
| C16 | 17.4 | 8.5 | 50 | 0.2 | 2.5 | 8 | 2.3 | 22.8 | 25.3 | 7.3 | 57 | 0.2 | 5.3 | 0 | 5.3 | 3.5 |
| C20 | 21.9 | 6.0 | 52 | 0.6 | 3.1 | 13 | 2.7 | 22.9 | 32.9 | 5.8 | 55 | 1.1 | 7.7 | 0 | 7.7 | 0.0 |

| | Season - Spring | | | | | | | | Season - Summer | | | | | | | |
|---|---|---|---|---|---|---|---|---|---|---|---|---|---|---|---|---|
| | aMTT | Skew | SD | SEM | gw gMTT | OL | gMTT | Yf | aMTT | Skew | SD | SEM | gw gMTT | OL | gMTT | Yf |
| unit | year | - | - | - | year | % | year | % | year | - | - | - | year | % | year | % |
| C1 | 5.2 | 6.8 | 19 | 0.5 | 1.0 | 0 | 1.0 | 24.9 | 8.4 | 4.5 | 25 | 0.8 | 0.9 | 0 | 0.9 | 18.6 |
| C2 | 1.6 | 6.1 | 3 | 0.2 | 0.7 | 0 | 0.7 | 25.5 | 2.7 | 4.2 | 8 | 0.6 | 0.7 | 0 | 0.7 | 6.0 |
| C4 | 5.7 | 10.8 | 27 | 1.5 | 1.2 | 44 | 0.7 | 52.7 | 9.1 | 6.1 | 39 | 2.0 | 0.7 | 0 | 0.7 | 39.2 |
| C5 | 9.9 | 9.3 | 50 | 1.5 | 1.2 | 57 | 0.5 | 65.6 | 11.3 | 7.6 | 52 | 1.7 | 0.8 | 8 | 0.8 | 38.4 |
| C6 | 9.1 | 9.5 | 45 | 1.0 | 1.0 | 42 | 0.6 | 58.0 | 9.9 | 8.2 | 42 | 1.0 | 0.8 | 7 | 0.8 | 33.5 |
| C7 | 5.5 | 11.2 | 19 | 0.6 | 1.1 | 18 | 0.9 | 36.6 | 7.5 | 7.5 | 27 | 0.8 | 0.9 | 0 | 0.9 | 26.9 |
| C9 | 8.2 | 10.8 | 31 | 0.4 | 1.3 | 24 | 1.0 | 40.7 | 11.2 | 7.0 | 34 | 0.5 | 1.2 | 5 | 1.1 | 24.3 |
| C10 | 8.0 | 8.2 | 32 | 0.4 | 1.1 | 31 | 0.8 | 46.6 | 9.0 | 6.3 | 31 | 0.4 | 0.9 | 0 | 0.9 | 30.5 |
| C12 | 8.2 | 7.6 | 29 | 0.3 | 1.2 | 21 | 0.9 | 38.5 | 10.2 | 5.3 | 29 | 0.3 | 1.1 | 1 | 1.1 | 26.1 |
| C13 | 7.8 | 10.8 | 30 | 0.3 | 1.2 | 18 | 1.0 | 36.7 | 12.5 | 8.8 | 45 | 0.4 | 1.2 | 3 | 1.2 | 23.1 |
| C14 | 12.2 | 10.6 | 45 | 0.3 | 2.1 | 21 | 1.6 | 32.0 | 16.3 | 7.9 | 54 | 0.4 | 1.8 | 11 | 1.6 | 20.5 |
| C15 | 9.2 | 11.4 | 34 | 0.2 | 1.2 | 22 | 0.9 | 41.4 | 12.4 | 9.2 | 41 | 0.2 | 1.2 | 9 | 1.1 | 27.3 |
| C16 | 11.4 | 11.0 | 40 | 0.1 | 1.7 | 20 | 1.4 | 35.1 | 15.6 | 8.9 | 48 | 0.2 | 1.8 | 11 | 1.6 | 22.9 |
| C20 | 12.2 | 10.5 | 39 | 0.8 | 2.6 | 27 | 1.9 | 35.8 | 20.4 | 5.4 | 59 | 1.3 | 1.6 | 9 | 1.5 | 24.3 |

Commented [EJS9]: New Table

**Figure 2 Simulated travel time distribution of the groundwater in Kryeklan.** C16 is used as a visual reference in all panels. The figure includes all 14 investigated sub-catchments, color-coded as Fig. 1 and displayed in the legend in size order from left to right with C2 being the smallest and C16 the largest sub-catchment. The figure is divided into (a) all sub-catchments, (b) the sub-catchments of C13, (c) the sub-catchments of C12 and, (d) the sub-catchments of C14 and C15.

**3.2 Stable isotopes and stream chemistry**

The simulated winter mean travel times (MTTw) were compared to the measured winter isotope signature for each site, as well as to the measured average winter cation concentration and pH, using linear regressions (Fig. 3). There was a significant correlation between the calculated mean winter travel times and both $\delta^{18}O$ (r=0.90, p<0.001) and $\delta^2H$ (r=0.90, p<0.001). Both $\delta^{18}O$ and $\delta^2H$ decreased with travel time, approaching the long-term precipitation average of -13.46 ‰ and -99.88 ‰, respectively. There was also a significant correlation between the measured winter base cation concentration and the simulated travel times (r=0.88, p<0.001; Fig. 3). pH had a similar behavior, but the correlation was somewhat weaker (r=0.73, p<0.001). The main outliers were the mire-dominated sub-catchments C4 and C5, which have high concentrations of organic acids that influence pH negatively. Note that there are isotope data for 13 sub-catchments and chemistry data for 14 sub-catchments (isotope data excludes C20, see Table 3). The yearly mean travel times (MTTy) were also compared to the yearly average of base cations and pH, with significant results for both pH (r=0.83, p<0.001) and base cations (r=0.90, p<0.001).

**Figure 3: Linear regressions of stream chemistry and calculated geometric mean travel times (MTT).** The black line is the regression line, and the green lines are the 95% prediction bands. The plots also show the SEM (standard error of the mean) of the calculated average travel time and the chemistry observations (see Supporting information 1). (a) and (b): Average winter isotope signature, $\delta^{18}O$, and $\delta^2H$ (‰), against MTTw. Here the long-term average of precipitation signature is shown as a horizontal blue line in each graph (-13.46 $\delta^{18}O$ and -99.88 $\delta^2H$ for the years 2007-2016). (c): Average winter base cation concentration (µeq/ L, mire adjusted according to table 3 and eq. 3) against MTTw. (d): Average winter pH against MTTw. E: Average yearly base cation concentration (µeq $L^{-1}$, mire adjusted according to table 3 and eq. 3) against MTTy. F: Average yearly pH against MTTy.

**3.3 Catchment characteristics**

There was no significant correlation between sub-catchment size and calculated mean annual travel time (MTTy) (Fig. 4), C20 being the main outlier. Furthermore, when comparing the MTTy to other catchment characteristics, there was no significant correlation to the proportion of mires, till or lake area. However, there were significant (although weak) correlations between MTTy and mean travel distance (MTD) (r=0.76, p<0.01) and MTTy and catchment slope (r=0.73, p<0.01). A strong significant correlation between the MTTy and the proportion of low conductive sediments (LCS) (r=0.90, p<0.001), was also found. By using multiple regression, two simple relationships could be established between the sub-catchments and three characteristics (further description of this relationship can be found in Supporting information 2). These show that although there is a correlation between the MTTy on one hand and slope (Eq. 4) or MTD (Eq. 5) on the other hand, the most significant parameter is the LCS:

$$MTT\ (y) = -0.33 + 0.31 * Slope[°] + 0.09 * LCS[\%] \ (R^2 = 0.90\ , p < 0.001) \qquad \text{Eq. (4)}$$

$$MTT\ (y) = 0.19 + 0.01 * MTD\ [m] + 0.08 * LCS[\%]\ (\ R^2 = 0.90, p < 0.001) \qquad \text{Eq. (5)}$$

**Figure 4: Linear regression of catchment characteristics and calculated mean annual travel times (MTTy).** The black line is the regression line, and the green lines are the 95% prediction bands. The plots also show the SEM (standard error of the mean) of the simulated average travel time (see Supporting information 1). (a): Catchment area against MTTy. (b): Mean travel distance of particles (MTD) against MTTy (c): Catchment slope (°) against MTTy. (d): Low conductive sediments (LCS) (%) against MTTy.

**aMTT** = arithmetic mean (year), **SD** = Standard deviation, **SEM** = standard error of the mean, **OL** = fraction overland flow (%), **gw gMTT**= geometric mean of the particle tracking (groundwater gMTT) (year), **gMTT** = geometric mean of the particle tracking adjusted for overland flow according to Eq. (4) (overall gMTT) (year), **Yf** = fraction of surface flow and groundwater less than three months (%) (Supporting information)

**3.2 Testing model results to stream chemistry**

Three distinct seasons were evaluated with regards to the stream chemistry: winter, spring, and summer. The isotopic composition was available for 13 out of 14 sub-catchments (C20 excluded because of short time-series), while the base cation (BC) data was available for all sub-catchments. According to the modelling results, sub-catchments receive older water when the average isotopic composition is closer to the long-term precipitation under winter conditions (Fig. 6a). Some of the larger sub-catchments have a signature close to the long-term precipitation average, suggesting that they have reached complete mixing (C15, C14, and C16). However, the negative correlation is significant (r=-0.80, P<0.01), with older stream water age being closer to the long-term precipitation average. The negative correlation between the $\Delta\delta O^{18}_{spring}$ and the young water fraction was also significant (r=-0.90, P<0.0001, Fig. 6c), following our conceptual model (Fig. 2a). Sub-catchments with a larger fraction of young water during the spring displayed a greater dynamic in the isotopic composition of the stream water. The opposite was also true for the summer since the precipitation was enriched compared to the baseflow. The positive correlation was weaker compared to the spring season, but still significant (r=0.80, P<0.001, Fig. 6e).

The overall gMTT always had a strong statistical significance to the BC concentration (Fig. 6 b, d, and e), generally agreeing with our conceptual model (Fig. 2b). The correlation between BC and gMTT was strongest during winter (r=0.88 P<0.0001) and weakest during summer (r=0.79, P<0.001). The sub-catchments with the oldest age and highest BC concentration include some of the largest sub-catchments of C14 and C16, but also C20, which is one of the smaller sub-catchments. These three sub-catchments have the largest portions of fluvial sediment deposits (Table 1). The youngest ages and lowest BC concentrations were connected to smaller sub-catchments in the till areas, such as C2 and C4.

Commented [EJS10]: Edited

[Figure]

**Figure 6: Results of seasonal water fraction compared to stream chemistry, $\delta^{18}O$, and base cation (BC) concentration.** Note that $\delta^{18}O$ results are for 13 sites, while the BC record comprises all 14 sites. The sub-plots (a) to

(f) show the $\delta^{18}O$ (winter) or $\Delta\delta^{18}O_{Spring/summer}$ and BC concentrations as a function of the overall gMTT during winter, spring, and summer, respectively. The standard error of the mean (SEM) is shown for the field observations.

**3.3 Model results compared to catchment characteristics**

The main catchment characteristics found to be correlated to gMTT were catchment size, the fraction of low conductive soils (LCS), and the fraction of mires.  The strongest positive correlation was found between the young water fraction and the proportion of sub-catchment mire coverage (r=0.96, P<0.0001), as well as gMTT and low conductive soils (LCS) (r=0.90, P<0.0001) (Figure 7). A larger fraction of mires increases the young water fraction, and a larger fraction of LCS increases gMTT. A positive correlation between catchment size and gMTT was also found. The correlation was relatively weak due mainly to one outlier, C20, yet significant (r=0.63, P<0.05) (Fig. 7). However, the catchment size is also correlated to the fraction of LCS, which may be the underlying reason for this correlation (Table 6) as C20 is the only relatively small monitored sub-catchment located in the sedimentary soil area. The annual and seasonal patterns are generally similar (Table 6). However, the positive correlation between mires and the young water fraction was lost during winter, presumably due to a lack of new precipitation input into the system. The gMTT and the young water fraction correlation followed the pattern of the correlation between gMTT and mires. A weak negative correlation between gMTT and the young water fraction was found in the annual average and during spring but was lost during summer and winter.

[Figure]

Commented [EJS11]: Edited

**Figure 7: Travel time important catchment characteristics.** The figure shows the annual averages. (a) Mires and young water fraction, (b) mires and gMTT (year), (c) low conductive soils (LCS) and gMTT (year), and (d) Catchment size and gMTT (year). The gMTT has been adjusted for the overland flow for each season, according to Eq. (4).

**Table 6: Correlation matrix – young water fractions, gMTT, and catchment characteristics.** The table includes yearly calculations (white), winter calculations (blue), spring calculations (green), and summer calculations (orange). Darker colours show when the absolute value of. r > 0.50 with the connected p-value according to [a] p>0.05, [b] a p<0.05, and c p<0.01.

**Annual**

| | Log C.-size | Mire (%) | LCS (%) | gMTT | Yf |
|---|---|---|---|---|---|
| **Log C.-size** | 1 | | | | |
| **Mire (%)** | 0.02 | 1 | | | |
| **LCS (%)** | 0.58 [b] | -0.37 | 1 | | |
| **gMTT (y)** | 0.63 [a] | -0.51 [a] | 0.90 [c] | 1 | |
| **Yf (%)** | -0.02 | 0.96 [c] | -0.39 | -0.53 [a] | 1 |

**Season - winter**

| | Log C.-size | Mire (%) | LCS (%) | gMTT | Yf |
|---|---|---|---|---|---|
| **Log C.-size** | 1 | | | | |
| **Mire (%)** | 0.02 | 1 | | | |
| **LCS (%)** | 0.58 [b] | -0.37 | 1 | | |
| **gMTT (y)** | 0.64 [b] | -0.34 | 0.92 [c] | 1 | |
| **Yf (%)** | -0.08 | -0.14 | -0.43 | -0.21 | 1 |

**Season - spring**

| | Log C.-size | Mire (%) | LCS (%) | gMTT | Ywf |
|---|---|---|---|---|---|
| **Log C.-size** | 1 | | | | |
| **Mire (%)** | 0.02 | 1 | | | |
| **LCS (%)** | 0.58 [b] | -0.37 | 1 | | |
| **gMTT (y)** | 0.55 [b] | -0.55 [b] | 0.92 [c] | 1 | |
| **Yf (%)** | 0.11 | 0.95 [c] | -0.29 | -0.52 [a] | 1 |

**Season - summer**

| | Log C.-size | Mire (%) | LCS (%) | gMTT | Yf |
|---|---|---|---|---|---|
| **Log C.-size** | 1 | | | | |
| **Mire (%)** | 0.02 | 1 | | | |
| **LCS (%)** | 0.58 [b] | -0.37 | 1 | | |
| **gMTT (y)** | 0.68 [c] | -0.50 [a] | 0.80 [c] | 1 | |
| **Yf (%)** | 0.20 | 0.91 [c] | -0.20 | -0.28 | 1 |

Commented [EJS12]: New Table

**4 Discussion**

Particle tracking in the Mike SHE model showed promising results in its ability to capture the travel times across the 14 Krycklan sub-catchments. Travel times of stream water contribution and young water fractions were related to stream winter $\delta^{18}O$ signatures, $\delta^{18}O$ seasonal shifts, and base cation (BC) concentrations. Particle tracking could therefore be a useful complementary tool to tracer-based studies of travel time, especially in snow-dominated catchments, areas with pronounced seasonality, and streams dominated by old groundwater (older than 4-5 years). In this study, we found the hydrologic conductivity of the soil to be the most important parameter for the water age and mires to be an important factor regulating the young water fraction.

**4.1 Model testing and uncertainties**

Particle tracking in Mike SHE is associated with some uncertainties and limitations. A comparison of the results from this study to previous studies of mean travel times (MTT) for one of the Krycklan sub-catchments (C7) shows, however, that the different approaches gave similar results. While our study estimated a MTT time to 1.1 years, Peralta-Tapia et al. (2016) calculated a ten-year average travel time of 1.8 (minimum 0.8 and maximum 3.3) years using isotopic data and a gamma transformation method. In a study using the same data in the Spatially distributed Tracer-Aided Rainfall–Runoff model, the median of the travel time distribution was approximated to 0.9 years for the same sub-catchment (Ala-aho et al., 2017). The close agreement with the previous model runs strengthen our results.

One limitation of our modelling approach is that particle tracking is restricted to the saturated zone. This restriction is primarily related to the overland flow component, most visible in mire dominated catchments in connection with the spring snowmelt. We accounted for this effect by assuming the age of the overland flow component to be zero days (Eq. (4)). If the age of the water – or its travel time – is the time it spends in the ground, this would be the actual age of the water. Alternatively, one could define the age as the time from when a water unit melted. However, that would add additional uncertainties, and for overland flow, it would most likely still only amount to an additional couple of days in most cases and would likely not influence the overall gMTT to any large extent. Counting the number of days from when the snow fell is not particularly meaningful from a hydrological point of view as the storage of snow in winter can last up to six months.

Another uncertainty in Mike-SHE is related to the travel time from infiltration under the unsaturated condition to groundwater recharge, which, due to technical limitations, was not accounted for in the particle tracking calculations. Particles are placed in the groundwater proportionally to the recharge (Fig. 3). Therefore, the main portion of particles is introduced to the model at high recharge when the groundwater level is shallow across the catchment. However, some particles are introduced when the groundwater level is lower, such as early snowmelt or after extended dry periods. In our simulations, the groundwater table varies between 0-3 m below the ground surface (Figure 3). While mires generally have an average groundwater table above 1 m, till areas range between 2-3 m. C14 is an exception:

here, a deep esker traversing the sub-catchment results in a lower water table than in other Krycklan locations. Accounting for the travel time from infiltration to recharge could impact the results and give especially C14 older MTT than if the groundwater level was at the same level throughout the whole catchment. This limitation primarily affects catchments with long MTTs and, therefore, does not question the general patterns that were observed.

We used the stream winter isotopic composition and BC concentration to test Mike- SHE's ability to capture the variability of  travel times in the 14 Krycklan sub-catchments. Based on our results, we found significant and robust correlations between the winter isotopic signature  $\delta^{18}$O  as well as the stream chemistry, on the one hand, and the calculated travel times on the other (Fig. 6). Theoretically, infinitely long travel time would result in a stream water isotopic signature approaching the long-term average precipitation input (Fig 2). In contrast, the BC concentration of the stream water would increase until it reaches thermodynamic equilibrium with the soil mineral composition (Erlandsson et al., 2016). The strong statistical agreement between both the observed winter isotopic composition and stream chemistry and the particle travel times on the other supports the credibility of the model results.

~~A comparison of these results to previous studies of MTT for one of the Krycklan sub-catchment (C7) shows that the different approaches gave similar results. While this study estimated the long-term mean travel time to 1.3 years, Peralta-Tapia et al. (2016) calculated a ten-year average travel time of 1.7 years using isotopic data and a gamma transformation method. In a non-distributed modeling study using the same data, but another travel time distribution technique, the median of the travel time distribution was approximated to 0.9 years in the same catchment (Ala-aho et al., 2017).~~

**4.1.1 Testing model results against isotopic composition**

According to the conceptual model (Fig. 2), older baseflow water should result in an isotopic signature closer to the precipitation average. There was a strong negative correlation between groundwater age and the streams isotopic signature during baseflow (Fig. 6a), suggesting that the model produces credible water age patterns for the winter season. The larger sub-catchments, including C14, C15, and C16, are close to the long-term precipitation average, which limits the ability to estimate the travel times using isotopes. Water older than 4-5 years is argued not to be accurately quantifiable using isotopes due to amplitude loss (Kirchner., 2016). These theoretical considerations strengthen our results of a winter MTT of 4-6 years for the larger sub-catchments and provided new insights into travel times for these systems.

In spring, the young water fraction was used to evaluate the proportion of water reaching the stream as overland flow. The difference between the previous winter baseflow and stream isotopic signature at snowmelt ($\Delta\delta O^{18}_{spring}$) is mechanistically related to the amount of young water reaching the stream (Laudon et al., 2007; Tetzlaff et al., 2009). We found a strong connection between $\Delta\delta O^{18}_{spring}$ and our calculated young water fractions (Fig. 6c). The larger young water fraction was generally found in mire dominated sub-catchments, such as C4 and C5. In contrast, equally sized sub-catchments without mires, such as C1 and C2, had a less $\Delta\delta O^{18}_{spring}$ and hence smaller young water fraction. Notably, these small, entirely forested catchments are the only ones with no overland flow during the spring flood,

which again emphasizes the importance of the mires for the hydrology of the boreal landscape (Fig. 5). These results are well in line with previous work in Krycklan using end-member mixing of new and old water in the same streams (Laudon et al. 2007; 2011). Those earlier results showed a large overland flow component in wetland catchments because of frozen conditions during spring flood with biogeochemical consequences during snowmelt.

In summer, the conceptual model predicted that $\Delta\delta O^{18}_{summer}$ should also be correlated to the young water fraction, but with the opposite sign, due to the enriched summer rain (Fig. 2). Inter-annual precipitation and evapotranspiration variability likely caused the relationship to be less evident compared to the spring flood results as the snowmelt conditions are more consistent from year to year. However, although less strong than compared to the spring $\Delta\delta O^{18}$, there was still a strong connection between the average summer $\Delta\delta O^{18}$ and the modelled young water fraction (Fig. 6e).

**4.1.2 Testing model results against base cation concentration**

The base cation (BC) concentration followed the same pattern throughout the year (Fig. 6b, 6d, and 6f), with increasing concentration strongly correlated to increasing age. Since the weathering rates generally are kinetically controlled, i.e., related to the travel time, such stream chemistry variables can be used as a relative indicator for stream water age ~ as long as the mineralogy remains comparatively homogenous (Erlandsson Lampa et al., 2020). This study showed that the modelled travel times were significantly correlated to the BC concentrations.  Reducing weathering to a matter of travel times  may be an oversimplification  as the rate is affected by, for example, chemical conditions, differences in mineralogy, and particle size distributions. However, previous research in the Krycklan catchment has indicated that the chemical composition of the local mineral soils is surprisingly homogeneous, even when comparing unsorted till and sorted sediments (Klaminder et al., 2011; Peralta-Tapia et al., 2015; Erlandsson et al., 2016; Lidman et al., 2016). Therefore, we do not expect mineralogical differences between soil types to have a significant impact on the release of cations. The one exception is peat deposits, which strongly affect the cation concentrations, and that was accounted for by adjusting the concentrations for the influence of the mires following Lidman et al. (2014). Differences in particle size distribution may be important because coarser soils will have less surface area per volume unit, therefore allowing for less weathering. However, such soils can also be expected to have higher hydraulic conductivities, leading to higher flow velocities and, consequently, less time available for weathering. Therefore, differences in area-volume ratios between different soil types would not counteract the effect of travel times on the weathering, rather enhance it. Accordingly, base cation concentrations should still be a useful indicator of transit times. ~~The pH of some sub-catchments has also been shown to be affected by mires, especially C4 and C5 (Buffam et al., 2007), due to high concentrations of organic acids that influence pH, especially when the buffer capacity is low (Hruska et al. 2003). This effect can be observed in the results of the mire dominated sub-catchments, which fall below the 95 % prediction line (Fig. 3D and 3F). Hence, we do not think that these deviations contradict the credibility of the model results.~~

It can also be argued that pH is not a mixable quantity and therefore unsuitable as a tracer. Still, the purpose of the comparisons to stream water chemistry was not to mechanistically explain the evolution of stream water chemistry, but instead to compare the modeling results to some parameters that could be expected to reflect the groundwater travel times. Such tests are crucial for the credibility of the model results. Because pH increases as a result of silicate weathering, it is likely that pH would increase with the groundwater travel time, albeit not necessarily in a linear manner. Complementing isotopic tracers with transported solutes for testing simulated travel times provide more insight into catchment processes. Despite the arguments that can be made against the use of pH and

Despite arguments that can be made against the use of cations as tracers, they still offer a complementary possibility to test the performance of the model. As emphasized by McDonnell and Beven (2014), the inclusion of tracers in hydrological models is necessary to ensure that a model does reproducereproduces the speed of flow, which is a crucialan important parameter when assessing travel time distributions. In catchment-scale models, this could be an isotopic tracer or a solute that is transported with the water (Hooper et al., 1988; Seibert et al., 2003; Fenicia et al., 2010; Hooper et al., 1988; Hrachowitz et al., 2013; Seibert et al., 2003). Although neither the travel time distribution nor the kinetics of weathering is fully understood, the strong agreement between the calculated travel times and the observed stream water chemistry strengthens the credibility of our model results and, the system understanding of catchment-scale travel times and their connection to biogeochemistry. More specifically, the chancesresults increase the likelihood that the model is producing credible results for the right reasons.

Strengthening the credibility of particle tracking in Mike SHE to produce travel time distributions enables the use of particle tracking as a useful complement to other similar studies in the future. For example, stable water isotopes and biogeochemical tracer tests can be affected by dilution or chemical reactions, and here particle tracking could be a useful complement. Further extensions in the Mike SHE family (Graham & Butts, 2005) also allow the incorporation of solutes or isotopes with more complex biogeochemical behaviour. These extensions could be used for further calibration and validation of the hydrological model (McDonnell and Beven, 2014) as well as investigation of biogeochemical processes in the catchment. A more mechanistic investigation of the relationship between groundwater travel times and stream water chemistry would require such extensions.

**4.2 Catchment slope and hydraulic conductivity control travel times**

Contrary to our hypothesis,

**4.2 Mean travel times, young water fractions, and catchment characteristics**

The main factor controlling the groundwater travel times was the hydraulic conductivity and slope of the catchment rather than the catchment size itself. In agreement with previous studies by Capell et al. (2012), Muñoz-Villers et al. (2016), and Tetzlaff et al. (2009), there were significant correlations between catchment slope and travel timecharacteristics found that affect the age and young water fraction were low conductive soils (LCS) and the fraction of mires (Fig. 4). However,7, Table 6). The most significant factor for the mean travel times was foundrelated to be the proportion of low conductive sedimentssoils (LCS), which overshadowed both the slopeimportance of the catchment and the travel distance of particlessize. Earlier studies in Krycklan by Peralta-Tapia et al. (2015) and Tiwari et al. (2017) have suggested that the MTT of groundwater is linked nonlinearly linked to the catchment size, i.e., the

travel times increases with the catchment size.. However, we found the silt-rich but relatively small C20 to be a distinct outlier to this pattern, indicating that catchment size may not be the underlying factor causing high MTTs (Fig. 4).6). As shown in Table 6, the catchment size is correlated to the fraction of LCS. In other words, there are few small catchments in the silt-areas with low conductivity. The reason is partly the setup of the Krycklan catchment study, which initially focused on the till areas, and partly the fact that the LCS are located in the lower parts of the Krycklan catchment so that all large catchments by necessity must contain at least some LCS. The long travel times in relation to the relatively small catchment size in C20 means that the groundwater flow velocity generally is lower than elsewhere. Nevertheless, the average catchment slope of C20 is steeper than in comparably sized sub-catchments in the till areas, so the topographical possibilities to build up high hydraulic gradients that can drive the water transport should be largerlarge (Table 1; and Fig. 4). This is probably related to the fact that C20 is the only relatively small sub-catchment (1.45 km²) in the area largely covered by LCS. However,). The fluvial sediment deposit fraction may also explain the relatively long travel times of C14 and C16. For example, Although C14 is smaller than C15, which mostly lacks LCS, it still has longer MTT.

In contrast, C15 is much closer to C12 and C13 in MTT, even though the C15 catchment is almost twice as large. This suggests (Table 5). The results suggest that the critical difference between these sub-catchments and other sub-catchments is related to the hydraulic conductivity of the regolith,soils rather than the catchment size. Without the contribution of water from headwater catchments with fine regolithsoils (such as C20), the MTT of sub-catchments like C14 and C16 would probably have a MTTbe much closer to that of the other smaller till dominated sub-catchments. The results, therefore, emphasize that one cannot generally assume that the travel time would increase with catchment size unless the distribution of regolith is comparable throughout the landscapesoils is comparable throughout the landscape. The effect of LCS is more prominent in winter than during the other seasons. For example, the difference between the winter and spring mean travel time is almost six years for C20 compared to two years for the similar-sized sub-catchment C6 and the mean travel time of C14 is four years compared to three years for the similar-sized sub-catchment C15.

**5 Summary remarks**

The Mike SHE model showed promising results in its ability to capture groundwater travel times, which was firmly related to winter stream isotope signatures. The simulated travel times were, in turn, well correlated to the base cation concentration and pH of the streams. In contrast to our hypothesis, we found that the catchment size itself is not the main factor determining groundwater travel times. Instead, we found the hydrologic conductivity of regolith to be the most important parameter, but also that the catchment slope and travel distance of the groundwater could have an impact. This essentially points back to Darcy's law, which states that the groundwater flow is governed by the pressure gradient (approximated by the catchment slope in this case) and the hydraulic conductivity of the medium (approximated by soil types). In that sense, the results are in line with theory. However, it is far from evident that precisely these catchment parameters would stand out as most important in the complex landscape of the Krycklan

Sub-catchments with mires receive the highest young water fraction during the spring snowmelt; however, the annual age of water is not as strongly connected to that landscape feature (Table 6). The main factor that controls this is the soil frost on the mires (Peralta-Tapia, 2014), which reduces the renewal of the groundwater at spring because a larger fraction of water flows directly to the stream as overland flow. For example, C2 and C4 have a similar catchment size and soil properties, with the main difference that C4 has a significant fraction of mires and a greater seasonality in travel times. Even though C2 and C4 have this landscape difference, they still have a similar annual age (Table 5). Besides the somewhat higher specific discharge from mires compared to forests (Karlsen et al. 2016), the main hydrological effect of mires consequently appears to be a redistribution of water between the seasons, causing younger runoff during the spring and older water during dry and cold seasons. In forest till soils, on the contrary, most of the snowmelt infiltrates the ground and instead displace older, pre-event water during the spring flood. The infiltration of snowmelt water leads to a replacement of older water by younger in the forest soils. This process called transmissivity feedback explains the younger water age during the rest of the year and the smaller seasonal variation of forested till-soil catchments (Bishop, 1991; Laudon et al. 2004). The process is a consequence of exponentially increasing hydraulic conductivity toward the soil surface in till soils.

**5 Summary remarks and implications**

Northern landscapes are sensitive to climate change (Tetzlaff et al., 2013; Sprenger et al., 2018). Climate predictions suggest that warming will affect higher latitudes to a disproportionally large extent, and hence soon begin to affect the annual snowpack, shorten the longevity of snow cover, increase the frequency of winter thawing episodes, reduce soil frost, and increase annual precipitation (IPCC., 2014; Jungqvist et al., 2014; Brown et al.,2017; Lyon et al., 2018). To foresee the implications of such changes, it is important to have a good baseline understanding, including both empirical data but also well-calibrated and tested models, upon which we can build future predictions of what such changes will mean to our water resources. In a mosaic landscape, such as the northern boreal landscape, distributed models can be of great value in this context since variable impacts on different landscape characteristics can be distinguished and disentangled.

The present study was based on the integration of a large dataset from a previously well-investigated catchment and an advanced distributed 3D hydrological model. The results showed that the groundwater travel times vary considerably on annual and intra-annual scales in the boreal landscape, both as an effect of physical differences between different types of catchments, most notably the hydrological conductivity of the soils, and the response of different landscape units to the changing of the seasons. Yet, the combination of stable water isotopes, stream water chemistry, and particle tracking provided a consistent picture of how the boreal landscape functions hydrologically and what processes and factors are of importance. Hence, this system approach not only strengthens the credibility of these specific modelling results but also more broadly confirms the applicability of process-based numerical modelling and particle tracking under the complex hydrological conditions with, for example, long dry winters, temporary soil frost, and intensive spring floods that prevail in the boreal region. 
[revised manuscript text omitted]

---

## Referee Report (RR1)

| | Annual | | | | |
|---|---|---|---|---|---|
| | Log C.-size | Mire (%) | LCS (%) | gMTT | Yf |
| **Log C.-size** | 1 | | | | |
| **Mire (%)** | 0.02 | 1 | | | |
| **LCS (%)** | 0.58 [b] | -0.37 | 1 | | |
| **gMTT (y)** | 0.63 [a] | -0.51 [a] | 0.90 [c] | 1 | |
| **Yf (%)** | -0.02 | 0.96 [c] | -0.39 | -0.53 [a] | 1 |
| **gMTT (y)** | 0.64 [b] | -0.34 | 0.92 [c] | 1 | |
| **Yf (%)** | -0.08 | -0.14 | -0.43 | -0.21 | 1 |
| **gMTT (y)** | 0.55 [b] | -0.55 [b] | 0.92 [c] | 1 | |
| **Yf (%)** | 0.11 | 0.95 [c] | -0.29 | -0.52 [a] | 1 |
| **gMTT (y)** | 0.68 [c] | -0.50 [a] | 0.80 [c] | 1 | |
| **Yf (%)** | 0.20 | 0.91 [c] | -0.20 | -0.28 | 1 |

---

## Author Response (AR2)

**Letter to Editor**

Dear Elin and co-workers,

Thank you again for the strong improvement of your manuscript. I have received and studied the reports of two referees. Moreover, I quickly went through your manuscript myself. Referee 1 raised a series of concerns to sharpen the structure and to strengthen your conclusions with respect to insights about the system – not the model. Referee 2 is more positive about the reached state. He/she points to the lack of structural detail in the model setup description.

I strongly agree with both referees and would like to ask you for another round of major revisions along the suggested lines. In my view, your study has a strong data-related focus, which is somehow difficult to trace and obscured when referring to your MikeSHE model and the added corrections. Moreover, I find it particularly difficult to draw the lines to catchment functioning beyond small vs. large or GW dominated vs. OF dominated. I have full confidence that you can and will turn your manuscript into a nice paper when adhering to and working clearly towards the key messages about landscape functioning.

I hope you find the reviews (incl. mine) helpful for this. If you have further questions, please contact me.

All the best. Many cheers.

Conrad

**Reply:** Dear Editor, we are grateful for your thorough work, both reading and discussing the manuscript, as well as your constructive criticism. It is obvious that you gave our manuscript a lot of time and effort, which we much appreciate, and we hope that our answers and changes are adequate and satisfactory. We received commented on the structure, especially regarding the method section about the Mike SHE model as well as the discussion section, both from you and reviewers. We hope that the updated version of the manuscript will remedy these concerns. The major changes in the method section include an extended description of the Mike SHE model, and an edited version of the section regarding the stream chemistry and seasonality. Our answer is built up as follows: a list of major changes, a reply to the editor and thereafter, a reply to reviewer #1 and #2 respectively. All questions and comments are stated unbridged, with our replies below each question in brown. Blue text was taken from the old version of the manuscript and red text was taken from the new version of the manuscript. Please note that the abbreviation for geometric mean of the travel time distribution is now shortened to $MTT_{geo}$.

**List of major changes**

The major changes to this version of the manuscript includes:

- A more in-depth description of the Mike SHE flow model in the method section
- A new figure describing Mike SHE
- An improved structure for the method section and the discussion section
- Updated figures and tables from input by Editor and reviewers
- Table 5 have been simplified and additional information was moved to appendix
- New abbreviation geometric mean of the travel time distribution ($MTT_{geo}$)

**Reply to Editors review**

**Here is my review:**

1. L115-124: After studying your methods and consulting your 2018 paper, I am not quite sure how you used MikeSHE and if your application can really hold as physical 3D GW, 1D vadose zone and surface water model. If I am not mistaken, MikeSHE allows to use different approaches to calculate the hydrological fluxes through the different model units – which are not necessarily "physically" defined.

**Reply:** Reviewer #1 and #2 also commented on this part of the method section, and we agree that a more in-depth description of Mike SHE could be useful. We, therefore, extended the flow model section according to the text example below (there is an extended version of this text in the manuscript). We also added a figure explaining the Mike SHE model. We hope that these additions will give sufficient information regarding the Mike SHE model setup.

Part of new method section added to the manuscript:

We applied the Mike SHE/Mike-11 hydrological modelling tools to quantify travel times in a pre-calculated 3D transient flow field. The simulated terrestrial hydrological system for the Krycklan catchment includes: the saturated and unsaturated flow, ET, snowmelt, overland flow, and streamflow processes. The fully distributed 3D modelling tool use topography, soil properties, and time-varying climate inputs to calculate the water fluxes throughout a catchment (Rahim et al., 2012; Sishodia et al., 2017; Wang et al., 2012; Wijesekara et al., 2014). The ET processes include canopy interception, open surface evaporation, root uptake, sublimation, and soil evaporation from the unsaturated zone based on a methodology developed by Kristensen and Jensen (1975). Flow in the saturated zone (SZ) is calculated in 3D by the Darcy equation. The flow in the unsaturated zone (UZ) is calculated in vertical 1D using the Richards equation, and overland flow (OL) is calculated using a horizontal 2D diffusive wave approximation in the Saint-Venant equations (Fig. 3)….

[Figure]

**(New) Figure 3: Schematic Mike SHE model set up.** Precipitation falls on the ground as rain or snow. Evapotranspiration (ET) processes include canopy interception, open surface evaporation, root uptake, and soil evaporation from the unsaturated zone (UZ). The overland flow (OL), saturated zone (SZ), and UZ interact depending on the saturation level. The SZ is divided into ten calculation layers (CL), while the UZ has a much finer description. Streamflow is modelled through Mike 11 and is not restricted to the Mike SHE resolution. The figure is used on the courtesy of SKB. Figure illustrator: LAJ.

2. Moreover you superimpose mixing assumptions (eq. 3 and 4), which tries to compensate for some model limitations. L123f is full of further claims, which I find difficult to recover in your manuscript. I suggest to really stick to the approach using data from 14 sub-basins and calculating seasonal gMTT from this data and the model and evaluation of this gMTT findings.

L123: The complexity of the model allows for an in-depth investigation of advective travel times by non-reactive particle tracking simulations in a transient flow field.

**Reply:** We hope that the extended version of the Mike-SHE model setup, is sufficient to allow this claim. Regarding the question of the specific equation see answer below:

**Equation 1-4:** We agree with later comments by the editor regarding the necessity to explicit write out these equations (equation 1 to 4). To simplify the manuscript the text has been modified so that these equations instead are written in text format in the method section.

**Equation 3:** Adjusted cation concentration = Observed cation concentration/ (1-fraction of mire coverage)

While we agree that Eq. 4 is an attempt to compensate for some limitations of the particle tracking in Mike SHE, we must emphasise that Eq. 3 is not. The purpose of using base cations and oxygen isotopes in the manuscript was to enable a comparison of the modelled transit times to something observable in order to strengthen the credibility of the performance of the model. However, it requires a fundamental understanding of how they work and, in many cases, also some adjustments in order to allow a fair and meaningful comparison. The underlying assumption concerning the base cations, as illustrated in Fig. 2, is that the kinetically controlled weathering of base cations from the mineral soil could act as a proxy for the contact time between the minerals and the groundwater, i.e., the groundwater transit time. The assumption is, however, only valid when the groundwater is in contact with mineral soils, which complicates things because mires are abundant in some of the investigated sub-catchments. Because no minerals are present in the peat the base cation concentration cannot be expected to increase during the time the groundwater spends there. Hence, a direct comparison between base cations and groundwater age in a mire-dominated would be misleading so this needs to be addressed somehow to allow a fair comparison.

The fact that mire-dominated sub-catchments have lower concentrations of base cations (and other weathering-derived elements) as a result of no weathering in the peat is well demonstrated and we refer particularly to Lidman et al. (2014), which is cited in the manuscript, for further details and discussions on this matter. All data presented in the manuscript come from the same catchments as were investigated in this study, so we believe that there is strong underlying evidence for Eq. 3. Again, this is not to correct or compensate the modelling results, but to adjust the observed cation concentrations in certain sub-catchments so that the more accurately reflect the transit time, thereby allowing some sort of validation of the model.

Furthermore, the effects of this adjustment should not be exaggerated, and this may be helpful to illustrate (Figure Y below). Because most investigated sub-catchments only contain a minor fraction of mires the correction will only have a major impact on a few sub-catchments, e.g. C4 and C5. Hence, the correlations and relationships we present would be valid without this adjustment, but they would be a bit weaker and, in our opinion at least, somewhat unfair because

the underlying conceptual model would not make full sense, since it would assume weathering of base cations in peat.

[Figure]

Figure Y: Winter Stream concentration before (x-axis) and after implementing equation 3 (y-axis)

**Equation 4:** $MTT_{geo}$=groundwater $MTT_{geo}$*(1- fraction overland flow).

There is a limitation with particle tracking in Mike-SHE, since it only functions in the saturated zone. This is due to that the particle tracking is based on saturated flow using the advection-dispersion ekvation based on the Darcy velocity. This correction can however be seen at the soil contact time, which is needed to compare the $MTT_{geo}$ to stream BC concentrations. The stream BC concentration is affected by the water-soil contact time and is diluted by water taking the overland route to stream. To account for the OL water, especially in spring, the $MTT_{geo}$ was reduced using the OL fraction as scaling factor.

However, for most catchments, almost all water comes to the streams as groundwater flow (for example, the full-scale catchment receives approximately 10% water as overland flow). The overland flow is also mainly linked to the spring snowmelt and to sub-catchments with mires. It has no impact on the winter gMTT (water only from groundwater) and very little impact on summer and annual $MTT_{geo}$. Please see the Fiq. X below, which compares the annaul adjusted $MTT_{geo}$ and unadjusted $MTT_{geo}$. Although we mainly discuss the adjusted values, we will state the unadjusted values in Appendix.

[Figure]

Figure X: Annual unadjusted MTT$_{geo}$ (x-axis) and after implementing equation 4 → annual MTT$_{geo}$ (y-axis)

3. Sec. 2.2: Despite being in favour for the reached brevity, the essential points are difficult to discern. Even after consulting your 2018 paper, I cannot really see, if the numerical discretisation and setup is as presented in Bosson et al. (2008) table 3-1 with a more fine vertical discretisation near the surface to enable a calculation of unsaturated flow (see Vogel and Ippisch, 2008, 10.2136/vzj2006.0182). L174 is rather confusing - especially when consulting Tab 2 with the greatest depth of surface soils to 7 m (out of 100 m) and the rare occasions of GW tables deeper than 10 m (Fig 3b).

**Reply**: The unsaturated zone and saturated zone have different discretization, with the unsaturated zone having a discretization from centimeters to meters while the saturated zone has a discretization from a meter to a few meters. Since the particle tracking occurs in the saturated zone, we believed that the most important information was the properties of the saturated zone. However, for clarity purposes, we added more information regarding the complete Mike SHE model according to the answer to question 1.

4. Also the actually used "physical" concepts remain unclear to me. In L180 you refer to depth-dependent drainage functions with given hydraulic conductivity values (Tab 2), which I could not fully understand from a quick scan of your 2018 paper and Bosson et al. (2008). I am under the impression, that this is some sort of Darcy flux with unit gradient.? Is this correct? How is infiltration calculated? Please do not get me wrong: I have no problem with your model selection and the used approaches. Moreover I like the brevity. However, I would suggest to clarify which "numerical engine" (to use the term from Graham and Butts (2005)), which concepts and which discretisation is used. Maybe tab 2 could also come as a figure for easier understanding and to get the model out of the way for the more interesting insights?

**Reply:**  The reason behind the exclusion of a more descriptive part regarding the flow model was that we did not want the flow model to be the focus point for this study. However, we understand

from the reviewers and the editor that a more explicit description of the Mike SHE model is needed. We therefore extended the part about the flow model in the method section. As the editor suggested, we also added a figure explaining the Mike-SHE model (see answer to question 1).

5. I got confused by the brief announcement of the particle tracking approach in Sec 2.2 (L190ff.) but the actual subsection 2.4 with the more detailed description. Also giving tab 3 without explanation how porosity affects the particle tracking added to my confusion. I suggest to tackle all particle tracking in sec. 2.4.

**Reply:** We agree that this part of section 2.2 should be moved to section 2.4 for better clarity. We also clarified in the text which equation is used for the particle tracking:

The advection-dispersion equation governs the transportation of particles for a porous medium. The Darcy velocity is divided by the porosity to calculate the groundwater velocity. Therefore, the only complementary input data needed to run the particle was porosity values (Table 4).

6. Fig 3 is insightful, however I find it difficult to understand the low number of particles. If you get about 20 particles per year and most particles are released rather quickly, can your really robustly sample the tails? I would also welcome a word about the idea behind only "releasing" particles in the saturated zone (and not simply adding markers to all incoming water).

**Reply:** The word "release" might be a somewhat misleading. Particles are added through the groundwater recharge. However, the number of particles released must be limited to run the model. With 0.5 particles per 10 mm recharge per cell, we are still able to capture the main recharge inputs, such as the spring snow melt (approximately 200 mm → 10 particles) and the larger precipitation events during the summer and autumn (150 mm → 8 particles). Note, that 20 refers to the number of particles per cell. Due to the model size of the catchment, there is in total more than ½ million particles injected in the model over a year, which is very computational heavy.

7. Sec. 2.3: I am slightly confused by the title and the following description of the system's seasonality. Maybe I went through the manuscript too quickly. I suggest to include a statement about the references ($\delta18O$ and BC) and the respective concepts (Fig 2) incl. the site seasonality before going into detail. L125-135 does not really serve this.

**Reply**: We changed the title to: "2.2 Linking seasonal base cation concentration and isotopic signature to stream water age". We also agree that a better statement about the seasonality could be useful in this section. Therefore, we changed the first paragraph to:

This study was focused on three seasons in Krycklan, winter, spring, and summer. We defined the winter to occur from late November to late March because it is characterized by an extensive and permanent snow cover with little groundwater recharge. We assumed that the winter stream composition reflects the chemistry of deeper groundwater (Fig. 2). Similarly, we defined spring by the rapid transition in hydrology and biogeochemistry in April-May. During snowmelt, ca 50 % of the annual precipitation leaves the system in a short period of time, diluting baseflow with new input of water. Finally, we defined the summer season as the period between July and September when the hydrology is characterised by rain, high ET and relatively little runoff. June and October were excluded because,

hydrologically, they are transition months between the three distinct seasons. This is because snowmelt can still influence runoff in June, and winter conditions (snowfall, soil frost, etc.) can sometimes begin to establish in October.

**Figure 2:** We received some additional suggestions from reviewer #1, which we also added. The main changes are, $\propto$ have been removed, $MTT_{geo}$ was removed and changed to mean soil contact time, and the line for precipitation was removed for clarity. We also added a description of $\delta^{18}O$ and BC directly in the figure, instead of mainly having it in the figure caption.

[Figure]

$\delta^{18}O_{winter}$ = average isotopic signature for winter streamflow, correlated to the groundwater age

$\Delta\delta^{18}O_{spring}$ = difference between average winter and average spring stream isotopic signature
the difference is correlated to the young water fraction
$\Delta\delta^{18}O_{summer}$ = difference between average winter and average summer stream isotopic signature
the difference is correlated to the young water fraction

8. With respect to the employed "correction terms" (eq. 1,3,4) I cannot really see how much this influences your results. At the moment it simply resides in the realm of post processing of the MikeSHE model output and the data. However, it comes with a series of assumptions.

**Reply:**

**Equation 1:** This method is standard when working with water isotopes (unless you specifically wish to trace the lake water). Incomplete evaporation from a water surface, e.g. a lake, will alter the isotopic composition of the water so this needs to be accounted for when looking at the isotopic signal from lake-influenced catchments (Peralta-Tapia et al., 2015).

Peralta-Tapia, A., Sponseller, R. A., Ågren, A., Tetzlaff, D., Soulsby, C., and Laudon, H.: Scale-dependent groundwater contributions influence patterns of winter baseflow stream chemistry in boreal catchments. Journal of Geophysical Research: Biogeosciences, 120(5), 847–858. https://doi.org/10.1002/2014JG002878, 2015.

**Equation 3 and 4** – Please see answer to question 2

9. For linear regressions (like eq 1) it might be sufficient to name the scaling factor and intercept and focus on what is behind the regression? (Is there seasonality in the signal… Shouldn't there be a weather dependency of this relationship (depending on actual evaporation)?)

**Reply:** The use of water isotopes in catchment hydrology is based on the seasonal fractionation, which is shown in Fig. 2. The signal is indeed strongly affected by the weather and in particular the temperature at which the phase transitions occur, e.g. water vapor to rain and rain to snow. This causes a distinct seasonal variation in the input signal to the system.

Further fractionation may occur also in the catchments when there are phase transitions. This is the case, for instance, when there is evaporation of inception on tree, but since this typically ends in complete evaporation of the water drops it does not alter the input signal to the system. However, if there is incomplete evaporation, i.e. parts of the evaporating water remain in the systems, this can cause a substantial alternation of the isotopic signature of the water. This is a problem mainly in lakes, where there is continuous evaporation of water from the lake surface, particularly during the summers. This is enough to significantly change the isotopic signature of the lake water. In all applications of water isotopes in catchment hydrology it is therefore customary to adjust the isotopic signature for the effects of evaporation from the lake (unless the purpose is to trace lake water), (Peralta-Tapia et al., 2015).

Peralta-Tapia, A., Sponseller, R. A., Ågren, A., Tetzlaff, D., Soulsby, C., and Laudon, H.: Scale-dependent groundwater contributions influence patterns of winter baseflow stream chemistry in boreal catchments. Journal of Geophysical Research: Biogeosciences, 120(5), 847–858. https://doi.org/10.1002/2014JG002878, 2015.

10. Eq 3 in my eyes says that the flux in a cell is averaged over all surface fractions but that the concentrations of non-mire surface fractions are diluted by the same flux of non-loaded water from mires. I see its practical meaning but I am not quite sure about its theoretical underpinning and if this has effect on the results.

**Reply:** The conceptual idea behind Eq. 3 is that weathering occurs in mineral soils but not in peat. Hence, base cations will be released to the groundwater, causing higher concentrations, as long as the groundwater remains in the mineral soils. Any period that the groundwater may spend in a mire would, however, not lead to higher concentrations because there are no minerals present that can release base cations. Hence, if a sub-catchment has large portion of mires where the groundwater may spend substantial amounts of time, the transit time of the groundwater in the system cannot be expected to be reflected in the base cation concentration. Hence, this undermines the assumption we need to make in order to test the modelled transit times against real data. Therefore, some sort of correction is needed.

While this is a conceptual idea, it was developed based on actual data from the very same sub-catchments as in this study. This has been presented in detail in above all Lidman et al. (2014), which is cited in the manuscript. This paper includes a wide range of weathering-derived elements, which all showed the same pattern of lower concentrations from mire-dominated catchments. Depending on how strongly different elements bind to organic matter there may also

be a substantial accumulation in the peat, but for the base cations this effect was minor. For example, if a mire-dominated catchment (like C4 in the manuscript) contains ca. 40% mires, the Na concentration in that stream would be 40% lower than in a 100% forested stream. This was statistically valid across a wide range of elements. That is the empirical underpinning of Eq. 3. Hence, we believe that it is firmly justified.

The effects of the correction on the results are shown in Fig. Y. As can be seen, the overall pattern is still the same because most sub-catchments do not contain so much mires, but above all the mire-dominated streams C4 and C5 deviate and display conspicuously low base cation concentrations given the modelled transit times. One possible interpretation of this figure would be that the model fails to accurately calculate the transit times of the water in the mires, but that would then rely on the erroneous assumption that there is widespread weathering of base cations in mires. Hence, it would be a misrepresentation of the results, and therefore we feel that this correction is relevant and justified.

11. Eq 4 is a similar scaling approach. This appears to have strong implications. After thinking about it I am not really sure what the propose of MikeSHE remains if groundwater flow is indeed calculated as "drainage function" based on conductivity and when overland flow is actually imposed as surface fraction. I assume that your

**Reply:**

**The drain constant:** The uppermost saturated calculation layer's depth, 2.5 m, was calibrated by studying the calculation layer thickness's influence on the groundwater table's location and dynamics. Above this depth, the ET processes were shown to have an impact on the groundwater table dynamics. If the groundwater table falls below the uppermost calculation layer, a more simplistic method, not taking capillary rise and all ET-processes into account, is applied. The top calculation layer's thickness results in all soils shallower than 2.5 meters being averaged into one soil type. Kh is averaged using the thickness of each soil layer. Vertical flows are more dependent on the lowest Kv. Therefore, the harmonic weighted mean value is used to calculate the new Kv instead. In MIKE SHE, a drain function can be applied to account for small ditches and drains in the landscape not captured by the model grid or the 1D river flow network. It has also been used to account for higher hydraulic conductivity in the uppermost part of the first calculation layer. The function is activated whenever the groundwater reaches 0.5 m below the ground surface, above which higher K-values have been observed.

**Equation 4:** As discussed above, Eq. 4 is an attempt to look at the implications of the limitations imposed by the particle tracking module in Mike SHE. Since the particle tracking is limited to the groundwater, the modelled transit times will of course be representative for the groundwater, but in cases where there is considerable overland flow, the age of the groundwater will not be the same as the age of the stream water. This is important above all during the spring flood and, in particular mire-dominated systems. We see this as a way to further test the performance of the model and to address the impact of overland flow on the overall transit time of the stream water.

We also wish to emphasize that Eq. 4 is not used to substitute the modelled groundwater transit times – these are still presented in the manuscript. Instead, it offers an additional metric, which deepens the understanding of the hydrology of these systems. Otherwise, the limitations of the particle tracking would only leave us with groundwater age, but for many applications a

quantification of the stream water age is also valuable. As shown in Fig. X above (question 2), the difference between the two is quite small.

12. Eq. 1-4 in general: I am not really convinced by the chosen notation with a mix of variables and names and everything. First of all, the equations are rather simple. Hence I could imagine that they are actually not needed. The calculation and averaging of $\Delta\delta18O$ is also quite straight forward (eq 2). Generally speaking, the average of an average is equal to the average of the whole set as long no weights are added. Hence I am not sure if this can be presented more easily?

**Reply:** We agree that the chosen notation was unnecessarily complicated and inconsistent. This has changed in the revised manuscript. The equations are of course relatively simple, but for the sake of clarity we believe that it might be worth to explicitly write them out.

We also realize that the delta-delta notation ($\Delta\delta^{18}O$) may seem a bit confusing. Lower case delta of course refers to the standard delta notation ($\delta^{18}O$) that is used for many isotope systems. What we then look in the testing of the model is the change in $\delta^{18}O$, and change is often denoted by upper case delta.

13. Sec 2.4: I find it very difficult to understand why particles are only traced in the GW and not form the entry into the system. I suspect this to be a limitation of the model. However, this forces the correction with eq. 4.?

**Reply**: Yes, this is a limitation in the model. Please see answer to question 2, regarding equation 4.

14. The second post processing step comes from the geometric vs. arithmetic MTT calculation. Obviously, this has dramatic impact. Using the median or a log(time) conversion could be other approaches. In the later course you refer to percentiles below 1 year… First of all, I would expect a little more reference to the literature about these steps. I suggest to clarify the used methods also with respect to your forthcoming comparison of $\Delta\delta18O$ and BC vs. gMTT. Maybe a more conceptual description about the theoretical similarities about the respective approaches would be helpful to prepare and clarify your findings? I would also appreciate to be able to discern what of your results can be attributed to the model or the decisions in your post processing.

**Reply:** It is indeed a complex question how to reduce a distribution to a single number in the most meaningful way. Both geometric and arithmetic means as well as median values and log time conversions are common choices (Destouni et al., 2001; Kaandorp et al., 2018; Massoudieh et al., 2012, 2017; Unlu etal., 2004) , which all have their strengths and weaknesses. None of them is, however, able to fully represent the underlying distribution, but the way we see it they are all valid metrics, which are ubiquitous in the literature. Figures A, B and C below show

different metrics of the distribution compared to the old groundwater fraction (groundwater older than 5 years). Since the BC concentration is thought to be well correlated to the age of the groundwater, we needed a stable measurement that can account for the tail, without giving the tail to great weight. The main reason we favour geometric mean over arithmetic mean is that the latter is highly sensitive to the tail of the distribution, and this is where one might suspect that there is considerable uncertainty in the model. Whether the oldest particles reach ages of say 100, 1000 or 5000 years, will have profound impact on the arithmetic mean, but less so on the geometric mean (Taagepera, 2008; Unlu etal., 2004; Zhang etal., 1996). For example, the a _MTT goes up to 20-30 years (Figure A, below), while the g_MTT only goes up to 3-4 years (Figure C, below), much closer to the median. However, compared to the median (Figure B, below), the geometric mean still considers the length of the tail. The median of the distributions becomes almost equal for most sub-catchments, making it hard to say anything about the difference of the distributions. However, the geometric mean, because it takes the length of the tail into account, gives a spread to the travel times, which can be directly linked to the tail. All metrics are however stated in Table 5 and Appendix.

[Figure]

Figure A, B and C: The old groundwater fraction (>5years) compared to distribution metrics of arithmetic mean (average), median, and geometric mean (geo).

The geometric mean is mainly recommended to use as a when the distribution is significantly skewed, i.e., when the standard deviation (SD) is more than half of the arithmetic mean (Taagepera, 2008). This is the case for the travel time distribution. However, as for the isotopic signal and the base cation (BC) concentration, the SD is much smaller than the arithmetic mean (Table 2). As for the comparisons to BC and oxygen isotopes, the choice of metric is not crucial because they are all strongly correlated to one another. We realise that we should have emphasised this better in the manuscript. We agree it is vital to make sure that the findings ultimately reflect an underlying pattern and are not merely an artefact from decisions in the post processing of the results. We hope that this is clear in the revised manuscript.

References:
Destouni, G., Simic, E. and Graham, W.: On the applicability of analytical methods for estimating solute travel time statistics in nonuniform groundwater flow, Water Resour. Res., 37(9), 2303–2308, doi:10.1029/2001WR000348, 2001.

Kaandorp, V. P., de Louw, P. G. B., van der Velde, Y. and Broers, H. P.: Transient Groundwater Travel Time Distributions and Age-Ranked Storage-Discharge Relationships of Three Lowland Catchments, Water Resour. Res., 54(7), 4519–4536, doi:https://doi.org/10.1029/2017WR022461, 2018.

Massoudieh, A., Sharifi, S. and Solomon, D. K.: Bayesian evaluation of groundwater age distribution using radioactive tracers and anthropogenic chemicals, Water Resour. Res., 48(9), doi:https://doi.org/10.1029/2012WR011815, 2012.

Massoudieh, A., Dentz, M. and Alikhani, J.: A spatial Markov model for the evolution of the joint distribution of groundwater age, arrival time, and velocity in heterogeneous media, Water Resour. Res., 53(7), 5495–5515, doi:https://doi.org/10.1002/2017WR020578, 2017.

Taagepera, R.: Making Social Sciences More Scientific: The Need for Predictive Models. Oxford: Oxford University Press. https://doi.org/10.1093/acprof:oso/9780199534661.001.0001, 2008.

Unlu, E. and Faller, J.F. (2001), Geometric mean vs. Arithmetic mean in extrusion residence time studies. Polym Eng Sci, 41: 743-751. https://doi.org/10.1002/pen.10770

Zhang, C., Zhang, S. A robust-symmetric mean: A new way of mean calculation for environmental data. GeoJournal 40, 209–212 (1996). https://doi.org/10.1007/BF00222547

15. As a minor point I was wondering if 1 particle per 20 mm GW recharge is really sufficient? That means that one year is represented by ±18 bins only…

**Reply**: Please see answer to question 6.

**Results: I will go through the remainder rather quickly.**

Fig 4: Where is the gMTT, MTT, median, percentiles etc.? How does this compare to estimates from data (before turning to the much more aggregated Fig. 6 and 7)? I think a brief summary of the nice water dynamics performance from your 2018 paper would be worth mentioning. (This could also come as staring point in your methods.)

**Reply**: We decided that the figure became confusing to showcase all these values in it. However, we agree that showcasing the values for at least one sub-catchment could be useful for a visual representation (see figure below). Most values can however be found in table 5 and Appendix Table A1. We compare our results with earlier findings in the discussion section. Here we have other studies made for C7. Otherwise, we are restricted to the other stream chemistry results shown in figure 6 and 7. We also added a brief summery if the results for the 2018 paper in the method section.

[Figure]

**Figure 5: Examples of particle tracking results.** The figure shows the age of particles reaching sub-catchment outlets. The solid line showcases the statistics for C16, including the 25th percentile, the median, the geometric mean, the arithmetic mean, and the 75th percentile (Appendix, A1). Moreover, the figure shows three other example distributions, including C2 (small till and forest-dominated catchment), C4 (small mire dominated catchment), C20 (small silt dominated catchment).

Tab 5 and Fig 5: I find it not easy to look through these many results with the few guidance offered. Likewise I found it difficult to reconnect your conclusions to the presented graphs and numbers.

**Reply:**

**Table 5:** Reviewer #1 also commented on Table 5, and we found that the explanatory text below the table was missing. We therefore placed this text above the table. We also made a shortened version of Table 5 with the most important information, and put the other information in Appendix, Table A1. We believe that A1 provides more in-depth information about the travel time distribution but this might not be necessary for the major results and discussion of this manuscript.

**Figure 5:** Fig. 5 contains plenty of information and they may therefore require some effort when looking through, but we agree that we as authors should provide the reader with sufficient guidance to make this as easy as possible. The inclusion of this figure was based on previous comments on the manuscript, where reviewers requested a more thorough presentation of the modelled distributions. To make figure 5 easier to understand, we added indicators for each season, starting with spring and ending with winter. We also added a descriptive text to the y-axis of the smaller chart, more descriptive titles for each sub-catchment and a line at 50 % so that the smaller charts more easily can be compared with each other. The figure text was also updated accordingly. See figure below:

[Figure]

**Figure 6: Seasonal fraction of discharge to streams.** The figure shows the proportion of annual stream discharge arriving as groundwater and overland flow. Four sub-catchments are exemplified, including (a) the small till and forest dominated C2, (b) the small mire dominated C4, (c) the silt dominated C20 and (d) the full-scale Krycklan catchment C16 with mixed mires and forests (extended version in Appendix Fig. A1). The figure showcases the water age fraction discharging to the streams. The fractions are both shown as part of the total annual discharge as well as the water composition. The bands below the months highlight the three investigated seasons, spring, summer, and winter.

(L344 how can I see deep and long flow paths?)

**Reply**: We chose to remove this sentence. It was supposed to say that they are more connected to sub-catchments with deeper soil. But it seemed to work better as a discussion point.

Fig 5: What is "discharge fraction" in the main panels referring to? (percentage of annual discharge?) Why is the overall shape of the curves of each GW fraction very much similar although the flow paths should be very different? Would it be an option to avoid the C# but to refer to real descriptors (like whole catchment, small forest subbasin)?

**Reply:** Yes, it is percentage of annual discharge. Note that therefore the actual discharge from the small sub-catchment of C2 is much smaller than the discharge from the entire catchment.

Although the main shape is similar for all sub-catchments, the age of the groundwater as well as the fraction of overland flow is very different. We also decided to showcase the annual discharge on a weekly basis, because the daily discharge graphs were hard to distinguish. The monthly discretization also makes the shapes similar because daily differences are not showcased. We like to keep the C# so that the study can be compared easily to other studies in Krycklan. These denotations are already well established in the literature from Krycklan, which is quite extensive. However, we have added the suggested descriptors since we agree that these would clarify the examples.

Fig 6: Why is not always gMTT used on x-axis? Would it be an option to give focus to some sub-basins, which I can find in each plot? Eg. C20 is in b, d, f but not in a, c, e. Also the others from Fig 5 are not to be found in all plots. Maybe this could also strengthen your story, when you can adhere to some examples motivated throughout the manuscript?

**Reply:** We agree that we could give a better focus to some sub-catchments. We have changed these figures accordingly and showcase C2, C4, C16 and C20. Note that we don't have isotopic data for C20, and we therefore cannot showcase it in all plots. The same is shown in Fig. 5. We explained in the method section that the $^{18}O$ is more related to the young water fraction rather than to the actual age of the water. Therefore, we showcase $MTT_{geo}$ in some plots and the young water fraction in some plots.

**Discussion: Again just quickly.**

I follow referee 1 that the focus should really be the data and the characteristics. When it come to the model, I do not see that there is sufficient basis about the employed concepts and numerics to attribute found behaviour to model specifics. Which are the most valuable and/or most delicate properties which will be affected by changes in snow melt and evaporation? What can we learn for other catchments with far less data than Krycklan can offer?

**Reply:** We agree with the editor and reviewer #1, that the discussion section could use some structural changes to make it clearer what the most important parts of the study has to offer. The first section is now dedicated to the limitations of the model. The second and third parts are dedicated to model and stream chemistry and their consistent picture of how the boreal landscape functions. The last section discusses the causes of intra-annual variations of $MTT_{geo}$ and young water fraction, and which catchment characteristics these can be linked to. Finding characteristics that increases or decreases discharging travel times or young water fraction are important to understand, especially for boreal catchments. Future land-use or climate changes may cause soil types and landscape types to respond differently in the future and therefore, to have a well-established baseline understanding of these areas is important for future predictions.

You state that soil hydraulic conductivity is essential. However, this appears to be somehow one of the few site parameters. I am inclined to claim this importance a model artefact - especially given the rather broad definition (in orders of magnitude).

**Reply:** One uncertainty is related to the calibration of the flow model, especially regarding depth and soil properties (Jutebring Sterte et al., 2018). However, the flow model was calibrated and evaluated based on daily discharge measurements for the 14-sub-catchments, groundwater measurements, and all other available data from the Krycklan site. We believe that the well-represented groundwater levels and the correlations to stream discharge amount and peak flows gave a credible model. It should be noted that the study is still based on a model, therefore, we also used chemistry based indications, such as $^{18}$O and BC, to ensure that the results are not related to a model artefact but instead related to the catchment functioning.

Following Darcy's law, which describes the theoretical understanding of the physics that underlies most geohydrological models, the hydraulic conductivity, soil porosity, and hydraulic gradient are the three parameters needed to model groundwater travel times, and of these the conductivity generally varies the most (orders of magnitude). We do not see this strong dependence on hydraulic conductivity as a model artefact, but as an expected outcome based on our physical understanding of groundwater flow. Interestingly, in previous studies for the site, this effect from soil properties on travel times wasovershadowed by the effect of the sizes of the different sub-catchments. One thing that the present study in combination with the new data from the C20 catchment has demonstrated is that small catchments can have long MTTs, but it all seem to depend on the soil type.

We also want to emphasize that the modelled travel times have a strong agreement with observable data, i.e., stream chemistry observations linked to travel times such as isotopic data and BC (without being calibrated to relate to these factors). Combined this indicates that the results are not just an artefact but is a result of a well calibrated physically based flow model whose results also can be linked to stream chemistry. Our study is also in agreement with previous studies which have shown that the main part of the groundwater discharged from the small till catchments is younger groundwater (Bishop, 1991; Kolbe et al., 2020; Laudon et al. 2011). Groundwater investigations have shown a much older age (i.e. several decade) already at a few meters depth below the water table compared to shallower groundwater, and a greater variability of the shallow groundwater isotopic signal. This demonstrates that the main part of water being discharged in the till dominated areas is relatively young groundwater. This has been linked to the decreasing hydraulic conductivity in the till soil, especially in the upper part of the till soil.

Bishop, K., Seibert, J., Nyberg, L., and Rodhe, A.: Water storage in a till catchment. II: Implications of transmissivity feedback for flow paths and turnover times. Hydrological Processes, 25(25), 3950–3959. https://doi.org/10.1002/hyp.8355, 2011.

Laudon, H., Berggren, M., Ågren, A. et al.: Patterns and Dynamics of Dissolved Organic Carbon (DOC) in Boreal Streams: The Role of Processes, Connectivity, and Scaling. Ecosystems 14, 880–893. https://doi.org/10.1007/s10021-011-9452-8, 2011.

Kolbe, T, Marçais, J, de Dreuzy, J-R, Labasque, T, Bishop, K.: Lagged rejuvenation of groundwater indicates internal flow structures and hydrological connectivity. Hydrological Processes; 34: 2176– 2189. https://doi.org/10.1002/hyp.13753, 2020.

.

You tackle the role of mires throughout the manuscript but it remains rather implicit. Maybe this could be worked out more clearly based on a hypothesis? Or is it really a side topic?

**Reply:** Wee agree that the fraction of mires could be stated earlier as an important factor to investigate. Therefore, we added a section to the introduction that states:

We focused especially on the catchment characteristics that have been suggested to be important factors for regulating stream chemistry of the Krycklan sub-catchments, including the areal coverage of mires, catchment size, soil properties, and seasonal changes in groundwater recharge (Karlsen et al., 2016; Klaminder et al., 2011; Laudon et al., 2007; Peralta-Tapia et al., 2015; Tiwari et al., 2017).

**Conclusions**

You state "Yet, the combination of stable water isotopes, stream water chemistry, and particle tracking provided a consistent picture of how the boreal landscape functions hydrologically and what processes and factors are of importance." I am not sure if these threads have been really brought together yet.

**Reply:** We like to thank the editor for this comment. We believe that this comment also highlights some of the issues with the structure of the discussion section. Therefore, we have re-written this section and re-structured it to strengthen the conclusion section. I.e., that the previous isotopic studies and our study link the young water fraction and variations of young water fraction to the fraction of mires, the mire frost, and the high hydraulic conductivity of the shallow part of the till soils. In turn the BC concentration can be linked to the discharging water age, which in turn can be linked to the soil properties and in part to the catchment size.

Moreover, the role of snow packs is not really worked out in your study (or I have missed it). Since it is also in the title I suggest to really give it more attention.

**Reply:** The main impact of the snowpack is the reduced groundwater recharge in winter, and the spring flood due to snowmelt in spring. We agree that the snowpack was given too much attention in the title, so we therefore changed the title accordingly.

**Minor things I spotted:**

L96f. Verb missing?

Snow-dominated landscapes have gained increasing attention due to their importance as water resources (Barnett et al., 2005) and their vulnerability to climate change the last decades (Tremblay et al., 2011; Aubin et al., 2018).

**Reply:** We appreciate that this error was noticed, and we have changed this sentence to:

Snow-dominated landscapes have received increasing attention in the last decades due to their importance as water resources (Barnett et al., 2005) and their vulnerability to climate change (Tremblay et al., 2011; Aubin et al., 2018).

Tab1 caption: Please clarify "Soil proportion based on the soil"

**Reply:** We have corrected this text according to:

Table 1: Sub-catchment characteristics. The list includes all 14 monitored sub-catchments in Krycklan, called C1 to C20, including the entire Krycklan catchment, C16. Different branches of the stream network are illustrated in

distinct colours in Fig. 1. The table includes the sub-catchment area, average elevation, and average slope. Further descriptions of these characteristics can be found in Karlsen et al. (2016). The table also includes soil proportion based on the soil map (1:100,000) from the Swedish Geological Survey (2014).

**Reply to reviewer #1**

**General comments:**

The paper has definitely seen some improvements during the last revision. The authors did a thorough job of considering and answering the reviewers questions and concerns. In particular, the introduction is more complete and the methods section contains more important details.

The addition of investigating the seasonal differences has definitely made the paper more interesting and impactful. Stressing the specific local settings (with a long, snowy, low-recharge winter period, mire abundance and regional soil types) and directing the investigation and discussion towards these factors moves the manuscript in the direction of presenting novel results and insights.

The presentation style and structure has, however, suffered from incorporating the new research. Often the manuscript is not that clearly written and the individual sentences are not connected in order to show relationships and tell a coherent story. Regularly there is a lack of precision in the wording that makes it hard to follow the discussion and arrive at an otherwise obvious conclusion.

The authors decided to move away from investigating the entire travel time distributions and concentrate on mean travel times and young water fractions alone. I find this to be a missed opportunity since the use of the distributed model allows for a more in-depth analysis of the catchment system dynamics (and travel time distributions contain a lot more information on that).

The discussion section is mainly filled with an evaluation of the catchment model (2.5 pages) while the new results concerning travel times only occupy 1 page. I would like to see a significant improvement of the way the discussion and conclusion is presented in terms of structure and writing style.

**Reply:** We would like to thank Reviewer #1 for his/her review of the manuscript. The constructive criticism concerns and suggestions have helped improve the new version of the manuscript even further. We agree that the introduction is at a higher level this time around and are pleased that the reviewer noticed the effort put into this section. We also appreciate that the reviewer thinks that the manuscript became more interesting due to the new focus on seasonal travel times. After reading the comments from the two reviewers and the editor we, however, agree that the text needs to be re-worked for better clarity. Here we used the specific comments to improve the language and clarity of the text. We especially addressed the concerns regarding the discussion and conclusion, with more focus on the actual results and we addressed the structure and writing style. Regarding the comment on the manuscript moving away from the entire travel time distribution is slightly confusing to us. The first iteration of the manuscript focused on the mean travel time of winter base flow. The distribution of the travel times was shown, but this is still the case in Figure 5 (new manuscript). We also believe that Figure 6 (new manuscript) gives a better representation of the travel time distribution for each stream and how it varies within a year, which the previous version of this study did not have.

Our explanations and responses to all Reviewer #1 comments and questions are listed below. All the comments have been included unabridged in this document, together with our response

following each comment in brown. Blue sections are text directly imported from the old version of the manuscript, and red was taken from the new version. Please note that the abbreviation of geometric mean of the travel time distribution and the young water fraction is now shorten to $MTT_{geo}$ and young water fraction respectively.

**Specific comments:**

**Key points**

Line 18: '…while THE FRACTION of mires…'

**Reply:** We have changed this to "areal coverage of mires"

**Abstract**

Line 29: Why do you write 'the following snowmelt'? Following what exactly?

In this study, a particle tracking model approach in Mike SHE was used to investigate the travel time, and pathway of water in 14 partly nested, long-term monitored boreal sub-catchments characterized by long and snow rich winters with little groundwater recharge and highly dynamic hydrology during the following snowmelt.

**Reply:** This sentence was changed to:

In this study, a particle tracking model approach in Mike SHE was used to investigate the travel time, and pathway of water in 14 partly nested, long-term monitored boreal sub-catchments. These sub-catchments are characterized by long and snow-rich winters with little groundwater recharge and highly dynamic runoff during spring snowmelt.

Line 30: Better describe what potentially causes the seasonality in MTTs (changes in precipitation, temperature, vegetation?). 'Seasonality' causing seasonal differences is not a good explanation…

The seasonality caused considerable variation in travel times between different seasons and landscape types, with winter mean travel times ranging from 1.2-7.7 years and spring mean travel times ranging from 0.5-1.9 years.

**Reply:** This sentence was changed to:

The variations were found to be related to the distribution of different landscape types and their different response to seasonal changes. Winter MTTgeo ranged from 1.2-7.7 years, while spring MTTgeo varied between 0.5-1.9 years.

Line 32: '…(YOUNGER than three months)...'

**Reply:** We have changed this to <3 months

Line 34: This is the only time in the entire manuscript that you mention the word 'synchronicity'. This should be explained in the results, discussion and/or conclusion section.

**Reply:** We appreciate that reviewer #1 noticed this, and we have addressed it in the discussion section.

Line 36: What about the mires affected the young water fraction? Was it the fraction of mires?

Mires were found to affect the young water fractions of the stream contribution

**Reply:** This sentence was changed to:

The areal coverage of mires was found to be especially important for the contribution of young water in spring …

Line 37: So catchments with mires cannot also be forest-dominated?

Mires were found to affect the young water fractions of the stream contribution (r=0.96, P<0.0001) by introducing larger differences in the mean travel times between the seasons compared to forest dominated sub-catchments.

**Reply:** Yes, that is true. Sub-catchments can have a minor mire areas but still be forest dominated. What we were trying to say here is that the more mires there are in a catchment, the more we can see a difference between seasonal travel times. We tried to clarify this by changing the sentence to:

The areal coverage of mires was found to be especially important for the contribution of young water in spring (r=0.96, P<0.0001). The main factor for this was attributed to extensive soil frost in mires, causing considerable overland flow during the snowmelt period…

**Introduction**

The introduction has improved a lot. Previous work is now covered more completely and the authors have obviously read and inserted the suggested additional references appropriately. Also the focus and objective of the study has been shifted towards exploring TTDs and their controls in different seasons which adds significantly more relevance to the results.

Line 105-106: A strange sentence: '…water that have spent…', '…sub-subsurface…'.

These specific conditions provide unique opportunities to study the source of water that have spent the longest time in the sub-subsurface environment.

**Reply:** This sentence was changed to:

Landscapes with long-lasting snow cover that often melts rapidly in the spring creates both opportunities and challenges for determining stream water age and pathways.

**Method**

Line 152: '…has been included…'?

The catchment is divided into 14 nested sub-catchments and has been included multi-disciplinary biogeochemical and hydrological research for more than 30 years (Laudon and Sponseller, 2018).

**Reply:** This sentence was changed to:

For more than 30 years, multi-disciplinary biogeochemical and hydrological studies have been conducted in Krycklan (e.g., Laudon and Sponseller, 2018). Streamflow is monitored in 14 nested sub-catchments, called C1 to C20, with the longest continuously monitored time-series starting at the beginning of the 1980s.

Line 183: '…gave a more precise location…'? What do you mean by this? Better spatial resolution in general? A more precise location of certain features? And what happened to 'the threshold level of the lake outlet of C5'?

**Reply:** Both the reviewers and the editor were confused by the Mike-SHE model section. We have therefore updated this section for more in-depth information regarding the model tool itself. Regarding the question at hand, streams are modelled using a 1D module for river hydraulics called Mike 11. This module also uses high-order dynamic wave formulations of the Saint-Venant equations. The Mike11 model is not restricted to the grid size of the Mike-SHE model. Instead, Mike 11 can be much more complex to allow for a more precise calculation of water levels and flow rates. If the cross section of the stream is smaller than the grid size the M11 cross section is placed between two adjacent grid cells in MIKE SHE, which is the case in the present study, the small forest streams in Krycklan are much smaller than the applied grid size of the MIKE SHE model. Due to this coupling between the MIKE SHE and the MIKE 11 model we could give the outlet of the sub-catchment a more precise location as well as the lake threshold level.

Line 193: What happens to particles that leave the saturated zone but re-enter it later (re-infiltration)? Is that somehow accounted for? If not, in which way would that influence your results?

**Reply:** This is a source of uncertainty in the particle tracking model. Particles that reach the unsaturated zone or the overland waters are removed from the calculations and are not used to calculate the mean travel times for the stream flow. If this part of water was accounted for it would generally increase the mean travel times for most sub-catchments. However, all recharge areas over an annual cycle, are included as sources of particles in the model. Which means that areas where water are discharging under certain conditions might turn into recharge during another time of the year. This means that we include most of the flow paths that annually contribute water to the streams.

Line 228: I would prefer a more unambiguous description of the isotopic value. Instead of 'more depleted' I would write 'more negative' because the signal could be more depleted in the heavy but also more depleted in the lighter isotope.

**Reply:** We agree that "more depleted" is an ambiguous term unless it is specified which isotope is depleted. It seems to be widely used when describing isotopic fraction, but it would clearer to use another description. Therefore, we changed this to "much lighter (depleted) signal".

Line 230: In the equation you write 'spring/summer', above and below you only use 'spring'.

**Reply:** We have corrected this in the new revision of the manuscript.

Line 273-274: Does this include snowmelt isotope measurements? If not, a huge amount of water input to the groundwater (with very different isotopic composition) would be neglected.

**Reply:** This data includes all precipitation, i.e. both rain and snow. We added a statement about this in the method section regarding the chemistry data.

Line 290: What about the time it takes to reach the water table? What deviations from the 'real', 'complete' MTT do you expect when using this simplification?

**Reply:** We addressed this uncertainty in the discussion section:

Another uncertainty related to the particle tracking model in Mike SHE is related to the travel time from the point of infiltration through the unsaturated soil horizons to the saturated groundwater. Due to technical limitations, this travel time cannot be accounted for in the particle tracking calculations. Particles are placed at the groundwater table proportionally to the groundwater recharge (Fig. 4). Therefore, the main fraction of particles introduced to the model occurs at high infiltration rates when the groundwater level is close to the soil surface. Under these conditions, the water has, in most cases, spent a relatively short time in the unsaturated zone. However, some particles are also introduced when the groundwater level is lower, such as early snowmelt or following extended dry periods. Under such conditions, the model uncertainty increase. In this context, the smallest potential uncertainty occurs in mires that seldom experience a groundwater table below one meter below the soil surface. The uncertainty becomes somewhat larger in the till areas where the unsaturated zone on average is above 1 m but can extend down to 3 m below the ground during low flow. C14 and the lower part of C16 are exceptions to these relatively shallow saturated conditions as a deep esker traverses the sub-catchments resulting in a groundwater level up to 10 m below the soil surface (Fig. 1). Accounting for the travel time from infiltration to recharge could impact the results and provide, especially for C14 and C16, longer MTT than if the groundwater level were at the same level throughout the whole catchment. This limitation primarily affects catchments with the longest MTTs and, therefore, does not seriously question the general pattern observed. The distance from the ground surface to the groundwater table is for most model cells much shorter than the distance to the nearest stream so most of the transit time should be related to the groundwater flow rather than to percolation. Although water, especially during dry conditions, no doubt can spend considerable time in the unsaturated zone, it must also be acknowledged that this water volume is small compared to the groundwater inventory in the saturated zone so its impact on the average MTTs should be relatively small.

Line 297: Can a standard deviation be 'high'. I'm not sure – but I'm not a native speaker.

**Reply**: We are not native speakers either, but to our knowledge, a high standard deviation indicates that the data is more spread out, while a low standard deviation means that the data is more grouped around the mean. Our impression is that "high" and "low" frequently are the words used to describe the standard deviation.

Line 298: The median would also be an appropriate measure that has been used frequently.

**Reply:** We agree that median also can be used to find the bulk of a population. However, it only states the middle value regardless of the size of the values to the right and left of the middle value, whereas the geometric mean will account for the size of the values. We discussed this in an answer to the Editor (see answer to Editor question 14) and extended the section regarding the use of the geometric mean in the method section.

Line 301: I am confused because I thought you wanted to look at MTTs for different seasons. What do you mean by '…when the median was stabilized for all sub-catchments'? Do you mean that the seasonal change in the median TT was stable for all sub-catchments?

To identify the minimum particle tracking time needed for robust travel time estimates, we compared median travel times for varying lengths of particle tracking. We assumed that the calculation was run for enough time when the median of the travel time was stabilized for all sub-catchments. The median stabilized after 500 years of simulation time, but in the end, we let the particle tracking run in total 1000 years to ensure that the results were stable for all parts of the catchments. Thereafter, we used all particles that reached a stream or lake to calculate mean travel times for each sub-catchment.

**Reply:** We agree that this part was confusing. We changed most of this section, so we cannot state the exact change, but this is the most relevant change in response to this question:

Particle tracking was used to assess groundwater travel times from groundwater recharge to stream runoff for each sub-catchment. The model was run for 1000 years to capture the travel times of all discharging groundwater for each sub-catchment. One year of simulated flow results from Jutebring Sterte et al. (2018) was cycled 1000 times to extend the particle tracking simulation. The year 2010 was selected, as the water balance was close to the long-term annual averages observed for the Krycklan catchment. All particles were released at the top of the transient groundwater table the first year. Numerical constraints restricted the number of particles released to 0.5 particles/10 mm modelled groundwater recharge, which corresponds to a total of approximately 0.6 million particles. This number of particles was assumed to be enough to capture the timing of recharge patterns (Fig. 4).

Line 306: Don't forget the time that water needs to percolate to the water table. This could be significant in certain situations (especially when it's dry).

**Reply**: We addressed this concern at the comment of line 290. Unfortunately, this is a limitation in the Mike SHE.

**Results**

Line 338-340: A bit confusing. So now it reads as if first you determined the mean travel time of all catchments then you determined the groundwater mean travel time and then you determined the groundwater geometric mean travel time? It would be easier to understand if you connected the sentences better…

The particle tracking model in Mike SHE was used to establish mean travel time in the 14 sub-catchments. The time from groundwater recharge until the groundwater reached a stream was used as an estimation of groundwater travel time. The geometric mean (gMTT) was used to describe the central tendency of travel times because of the skewed distribution (Table 5, Fig. 4).

**Reply**: We like to thank the reviewer for this comment. We tried to clarify this section and have changed it to:

The particle tracking results were used to establish travel time distributions and MTT of stream water runoff of the 14 sub-catchments in Krycklan. Since the travel time distributions were significantly skewed, we assumed that the geometric mean of the travel time distributions provided the best representation of MTT (Table 5, Fig. 5). However, all metrics are stated in the Appendix, Table A1. The annual MTTgeo for all sub-catchments ranged from 0.8 to 3.1 years (Table 5). Most groundwater discharging to a stream had a travel time of less than one year (34% to 54%). The longest stream MTTs were connected to silt dominated catchments such as C16 and C20. We used some sub-catchments for result representation, but all results are provided in Table 5 and Appendix A1. The displayed sub-

catchments were: C2 (small till and forest dominated catchment), C4 (small mire dominated catchment), C20 (small silt dominated catchment), and C16 (the full-scale Krycklan catchment).

Line 350: Enhance or dilute? These are not antonyms. What do you mean exactly? Also it is strange to mention that they are affecting stream solutes in 'different ways' without giving any more details…

On an annual basis, a fraction of the water reached a stream as overland flow, which may enhance or dilute various stream solutes in different ways.

**Reply**:  What we meant to say by "enhance or dilute" was the the concentration of a certain solute might *either* increase *or* decrease depending on the difference in concentration between the stream water and the overland flow. For clarification we change this sentence to:

On an annual basis, a fraction of water reached the streams as overland flow. A major part of the overland flow occurred during the snowmelt in spring, especially in sub-catchments with mires such as C4 (Fig. 6).

Line 353: You are using age and gMTT interchangeably. Please stick to one term for clarity. Also, I think you want to say that the 'discharging' groundwater is youngest. There could be very old groundwater in the system that does not discharge to the stream during that period of time.

Line 354: Try to be clearer in your use of language. It is not the groundwater that 'increases'. It is either the age of the groundwater or the gMTT of the groundwater that increases.

Line 355: Again, more clarity is required. Travel times are not 'old', travel times are 'short' or 'long'. Discharging water can be 'old' or 'young'.

Line 353 to 355: On an annual basis, a fraction of the water reached a stream as overland flow, which may enhance or dilute various stream solutes 350 in different ways. The major part of the overland flow occurs during the snowmelt in April to May, especially in sub-catchments with mires such as C4 (Fig. 5). Each site has the oldest age during the winter season (1.2-7.7 years) and the youngest age in spring and summer (0.5-1.9 years). The input of new water is also reflected in the seasonal groundwater gMTT. The groundwater is youngest in connection with the snowmelt during late spring, then increases during the summer period with little groundwater recharge (Jun-Jul). The oldest groundwater travel times occur during the winter, before the beginning of snowmelt in late March 355 or early April.

**Reply:** These comments regarding Line 353-355 are connected to the same section. Therefore, we will address them at the same time. We agree that this section could be re-written for clarity. We agree with the suggestions by the reviewer and have used them to change the section to:

On an annual basis, a fraction of water reached the streams as overland flow. A major part of the overland flow occurred during the snowmelt in spring, especially in sub-catchments with mires such as C4 (Fig. 6). Both the fraction of young water reaching the streams and the MTTgeo displayed strong seasonal trends. The longest seasonal MTTgeo, 1.2-7.7 years, and the smallest young water fraction were found during the winter season. In winter, the fraction of older water successively increased until the spring snowmelt began in early April. Conversely, the smallest fraction of old discharging water and short MTTgeo, 0.5-1.9 years, were connected to events of larger groundwater recharge, such as the spring snowmelt and heavy summer rains.

Line 361: Why do you write 'despite a larger young water fraction'? Since this larger fraction is not considered in the particle tracking, it should not have a big influence on the groundwater

gMTT. Also, why do you write 'even more'? The decrease in groundwater gMTT for C4 is smaller than for C2.

Mire sub-catchments have the youngest mean travel time during spring snowmelt. However, as exemplified by the similar-sized sub-catchments of C2 (forest) and C4 (mire), groundwater is not renewed to the same extent in mires as in forested sub-catchment (Table 5). The groundwater gMTT of C2 was reduced from 1.2 years to 0.7 years from winter to spring. In C4, groundwater gMTT 360 was reduced from 1.5 years to 1.2 years, despite a larger young water fraction. The overall gMTT of C4 decreased even more, from 1.5 years to 0.7 years. A more pronounced seasonality in mean travel times also occurs for catchments with a larger proportion of mires combined with low conductive soils (LCS). For example, C20 had an overall gMTT that reduced from 7.7 years to 1.9 years from winter to spring, while the overall gMTT of the similar-sized till sub-catchment C6 only changed from 2.8 to 0.6 years.

**Reply**: We tried to refine this section and clarify which numbers in table 5 we refer to and why. Table 5 was also shortened to the values we discuss most. However, all other information is still available in appendix. We changed this section to:

In spring, mire sub-catchments had the shortest MTTgeo. However, as exemplified by the similar-sized C2 and C4 sub-catchments, groundwater was not renewed to the same extent in mire dominated systems due to a larger fraction surface runoff (Fig. 6). Mire dominated sub-catchments (like C4) displayed stronger seasonal variations in MTTgeo, with shorter MTTgeo than till dominated sub-catchments (like C2) in spring and longer MTTgeo than C2 in winter (Table 5). In C4, the MTTgeo reduced from 1.5 to 0.7 years from winter to spring, while the corresponding change in C2 was 1.2 to 0.7 years. The seasonality of MTTgeo was even more pronounced for catchments with a larger areal coverage of mires combined with a larger areal coverage of silt. For example, C20 had an MTTgeo that reduced from 7.7 years to 1.9 years from winter to spring, compared (Table 5).

Line 390: 'strong statistical significance' or 'significant statistical correlation'?

The overall gMTT always had a strong statistical significance to the BC concentration (Fig. 6 b, d, and e), generally agreeing with our conceptual model (Fig. 2b).

**Reply**: We have clarified this sentence to:

The MTTgeo had a significant correlation to the BC concentration during all seasons (Fig. 7 b, d, and e), again agreeing with our conceptual model (Fig. 2b).

**Discussion**

I would wish for a careful revision of the style and structure used throughout the discussion. Often there is a lack of cohesion between the individual statements and it is not explained why statement a) would support statement b). One example of this can be found at lines 471-473.

According to the conceptual model (Fig. 2), older baseflow water should result in an isotopic signature closer to the precipitation average. There was a strong negative correlation between groundwater age and the streams isotopic signature during baseflow (Fig. 6a), suggesting that the model produces credible water age patterns for the winter season. The larger sub-catchments, including C14, C15, and C16, are close to the long-term precipitation average, which limits the ability to estimate the travel times using isotopes. Water older than 4-5 years is argued not to be accurately quantifiable using isotopes due to amplitude loss (Kirchner., 2016). These theoretical considerations strengthen our results of a winter MTT of 4-6 years for the larger sub-catchments and provided new insights into travel times for these systems.

**Reply:** We have tried to restructure the discussion to get a better and easier to follow structure and style. Regarding the example, we changed this paragraph to:

Following the conceptual model (Fig. 2), patterns in stream isotopic signatures can be explained by seasonal changes in travel times. The modelling results show that all sub-catchments discharged the oldest water in winter, somewhat younger water in summer, and water of the youngest age in spring. When winter arrived, the main precipitation was snow resulting in that groundwater recharge effectively ceased, which caused an increasing proportion of old groundwater discharging into the streams (Fig. 6). In agreement with our conceptual model (Fig. 2), a strong negative correlation between winter MTTgeo and the isotopic stream signatures during winter baseflow was observed (Fig. 7a). At an average water age older than four years, it can be expected that the groundwater has reached full mixing. Hence, older water can no longer be accurately quantified using water isotopes only due to amplitude loss (Kirchner., 2016). These theoretical considerations strengthen the results of a winter MTTgeo between four and six years for the larger sub-catchments as their stream isotopic signatures were close to the long-term precipitation average and, therefore, should have reached complete mixing.

Line 430: '…for the MEAN water age…'.

**Reply:** We changed this sentence accordingly.

Line 437: I would add the acronym of the model for better recognition (STARR or something like that).

**Reply:** We changed this sentence accordingly.

Line 450-459: I understand your reasoning. However, especially in catchments where the soils have low conductivities, the difference between groundwater TTs and overall TTs could be quite large. Can you at least try to give some estimations on how long the percolation through the different soils to the groundwater table would take?

**Reply:** The hydraulic conductivity in the unsaturated zone varies with level of saturation, which in turn varies every time step. This makes it difficult to estimate travel times for the unsaturated zone. We therefore focused on the groundwater recharge and saturated zone flow and not unsaturated flow. Given that the distance between the ground surface and the groundwater table is so short compared to the average distance from a cell in the model to the nearest stream we would expect that most of the transit time in the soil is spent in the saturated zone. However, there are no doubt water that also can spend considerable time in the unsaturated zone, but when looking at the average transit time we must also acknowledge that the volume of water in the unsaturated zone is small compared to the groundwater compartment. Therefore, we would envisage that this limitation in the particle tracking in most cases should not have dramatic consequences for the modelled transit times. We also added a discussion part about this in the discussion section 4.1.

Line 472: How does the negative correlation suggest that the model produces credible patterns?

According to the conceptual model (Fig. 2), older baseflow water should result in an isotopic signature closer to the precipitation average. There was a strong negative correlation between groundwater age and the streams isotopic signature during baseflow (Fig. 6a), suggesting that the model produces credible water age patterns for the winter season.

**Reply:** Please see the first answer for the discussion section

Line 484: 'less' is not the correct descriptor for a difference.

The larger young water fraction was generally found in mire dominated sub-catchments, such as C4 and C5. In contrast, equally sized sub-catchments without mires, such as C1 and C2, had a less $\Delta\delta O_{18spring}$ and hence smaller young water fraction.

**Reply:** We agree that this is not a correct descriptor. However, the specific sentence was removed.

Line 489: 'biogeochemical consequences'?

Those earlier results showed a large overland flow component in wetland catchments because of frozen conditions during spring flood with biogeochemical consequences during snowmelt.

**Reply:** This section was changed and biogeochemical consequences, is no longer used in the text.

Line 491: In general, try to be more precise. Instead of writing: 'Inter-annual precipitation and evapotranspiration variability likely caused the relationship to be less evident…', better write 'A larger inter-annual precipitation and evapotranspiration variability likely caused the relationship to be less evident…'. This also applies to the variable names – one time you write 'the summer $\Delta\delta O18$', another time you write '$\Delta\delta O18summer$'. Be more consistent.

In summer, the conceptual model predicted that $\Delta\delta O18summer$ should also be correlated to the young water fraction, but with the opposite sign, due to the enriched summer rain (Fig. 2). Inter-annual precipitation and evapotranspiration variability likely caused the relationship to be less evident compared to the spring flood results as the snowmelt conditions are more consistent from year to year. However, although less strong than compared to the spring $\Delta\delta O18$, there was still a strong connection between the average summer $\Delta\delta O18$ and the modelled young water fraction (Fig. 6e).

**Reply:** We agree to these changes. We have applied them to the section in question and checked the naming of $\Delta\delta^{18}O_{summer}$ I the entire manuscript.

Similar to the conditions in spring, the conceptual model predicted that the difference in stream isotopic signature between winter baseflow and summer flow, $\Delta\delta18Osummer$, should be correlated to the young water fraction in summer, but with the opposite sign, due to isotopically heavier summer rains (Fig. 2). A larger inter-annual variation in precipitation and high ET likely caused the relationship to be less evident compared to the spring results as the snowmelt conditions are more consistent from year to year. However, although less evident than compared to the $\Delta\delta18Ospring$, there was still a significant correlation between the average $\Delta\delta18Osummer$ and the modelled young water fraction (Fig. 7e).

Line 525: Again, more precision in how you describe your results is necessary: It is 'the fraction of low conductive soils' and not just 'low conductive soils'.

**Reply:** We have checked the manuscript for this error and have changed it accordingly.

Line 550: They don't 'receive', they 'produce'. Also, what exactly is the 'annual age of water'?

Sub-catchments with mires receive the highest young water fraction during the spring snowmelt; however, the annual age of water is not as strongly connected to that landscape feature (Table 6).

**Reply:** We have checked the manuscript for the word receive and changed it to produces whenever it seemed appropriate. We agree with the specific line change, however, the specific sentence was removed in the new version.

Line 555: A 'similar annual age'? Please be more precise.

**Reply:** This specific sentence was removed in the new revision of the manuscript. However, it was meant to be the mean annual age expressed as $MTT_{geo}$, Table 6.

Line 558: Does it displace the older water or is it just added to the older water (and mixed)? One would cause the stream water to first become very old and then very young, the other one would cause the stream water to become somewhat younger on average.

Line 559: What you describe is not called 'transmissivity feedback'. Transmissivity feedback describes the process that discharge from hillslopes increases over-proportionally when the storage in the hillslope increases - due to the fact that soil layers closer to the surface have higher hydraulic conductivity. Sloppy and makes me question the general understanding of the processes taking place.

**Reply:** Line 558-559: We understand that this section needed to be restructured, since the message was lost. As regards the "transmissivity feedback" we believe we fully agree with the reviewer on what it means. We regret that we expressed ourselves in a manner that led the reviewer to think otherwise and hope that this matter will be clearer in the revised manuscript. We hope the new section is clearer:

Earlier studies have demonstrated that fluxes of old groundwater are more stable throughout the year than younger groundwater showing a more variable temporal pattern (Rinaldo et al., 2011; van der Velde et al., 2015; Kaandrop et al., 2018). In our system, such a pattern can mechanistically be linked to till soils dominating most sub-catchments, where the groundwater response to precipitation events can be described by transmissivity feedback processes (Bishop, 1991), caused by the fact that the hydraulic conductivity increases exponentially towards the soil surface. When water infiltrates the ground, the water table rises and activates more conductive soil layers, resulting in rapid increases in the lateral flow. This implies that much of the water transport in till soil occurs relatively close to the surface, while the groundwater in deeper layers is more stagnant, which further explains the relatively short and consistent MTTs of till soils. Measurements of chlorofluorocarbons (CFCs) further support that deeper groundwater water transport in till soils in Krycklan is slow. Not far below the groundwater table CFCs have indicated that the groundwater can be several decades old, suggesting that most of the groundwater transport occurs close to the surface (Kolbe et al., 2020). Consistent with this explanation, silt dominated areas, that have more consistent hydrological conductive with soil depth, had much longer MTTs than comparatively sized sub-catchments underlain by till soils (Fig 6, Fig. 8 and Appendix Fig. A1).

**Figures:**

Figure 1: Catchment topography is shown in panel (d) – not (e).

**Reply**: Thank you for noticing this error. The error was corrected.

Figure 2: (a) The use of the symbol ∝ is incorrect here. It means 'proportional to'. The difference in the isotopic signature can only be proportional to the fraction of young water. Also I think you are missing a Δ before baseflow MTT. (b) You haven't explained yet what gMTT is. Also, there should not be a line for the precipitation BC concentration (it is not connected to any MTT), you can instead mark it on the y-axis.

**Reply**: We have changed this figure according to both the editor and reviewer 1# suggestions. The new figure (a and b) is displayed below. The main changes are, ∝ have been removed, MTTgeo was removed and changed to groundwater age, and the line for precipitation was removed for clarity. We also added a description of $\delta^{18}O$ and BC directly in the figure, instead of mainly having it in the figure caption

[Figure]

Figure 3: Better write '(a) Steps of the particle tracking'. In panel (b) write 'Average depth to the groundwater table'. I don't understand what you want to say here: 'Note that the top vertical layering of the saturated zone was set to 2.5 m below the ground surface, and the thickness thereafter follows the soil layers (thickness increasing with depth).' Can you add another panel to the figure to show what you mean?

**Reply:** We changed the figure and caption according to the reviewer's suggestions. The text in question was, however, removed from the caption and better described in the method text.

Figure 4: The figure shows the 'age' of the particles reaching the outlet, not the 'timing' as you write in the caption.

**Reply**: We agree that "timing" should be changed to "age"

**Tables**

Table 1: What are the numbers in parentheses in the 'Sorted sediments' column?

**Reply**: Thank you for noticing this error. Some text has disappeared in an earlier iteration of the manuscript. We have added this yet again to the table text:

**Table 1: Sub-catchment characteristics.** The list includes all 14 monitored sub-catchments in Krycklan, called C1 to C20, including the entire Krycklan catchment, C16. Different branches of the stream network are illustrated in distinct colours in Fig. 1. The table includes the sub-catchment area, average elevation, and average slope. Further descriptions of these characteristics can be found in Karlsen et al. (2016). The table also includes soil proportion based on the soil map (1:100,000) from the Swedish Geological Survey (2014).

Table 2: One row just reads: 'Soil frost'. What does that mean?

**Reply**: This part of the table was removed and written as text in the method section for clarity. We applied soil frost on the mires since it was an important part of the hydrological functioning of the catchment, which mean reduced horizontal flow and infiltration at freezing temperatures and at the start of the snowmelt.

Table 5: Is aMTT the arithmetic mean of the 'particle tracking (groundwater gMTT)' or of the 'particle tracking adjusted for overland flow according to Eq. (4)'? Please clarify.

**Reply**: We apologize. The explanation for these was stated below the table. We found that this information easily was missed when looking at the table. Therefore, we moved it to above the table. We also made the table smaller to just include the most important parts discussed in the manuscript. However, all data are still available in the Appendix.

Table 6: I guess something went wrong in the formatting of the caption 'according to a p>0.05, b a p<0.05, and c p<0.01'.

**Reply**: Thank you for noticing this error. It has been corrected.

An idea: Since the upper two rows of each table are identical, why not make one bigger table out of the four smaller tables? Something similar to the attached table (or transposed).

**Reply:** Wee agree that this table could be smaller, and we like the suggestion from the reviewer. We used some of the ideas and remade the table:

**Table 6: Correlation matrix – young water fraction (YWF), geometric mean travel time (MTT$_{geo}$), and catchment characteristics.** The catchment characteristics include the log catchment size, km$^2$ (Log C.-size), the areal coverage of mires, and the areal coverage of silt. The table includes annual (grey), winter (blue), spring (green), and summer (orange) results. Darker colours show $|r| > 0.5$ with the connected p-value according to [a] $p<0.05$ and [b] $p>0.05$.

| | Winter season | | | | | Summer season | | | | |
|---|---|---|---|---|---|---|---|---|---|---|
| | Log C.-size | Mire (%) | Silt (%) | MTT$_{geo}$ | YWF | Log C.-size | Mire (%) | Silt (%) | MTT$_{geo}$ | YWF |
| Log C.-size | 1 | | | | | 1 | | | 0.91 [a] | |
| Mire (%) | | 1 | 0.92 [a] | | 0.64 [a] | | 1 | 0.80 [a] | -0.50 [b] | 0.68 [a] |
| Silt (%) | 0.58 [a] | | 1 | | 0.58 [a] | 0.58 [a] | | 1 | | 0.58 [a] |
| MTT$_{geo}$ | 0.63 [b] | -0.51 [b] | 0.90 [a] | 1 | | 0.55 [a] | -0.55 [a] | 0.92 [a] | 1 | |
| YWF (%) | | 0.96 [a] | | -0.53 [b] | 1 | | 0.95 [a] | | -0.52 [b] | 1 |
| | Annual | | | | | Spring season | | | | |

**Appendix**

Line 835-837: What is 'ai'?

**Reply**: We added this text for clarification: whereas: ai= data set values, n=number of values.

Line 841: 'is Krycklan'?

**Reply**: This has been corrected to: "in Krycklan"

**Supplements**

The supplement was supposed to be deleted, I guess. It is still available with the manuscript though. Don't forget to delete it, otherwise readers will be confused.

**Reply**: Yes, we chose to remove the supplements and have all necessary information in the manuscript and the Appendix. We thank the reviewer for pointing it out and we contacted the editorial support team about this issue. They in turn told us not to upload a new version of the

supplements. Hopefully, this issue will therefore be resolved for the next version of the manuscript.

**Reply to reviewer #2**

**General comments:**

I would like to commend the authors on the extensive revisions they undertook in response to the two reviews in the first round. The most significant concern raised by myself, and the other reviewer, was that the organization around the catchment size hypothesis lacked novelty. I commend the authors for taking this comment to heart and significantly revising the text to re-focus their analysis on the aspects of their study that are more novel: (1) their unique study location and (2) the ability to explore seasonal differences in MTTs within this setting. I think this change along with the expanded literature discussion greatly improved the manuscript. And I agree with the authors that this new focus brings much more novelty to the study.

**Reply:** We thank reviewer #2 for his/her time and effort to read, comment, critique and discuss our manuscript. We are delighted to read that reviewer #2 agrees with most changes of the manuscript compared to its earlier state, especially with regards to the new study focus. The remaining concerns from reviewer #2 has been commented on below (brown). Text from the new manuscript was marked red. We hope that the new revised manuscript and the answers to all reviewers and editor concerns and remaining critiques will be sufficient for publication.

I also appreciate the addition of Figure3. This makes the approach much more clear and addresses some of my concerns about the confusion of the methodology.

**Reply:** Thank you for the comment, and we agree that the Figure 3 (old manuscript) made the method section clearer.

My only remaining concern is in the description of the MIKE-SHE model itself. While I realize that most of this is published in other locations, I was hoping that the authors would provide a better description of how the model is actually working -- ie. how physical processes and interactions are represented. Section 2.2 was modified but is still mostly about the parameterization of the layers. I think this section should be updated to provide a more comprehensive overview of the model for those who are not familiar.

**Reply:** We agree that a description of a more in-depth description of Mike SHE could be useful. This was also addressed by reviewer 1# and the editor. The section about the Mike SHE model was therefore re-written in the updated version of the manuscript.

Also a more detailed comment on this section: This description of the upper most soil layer thickness (line 175) is confusing and doesn't match with the table where it looks like this is only true in the case of till? Sandy sediments appear to have a thickness of 0.8 for the top layer? Can you clarify this?

**Reply:** We agree that this section was a bit confusing. Therefore, we simplified the table and changed the text in the method section of Mike SHE. We hope that these changes will make it clearer. As stated in the new version of the manuscript:

For the Krycklan model, the horizontal grid was set to 50*50 m. Vertically, the model is divided into ten calculation layers (CL) and extends to a depth of 100 m below ground. The CLs of the SZ vary with depth and are thinner closer to the soil surface; the first CLs extend to 2.5, 3, 4, and 5 meters below the ground surface, with the soil properties and depth extension following the stratigraphy (Table 3). The ET and UZ processes are only fully active in the uppermost SZ-CL, and here the ET and UZ are calculated at a finer resolution, leading to a detailed calculation of the groundwater table level. If the groundwater table falls below the first SZ-CL, a more simplistic method, not taking capillary rise and all ET-processes into account, was applied. The depth of the first SZ-CL was set to 2.5 m and was calibrated using the influence of the CL thickness on groundwater table level, UZ, and ET dynamics…

---

## Author Response (AR3)

**Answer to Editor**

Comments to the Author:

Dear Elin and co-workers,

I have received a review for your manuscript, which I have little to add to. I have to agree with the referee that we are now actually on track to make it a decent publication and that the review process did cover quite a bit of the work that could be expected within your team, indeed. Please do not overlook the ironic tongue in some of the comments.

To avoid further complication I refrain from adding my own comments or a second review. Please carefully consider the referee's comments for another round of revisions including the overall logical coherence. I suspect some of the remaining confusion to be easily sorted when the respective logical lines of foci are strengthened.

I also encourage you, to really seek the expertise within your group of co-authors. I have full confidence that you will succeed and that your team has excellent capacities to make your paper a nice contribution to our science.

I look forward to receive your revised manuscript. All the best.

Conrad

**Answer:** Thank you for your consideration of our manuscript to Hydrology and Earth System Sciences (HESS). We have, as you suggested, gone through all the comments from the referee and especially looked into the comments regarding the structure and logical order of the text. We hope that the new version of the manuscript will be to your satisfaction. Our reply is structured as follows: answers and changes in brown and text from the original manuscript in blue**.**

**List of major changes**

The major changes to this version of the manuscript include:

- The title was changed to "How catchment characteristics influence hydrological pathways and travel times in a boreal landscape" based on the comments from the reviewer.
- Made several structural and clarifying changes of the text following comments from the reviewer
- Added information in the method section regarding the construction of the travel time distributions of Figure 5
- The order of catchments in the tables is now colour based.

**Ref 1**
**General comments:**

The authors did a good job in improving the readability and overall quality of the manuscript in the last iteration. The authors' response was considerate and explained the reasoning behind all relevant changes and decisions. The fact that I still have a lot of questions and comments is not a bad sign because I see that my former comments were appreciated and helped in improving the paper. Therefore I am sure that the authors will also be able to use the new review to the benefit of the manuscript.

**Reply:** Thank you for your constructive criticism, suggestions, and questions. We are very thankful for the time and effort to read and comment on our manuscript, and we hope that you will be pleased with our changes and answers to the next version. We have answered all your questions in brown below. Blue text marked "Original text" was taken directly from the previous version of the manuscript. All reviewer and editor questions and comments are stated unbridged in black, and our reply and changes are stated below each question/comment.

**Main question**

My main questions that need attention concern the methods section. Some important details are still unclear to me:

1. For constructing the TTDs you show in Figure 5 do you take all the particles arriving in a stream at the catchment outlet over the entire modeling period? So these are not individual backward TTDs for a single moment in time? They are, however, similar to flow weighted averages of all individual backward TTDs, is that right? Accordingly, your annual and seasonal MTTs are flow-weighted, right? You take all the particles arriving in a stream within a certain period of time (e.g. spring) and take the average of the distribution.

**Original text:** Section 2.5 The time it took for particles to reach a stream or lake via groundwater (hereafter called 'travel time') was calculated for each sub-catchment both annually and for each season. The calculated travel time distributions were analysed using four statistical measurement tools, the arithmetic mean, the geometric mean, median, and the standard deviation (SD).

**Original text:** Figure 5: The figure shows the age of particles reaching sub-catchment outlets….

**Reply:** We like to answer these questions together because they handle the same issue. Figure 5 shows the distribution of all the particles exiting the model though a catchment stream over the entire modelling period. To clarify this, we changed the figure text to state that the statistics are based on all particles over the entire modelling period. We also added a part in section 2.5 to clarify the creation of the travel time distributions. It is correct that we take all the particles exiting a stream within a certain period of time and take the average of that distribution.

**Text changed to:** Section 2.5: The time it took for particles to reach a stream or lake via groundwater (hereafter called 'travel time') was calculated for each sub-catchment. The calculated travel time distributions were based on all particles arriving in a stream within a certain period of time, either annually or for a specific season, for the entire modelling period. The distributions were analysed using four statistical measurement tools, the arithmetic mean, the geometric mean, the median, and the standard deviation (SD).

**Text changed to:** Figure 5: The figure shows the distribution of all particles reaching the different streams for the entire modelling period….

2. How exactly do you account for the reduction of travel times due to overland flow? Please show the equation.

**Reply:** As we stated in the discussion and the method section, we reduced the age using the amount of overland flow as a scaling factor. We showed the equation in previous iterations of the manuscript, but we received comments that it was not necessary to show because it's such a simple equation. However, we agree that showing the amount of overland flow for different seasons would be a good addition. We added this in the Appendix, Table A1. We also added the equation in the Appendix (Eq. A6).

**Text changed to:** $\text{MTT} = \text{MTT}_{Particle\ tracking}\ (1 - \text{fraction OL})$         Eq. A6

3. Please make sure to add all this relevant information to your methods section.

**Reply:** We hope that these additions to the method section and Appendix will clarify and remedy these concerns.

**Specific comments:**

**Title**

1. It includes everything. Maybe a focus on the most important issue would be better. Is isotopic composition not part of the stream chemistry? For example, a more logical title would be: How catchment characteristics influence flow pathways, travel times and stream chemistry in a boreal landscape

**Reply:** Thank you for this suggestion. We understand the title concern that it should be more focused, and we liked the suggestion. Therefore, we changed it, but we adjusted it a bit further to:

**Text changed to:** The title now reads "How catchment characteristics influence hydrological pathways and travel times in a boreal landscape"

**Key points**

1. The last key point is confusing. The greatest seasonality is observed in silty soils because they contribute a larger amount of old water during winter baseflow compared to other soils? And why do you mention the mires in the same key point? Are they also related to the seasonality in catchments with silty soils?

**Original text:** The greatest seasonality in mean travel times was found in sub-catchments dominated by silty soils contributing with old water during winter baseflow, while mires contributed the largest fraction young water during spring snowmelt.

**Reply:** We have changed this key point for clarification.

**Text changed to:** The greatest seasonality in mean travel times was found in catchments dominated by silty soils because of the long travel times in winter and relatively short in spring.

**Abstract**

1. Line 25: '…the pathways and associated travel times of water in 14 partly nested…'. First you use the model to investigate the pathways. Each of the pathways has an associated travel time. This is the logical order you could follow throughout the manuscript.

**Original text:** In this study, a particle tracking model approach in Mike SHE was used to investigate the travel time, and pathway of water in 14 partly nested, long-term monitored boreal sub-catchments.

**Reply:** We agree that this is the logical order to follow and changed the mentioned text accordingly, and we have also checked other parts of the manuscript for this logical error. Please see our change below:

**Text changed to:** In this study, a particle tracking model approach in Mike SHE was used to investigate the pathway and its associated travel time, of water in 14 partly nested, long-term monitored boreal sub-catchments (0.12-68 km2).

2. Line 29: Which seasonal changes? Meteorological changes?

**Original text:** The variations were found to be related to the distribution of different landscape types and their different response to seasonal changes.

**Reply:** We agree that this sentence should be clarified. We changed it to:

**Text changed to:** The variations were related to the distribution of different landscape types and their varying hydrological response during different seasons.

3. Line 31: By 'groundwater age' you mean what exactly? The residence time of the groundwater, the travel time of the groundwater to the stream? The age of the fraction of streamflow fed by groundwater?

**Original text:** The groundwater age was positively correlated to the areal coverage of low conductive silty sediments

**Reply:** We agree that the use of groundwater age was confusing here. We changed this sentence accordingly.

**Text changed to:** The travel time of water to streams was positively correlated to the area coverage of low conductive silty sediments

**Introduction**

1. Line 41-45: Here you introduce the terms 'age', 'residence time' and 'travel time' somewhat interchangeably and without proper explanation. Also, I would mention flow pathways first (they cause/control the travel times and not the other way around). For example: 'The pathways of water through the terrestrial landscape to stream networks and associated age is a…'.

**Original text:** The age and pathway of water through the terrestrial landscape to stream networks is a widely discussed topic in contemporary hydrology.

**Reply:** We agree with the suggestion and have changed the sentence accordingly. This was also mentioned in point 7 of the method questions, where we answered more thoroughly and checked the whole manuscript for these types of inaccuracies.

**Text changed to:** The pathways and associated travel times of water through the terrestrial landscape to stream networks is a widely discussed topic in contemporary hydrology.

2. Line 55: Only if you assume that the groundwater is fully mixed. If not, even groundwater can still be associated with certain hydrologic events.

**Original text:** Stream water consists of a blend of overland flow and groundwater of different ages. The mean travel time (MTT) to streams is calculated as the average age of this mix (McGuire et al., 2006). The baseflow is the part of stream groundwater contribution that is not linked to a specific hydrological episode.

**Reply:** We agree the sentence was a bit unclear. We changed it as follows:

**Text changed to:** Stream water consists of a blend of overland flow and groundwater of different ages. The mean travel time (MTT) to streams is calculated as the average age of this mix (McGuire et al., 2006). The baseflow is the part of stream groundwater contribution that generally has travelled the furthest and is the oldest (Klaus et al., 2013; Hrachowitz et al., 2016). Specific hydrological episodes become harder to distinguish the older and the more well-mixed the baseflow is. In contrast, young stream water is typically connected to overland flow or fast and shallow groundwater pathways, which mainly can be seen at times with large rain or snowmelt inputs…

3. Line 56-57: You should at least mention that a large fraction of young stream water can also be derived from shallow subsurface flow in the soil or at the soil bedrock interface (some call this 'interflow').

**Original text:** In contrast, young stream water is typically connected to overland flow or fast and shallow groundwater, which mainly can be seen at times with large rain or snowmelt inputs

**Reply:** We do not understand this question, since we believe we already mentioned shallow groundwater flow paths, but we tried to make the text clearer.

**Text changed to:** In contrast, young stream water is typically connected to overland flow or shallow subsurface pathways, which mainly can be seen at times with large rain or snowmelt inputs.

4. Line 65: It is the dampening of the isotopic tracer signal that provides MTT estimates.

**Original text:** Isotopic tracer dampening can provide an estimate of MTT

**Reply:** We changed this sentence to make it clearer.

**Text changed to:** Isotopic tracer signal dampening can provide an estimate of MTT

5. Line 68: Precipitation-discharge relationships are not similar to the mentioned gamma distribution transfer functions. I would leave them out of this argument.

**Original text:** Theoretical transfer functions, such as the gamma distribution model, can also be used by relating input and output signals of isotopes, such as precipitation-discharge relationships

**Reply:** We agree to leave it out of the argument and decided on removing the sentence completely.

6. Line 86: How are 'isotope models' a method to calculate travel times? I understand how particle tracking can be used to calculate TTDs but 'isotope models'? Maybe you are talking about models that use solute transport routines (assuming conservative non-reactive tracers) to trace water pulses through a catchment system? Fitting references would include Remondi et al. (2018) and Heidbüchel et al. (2020).

**Original text:** Two common methods to calculate travel times using numerical methods includes isotope models and particle tracking (Hrachowitz et al., 2013; Ameli et al., 2016; Kaandorp et al., 2018, Yang et al., 2018).

**Reply:** Thank you for the reference suggestions. Using your advice, this sentence was changed to:

**Text changed to:** Common methods to calculate travel times using numerical methods includes models using solute transport routines, and particle tracking (Hrachowitz et al., 2013; Ameli et al., 2016; Kaandorp et al., 2018; Remondi et al., 2018; Yang et al., 2018; Heidbüchel et al. 2020).

7. Line 117: Age distributions or travel time distributions? Please be consistent in your terminology.

**Original text:** The main objective of this study was to quantify annual and seasonal travel time distributions and calculate MTT of water runoff to streams of the Krycklan sub-catchments to disentangle how these are related to physical landscape characteristics and seasonality.

**Reply:** For clarification, this sentence was changed to:

**Text changed to:** The main objective of this study was to quantify annual and seasonal (winter, spring, and summer) travel time distributions and calculate MTT of water runoff to streams of the Krycklan sub-catchments to disentangle how these are related to physical landscape characteristics and seasonal variation of groundwater recharge…

8. Line 118: Seasonality in what? Temperature? Precipitation amount? Weather in general?

**Original text:** The main objective of this study was to quantify annual and seasonal travel time distributions and calculate MTT of water runoff to streams of the Krycklan sub-catchments to disentangle how these are related to physical landscape characteristics and seasonality.

**Answer:** We changed this sentence to include an explanation to your question:

**Text changed to:** The main objective of this study was to quantify annual and seasonal (winter, spring, and summer) travel time distributions and calculate MTT of water runoff to streams of the Krycklan sub-catchments to disentangle how these are related to physical landscape characteristics and seasonal variation of groundwater recharge…

9. Line 119: How do you compare calculated travel times to 10-year seasonal isotope signatures? Do you compare the relative damping of the signatures for different years to the calculated travel times?

**Original text:** Firstly, the credibility of the model results was tested by comparing calculated travel times for the 14 sub-catchments to ten-year seasonal isotope signatures and base cation concentrations record from the Krycklan network.

**Reply:** We changed this sentence slightly to be more precise but kept it short since this part belongs to the introduction.

**Text changed to:** Firstly, the credibility of the model results was tested by comparing calculated travel times for the 14 sub-catchments to ten-year observational records from Krycklan, including average seasonal changes of stream isotope signatures and base cation concentrations

**Method**

1. Line 169: What is a 'rapid transition in hydrology'?

**Original text:** Similarly, we defined spring by the rapid transition in hydrology and biogeochemistry in April-May

**Reply:** We changed this sentence for clarification

**Text changed to:** Similarly, we defined spring as the hydrological period directly influenced by the snowmelt, in April-May.

2. Line 177: 'measurements' instead of 'results'.

**Original text:** Ten years of δ18O results for 13 of the 14 sub-catchments.

**Reply:** Thank you for noticing this error, we have corrected it.

**Text changed to:** Ten years of $\delta^{18}$O measurements for 13 of the 14 sub-catchments

3. Line 177-179: Comprehensibility would benefit enormously from a more concise writing style. This sentence is a good example: Instead of writing a long and convoluted sentence like this one: 'Some of the sub-catchments are affected by evaporation from lake surfaces that result in isotopic fractionation that, consequently, affected the signal (Leach and Laudon, 2019)'. Better write: 'Isotopic fractionation caused by lake surface evaporation affects the isotopic signal of some of the sub-catchments (Leach and Laudon, 2019).

**Reply:** We checked the text for more example like this and tried to remedy as many as we could find. In this specific case, we used the suggested sentence in the new manuscript.

4. Line 179: '…was corrected by accounting for the percentage…'

**Original text: …**was corrected for the percentage of …

**Reply:** We agree with this change.

**Text changed to:** …was corrected by accounting for the percentage…

5. Line 183: This is not a logical conclusion. Just because there is no groundwater recharge during winter this does not mean automatically that the stream is only fed by groundwater only. It could still be fed by overland flow from meltwater or from rain on snow...

**Original text:** The comparison of the modelling results to observations of δ18O was based on a conceptual model of the seasonal variability and differences between precipitation and runoff (Fig. 2a). Because there is no or very little groundwater recharge during winter because almost all precipitation inputs arrive and accumulates as snow, we assume the stream isotopic signature originates from groundwater only (Laudon et al., 2007). Hence, we assumed that δ18O during winter baseflow should be statistically related to the average travel time of the groundwater to the streams (up until the point where full mixing is reached)

**Reply:** It is true that meltwater or rain on snow could still feed the streams with water. However, we chose the winter season, specifically when the temperature is below zero, resulting in very little to no meltwater and unfrozen precipitation. We have tried, unconvincingly till now, to make the argument that the setting of this site with long and cold winter without much mid-winter melting and rain on snow allows making the case we have made. The text was changed as follows:

**Text changed to:** The comparison of the modelling results to observations of δ18O was based on a conceptual model of the seasonal variability and differences between precipitation and runoff (Fig. 2a). The precipitation signal varies on a seasonal basis, creating an amplitude difference. This amplitude is reduced due to groundwater mixing until complete mixing is reached and the groundwater receives the same signal as the long-term precipitation average. There is no or little groundwater recharge during winter because almost all precipitation inputs arrive and accumulates as snow. Hence, we assume that the stream isotopic signature originates from groundwater only (Laudon et al., 2007; Peralta et al., 2015). Consequently, the closer the stream signature comes to the long-term precipitation average, the more the groundwater has been mixed. The groundwater isotopic signature, in turn, should be correlated to the travel time to stream until full mixing of the precipitation signal is reached.

6. Line 184: What aspect of the isotope (measurements? time series?) should be statistically related to average age of the groundwater?

**Original text:** The comparison of the modelling results to observations of $\delta^{18}$O was based on a conceptual model of the seasonal variability and differences between precipitation and runoff (Fig. 2a). Because there is no or very little groundwater recharge during winter because almost all precipitation inputs arrive and accumulates as snow, we assume the stream isotopic signature originates from groundwater only (Laudon et al., 2007). Hence, we assumed that $\delta^{18}$O during winter baseflow should be statistically related to the average travel time of the groundwater to the streams (up until the point where full mixing is reached)

**Answer:** We have changed this section to include an answer to this question. The closer stream signature during winter baseflow comes to the long-term precipitation average, the more the groundwater has been mixed and should be correlated groundwater travel time to streams. This assumption is also supported by that the long-term isotopic signature is very close to the signature of the deep groundwater (old water) in Krycklan.

**Text changed to:** The comparison of the modelling results to observations of δ18O was based on a conceptual model of the seasonal variability and differences between precipitation and runoff (Fig. 2a). The precipitation signal varies on a seasonal basis, creating an amplitude difference. This amplitude is reduced due to groundwater mixing until complete mixing is reached and the groundwater receives the same signal as the long-term precipitation average. There is no or little groundwater recharge during winter because almost all precipitation inputs arrive and accumulates as snow. Hence, we assume that the stream isotopic signature originates from groundwater only (Laudon et al., 2007; Peralta et al., 2015). Consequently, the closer the stream signature comes to the long-term precipitation average, the more the groundwater has been mixed. The groundwater isotopic signature, in turn, should be correlated to the travel time to stream until full mixing of the precipitation signal is reached.

7. Line 184: Are you aware of the fact that the average age of the groundwater is the mean residence time and not the mean travel time of the water reaching the stream? They can be very different. You need to be very careful with the terminology you are using. Please go through the manuscript and check each occurrence of the words 'age', 'MTT', 'travel time', 'residence time'. Don't use two terms for one and the same concept – like 'average streamflow age' and 'MTT' or 'average groundwater age' and 'mean residence time'.

**Reply:** We agree that these concepts were mixed up at some places in the text. We searched for them and corrected the issues we found. This is how we used the concepts:

- MTT – mean travel time, the mean of the travel time distribution based on the model ages of all the released particles reaching the stream. Particles are "born" when introduced into the model and receive an increasing age with time.
- Travel time – in this manuscript, it is the time it takes a particle to reach a stream, i.e., equal to the end age of a particle. This could also be water soil contact time of water discharging to streams
- Residence time – the average time a particle (or in this case water molecule) spends within a system. We only look at the water that exits the system via streamflow. Therefore, residence time should not be used in our case. We have removed the instances where this have been used.
- Streamflow and groundwater age – We agree that the use of these in the text is misleading. We do not look at the age of stream water, nor the age of the groundwater. We look at the travel time of water to the streams. We have checked the manuscript for such errors and removed them.

8. Line 185: Up until the point where full mixing is reached. Full mixing of what?

**Original text:** The comparison of the modelling results to observations of δ18O was based on a conceptual model of the seasonal variability and differences between precipitation and runoff (Fig. 2a). Because there is no or very little groundwater recharge during winter because almost all precipitation inputs arrive and accumulates as snow, we assume the stream isotopic signature originates from groundwater only (Laudon et al., 2007). Hence, we assumed that δ18O during winter baseflow should be statistically related to the average travel time of the groundwater to the streams (up until the point where full mixing is reached)

**Answer:** The precipitation signal varies on a seasonal basis, creating an amplitude difference. This amplitude reduces due to mixing until complete mixing is reached and the groundwater gets the same

signal as the long-term precipitation average (which also is the same as the old and deep groundwater). The difference in $\delta^{18}$O between winter baseflow and the long-term average of the precipitation should therefore be linked to the groundwater travel time to streams.

**Text changed to:** The comparison of the modelling results to observations of δ18O was based on a conceptual model of the seasonal variability and differences between precipitation and runoff (Fig. 2a). The precipitation signal varies on a seasonal basis, creating an amplitude difference. This amplitude is reduced due to groundwater mixing until complete mixing is reached and the groundwater receives the same signal as the long-term precipitation average. There is no or little groundwater recharge during winter because almost all precipitation inputs arrive and accumulates as snow. Hence, we assume that the stream isotopic signature originates from groundwater only (Laudon et al., 2007; Peralta et al., 2015). Consequently, the closer the stream signature comes to the long-term precipitation average, the more the groundwater has been mixed. The groundwater isotopic signature, in turn, should be correlated to the travel time to stream until full mixing of the precipitation signal is reached.

9. Line 187: But now imagine a year where the average input signal is equal to the average annual precipitation signal (that can also be found in the deeper groundwater). In this case you cannot draw a conclusion about how old the winter groundwater signal is because it will be close to the long-term average no matter if there is more young recently recharged or more old groundwater in the mix. Maybe you can use the variability in the winter stream isotope signal. But I would like to see a better explanation on why this method would work.

**Reply:** In Krycklan we have approximately six months of accumulating snowpack during winter and only a few instances of precipitation falling as rain nor snowpack melting. Of course, it can occur, but it's rare, and one can assume that during these months no recharge occurs to the groundwater. We hope that the changed text below will better describe our conceptual model (Peralta et al., 2015). Our concept builds on:

1. We know from empirical evidence that deep groundwater (< 10 m soil depth) is equal to long-term average precipitation inputs.

2. All recharge that has occurred the month after the last spring snowmelt and before the winter is isotopically heavier.

3. The stream mix we measure during winter is hence primarily a combination of deep groundwater and more recent summer/autumn precipitation water. Hence, we can assume that the deviation between long-term average precipitation and winter streamflow is a good proxy for the travel time.

**Text changed to:** The comparison of the modelling results to observations of δ18O was based on a conceptual model of the seasonal variability and differences between precipitation and runoff (Fig. 2a). The precipitation signal varies on a seasonal basis, creating an amplitude difference. This amplitude is reduced due to groundwater mixing until complete mixing is reached and the groundwater receives the same signal as the long-term precipitation average. There is no or little groundwater recharge during winter because almost all precipitation inputs arrive and accumulates as snow. Hence, we assume that the stream isotopic signature originates from groundwater only (Laudon et al., 2007; Peralta et al., 2015). Consequently, the closer the stream signature comes to the long-term precipitation average, the more the groundwater has been mixed. The groundwater isotopic signature, in turn, should be correlated to the travel time to stream until full mixing of the precipitation signal is reached.

Peralta-Tapia, A., Sponseller, R. A., Ågren, A., Tetzlaff, D., Soulsby, C., and Laudon, H.: Scale-dependent groundwater contributions influence patterns of winter baseflow stream chemistry in boreal catchments. Journal of Geophysical Research: Biogeosciences, 120(5), 847–858. https://doi.org/10.1002/2014JG002878, 2015.

10. Line 191: But does the preceding spring not influence the summer signature also? So how can you assume that the difference between winter and summer signature gives you the young water fraction in the summer if there is another overlaying signal from the spring? Somehow you have to take this into account.

**Reply:** What you state is true and is a weakness with this concept. To account for this as much as possible, we use the relative change, the ratio between the winter/spring and winter/summer, that gives a

relatively good proxy of change. We also excluded June from the summer evaluation, because according to stream chemistry observations at the site, one can still see the impact of the spring snowmelt in most streams during this month. However, we added a section regarding the summer season issue in the discussion section 4.2.

**Text changed to:** Similar to the conditions in spring, the conceptual model predicted that the difference in stream isotopic signature between winter baseflow and summer flow, $\Delta\delta^{18}O_{summer}$, should be correlated to the young water fraction in summer, but with the opposite sign, due to isotopically heavier summer rains (Fig. 2). A larger inter-annual variation in precipitation and high ET likely caused the relationship to be less evident compared to the spring results as the snowmelt conditions are more consistent from year to year. The groundwater signal reaching the streams during the summer season may also be affected by a lingering signal from the snowmelt. However, although less evident than compared to the $\Delta\delta18O_{spring}$, there was still a significant correlation between the average $\Delta\delta^{18}O_{summer}$ and the modelled young water fraction (Fig. 7e).

11. Line 253: Why are the UZ processes only active in the SZ? Should they not be active first and foremost in the UZ?

**Original text:** The ET and UZ processes are only fully active in the uppermost SZ-CL, and here the ET and UZ are calculated at a finer resolution, leading to a detailed calculation of the groundwater table level. If the groundwater table falls below the first SZ-CL, a more simplistic method, not taking capillary rise and all ET-processes into account, was applied

**Reply:** Apart from the UZ itself, the interaction between UZ and SZ processes (including influence from ET) is only fully active in the uppermost SZ calculation layer. If the groundwater table falls below the lower level of the uppermost calculation layer, a more simplistic method is used. The main result of this simplification is that capillary rise is not accounted for when the groundwater table falls below the uppermost SZ calculation layer.

**Text changed to:** The SZ-CLs vary with depth and are thinner closer to the soil surface; the first CLs extend to 2.5 m, 3 m, 4 m, and 5 m, respectively, below the ground surface, with the soil properties and depth extension following the stratigraphy (Table 3). The UZ and SZ interact throughout the soil. If the soil is unsaturated, the UZ discretisation and equations are used. The influence from ET and UZ processes on the SZ is only fully active to the depth of the uppermost SZ-CL. Here, the ET and UZ are calculated at a finer resolution, leading to a detailed calculation of the groundwater table level.

12. Line 259: Do you mean all 'soil layers' not deeper than 2.5 m below the ground surface being average into one soil type?

**Original text:** The thickness of the first SZ-CL in the Krycklan model results in all soils shallower than 2.5 meters being averaged into one soil type.

**Reply:** Yes, you are correct, and we, therefore, changed the sentence for clarification.

**Text changed to:** Following the thickness of the SZ-CL in the Krycklan model, all soils above 2.5 m depth are prescribed as one soil type with hydraulic properties being an average of all the soil types throughout the vertical profile from the ground surface to 2.5 m depth.

13. Line 259-265: You used 'In the Krycklan model…' at the start of sentences three times in this short paragraph.

**Reply:** Thank you for noticing it. We have changed this paragraph to remedy this.

14. Line 276: Streamflow accumulated of what period of time? It makes a big difference if it is accumulated over a day or a year. So I would write '…able to reproduce daily accumulated discharge…'.

**Original text:** The Krycklan flow model was able to reproduce observed stream accumulated discharge, groundwater levels, and timing of precipitation events

**Reply:** We agree with this change.

**Text changed to:** The Krycklan flow model was able to reproduce daily accumulated stream discharge, groundwater levels, and timing of precipitation events

15. Line 306: Numerical constraints restricted the number of particles released to 0.5 particles/10 mm modelled groundwater recharge PER GRID CELL, which corresponds to a total of approximately 0.6 million particles FOR THE ENTIRE MODELED AREA IN THE FIRST YEAR.

**Original text:** Numerical constraints restricted the number of particles released to 0.5 particles/10 mm modelled groundwater recharge, which corresponds to a total of approximately 0.6 million particles. This number of particles was assumed to be enough to capture the timing of recharge patterns (Fig. 4).

**Reply:** We agree with this change.

**Text changed to:** Numerical constraints restricted the number of particles released to 0.5 particles/10 mm modelled groundwater recharge per grid cell, which corresponds to a total of approximately 0.6 million particles for the entire modelled area in the first year. This number of particles was assumed to be enough to capture the timing of recharge patterns (Fig. 4).

16. Line 318: However, that does not mean that the distribution is significantly skewed if the SD is larger than half of the average. I think what you want to express is that if the SD is smaller than half of the average, the distribution is not significantly skewed.

**Original text:** If the distribution is significantly skewed, the SD is larger than half of the average

**Reply:** Thank you for the suggested correction. The text was changed accordingly.

**Text changed to:** If the distribution is not significantly skewed, the SD is smaller than half of the average

17. Line 324: Good job explaining why you chose to use the geometric mean. Makes sense to me.

**Reply:** Thank you, we hoped that our choice would make more sense with this addition to the text.

**Result**

1. Line 354-355: Be more consistent in the way you describe the catchments (size, soil, landuse). For example you say that C20 and C16 are both silt-dominated, yet in the parentheses you only repeat this for C20.

**Original text:** The longest stream MTTs were connected to silt dominated catchments such as C16 and C20. We used some sub-catchments for result representation, but all results are provided in Table 5 and Appendix A1. The displayed sub-catchments were: C2 (small till and forest dominated catchment), C4 (small mire dominated catchment), C20 (small silt dominated catchment), and C16 (the full-scale Krycklan catchment).

**Reply:** Thank you for noticing this error. C16 was supposed to be an example of a larger catchment, while C20 was supposed to be an example of a silt-dominated catchment. We changed the text accordingly.

**Text changed to:** The longest stream MTTs were connected to the larger catchments, such as C16, and the silt dominated catchments such as C20. We used some sub-catchments for result representation, but all results are provided in Table 5 and Appendix A1. The displayed sub-catchments were: C2 (small till and forest dominated catchment), C4 (small mire dominated catchment), C20 (small silt dominated catchment), and C16 (the full-scale Krycklan catchment).

Line 371: However, the groundwater MTT of mire-dominated catchments supposedly showed less variation due to the smaller amount of groundwater recharge in your (not-overland-flow-corrected) particle tracking model. Is that the case?

**Original from text from the discussion:** The lack of synchronicity between mire and silt areas caused greater annual MTTgeo variation for sub-catchments with both features. For example, the MTTgeo for C4, dominated by mires, decreased from 1.5 years to 0.7 years from winter to spring. In contrast, winter MTTgeo for the C20 catchment dominated by silt was 7.7 years, which decreased to 1.5 years in spring. The results also show that groundwater recharge is affected by the soil frost in mires. For example, C4 showed more variations in its seasonal MTTgeo, although C2 (dominated by forest and till) and C4 (dominated by mires) had almost an equal annual MTTgeo (Table 5). In spring, the MTTgeo was shorter in C4 than in C2 due to the surface runoff from the frozen mire. Comparatively, in winter, the MTTgeo of C4 was longer than C2 due to the lower recharge and displacement of older water during the spring. Besides the slightly higher specific discharge from mires (Karlsen et al. 2016), empirical-based studies suggest that the soil frost on mires causes a large fraction of overland flow (Laudon et al. 2007; 2011).

**Reply:** True. For example, in C4, MTT$_{geo,}$ including OL, goes from 1.5 to 0.7 years from winter to spring, while C2 goes from 1.2 to 0.7 years. Only looking at the particle tracking, the MTT$_{geo}$ goes from 1.5 to 1.2 years, i.e., less variation due to smaller amount of groundwater recharge. These results can be found in the Appendix, Table A1. However, C4 shows more variation than C2 in both cases due to an older age in winter, due to less groundwater renewal in spring. Previously the result text got confusing by introducing both OL adjusted and not OL adjusted in the text. We will, however, add a part about this in the discussion section.

**Change in discussion:** The effect of the areal coverage of silty sediments was especially prominent in winter when the range in MTTgeo is between one and almost eight years. The change in seasonal MTTgeo from winter to spring was also largest for the silt dominated catchments. For example, there was a six-year difference for C20 compared to two-year difference for the similar-sized till dominated sub-catchment C6. These intra-annual variations can also be linked to another landscape feature, namely the areal coverage of mires. Mires affected the young water fraction only when new precipitation or snowmelt input into the system occurred in spring and summer. The contrasting hydrological response of mire and silt areas, respectively, caused greater annual MTTgeo variation for sub-catchments with both features. For example, the MTTgeo for C4, dominated by mires, decreased from 1.5 years to 0.7 years from winter to spring. In contrast, winter MTTgeo for the C20 catchment dominated by silt was 7.7 years, which decreased to 1.5 years in spring. The results also show that groundwater recharge is affected by the soil frost in mires. For example, C4 showed more variations in its seasonal MTTgeo, although C2 (dominated by forest and till) and C4 (dominated by mires) had almost an equal annual MTTgeo (Table 5). In spring, the MTTgeo was shorter in C4 than in C2 due to the surface runoff from the frozen mire. Besides the slightly higher specific discharge from mires (Karlsen et al. 2016), empirical studies suggest that the soil frost on mires causes a large fraction of overland flow (Laudon et al. 2007; 2011). Looking only at the part of runoff originating from groundwater (Appendix, Table A1), the MTTgeo for C4 decreases from 1.5 years to 1.2 years. However, C4 still showed more variation than C2, due to longer travel times in winter. The results suggest that the longer MTTgeo in winter was caused by the reduced groundwater renewal of mires in spring, because the main difference between the two being the mire soil frost of C4.

Line 393: How are the annual and seasonal MTTgeo values computed? Are they flow weighted or simple time averages?

**Reply:** We hope that the clarification of the method text 2.5 will suffice. MTTgeo values are calculated from the travel time distributions. The calculated travel time distributions were based on all particles arriving in a stream within a certain period of time, either annually or a specific season, for the entire modelling period. Since the particles are released with the rate of groundwater recharge, these become flow weighted in turn.

**Discussion**

1. Line 445: What do you mean by 'gamma transformation method'?

Original text: …years using long-term isotopic data and a gamma transformation method…

**Reply:** This sentence was changed to make it clearer.

**Text changed to:** … by applying a mathematical method for isotopic dampening to fit a model to the observed stream isotopic response.

2. Line 453: Do you show this scaling function anywhere? Would be good to see it written out.

**Original text:** Therefore, to allow for actual MTTgeo estimates, we corrected the results by reducing the estimated MTTgeo using the overland flow from the flow model as a scaling factor

**Reply:** We answer this question in The main question 2: As we stated in the discussion and the method section, we reduced the age using the amount of overland flow as a scaling factor. We showed the equation in previous iterations of the manuscript, but we received comments that it was not necessary to show because it's such a simple equation. However, we agree that showing the amount of overland flow for different seasons would be a good addition. We added this in the Appendix, Table A1. We also added the equation in the Appendix.

**Text changed to:** $\text{MTT} = \text{MTT}_{Particle\ tracking}\ (1 - \text{fraction OL})$  Eq. A6

3. Line 483-484: These theoretical consideration strengthen the results of a winter MTTgeo being LONGER OR EQUAL TO four years. Are they also telling you that winter MTTgeo is BETWEEN four AND SIX years?

**Original text:** At an average water travel time older than four years, it can be expected that the groundwater has reached full mixing. Hence, older water can no longer be accurately quantified using water isotopes only due to amplitude loss (Kirchner., 2016). These theoretical considerations strengthen the results of a winter MTTgeo between four and six years for the larger sub-catchments as their stream isotopic signatures were close to the long-term precipitation average and, therefore, should have reached complete mixing.

**Reply:** Reading the text again, it might have been misleading. What we are trying to say is that the mix of the isotopes suggests that the water is older or equal to four years and that this is consistent with the calculated MTT, because they are equal to or older than four years.

**Text changed to:** At an average water travel time older than four years, it can be expected that the groundwater has reached full mixing. Hence, older water can no longer be accurately quantified using amplitude dampening of the water isotope signal (Kirchner., 2016). These theoretical considerations strengthen the results of a winter MTTgeo older than four years for some sub-catchments, since their stream isotopic signatures were close to the long-term precipitation average and, therefore, should have reached complete mixing.

4. Line 529: What exactly do you mean by 'synchronicity'? The seasonal patterns? Do you mean that the changes occur at exactly the same time during a year?

**Original text:** All sub-catchments showed similar synchronicity in the seasonal patterns in MTTgeo and young water fraction, but catchment characteristics influenced the magnitude of the seasonal patterns across the landscape. On a landscape level, the main causal mechanism determining the annual MTTgeo was the areal coverage of silt, which overshadowed the importance of other catchment characteristics (Fig. 8, Table 6). This finding stands in contrast to earlier studies in Krycklan by Peralta-Tapia et al. (2015) and Tiwari et al. (2017) that suggested that the groundwater travel times are nonlinearly linked to the catchment size. We found that the small silt dominated C20 catchment was a distinct outlier to such a scale-dependent pattern, indicating that catchment size may not be the primary factor determining the variability (Fig. 6).

**Reply:** The text was re-written to include the answer to your question. What we mean is that all sub-catchment had their shortest $\text{MTT}_{geo}$ during the spring snowmelt and the longest in winter. We removed the word "synchronicity" for clarification.

**Text changed to:** All sub-catchments showed similar seasonal patterns in MTTgeo and young water fraction, manifested as water with shorter travel times discharging in spring and water with longer travel times discharging in winter. Some of the catchment characteristics influenced the

magnitude of these seasonal patterns across the landscape. On a landscape level, the main causal mechanism determining the annual MTTgeo was the areal coverage of silty sediments (Table 1), which largely overshadowed the importance of other catchment characteristics (Fig. 8, Table 6).

5. Line 531: 'the areal coverage of silt'? Do you mean 'the spatially averaged silt content of the soil layer'? Or 'the areal fraction of soils with a certain silt content'?

**Original text:** On a landscape level, the main causal mechanism determining the annual MTTgeo was the areal coverage of silt, which overshadowed the importance of other catchment characteristics (Fig. 8, Table 6).

**Reply:** What we meant by "the areal coverage of silt", was the areal proportion covered by silt soils, referring to the silty sediment column in Table 1. We have tried to clarify this in the text.

**Text changed to:** On a landscape level, the main causal mechanism determining the annual MTTgeo was the areal coverage of silty sediments (Table 1), which largely overshadowed the importance of other catchment characteristics (Fig. 8, Table 6).

6. Line 535: The variability of what?

**Original text:** We found that the small silt dominated C20 catchment was a distinct outlier to such a scale-dependent pattern, indicating that catchment size may not be the primary factor determining the variability

**Reply**: Thank you for noticing that something was missing in this sentence. We have changed it accordingly.

**Text changed to:** The importance of the silt, rather than the catchment area, for the groundwater travel time is most clearly illustrated by the small silt dominated C20 catchment, which was a distinct outlier to such a scale-dependent pattern.

7. Line 532-543: I would also mention the correlation between catchment size and silt content you observed when discussing this issue.

**Reply:** We had a discussion regarding this topic in an earlier version of the manuscript, but it was removed for clarity purposes. We added this section again, in section 4.4., however, we re-wrote the section to work better with the new text.

**Text changed to:** However, it seems that this a spurious relationship, since there is a correlation between the catchment size and the areal coverage of silty sediments (Table 6). The reason is that the silty sediments are located in the lower parts of the Krycklan catchment, which implies that all large catchments contain at least some proportion of silty sediments. The importance of the silt, rather than the catchment area, for the groundwater travel time is most clearly illustrated by the small silt dominated C20 catchment, which was a distinct outlier to such a scale-dependent pattern. This indicates that catchment size may not be the primary factor determining the variability of travel times of different catchments (Fig. 6). Instead, the long travel times in C20 suggest that the groundwater flow velocity is slower in the silt areas than elsewhere in Krycklan, even though the average catchment slope is steeper than comparably sized sub-catchments in till areas (Table 1 and Fig. 1).owever, we found a correlation between the catchment size and the areal coverage of silty sediments. The reason is that the silty sediments are located in the lower parts of the Krycklan catchment, resulting in that all large catchments contain at least some proportion of silty sediments. However, we found that the small silt dominated C20 catchment was a distinct outlier to such a scale-dependent pattern, indicating that catchment size may not be the primary factor determining the variability of travel times of different catchments (Fig. 6). Hence, the long travel times in C20 suggest that the groundwater flow velocity is slower than elsewhere in Krycklan, even though the average catchment slope is steeper than comparably sized sub-catchments in till areas (Table 1 and Fig. 1). The areal coverage of silty sediments may also explain the relatively long travel times of other catchments such as C14. Although C14 is smaller than C15, it still has longer MTT.

8. Line 545: 'Silt fraction effect' is unprecise wording. What fraction? In the soil? Across the landscape?

**Reply:** This section was changed to:

**Text changed to:** The effect of the areal coverage of silty sediments

Line 570: What do you mean by '…silt dominated areas, that have more consistent hydrological conductive with soil depth…'?

**Reply:** The till soils show a greater decrease in hydraulic conductivity than other soils in the area, with fast-flowing water in the shallow part of the soil and slow-moving water in the deeper parts of the soil. This sentence was re-written to:

**Text changed to:** Consistent with this explanation, silt dominated areas had much longer MTTs than comparatively sized sub-catchments underlain by till soils, because till soils have a greater decline in hydrologic conductivity with soil depth (Fig 6, Fig. 8 and Appendix Fig. A1).

**Conclusions**

Line 583-585: This is interesting and you should explain what this would entail. Where and in which way would transit times change in a warmer climate?

**Original text:** In contrast, mires lead to increased fractions of young water, and hence shorter travel times, but mainly in spring when the soil was frozen. As a result of the lower groundwater recharge during the snowmelt, however, the MTTs in mires were, in turn, longer than in forests during the winter. In a warmer climate with reduced soil frost and decreased snowmelt input, we would expect the effect of mires to be reduced while the impact of till and silty sediment soils likely will remain relatively unaffected.

**Reply:** We believe that the section covers part of your question, but it was a bit "wordy" and we hope our changes make the statement clearer. We also extended the discussion regarding climate change.

**Text changed to:** In contrast, mires led to increased fractions of young water, and hence shorter travel times, but mainly in spring when the soil was frozen. However, mire dominated catchments experience longer travel times than similar-sized forested catchments in winter. Generally, for the boreal landscape, a warmer climate is predicted with reduced snow cover and snow duration, accompanied by increases in the frequency of winter thawing episodes and reduction in soil frost (IPCC., 2014; Jungqvist et al., 2014; Brown et al.,2017; Lyon et al., 2018). Our results suggest that these changes would reduce the intra-annual variations of MTT created by the freezing of mires, while the impact on other parts of the landscape would remain relatively low.

**Ref:** IPCC: Climate Change 2014: Synthesis Report. Contribution of Working Groups I, II and III to the Fifth Assessment Report of the Intergovernmental Panel on Climate Change. IPCC, Geneva, Switzerland, 151 pp, 2014.

Jungqvist G., Oni S. K., Teutschbein C., and Futter M. N.: Effect of Climate Change on Soil Temperature in Swedish Boreal Forests. PLoS ONE 9(4): e93957. https://doi.org/10.1371/journal.pone.0093957, 2014.

Brown, R., Vikhamar-Schuler, D., Bulygina, O., Derksen, C., Luojus, K., Mudryk, L., Wang, L., and Yang, D.: Arctic terrestrial snow cover. Chapter 3 in: Snow, water, ice and permafrost in the arctic (SWIPA) 2017, pp. 25–64, Arctic Monitoring and Assessment Programme (AMAP), Oslo, Norway, 2017

Lyon, S.W., Ploum, S.W., van der Velde, Y., Rocher-Ros, G., Mörth, C.-M., and Giesler, R.: Lessons learned from monitoring the stable water isotopic variability in precipitation and streamflow across a snow-dominated subarctic catchment. Arctic, Antarct. Alp. Res. 50, e1454778. https://doi.org/10.1080/15230430.2018.1454778, 2018.

**Figures:**

1. Figure 1: In the caption you use strange sentences ('(b) The soil map used in the Mike SHE flow model and is based on the soil map…', '(c) Depth to bedrock from the Swedish Geological Survey (2014) is shown in meters…').

**Original text:** Figure 1: The Krycklan catchment. (a) Location of sub-catchment and their outlets. The areas are color-coded based on their stream network connections, e.g., all sub-catchments of one colour connect before reaching the white area. For further details of the catchment characteristics, see Table 1. (b) The soil map used in the Mike SHE flow model and is based on the soil map (1:100,000) from the Swedish Geological Survey (2014), combined with field investigations. (c) Depth to bedrock from the Swedish Geological Survey (2014) is shown in meters below the ground surface. (d) Catchment topography, shown as meters above sea level (m.a.s.l.).

**Reply:** Thank you for noticing this error. We have changed this text to:

**Text changed to:** Figure 1: The Krycklan catchment. (a) Location of sub-catchment and their outlets. The areas are colour-coded based on their stream network connections, e.g., all sub-catchments of one colour connect before reaching the white area. For further details of the catchment characteristics, see Table 1. (b) The figure shows the soil map used in the Mike SHE flow model which is based on data from the Swedish Geological Survey soil map (1:100,000), 2014 and field investigations. (c) Soil depth to bedrock map taken from the Swedish Geological Survey, 2014, and is shown in meters below the ground surface (m.b.g.s.). (d) Catchment topography, shown as meters above sea level (m.a.s.l.).

2. Figure 3: There are no labels on the graphic. Where is the UZ, the SZ? I don't see the ten calculation layers either.

**Reply:** The figure is a schematic figure and not adapted to the specific model for the Krycklan catchment and its associated calculation layers. The ten calculations layer would not be possible to illustrate in a figure at this scale since they are so thin in comparison to the total soil depth. The purpose of the figure is to show the main interactions between the different parts of the system (ET, snow, precipitation, surface water and groundwater. Since the groundwater table is allowed to fluctuate from totally saturated and ponded conditions to large depth, the boundary between the UZ and SZ is not defined. However, as suggested, we added some labels to the figure describing the main parts of the model. See figure below.

[Figure]

**Figure 3 Schematic of a general Mike SHE model set up.** Precipitation falls on the ground as rain or snow. Evapotranspiration (ET) processes include canopy interception, open surface evaporation, root uptake, and soil evaporation from the unsaturated zone (UZ). The overland flow (OL), saturated zone (SZ), and UZ interact depending on the saturation level. The SZ is divided into ten calculation layers (CL), while the UZ has a much finer description. Streamflow is modelled through Mike 11 and is not restricted to the Mike SHE resolution. The figure is used on the courtesy of SKB. Figure illustrator: LAJ.

3. Figure 5: Are these TTDs specific for a certain point in time? Are they forward or backward TTDs? Are they weighted averages of multiple TTDs (master distributions)?

**Reply**: Please see the answer to Question 1 of the Main questions: We like to answer these questions together because they handle the same issue. Figure 5 shows the distribution of all the particles exiting the model though a catchment stream over the entire modelling period. To clarify this, we changed the figure text to state that the statistics are based on all particles over the entire modelling period. We also added a part in section 2.5 to clarify the creation of the travel time distributions. It is correct that we take all the particles exiting a stream within a certain period of time and take the average of that distribution.

**Text changed to:** Section 2.5: The time it took for particles to reach a stream or lake via groundwater (hereafter called 'travel time') was calculated for each sub-catchment. The calculated travel time distributions were based on all particles arriving in a stream within a certain period of time, either annually or for a specific season, for the entire modelling period. The distributions were analysed using four statistical measurement tools, the arithmetic mean, the geometric mean, the median, and the standard deviation (SD).

4. Figure 6: The blue outline of the surface runoff gives the impression that there is surface runoff happening constantly in b), c) and d). Rather make it a blue fill without an outline (like the green fill for the groundwater contributions).

**Reply:** We agree with this change and believe that removing the blue outline made the figure easier to understand.

5. Figure 7: I recommend using the same scale for MTT, BC and YWF for all axes. This way the different relationships (as well as the season-specific variabilities) become much clearer.

**Reply:** We understand the reviewer's point of view, and we have changed the figure accordingly. Although the individual point was easier to see in the original plot, this version shows better the seasonal changes.

[Figure]

6. In the caption better write something along the lines of 'Relationships of seasonal MTTgeo and young water fractions with seasonal stream isotopic composition and base cation concentration'.

**Original text:** Figure 7: Results of seasonal young water fraction (YWF) and MTTgeo compared to stream isotopic composition and base cation concentration. Note that δ18O results are for 13 sites, while the BC record comprises all 14. The sub-plots (a) to (f) show the δ18O (winter) or Δδ18Ospring/summer and BC concentrations as a function of the MTTgeo in winter, spring, and summer, respectively. The standard error of the mean (SEM) shown as whiskers denotes variations in field observations.

**Reply:** We agree to the suggested changed and the new figure text is written as:

**Text changed to:** Relationships of seasonal MTT$_{geo}$ and young water fractions (YWF) with seasonal stream isotopic composition and base cation concentration. Note that δ$^{18}$O results are for 13 sites, while the BC record comprises all 14. The sub-plots (a) to (f) show the δ$^{18}$O (winter) or Δδ$^{18}$Ospring/summer and BC concentrations as a function of the MTT$_{geo}$ in winter, spring, and summer, respectively. The standard error of the mean (SEM) shown as whiskers denotes variations in field observations.

**Tables**

1. Table 1 and 2: Just a suggestion: Why not grouping the catchments and nested sub-catchments instead of listing by catchment number (so all the green catchments together and all the red and blue catchments together as well)? The numbering is somehow arbitrary.

**Reply**: This has changed from iteration to iteration because some people want the tables in name order, others in size order and yet others in colour order. We do agree to collect them in colour order is the neatest version, so we changed the tables to firstly be ordered in colour order, thereafter in name order. The numbering may indeed come across as somewhat arbitrary, but they are long-established in publications from Krycklan.

2. Table 5: I would rearrange the order in all the tables and group the catchments as mentioned earlier. In the caption rather write '…that is younger than three months…' instead of '…that is less than three months…'.

**Original text:** The geometric mean of the travel time distribution ($MTT_{geo}$) is adjusted for the overland flow. The young water fraction (YWF) includes overland flow and groundwater that is less than three months (%). An extended version of the results, including arithmetic mean, median, and SD, is included in the Appendix, Table A1.

**Reply:** We agree with this change and we changed the text accordingly. See our answer regarding colour order for table 1 and 2. The tables were changed to order of colour.

**Text changed to:** The geometric mean of the travel time distribution ($MTT_{geo}$) is adjusted for the overland flow. The young water fraction (YWF) includes overland flow and groundwater younger than three months (%). An extended version of the results, including arithmetic mean, median, and SD, is included in the Appendix, Table A1.

3. Table 6: Why did you remove the values that are smaller than 0.5? Please add them again - in the caption you are still mentioning the darker colors. Otherwise I like the new design (just make sure you either close all or none of the boxes around the seasons).

**Reply:** Thank you for this comment. We agree that the new design works better. In the new design, we tried to make the table smaller, yet with all the necessary information. However, we agree that we should add all values again, and we changed the table accordingly. We noticed an error in the table as well, and this error was also corrected.

**Table 6: Correlation matrix – young water fraction (YWF), geometric mean travel time (MTT$_{geo}$), and catchment characteristics.** The catchment characteristics include the log catchment size (Log A), the areal coverage of mires (Mire), and the areal coverage of silt (Silt). The table includes annual (grey), winter (blue), spring (green), and summer (orange) results. Darker colours show |r| > 0.5 with the connected p-value according to [a] p<0.05 and [b] p>0.05.

|  | **Winter season** | | | | | **Summer season** | | | | |
|---|---|---|---|---|---|---|---|---|---|---|
|  | **Log A (km²)** | **Mire (%)** | **Silt (%)** | **MTT$_{geo}$ (year)** | **YWF (%)** | **Log A (km²)** | **Mire (%)** | **Silt (%)** | **MTT$_{geo}$ (year)** | **YWF (%)** |
| **Log A (km²)** | 1 | 0.02 | 0.58 [a] | 0.64 [a] | -0.08 | 1 | 0.02 | 0.58 [a] | 0.68 [a] | 0.20 |
| **Mire (%)** | 0.02 | 1 | -0.37 | -0.34 | -0.14 | 0.02 | 1 | .0.37 | -0.50 [b] | 0.91 [a] |
| **Silt (%)** | 0.58 [a] | -0.37 | 1 | 0.92 [a] | -0.43 | 0.58 [a] | -0.37 | 1 | 0.80 [a] | -0.20 |
| **MTT$_{geo}$ (year)** | 0.63 [b] | -0.51 [b] | 0.90 [a] | 1 | -0.21 | 0.55 [a] | -0.55 [a] | 0.92 [a] | 1 | -0.28 |
| **YWF (%)** | -0.02 | 0.96 [a] | -0.39 | -0.53 [b] | 1 | 0.11 | 0.95 [a] | -0.29 | -0.52 [b] | 1 |
|  | **Annual** | | | | | **Spring season** | | | | |

**Appendix**

Line 869-870: 'whereas'?

**Reply:** This word was left from an earlier iteration but is not necessary in the text as it stand now. This word war removed.

**References**

Remondi, F., Kirchner, J. W., Burlando, P., & Fatichi, S. (2018). Water flux tracking with a distributed hydrological model to quantify controls on the spatio-temporal variability of transit time distributions. Water Resources Research, 54(4), 3081-3099.

Heidbüchel, I., Yang, J., Musolff, A., Troch, P., Ferré, T., & Fleckenstein, J. H. (2020). On the shape of forward transit time distributions in low-order catchments. Hydrology and Earth System Sciences, 24(6), 2895-2920.

**Reply:** Thank you for the suggested references. These have been used in the introduction and added to the reference list.

---

## Author Response (AR4)

**Minor Revision**

Comments to the Author:

Dear Elin and co-workers,

Congratulations for your last revisions. Thank you for all the efforts which went into your publication in our special issue. I have to apologise for the delay in handling your latest version, which was mainly due to my own uncertainty how to proceed. While in my view your paper is ready for publication in general, I came across a couple of unnecessarily sloppy details, which might be easy to resolve. Moreover, I am under the impression that your publication could be a very nice example for transparent science including the fundamental analysis code. Hence I decided to ask you once more for a quick feedback in a round of minor revisions before final acceptance.

I hope you find my comments helpful to fortify your paper and its outreach. I would be very happy if you can find a way to provide a more specific and complete reference to the data and at best even a condensate of your code.

All the best. Thank you again.

Conrad

**Reply:** Thank you for this positive response, your hard work and consideration of our manuscript. We have answered below each of your respective questions in brown. The major changes are also listed in the "Table of major changes". We removed all colors from the tables as instructed by the editorial support team of Copernicus Publications We also uploaded the model code and set up to Safe Deposit. Safe Deposit (managed at Svartberget research station) is a system for collection of forest research and materials, for past and ongoing projects. The intention is to have a long-term storage solution that is accessible for project colleagues and the public.

**Table of major changes**

- Data, including model inputs, model set up and main chemistry data was uploaded to Safe deposit
- Due to restrictions by HESS, colors were removed from all tables

**Major comments**

1. First of all the data and code: Your statement links to the Krycklan data website, which is fine. I made the effort to download the data and do some very quick plotting. What I could not find is data about the isotopic concentration in the precipitation. Further the adjusted 18O observations appear to be lacking.

**Reply:** We apologies that this data was missing from the Krycklan database (Krycklan Database, 2021). The website is undergoing updates, so all data are not re-uploaded there yet, but the data will be uploaded shortly. However, all chemistry data for this study can in the meantime be acquired from Safe Deposit (Jutebring Sterte et al., 2021).

The original $^{18}O$ data for all streams are available from the Krycklan database and Safe Deposit. $^{18}O$ data were adjusted in this study using the Fractionation Correction from Peralta-Tapia et al. 2015. Reviewers of earlier versions of the manuscript thought the equation was not necessary to include in the manuscript. However, we added the regression equation in the Appendix, Eq. A7.

Peralta-Tapia, A., Sponseller, R. A., Ågren, A., Tetzlaff, D., Soulsby, C., and Laudon, H.: Scale-dependent groundwater contributions influence patterns of winter baseflow stream chemistry in boreal catchments. Journal of Geophysical Research: Biogeosciences, 120(5), 847–858. https://doi.org/10.1002/2014JG002878, 2015.

Krycklan Database, Hydrological Research at Krycklan Catchment Study, available at: https://www.slu.se/Krycklan, last access March 2021

DHI, Mike powered by DHI - MIKE software, available at: https://www.mikepoweredbydhi.com, last access March 2021

Jutebring Sterte, E., Johansson, E., Sjöberg, Y., Huseby Karlsen, R., and Laudon, H.: Krycklan Mike SHE 2020; Safe Deposit, available at: https://www.safedeposit.se/projects/166, last access March 2021.

2. Moreover, I do not see how I could possibly rework your results when starting MikeSHE from scratch and without the geo data. Since it is the year 2021 and since our special issue is also about transparency in data and code, I would like to persuade you to scope your options to publish some sort of repository alongside your paper. I simply see that your study is a very nice example of TTD and MTT analysis and people will hook to your work much more easily, when it is possible to really follow your calculations. I fully understand that the data policy of the Krycklan data explicitly seeks to avoid a copy of this data. This is fine. But for your further data and/or model setup/model results this could be easily feasible through github or any other DOI providing repository.

**Reply:** Thank you for this comment. We have now made all the model input data, including geological layers and input files, available at Safe Deposit (Jutebring Serte et al., 2021). The GIS, chemistry, and environmental data for Krycklan is available at the Krycklan database, under "Data service", "Krycklan chemistry and environmental data" and "Krycklan GIS data" respectively (Krycklan Database, 2021). Chemistry data is also available through Safe Deposit. Note that licenses and software necessary to run the model can be acquired from DHI.

Krycklan Database, Hydrological Research at Krycklan Catchment Study, available at: https://www.slu.se/Krycklan, last access March 2021

DHI, Mike powered by DHI - MIKE software, available at: https://www.mikepoweredbydhi.com, last access March 2021

Jutebring Sterte, E., Johansson, E., Sjöberg, Y., Huseby Karlsen, R., and Laudon, H.: Krycklan Mike SHE 2020; Safe Deposit, available at: https://www.safedeposit.se/projects/166, last access March 2021.

**Minor comments**

**Abstract:**

1. Why is the "Krycklan" catchment name avoided until L112? I suggest to include it in the abstract near L25, since it is a well known experimental catchment. This could also strengthen the reference of the third sentence. It will also help meta analyses.

**Reply:** We agree to this change and "Krycklan" was added to L25.

**Changed to:** In this study, a particle tracking model approach in Mike SHE was used to investigate the pathway and its associated travel time, of water in 14 partly nested, long-term monitored boreal sub-catchments of the Krycklan catchment (0.12-68 km$^2$).

2. L31ff. The sentence was modified and is fragmented now. Maybe: Catchments with mixed soil landscape settings typically displayed larger variability in seasonal MTTgeo, as contrasting hydrological responses between different soils (e.g., mires, till and silty sediments) are integrated. One more detail: I am not sure if mires and till can count as "soil types" (which may be a matter of languages used in the different classifications). Maybe simply "soils" would solve the issue?

**Reply:** We agree to the suggestion and have applied it to the new version of the manuscript accordingly.

We also agree that "mire" is not a soil type – or even a soil. Peat would be the appropriate term to describe the soil type in mires so this has been changed. However, we believe that "soil type" is a better and more specific word than just "soil" to describe the relevant differences in the landscape. Soil would rather refer to, for example, spodosol, histosol etc., which is not what we mean. Soil type is the appropriate term in soil science to describe the texture and composition of a soil. It was also a classification of soil types (and not soils) that was used to parameterize the model.

There are several national and international systems that are used to classify soil types, but most (if not all) in some way recognize the difference between sorted and unsorted soils. Especially in areas like Scandinavia, where there has been profound glacial influence on formation of soils, the perhaps most important distinction between different types mineral soils is that between unsorted glacial sediments (i.e. till) and sorted fluvial or glaciofluvial sediments. This is therefore how soil types are named and classified at least in a Swedish context, and that includes the maps from the Swedish Geological Survey that were used to set up the model. The soil type can then be described in more detail as, for example, sandy till or clay till, but in the available data they were just classified as till. Accordingly, we believe that till is the adequate way to describe the soil type in this case.

**Main Document:**

3. L135f. Maybe: the characteristic VEGETATION of this boreal landscape are Scots pine and Norway spruce… I would not see the vegetation as characteristic features.

**Reply:** We agree, this is a great improvement

4. L141 Why is the hydraulic conductivity given here? 5e-5 m/s is not coherent with table 3. Moreover it is difficult to refer to, when the other soil landscape characteristics are left for sec 2.3. Further a decreasing hydraulic conductivity is nothing specifically rare… I suggest to check again, what information is really characterising the different soil landscapes.

**Reply:** We agree to give a specific number here was confusing and have changed this sentence accordingly. However, the decreasing hydraulic conductivity is characteristic for glacial till compared to other soil types, and we believe that this is the key parameter that can explain much of the variation in transit times in the landscape. The other soil types in the catchment, such as the sandy and silty sediments have mainly been compacted by their own weight. This is probably true also for the upper part of the glacial till, which is so-called ablation till. However, beneath the top layer of the glacial till, there is basal till. The basal till was deposited underneath the glacier, which in turn gives the deeper till a significant compaction with soil depth compared to other soil types. We changed this section to clarify this.

**Change to:** Krycklan has a landscape distinctively formed by the last ice age (Ivarsson and Johnsson, 1988; Lidman et al., 2016). At the higher elevations to the northwest, located above the highest postglacial coastline, the soils can reach up to 15-20 m in thickness. Here, the soil primarily consists of glacial till, and the landscape is intertwined with lakes and peatlands. The deeper soils consist of basal till, which was deposited and compacted under the moving ice. In contrast, the shallower till layers, consist primarily of ablation till, which is less compact, since it mainly has been compacted by its own weight (Goldthwait, 1971). This causes a decreasing hydraulic conductivity with depth, which is characteristic for glacial till in northern Sweden (Bishop et al., 2011; Nyberg, 1995; Seibert et al., 2009). At lower elevations, the soils consist of fluvial and glaciofluvial deposits of primarily sandy and silty sediments. Compared to the soil at higher elevations in the catchment, these deposits can reach thicknesses up to approximately 40 to 50 m and have a hydrological conductivity that is more constant with depth because these soil types mainly been compacted only by their own weight.

5. Fig. 2: I still struggle to read this conceptual figure – not because it was too complex but rather not really in touch with the data. As mentioned, I tried to catch up with your approach by downloading some data from the database. I see your point in using the difference from the $\partial 18O$ winter. But what I could not really trace back is if the winter streamflow covers a range and the difference is calculated to either upper and lower bounds of this range, as the figure suggests. Your text does refer to the averages without any distribution. When checking in with the data, I could see the spring event as very pronounced signal while the summer is far less depicted. I was wondering if this figure might benefit from using real data or at least a curve for an idealised streamflow? Could you please clarify the winter streamflow range reference? Did I understand correctly that the isotopic concentration "raw data" from the Krycklan data base was adjusted (L179) and that the final merged product of 18O concentrations was used as reported in Table 2?

**Reply:** We tried to make the figure clearer by including the streamflow data of the sub-catchment C4, see figure below. Displayed in Table 2 are the average annual winter signatures ($\partial^{18}O$ winter). For example, on average C4 signature is approximately -13.1 ‰, for the years 2008-2018. The closer this average is to the long-term precipitation the older the winter baseflow is.

Every year, there is a change from winter baseflow to spring flood, i.e., winter -spring. The values listed in table 2 are the mean of the difference between the average spring signature and average winter signature for all years (2008-2018). I.e., on average, the change from winter to spring signature is -1.1 for C4. The same method was used for summer, but here the difference is made from winter and summer signature averages. The fact that the difference between winter and summer is less distinct than winter and spring, is also reflected in the results (Fig. 7c and e).

We also clarified the lengths of the winter, spring, and summer seasons. The winter season occurs early December to late February, from negative air temperatures, until increasing temperatures causes small snowmelt events. The spring season is the period of the main spring flood. The isotopic signal is more sensitive than the BC data. Even though the BC data is less affected by early snowmelt in March and small precipitation inputs in November, we decided to use the same dates for evaluation of 18O and BC for clarification. We also added these dates in a table (now called A1) in Appendix.

**Table A1: Dates used for chemistry investigation.** The dates are start and end dates for observations within the seasons classified as winter, spring, and summer respectively. Note that BC only includes dates from 2008-2016.

| Year | Season - Winter | | Season - Spring | | Season - summer | |
|---|---|---|---|---|---|---|
| | Start date (year-month-day) | End date (year-month-day) | Start date (month-day) | End date (month-day) | Start date (month-day) | End date (month-day) |
| 2008 | 2008-01-16 | 2008-02-13 | 04-18 | 05-12 | 07-01 | 09-28 |
| 2009 | 2008-12-09 | 2009-02-12 | 04-20 | 05-12 | 07-08 | 09-15 |
| 2010 | 2009-12-15 | 2010-02-10 | 04-15 | 05-14 | 07-06 | 09-28 |
| 2011 | 2010-12-16 | 2011-02-21 | 04-18 | 05-09 | 07-04 | 09-28 |
| 2012 | 2011-12-20 | 2012-02-14 | 04-17 | 05-14 | 07-03 | 09-25 |
| 2013 | 2012-12-18 | 2013-02-20 | 04-17 | 05-10 | 07-04 | 09-20 |
| 2014 | 2013-12-17 | 2014-02-25 | 04-22 | 05-13 | 07-08 | 09-29 |
| 2015 | 2014-12-16 | 2015-02-17 | 04-17 | 05-12 | 07-14 | 09-22 |
| 2016 | 2015-12-15 | 2016-02-15 | 04-18 | 05-12 | 07-12 | 09-20 |
| 2017 | 2016-12-06 | 2017-02-08 | 04-18 | 05-09 | 07-11 | 09-21 |
| 2018 | 2017-12-05 | 2018-02-13 | 04-17 | 05-07 | - | - |

$\delta^{18}O_{winter}$ = average isotopic signature for winter streamflow, correlated to the groundwater age

$\Delta\delta^{18}O_{spring}$ = difference between average winter and average spring stream isotopic signature the difference is correlated to the young water fraction

$\Delta\delta^{18}O_{summer}$ = difference between average winter and average summer stream isotopic signature the difference is correlated to the young water fraction

**Figure 2: Conceptual figure of travel time to stream vs stream isotopic signature (a and b), and stream base cation concentration (c).** (a) The connection between $\delta^{18}O$ and travel time to stream, where the sine curve shows the annual variations of $\delta^{18}O$ in precipitation, and approximate seasonal winter, spring, and summer stream compositions are marked and exemplified by the average annual changes of C4. In winter, the travel times are related to the average deviation in the isotopic signature between the winter baseflow and the long-term precipitation. In spring, the fraction of young water is correlated to the difference between the average spring stream signature and the average winter baseflow. In summer, the fraction of young water is correlated to the difference between the average summer stream signature and average winter baseflow. (b) Seasonal $\delta^{18}O$ averages for three example streams: C2, C4 and C16. (c) The connection between base cation (BC) concentration and soil contact time. The longer time the water spent in the mineral soil, the higher the stream concentrations of BCs will be due to soil weathering.

Table 2: I could not see any isotopic concentration records of the precipitation in the Krycklan database.

**Reply:** We apologies that this data was missing from the Krycklan database. However, as previously stated, all data can in the meantime be acquired from Safe Deposit. Here, we also added the specific chemistry data used in this study in a zip-folder together with the model set up and input files.

L285f. Did I understand correctly that until here, MikeSHE was setup based on common rainfall, runoff and groundwater dynamics but without information about the isotopic concentration dynamics or BC? Maybe referring to "all available data" is sparking some confusion?

**Reply:** We understand the confusion. We removed the "all available data" comment in the description of the Mike SHE model. No chemistry data was used to drive the Mike SHE flow or particle tracking. The chemistry data was only used to evaluate the model results.

Fig. 7: Here the concept explained in Fig. 2 is applied to evaluate the model performance. Referring to the last comment, 18O and BC are only used as evaluation reference, correct? I suggest to either include the Δ in the subfigure titles in c and e. Or even more easy to digest why not printing $\partial 18O$ Winter $- \partial 18O$ Summer?

**Reply:** You are correct. No chemistry data was used to drive the Mike SHE flow or particle tracking. The chemistry data was only used to evaluate the model results. We liked your first suggestion and added Δ in the subfigures c and e, see figure below.

[Figure]

L476f.: Did I understand correctly that MikeSHE simply adds a particle for every 20 mm recharge but that the isotopic concentration dynamics of the precipitation was not used? Does this imply that the percolation is actually neglected? In addition I find your argument slightly self-fulfilling: Since you focus your analysis on groundwater dynamics you can only retrieve dominance of the groundwater domain. All interpretation whether percolation and unsaturated zone water is negligible or not is actually just reiterating your assumption. I think this could be formulated more clearly here.

**Reply:** The chemistry data was only used for model evaluation. It was not used as a driver of the model.

Yes, the percolation is neglected. However, as stated in section 2.5 we do account for the overland flow when assessing MTT. Furthermore, the uncertainty of the time spent in the unsaturated zone is reduced whenever the groundwater table and recharge rates are high, which is when most particles are placed in the groundwater.

L542: I suspect "this" refers to the finding about catchment size. Maybe something like this is more clear? "However, a correlation of travel times to catchment size appears to be a spurious relationship."

**Reply:** We liked the suggested change and have applied it to the new version of the manuscript.

L602: why is there a dot after IPCC?

**Reply:** Thank you for noticing this error, it has been fixed and the dot was removed.